# From Rashomon Theory to PRAXIS: Efficient Decision Tree Rashomon Sets

**Zakk Heile** [*1]  **Hayden McTavish** [*1]  **Varun Babbar** [1]  **Margo Seltzer** [2]  **Cynthia Rudin** [1]

## Abstract

Standard machine learning pipelines often admit many near-optimal models. These "Rashomon sets" pose a range of challenges and opportunities for uncertainty-aware, robust decision making. They allow users to incorporate domain knowledge and preferences that would otherwise be difficult to specify directly in an objective, and they quantify diversity among valid models for a given training dataset and objective function. However, computation of Rashomon sets, even for simple, interpretable model classes such as sparse decision trees, continues to require immense memory and runtime resources. We present PRAXIS, an algorithm to approximate this Rashomon set with orders of magnitude improvement in runtime and memory usage. We validate that PRAXIS regularly recovers almost all of the full Rashomon set. PRAXIS allows researchers and practitioners to scalably model the Rashomon set for real-world datasets.

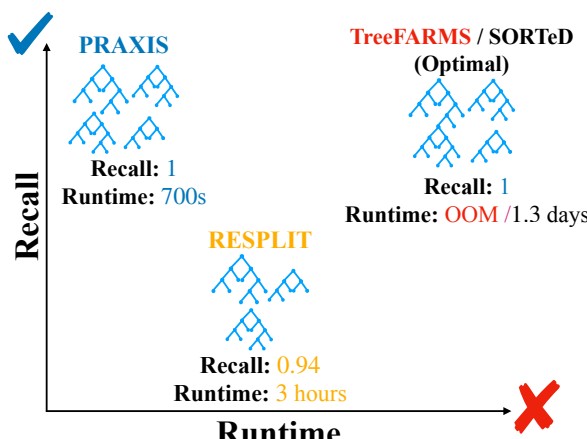

*Figure 1.* An illustration of PRAXIS and other Rashomon set algorithms on the News dataset (Fernandes et al., 2015b), $\lambda = 0.02, \varepsilon = 0.03$, depth $= 5$. PRAXIS runs orders of magnitude faster than competitors while ensuring near-perfect recall relative to optimal methods.

## 1. Introduction

Model selection is crucial to any machine learning pipeline. The Rashomon effect (Breiman, 1984) refers to the phenomenon that many models can be well-justified for a given dataset and objective. A recent framework for model selection in the presence of this effect is the *Rashomon set paradigm* (Xin et al., 2022; Rudin et al., 2024), whereby an algorithm first enumerates or represents the set of all plausible models that fit the data well (i.e., the Rashomon set), and then the user interacts with the Rashomon set to select the model that aligns best with their needs. This paradigm

---

[*]Equal contribution [1]Department of Computer Science, Duke University, Durham, USA [2]Department of Computer Science, University of British Columbia, Vancouver, Canada. Correspondence to: Zakk Heile <zakk.heile@duke.edu>, Hayden McTavish <hayden.mctavish@duke.edu>.

*Proceedings of the 43$^{rd}$ International Conference on Machine Learning*, Seoul, South Korea. PMLR 306, 2026. Copyright 2026 by the author(s).

Code for PRAXIS is available at https://github.com/zakk-h/PRAXIS

is particularly useful when users are unable to formalize all of their goals and constraints in advance. Users may want to find a model that maximizes fairness, obeys causal hypotheses, uses certain features, and/or follows specific structural constraints. These modeling goals become simpler when the Rashomon set has been enumerated, because only a simple loop through the set is required to optimize any secondary objective. The method for uncovering this Rashomon set depends on the model class being considered. For instance, Rashomon sets of generalized additive models (GAMs) can be efficiently approximated by sampling around a convex hyperboloid centered at the optimal weight vector (Zhong et al., 2023). Uncovering the Rashomon set for discrete, non-parametric model classes such as decision trees, however, can be a massive computational undertaking in both runtime and memory, as even finding a single optimal decision tree is NP-hard. As an example, Hu et al. (2019b) show that the size of the search space of decision trees of depth 4 with only 20 features is $\approx 8.4 \times 10^{18}$ trees.

Although two recent algorithms (Xin et al., 2022; Arslan et al., 2026) can exactly uncover the Rashomon set for sparse decision trees, their runtime and memory requirements scale exponentially with the depth budget and number of features.

RESPLIT (Babbar et al., 2025) provides an approximate Rashomon set, at significant cost to approximation quality and still with worst-case exponential cost per tree recovered.

We propose a new family of Rashomon set approximation algorithms with polynomial computation time per member of the Rashomon set. Our algorithms use a proxy subroutine for evaluating feature splits; this subroutine efficiently computes the objectives of high quality individual trees on a given subproblem.

Our approach, PRAXIS (**P**roxy-guided **R**ashomon set **A**ppro**X**imat**I**on**S**), achieves orders-of-magnitude improvements in both runtime and memory efficiency compared to state-of-the-art exact and approximate approaches while still recovering nearly the full Rashomon set.

Our contributions are as follows.

(1) We introduce a novel, flexible framework for efficient approximation of the Rashomon set of decision trees.

(2) We demonstrate reliable recovery of the Rashomon set with our approximations and discuss theoretical conditions under which this recovery is guaranteed.

(3) We demonstrate that our method is orders of magnitude faster and more memory efficient than the previous state-of-the-art methods and provide asymptotic analysis to match.

## 2. Related Work

**Optimal Individual Trees.** Decision trees have been established for decades as interpretable, scalable classifiers, with widely cited implementations such as CART (Breiman, 1984) and C4.5 (Quinlan, 2014). Early decision tree approaches were greedy, building trees from the root, adding splits one at a time according to heuristics, without looking back to see if the splits could be improved. Researchers have substantially improved the accuracy of individual sparse decision trees via global optimization of performance and sparsity, alongside a range of techniques for computational efficiency (Lin et al., 2020; Aglin et al., 2020; Hu et al., 2019a; Demirović et al., 2022; van der Linden et al., 2023; Bertsimas & Dunn, 2017). A recent survey found that many of the most scalable approaches succeed by leveraging tree-specific branch and bound logic with dynamic programming (Costa & Pedreira, 2023). Several tree optimization works have observed that such tree-specific approaches can be framed as search over an AND/OR graph (Sullivan et al., 2024; Chaouki et al., 2025); our algorithm also corresponds to an AND/OR graph search (see Appendix B.2).

**Optimal Rashomon Sets.** Several approaches for finding optimal single trees can be extended to find all trees within a small multiple of the optimal objective – that is, the Rashomon set of trees (Xin et al., 2022; Arslan et al.,

2026). Additional works exist for special cases of trees (Mata et al., 2022; Ciaperoni et al., 2024; Babbar et al., 2026). The ability to capture Rashomon sets enables a range of powerful downstream applications, such as adding robustness (Hsu et al., 2026), estimating variable importance (Donnelly et al., 2023; 2026), or providing customization and control to domain experts and practitioners (Rudin et al., 2024). These advances are significant but struggle to scale to larger practical datasets: they are combinatorially complex in memory and runtime, particularly with respect to the number of features in the dataset.

**Individual Tree Approximations.** Several approaches have improved the scalability of optimal and approximately optimal decision tree algorithms through better handling of continuous features (Mazumder et al., 2022; Brița et al., 2025) or incorporating carefully founded heuristics (McTavish et al., 2022; Blanc et al., 2024; Demirović et al., 2023; Kiossou et al., 2022; 2024; Kiossou & Schaus, 2026). Of particular relevance to our work is the LicketySPLIT algorithm (Babbar et al., 2025). Starting at the root, this approach considers all possible splits and evaluates each based on completions of the tree with a greedy algorithm. It selects the best initial split conditioned on greedy completions, then recurses. This split-selection heuristic is a rollout procedure (Bertsekas et al., 1997); consequently, the overall LicketySPLIT algorithm belongs to the class of pilot algorithms (Duin & Voß, 1994; Voß et al., 2005). Our method PRAXIS can be viewed as a pilot method that outputs not a single solution, but a set of near-optimal solutions. Like any pilot method, it uses a subroutine for heuristic completions. For ease of nomenclature, we call this subroutine a *proxy method* (defined in Definition 3.1), though this proxy can be quite complicated. Empirically, we find that the objective of a single-tree pilot method approach like LicketySPLIT serves as an excellent proxy, both for accurate results and for caching efficiency.

**Rashomon Set Approximations** Relatively few works have focused on computational benefits for approximate Rashomon sets, which can be complex given the distinct nature of the optimization task. Babbar et al. (2025) provide RESPLIT, a scalable Rashomon set approximation based on a hybridization of greedy heuristics and branch-and-bound search. RESPLIT's optimization strategy is orthogonal to our approach: it solves simpler subproblems optimally and then combines them approximately, whereas we instead approximate the full problem directly. In addition, a commonly used baseline is to take repeated bootstraps of a dataset and run single tree optimization on each bootstrap; this is explored in both exact decision tree Rashomon set works (Arslan et al., 2026; Xin et al., 2022). Both papers find bootstrapping to be inefficient, leading to many missed Rashomon set members, both when running an opti-

mal method on each bootstrap and when running a greedy method like CART (Breiman, 1984). We also compare to bootstrapping our proxy algorithm, LicketySPLIT, showing that our approach (PRAXIS) yields a dramatically better approximation of the Rashomon set.

## 3. Methodology

### 3.1. Notation

Let $D = \{(x_i, y_i)\}_{i=1}^n$ be a dataset of size $n$, where $y_i \in \{0, 1\}$ and each $x_i \in \{0, 1\}^k$ has $k$ binary features (possibly binarizations of continuous or categorical features). A binary decision tree $t \in \mathcal{T}$ is a function $\{0, 1\}^k \to \{0, 1\}$. A depth 0 tree, i.e., a leaf, makes a single point prediction for any sample. Any other tree has a simple splitting function (defined as $\text{split}_t(x_i) = x_{ij}$ for some $j \in [1, k]$), as well as two subtrees $t_{\text{left}} \in \mathcal{T}, t_{\text{right}} \in \mathcal{T}$ s.t. $\text{depth}(t) = 1 + \max(\text{depth}(t_{\text{left}}), \text{depth}(t_{\text{right}}))$; it returns:

$$t(x_i) = t_{\text{left}}(x_i)\text{split}_t(x_i) + t_{\text{right}}(x_i)(1 - \text{split}_t(x_i)).$$

That is, a tree returns the prediction of its left or right subtree based on a queried feature. We denote $D_t$ as the subset of dataset $D$ for which tree $t$ is assigned to make predictions; for the root tree, this is equal to $D$, and this is then recursively defined as

$$D_{t_{\text{left}}} = \{(x_i, y_i) \in D_t \text{ s.t. split}_t(x_i) = 1\}$$

$$D_{t_{\text{right}}} = \{(x_i, y_i) \in D_t \text{ s.t. split}_t(x_i) = 0\}$$

Let $|t|$ denote the number of leaves in tree $t$. We use an objective that penalizes both the number of misclassifications and the number of leaves, with a per-leaf miss penalty, $\gamma$:

$$\text{Obj}(t, D, \gamma) = \gamma|t| + \sum_{i=1}^n \mathbb{1}\{t(x_i) \neq y_i\}.$$

This objective is analogous to prior Rashomon set formulations (Xin et al., 2022; Arslan et al., 2026), which use a regularization parameter $\lambda$; the two are related by a scaling of $|D|$, with $\gamma = \lambda|D|$. In our implementation, we constrain $\gamma$ to be a whole number to avoid floating point calculations, with negligible effect on objective expressiveness (discussed in Appendix B).

Let $\mathcal{T}_d = \{t \in \mathcal{T} \mid \text{depth}(t) \leq d\}$; The Rashomon set for dataset $D$, depth bound $d$, and penalty $\gamma$ can be defined as

$$\mathcal{R}_{\varepsilon_{\text{abs}}}(D, d, \gamma) := \{t \in \mathcal{T}_d \mid \text{Obj}(t, D, \gamma) \leq \varepsilon_{\text{abs}}\},$$

with cardinality $|\mathcal{R}_{\varepsilon_{\text{abs}}}(D, d, \gamma)|$ (abbreviated to $|\mathcal{R}_{\varepsilon_{\text{abs}}}|$ for notational simplicity). It is often convenient to parameterize this using a fractional tolerance $\varepsilon_{\text{mult}}$, setting $\varepsilon_{\text{abs}} = (1 + \varepsilon_{\text{mult}}) \min_{t \in \mathcal{T}_d} \text{Obj}(t, D, \gamma)$.

As in prior work, we maintain a search graph representation of the Rashomon set, from which all valid trees can be enumerated after the algorithm terminates. Algorithms 8 and 9 in Appendix B specify the operations used to extend this graph with new leaves and splits, respectively, and populate the minimum objective of trees rooted at that subproblem to be used throughout our algorithm.

### 3.2. Proxy Optimizer Framework

Our goal is to approximate the Rashomon set efficiently. To that end, we utilize proxy algorithms (Definition 3.1) as efficient subroutines to find high-quality individual trees. We use the objectives incurred by these trees to prune our search space.

**Definition 3.1** (Proxy Algorithm). Given a dataset $D$, depth budget $d$, and regularization parameter $\gamma$, a *proxy algorithm* returns the objective of some tree $t \in \mathcal{T}_d$:

$$\text{PROXY}(D, d, \gamma) = \text{Obj}(t, D, \gamma).$$

If $t$ is not a leaf, then its children must satisfy:

$$\text{PROXY}(D_{t_{\text{left}}}, d - 1, \gamma) \leq \text{Obj}(t_{\text{left}}, D_{t_{\text{left}}}, \gamma)$$
$$\text{PROXY}(D_{t_{\text{right}}}, d - 1, \gamma) \leq \text{Obj}(t_{\text{right}}, D_{t_{\text{right}}}, \gamma)$$

Many algorithms in the literature can satisfy this property, and therefore can be a proxy algorithm (e.g., Breiman, 1984; Lin et al., 2020). If a decision tree algorithm does not satisfy this refinement property, we can easily modify it to meet the above criteria by memoizing subtree objectives (see Appendix B.6). If an algorithm exactly preserves the left and right subtree objectives, we can save runtime with caching, as also discussed in Appendix B.6.

Algorithm 1 presents our approach. At each subproblem (starting from the root), we construct nodes to represent each of the possible actions, i.e., leaf predictions or feature splits, for extending a tree while remaining within the budget $\varepsilon_{\text{abs}}$. We add any leaf predictions that fit within the budget (lines 2-8), and, if we have not hit a depth limit (lines 9-11), evaluate all possible splits, pruning those whose proxy-completed objectives exceed $\varepsilon_{\text{abs}}$ (lines 12-21). For each remaining split, we recursively approximate the sets of feasible left and right subtrees using the budget refinement procedure in Algorithm 3 (lines 22-23).

When the proxy algorithm is itself *exactly* optimal, PRAXIS recovers the Rashomon set in a manner similar to existing state-of-the-art solvers (Xin et al., 2022; Arslan et al., 2026). Using an approximate proxy algorithm, however, leads to many practical benefits.

Under some non-optimal settings of this proxy optimizer, we can still recover exact guarantees around our approximation. For example, Theorem A.12 in Appendix A guarantees that, under a small set of modifications to PRAXIS, the returned

set contains the full Rashomon set of simple rule lists (trees that have at least one leaf child for each split) for the given features and depth.

In practice, we find that strong, approximate proxy algorithms, as well as stronger pruning, yield efficient and reliable Rashomon approximations. Algorithm 2 presents our default proxy algorithm for the experiments presented in the main body of this paper. It corresponds to a substantially modified version of the LicketySPLIT algorithm (Babbar et al., 2025). At each subproblem, we return a leaf if the leaf objective and $\gamma$ penalty guarantee no further splits will improve the objective (lines 2-5); otherwise, we find a single split minimizing the objective of a greedy tree completion on each subproblem (line 6) and recursively call our algorithm on the resulting two subproblems (lines 7-9). We then compare that recursive solution to our leaf objective, and return whichever leads to a better tree (line 10). By contrast, LicketySPLIT compares the leaf objective to the greedy objective before recursion. Either choice leads to the same worst-case asymptotic complexity, but avoiding that prepruning step allows us to improve the objective of recovered trees. Several other important distinctions from LicketySPLIT that improve the proxy are discussed in Appendix B.5. Most significantly, we cache subproblem solutions encountered in GREEDY, which guarantees some cache reuse in later subproblems, and the greedy solver directly finds the accuracy-maximizing split when the remaining depth limit is one (rather than minimizing weighted child entropy). Appendix B.5 defines a family of proxy algorithms ranging from greedy to optimal, and Appendix D.5 provides empirical results for these alternatives.

### 3.3. Algorithm details

**Budget.**  One of the core details around PRAXIS is the setting of the budget $\varepsilon_{\text{abs}}$, which maps to the number of errors (and additional leaves) the algorithm can make while remaining in the Rashomon set. At the root, the budget can be based on a user-specified value, or it can be obtained by treating the proxy tree at the root as the reference solution and applying a multiplicative factor of $(1 + \epsilon_{\text{mult}})$. In the latter case, the output can also be filtered at the end to be $(1 + \epsilon_{\text{mult}})$ times the best tree recovered (this filtering is trivial since we enumerate the set in sorted order of objective).

To propagate the budget to child subproblems, a key subroutine in PRAXIS is Algorithm 3, which efficiently propagates the budget down the search space using the initial budget and the proxy algorithm's estimates of split quality. Given the two sibling subproblems resulting from a given split, the algorithm initially allocates budget for the left subproblem so that any solution found can be combined with the proxy algorithm's solution for the right subproblem while remaining within the parent budget $\varepsilon_{\text{abs}}$ (line 3). This is

---

**Algorithm 1** PRAXIS$(D, d, \gamma, \varepsilon_{\text{abs}})$

**Require:** Subproblem dataset $D$, remaining depth $d$, per-leaf penalty $\gamma$, budget $\varepsilon_{\text{abs}}$

1: Let $G \leftarrow \text{ORNODE}(\varepsilon_{\text{abs}})$ {Initialize subgraph for subtrees found within budget $\varepsilon_{\text{abs}}$ (See Appendix B.2)}
2: **for** $b \in \{0, 1\}$ **do**
3:     {For each possible leaf prediction $b$, set $C_b$ to the corresponding objective: $\gamma$ + misclassification error.}
4:     $C_b \leftarrow \gamma + \big| \{(x_i, y_i) \in D : y_i \neq b\} \big|$
5:     **if** $C_b \leq \varepsilon_{\text{abs}}$ **then**
6:         ADDLEAF $(G, b, C_b)$ {See Appendix B.2}
7:     **end if**
8: **end for**
9: **if** $d = 0$ **or** $\varepsilon_{\text{abs}} < 2\gamma$ **then**
10:     **return** $G$ {Either no remaining depth or any split would exceed the budget}
11: **end if**
12: **for each** feature $j$ **do**
13:     $(D_L, D_R) \leftarrow \text{PARTITION}(D, j)$
14:     **if** $D_L = \emptyset$ **or** $D_R = \emptyset$ **then**
15:         **continue** {Skip degenerate splits}
16:     **end if**
17:     $P_L \leftarrow \text{PROXY}(D_L, d - 1, \gamma)$ {Proxy cost on left}
18:     $P_R \leftarrow \text{PROXY}(D_R, d - 1, \gamma)$ {Proxy cost on right}
19:     **if** $P_L + P_R > \varepsilon_{\text{abs}}$ **then**
20:         **continue** {Prune split if proxy completions exceed the budget}
21:     **end if**
22:     $G_L, G_R \leftarrow \text{SOLVE\_SIBLINGS}(D_L, D_R, d-1, \gamma, \varepsilon_{\text{abs}}, P_R)$ {Find subgraphs for the left and right subproblems of this split; described in Algorithm 3}
23:     ADDSPLIT$(G, j, G_L, G_R)$ {Add $G_L, G_R$ as a split for the current subgraph $G$; see Appendix B.2}
24: **end for**
25: **return** $G$ {Rashomon graph for the subproblem}

**Usage Note:** If provided with only $\varepsilon_{\text{mult}}$ and not $\varepsilon_{\text{abs}}$, set Rashomon budget relative to proxy:

$$\varepsilon_{\text{abs}} \leftarrow \big(1 + \varepsilon_{\text{mult}}\big) \cdot \text{PROXY}(D, d, \gamma)$$

---

potentially an underestimate – if there are completions of the right subproblem that perform better than the proxy, then additional budget should be provided to the left subproblem. To handle this, the algorithm repeatedly refines (loosens) the left and right subproblem budgets if a better tree is found for the other subproblem (lines 5-12). This iterative budget refinement procedure continues until no better tree is found.

**Runtime and Memory Requirements.**  Whereas normally the memory and runtime requirements on Rashomon set construction can be arbitrarily larger than the size of the Rashomon set, PRAXIS takes memory and runtime lin-

**Algorithm 2** Default Algorithm for PROXY($D', d, \gamma$)

**Require:** Subproblem dataset $D'$, remaining depth $d$, leaf penalty $\gamma$
1: $n' \leftarrow |D'|$, $\quad p \leftarrow |\{(x_i, y_i) \in D' : y_i = 1\}|$
2: leaf_obj $\leftarrow \gamma + \min\{p, n' - p\}$
3: **if** $d = 0$ **or** leaf_obj $\leq 2\gamma$ **then**
4: $\quad$ **return** leaf_obj
5: **end if**
6: $j^\star \leftarrow$ feature whose split minimizes

$$\text{GREEDY}(D'_L, d-1, \gamma) + \text{GREEDY}(D'_R, d-1, \gamma)$$

7: Split $D'$ by $j^\star$ into $(D'_L, D'_R)$
8: $L \leftarrow \text{PROXY}(D'_L, d-1, \gamma)$
9: $R \leftarrow \text{PROXY}(D'_R, d-1, \gamma)$
10: **return** $\min\{\text{leaf\_obj}, L + R\}$

---

**Algorithm 3** SOLVE_SIBLINGS($D_L, D_R, d, \gamma, \varepsilon_{\text{abs}}, P_R$)

**Require:** Left/right datasets $D_L, D_R$, remaining depth $d$, regularization $\gamma$, parent budget $\varepsilon_{\text{abs}}$, proxy cost $P_R$
1: $\varepsilon_L \leftarrow -\infty$ {largest budget used for solving $D_L$ with PRAXIS ; currently we have not yet run on $D_L$ at all.}
2: $\varepsilon_R \leftarrow -\infty$ {largest budget used for solving $D_R$ with PRAXIS ; currently we have not yet run on $D_R$ at all.}
3: $\varepsilon_L^{(\text{new})} \leftarrow \varepsilon_{\text{abs}} - P_R$ {new budget to use for $D_L$ with PRAXIS }
4: **while** $\varepsilon_L^{(\text{new})} > \varepsilon_L$ **do**
5: $\quad \varepsilon_L \leftarrow \varepsilon_L^{(\text{new})}$
6: $\quad G_L \leftarrow \text{PRAXIS}(D_L, d, \gamma, \varepsilon_L)$
7: $\quad \varepsilon_R^{(\text{new})} \leftarrow \varepsilon_{\text{abs}} - G_L.min\_objective$
8: $\quad$ **if** $\varepsilon_R^{(\text{new})} > \varepsilon_R$ **then**
9: $\quad\quad \varepsilon_R \leftarrow \varepsilon_R^{(\text{new})}$
10: $\quad\quad G_R \leftarrow \text{PRAXIS}(D_R, d, \gamma, \varepsilon_R)$
11: $\quad\quad \varepsilon_L^{(\text{new})} \leftarrow \varepsilon\_abs - G_R.min\_objective$
12: $\quad$ **end if**
13: **end while**
14: **return** $G_L, G_R$

---

ear in the size of the Rashomon set approximation it finds. We demonstrate this in the following theorems (which are proven in Appendix A).

**Theorem 3.2** (PRAXIS Runtime). *Given a dataset $D$ with $n$ samples and $k$ features, depth limit $d$, leaf penalty $\gamma$, and Rashomon budget $\varepsilon_{abs}$, PRAXIS($D, d, \gamma, \varepsilon_{abs}$) computes an estimate of the Rashomon set $R_{\varepsilon_{\text{abs}}}(D, d, \gamma)$ in time $\mathcal{O}(|R_{\varepsilon_{\text{abs}}}| \, k \, d \cdot \text{PROXYCOMPUTE}(n, k, d))$, assuming the runtime of our proxy algorithm is bounded: $\text{PROXYCOMPUTE}(n, k, d) \in \Theta(n \, g(k, d))$ for some function $g(k, d) \in \Omega(1)$ that is also nondecreasing in $d$.*

Theorem 3.2 shows that for a polynomial-time proxy algorithm that is linear in $n$, PRAXIS performs only polynomial work per actual tree in the Rashomon set. This is a substantially stronger guarantee than that of TreeFARMS (Xin et al., 2022), SORTeD (Arslan et al., 2026), and RESPLIT (Babbar et al., 2025), which, in the worst case, require exponential time per tree they output. When using our default proxy algorithm (based on LicketySPLIT from Babbar et al., 2025), we have a runtime of $\mathcal{O}(|R| \, n \, k^3 d^3)$, i.e., we spend time that is only a polynomial multiple of the size of the Rashomon set that we need to output.

By scaling linearly in the output size, Theorem 3.2 is a substantial improvement relative to optimal solvers in the often-encountered scenario where the Rashomon set is a tiny fraction of the overall search space of trees. For instance, Semenova et al. (2022) show that even large Rashomon sets (allowing for 5% additive error) are on the order of $10^{-37}\%$ or $10^{-38}\%$ of the hypothesis space. Corollary 3.3 gives a condition on the Rashomon set size such that even finding a single optimal tree (which is a precursor to finding the entire Rashomon set, Xin et al., 2022; Arslan et al., 2026) with standard branch-and-bound is super-polynomially slower than PRAXIS, even though PRAXIS finds an approximation

of the entire Rashomon set.

**Corollary 3.3** (Super-polynomial speedup over optimal dynamic programming). *Consider a dataset $D$ of size $n \times k$, and let $d$ be the depth budget. Setting $\gamma = 0$ for simplicity and assuming there is at least one tree in the Rashomon set, if*

$$\forall q, |R_{\varepsilon_{abs}}(D, d, \gamma)| \in o\left(\frac{\binom{k}{d}}{(kd)^q}\right),$$

*then a full dynamic programming search (i.e., Algorithm 1 from Demirović et al., 2023) for a single optima will have worst-case runtime that is a super-polynomial factor of the Rashomon set size $|R_{\varepsilon_{abs}}|$, but PRAXIS with a polynomial-time proxy algorithm satisfying Theorem 3.2 will not, i.e:*

$$\forall q, \frac{\textit{Worst-Case Runtime}(\texttt{OPT})}{\textit{Worst-Case Runtime}(\texttt{PRAXIS})} \in \omega\left((kd)^q\right).$$

In addition to the runtime efficiency discussed above, Theorem 3.4 below establishes that PRAXIS's memory complexity scales linearly with the size of the input and the set of trees it finds, for any memory-efficient proxy algorithm.

**Theorem 3.4** (PRAXIS Memory Usage). *Let $d$ be a depth budget, and $\gamma$ be a leaf penalty. Given a dataset $D$ of size $n$ with $k$ features and a proxy algorithm with $\mathcal{O}(nk)$ memory usage, a memory-efficient implementation of PRAXIS can find an estimate of the Rashomon set $R_{\varepsilon_{abs}}(D, d, \gamma)$ using $\mathcal{O}\left(nk + \sum_{t \in R_{\varepsilon_{abs}}} |t|\right)$ memory (where $|t|$ is the # of leaves in tree $t$), with the runtime complexity in Theorem 3.2.*

In practice, our implementation trades additional memory

for faster runtime by caching and reusing solutions to repeated subproblems.

**Caching.** Rashomon set algorithms for decision trees make use of caching to allow subproblem reuse. The subproblem representation of Xin et al. (2022) requires a bitvector of length $n$ to represent the subset of data points contained in a subproblem (and the subproblem depth), which can be memory intensive. Alternatively, subproblems may be represented by the set of feature splits (and branch directions) used to reach them (Arslan et al., 2026). This representation is substantially more memory efficient, but it cannot detect when different sets of splits correspond to the same subset of data points, which can lead to redundant work. This creates a space–time tradeoff (see Appendix B.7).

In PRAXIS, we combine the memory efficiency of split-based representations (Arslan et al., 2026) with the reuse benefits of bitvector representations (Xin et al., 2022). We hash each bitvector representing the samples in a subproblem (and the depth budget) to a 64-bit fingerprint and use this fingerprint as the cache key. Thus, when our modified LicketySPLIT proxy is called, we cache the solution for every recursive subproblem it solves while computing the objective of the tree it implicitly enumerates; we apply the same caching strategy to each subproblem solved by the greedy algorithm. In Appendix B.7, we discuss and show the benefits of this approach. For well-behaved hash functions, the probability of an erroneous cache hit is vanishingly small; in practice, this saves orders of magnitude of memory without introducing errors.

**Recovery Conditions.** All trees returned by PRAXIS are guaranteed to lie within the specified budget $\varepsilon_{\text{abs}}$, meaning precision is easy to guarantee even when using an approximate proxy. However, perfect recall is not guaranteed: a proxy may exceed the budget on a subproblem even when an optimal subtree for that subproblem would remain feasible. In this case, some valid Rashomon set members will not be recovered. Theorems 3.5 and 3.6 provide sufficient conditions to guarantee a decision tree is recovered by PRAXIS.

**Theorem 3.5** (Frontier cut recovery condition). *Let $t$ be a tree within the root budget $\varepsilon_{\text{abs}}$. Fix any root or internal node $n_{d'}$ of $t$ at depth $d'$, and let $n_1, n_2, \ldots, n_{d'-1}, n_{d'}$ be the nodes on the path from the root to $n_{d'}$ (excluding root $r$). For each $n_i$ on this path, let $s_i$ denote its sibling in $t$, and let $(n_{d'})_{left}$ and $(n_{d'})_{right}$ denote the two children of $n_{d'}$. Let $d_u$ denote the remaining depth budget for internal node $u$. Suppose that*

$$\sum_{u \in \{s_1, \ldots, s_{d'}, (n_{d'})_{left}, (n_{d'})_{right}\}} \text{PROXY}(D_u, d_u, \gamma) \;\leq\; \varepsilon_{\text{abs}}.$$

*Then, PRAXIS will not prune exploration of $(n_{d'})_{left}$ and $(n_{d'})_{right}$. If this holds for all internal nodes, then $t$ will be*

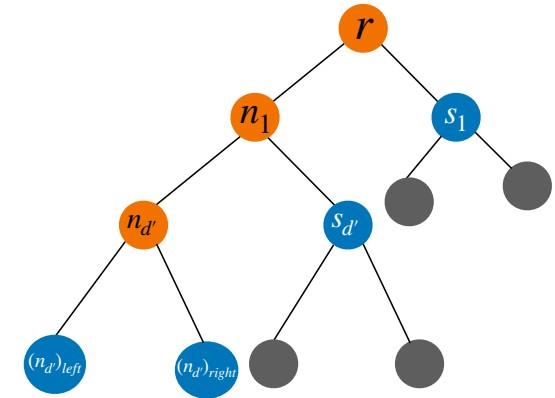

*Figure 2.* Frontier cut for internal node $n_{d'}$. The path to the node is shaded in orange. The nodes that are summed in the frontier cut are depicted in blue.

*in the set returned by* PRAXIS.

To illustrate the condition in Theorem 3.5, we introduce the notion of a *frontier cut*, depicted in Figure 2. A frontier cut for a node $N$ of a given tree corresponds to those subproblems not already encountered along the path from the root to node $N$. Theorem 3.5 tells us that if, for each internal node of a tree, proxy algorithm completions along that node's frontier cut remain within $\varepsilon_{\text{abs}}$, then we are guaranteed to include that tree in PRAXIS's output.

Intuitively, we would expect that conditioning on splits from trees that are in the Rashomon set should generally improve the performance of the proxy. Thus, we expect the condition from Theorem 3.5 to be satisfied for most trees in the Rashomon set. Even when some tree in the Rashomon set does not satisfy this condition, however, we are often still able to recover the tree, thanks to Algorithm 3. The repeated passes in that subroutine allow PRAXIS to widen the budget provided in subproblem exploration beyond what the initial proxy completions along a frontier cut would allow. Corollary A.16 in the appendix formalizes some of the objective improvements that will occur along sibling nodes as a consequence of these repeated passes.

A related condition for recovering a tree in the Rashomon set comes from Corollary 3.6. This corollary implies that if the gap between the proxy and an optimal tree completion decreases (or does not increase too much) further down a tree, then our approach will recover that tree. This aligns with observations about monotonically decreasing optimal-greedy gaps in Rashomon sets (Babbar et al., 2025). It also provides a multiplicative factor $\sigma$ that, when set appropriately, guarantees recovery of all trees in $\mathcal{R}_{\varepsilon_{\text{mult}}}$, even if the earlier condition does not hold.

**Corollary 3.6** (Slack needed for recovery under approximate proxies). *For leaf penalty $\gamma$, depth $d$ and dataset $D$,*

*fix a tree $t_r \in \mathcal{R}_{\varepsilon_{mult}}(D, d, \gamma)$ and initialize PRAXIS with*

$$\varepsilon_{abs} \leftarrow \sigma\left(1 + \varepsilon_{mult}\right) \text{PROXY}(D, d, \gamma).$$

*Define*

$$\alpha = \frac{\text{PROXY}(D, d, \gamma)}{\min_{t \in \mathcal{T}_d} \text{Obj}(t, D, \gamma)},$$

$$\beta = \max_{u \in \text{InternalNodes}(t_r)} \left\{ \frac{\text{PROXY}(D_u, d_u, \gamma)}{\min_{t \in \mathcal{T}_{d_u}} \text{Obj}(t, D_u, \gamma)} \right\},$$

$$\eta_r = \frac{\text{Obj}(t_r, D, \gamma)}{(1 + \varepsilon_{mult}) \min_{t \in \mathcal{T}_d} \text{Obj}(t, D, \gamma)} \in \left[\frac{1}{1 + \varepsilon_{mult}}, 1\right].$$

*Then,* PRAXIS *will return $t_r$ with $\sigma = \max\left\{1, \frac{\beta}{\alpha}\eta_r\right\}$*

In the worst case, $\alpha = 1$ and $\beta > 1$, meaning the proxy is exact at the root but not later down the tree. In this setting, PRAXIS still recovers every tree in $\mathcal{R}_{\varepsilon_{mult}}$ provided the Rashomon bound is rescaled by $\sigma \geq \beta$; a looser choice is to set $\sigma$ to be the proxy algorithm's global approximation ratio.

Based on the preceding theorems, recovery can be improved either by introducing slack into the root budget (Corollary 3.6) or by using a stronger proxy algorithm, which yields better estimates of the optimal subproblem objectives (Theorem 3.5). Both adjustments incur additional runtime cost, but can be applied after the initial execution of PRAXIS, reusing previous computation.

## 4. Experiments

Our experimental evaluation stress-tests PRAXIS across 50 dataset-binarization combinations (with up to two binarizations per dataset), spanning up to 11 million samples and 472 binary features. We primarily binarize continuous features using the threshold-guessing technique in McTavish et al. (2022) (Appendix C has additional information on datasets and binarizations). The goal of these experiments is to assess whether PRAXIS can scale to the most demanding and computationally challenging Rashomon-set problems encountered in practice.

In our experiments, we evaluate three key dimensions of performance: (1) time and memory usage required to generate a Rashomon set; (2) approximation quality, measured primarily through recall;[1] and (3) ability to recover optimal decision trees. We compare PRAXIS to SORTeD (Arslan et al., 2026) and TreeFARMS (Xin et al., 2022)—the two

existing exact Rashomon-set enumeration algorithms—as well as RESPLIT (Babbar et al., 2025), a state-of-the-art approximation algorithm for Rashomon sets. We also include a baseline that repeatedly trains the proxy on bootstrapped versions of the dataset (for as long as PRAXIS ran) to quantify how much PRAXIS's search procedure improves on the underlying proxy algorithm's performance.

**Timing and Memory.** Table 1 contains timing and memory profiles from a large range of datasets. The results for the full list of over 50 datasets are given in the Appendix D.

On all datasets with 100 or fewer binary features (except for Higgs or Covertype, which have large sample sizes), PRAXIS finished in 32 seconds or less. In contrast, SORTeD took up to 18 hours, and TreeFARMS frequently did not finish, because it ran out of memory. On Higgs, no other method finished within 90 hours or 200 GB of RAM, yet PRAXIS finished in 40 minutes with no extra memory beyond what was required to load the dataset. The difference becomes more dramatic as the number of features increases: PRAXIS takes 35 seconds on Churn, whereas SORTeD takes nearly 35 hours.

These results demonstrate that PRAXIS significantly improves the scalability of Rashomon set computation by orders of magnitude, outperforming RESPLIT by up to 2 orders of magnitude, TreeFARMS by up to 5 orders of magnitude (and handling many datasets that TreeFARMS cannot), and SORTeD by up to 3 orders of magnitude. In terms of memory consumption, PRAXIS is typically 5 × more memory efficient than RESPLIT, up to 2 orders of magnitude more memory efficient than SORTeD, and up to 4 orders of magnitude more memory efficient than Tree-FARMS. Appendix D.9 shows that PRAXIS scales much better with depth than approximate methods, achieving up to four orders of magnitude gains in runtime and memory for depth-7 Rashomon sets versus RESPLIT, while improving approximation quality. In this deeper regime, SORTeD fails to complete within 150 hours on these datasets, whereas PRAXIS finishes in only 11 seconds.

**Approximation Quality.** We now establish that PRAXIS approximates the Rashomon set with near-perfect recall across all tested datasets and far better than existing approximations. Table 2 shows the recall of PRAXIS on datasets where we can directly compare to the ground truth Rashomon set. We see that PRAXIS frequently achieves a recall of 1.0 across all bootstraps of datasets. Even when it does not, it is always above 0.98 on average, highlighting that the outcomes of downstream tasks are unlikely to change when using this subset of trees.

**Benchmarking Trees Found.** To investigate settings where exact Rashomon set computations are prohibitively

---

[1] We focus on recall rather than precision, because precision in this context amounts to a simple check of whether returned trees' training objectives are sufficiently low; since we return our trees in sorted order, we can achieve perfect precision without any drop to recall for any user-provided absolute threshold.

| Dataset | $n$ | $k$ | PRAXIS Time (s) | PRAXIS Peak MB | TreeFARMS Time (s) | TreeFARMS Peak MB | SORTeD Time (s) | SORTeD Peak MB | RESPLIT Time (s) | RESPLIT Peak MB |
|---|---|---|---|---|---|---|---|---|---|---|
| Churn | 5000 | 472 | **34.84** | **279.41** | – | – | 123776.11 | 22012.99 | 2564.30 | 2792.24 |
| Madeline | 3140 | 451 | **2971.65** | 29094.20 | – | – | – | – | 133189.38 | **26621.99** |
| Electricity | 38474 | 264 | **306.94** | **692.10** | – | – | 114325.61 | 23777.65 | 7619.63 | 6402.99 |
| Shopping | 12330 | 243 | **58.88** | **470.00** | – | – | 24151.41 | 7957.33 | 1760.04 | 2195.54 |
| Christine | 5418 | 231 | **944.27** | 10439.32 | – | – | 38970.60 | 12709.88 | 12625.37 | **3096.83** |
| Credit | 30000 | 225 | **26.03** | **249.60** | – | – | 65145.23 | 14300.78 | 2316.15 | 4355.73 |
| Adult | 48842 | 209 | **272.53** | **682.15** | – | – | 56440.23 | 12079.80 | 5494.42 | 5782.34 |
| Jasmine | 2984 | 207 | **22.77** | **451.61** | – | – | 5211.85 | 2705.49 | 517.17 | 745.36 |
| News | 39644 | 196 | **596.47** | **1480.48** | – | – | 114514.40 | 30426.06 | 13544.76 | 6082.23 |
| Bike | 17379 | 164 | **35.40** | **359.94** | – | – | 6533.75 | 3546.60 | 1022.18 | 1302.75 |
| Helena | 65196 | 156 | **4.74** | **290.96** | 38.32 | 1983.32 | 100.29 | 779.31 | 564.13 | 1755.30 |
| Jannis | 57580 | 106 | **327.54** | **1458.63** | – | – | 5277.68 | 5137.22 | 5587.19 | 2462.35 |
| Bank | 45211 | 97 | **2.43** | **195.74** | – | – | 1808.61 | 1189.15 | 164.34 | 1468.21 |
| Covertype | 581012 | 96 | **358.70** | **1301.23** | – | – | 64672.88 | 16107.48 | 10101.87 | 8631.58 |
| Droid | 29332 | 84 | **0.78** | **165.04** | – | – | 882.65 | 697.19 | 78.89 | 730.01 |
| Higgs | 11000000 | 84 | **2374.91** | **21537.79** | – | – | – | – | – | – |
| Magic | 19020 | 80 | **31.05** | **446.82** | – | – | 268.66 | 850.65 | 595.96 | 527.28 |
| Madelon | 2000 | 73 | **8.19** | 292.36 | – | – | 97.75 | 409.84 | 204.01 | **276.30** |
| Heloc | 2502 | 65 | **0.38** | **140.59** | – | – | 52.34 | 313.95 | 8.79 | 205.18 |
| Wine | 6497 | 64 | **0.24** | **135.34** | – | – | 62.98 | 307.63 | 12.54 | 255.48 |
| Compas | 4966 | 44 | **0.09** | **130.47** | 65.08 | 3742.00 | 7.23 | 163.50 | 11.90 | 159.61 |
| Poker | 1025010 | 40 | **13.71** | **952.10** | – | – | 7370.45 | 7475.91 | 657.48 | 5958.07 |
| Diabetes | 253680 | 33 | **0.79** | **291.69** | 553.31 | 28336.29 | 683.31 | 1067.69 | 48.68 | 1049.81 |
| Taxi | 1224158 | 27 | **1.18** | **894.65** | 12.50 | 2874.38 | 364.85 | 3318.65 | 32.35 | 2030.43 |

*Table 1.* Runtime and peak memory usage at $\lambda = 0.02$, $\varepsilon = 0.03$, depth $= 5$. Peak MB reports the peak resident set size (RSS) during the script to load packages, the dataset, and compute the Rashomon set. '–' indicates that the method did not finish in 90 hours or with 200GB of RAM.

| Dataset | Recall | |
|---|---|---|
| | $\lambda=0.005$ | $\lambda=0.02$ |
| Churn-472 | 0.997±0.005 | 1.000±0.000 |
| Electricity-264 | 0.994±0.003 | 1.000±0.000 |
| Shopping-243 | 0.984±0.034 | 1.000±0.000 |
| Christine-231 | 1.000±0.000 | 0.994±0.011 |
| Credit-225 | 1.000±0.000 | 1.000±0.000 |
| Adult-209 | 1.000±0.000 | 1.000±0.000 |
| Jasmine-207 | 0.993±0.009 | 1.000±0.000 |
| News-196 | 1.000±0.000 | 1.000±0.000 |
| Bike-164 | 0.986±0.010 | 1.000±0.000 |
| Helena-156 | 1.000±0.000 | 1.000±0.000 |
| Jannis-106 | 0.995±0.008 | 1.000±0.000 |
| Bank-97 | 1.000±0.000 | 1.000±0.000 |
| Covertype-96 | 1.000±0.000 | 1.000±0.000 |
| Droid-84 | 1.000±0.000 | 1.000±0.000 |
| Magic-80 | 0.997±0.004 | 0.998±0.004 |
| Madelon-73 | 1.000±0.000 | 1.000±0.000 |
| Heloc-65 | 0.999±0.000 | 1.000±0.000 |
| Wine-64 | 1.000±0.000 | 1.000±0.000 |
| Compas-44 | 1.000±0.000 | 1.000±0.000 |
| Poker-40 | 1.000±0.000 | 1.000±0.000 |
| Diabetes-33 | 1.000±0.000 | 1.000±0.000 |
| Taxi-27 | 1.000±0.000 | 1.000±0.000 |

*Table 2.* Rashomon set recall (mean ± standard deviation over 5 bootstraps) of PRAXIS relative to ground truth enumeration for $\varepsilon_{\mathrm{mult}} = 0.03$ and depth $d = 5$. Format: Dataset-NumBinaryFeatures.

slow/memory intensive but PRAXIS can still complete execution for a reasonable time and memory usage, we compare our approximation quality to RESPLIT (run to completion) and bootstrapping our proxy algorithm (LicketySPLIT, run for as long as PRAXIS ran). Figure 3 shows the cumulative number of trees found by each method with objectives less than or equal to a given value. These are representative results from large datasets; the results for additional datasets can be found in Appendix D.11. In the case of the Churn and Electricity datasets, PRAXIS finds more than one million trees that are better than the best tree RESPLIT finds; moreover, none of the trees found by RESPLIT lie in the desired Rashomon set. Across all 4 cases displayed in Figure 3, RESPLIT and bootstrapped LicketySPLIT returned 0 or a small fraction of the estimated Rashomon set.

**Finding Optimal Trees.** While the primary focus of our framework is to approximate Rashomon sets, our work also functions as a way to find near-optimal individual decision trees (by taking the first tree returned by PRAXIS).

Across regularization values $\lambda \in \{0.005, 0.01, 0.02\}$, and up to 50 dataset–binarization pairs for which both the globally optimal solver STreeD (van der Linden et al., 2023) and PRAXIS (with $\varepsilon_{\mathrm{mult}} = 0.03$) completed successfully (the number of completed runs varies slightly across $\lambda$), we observe exact agreement in the minimum objective value between STreeD and PRAXIS on every dataset. Even when setting $\varepsilon_{\mathrm{mult}} = 0$, PRAXIS recovers the optimal tree in the vast majority of cases. Specifically, for $\lambda = 0.02$, PRAXIS ($\varepsilon_{\mathrm{mult}} = 0$) recovers the optimal tree on all 50 datasets; for $\lambda = 0.01$, it fails on only two datasets; and for $\lambda = 0.005$, it fails on only three datasets. In the rare failure cases, the returned objective exceeded the optimum by only a very

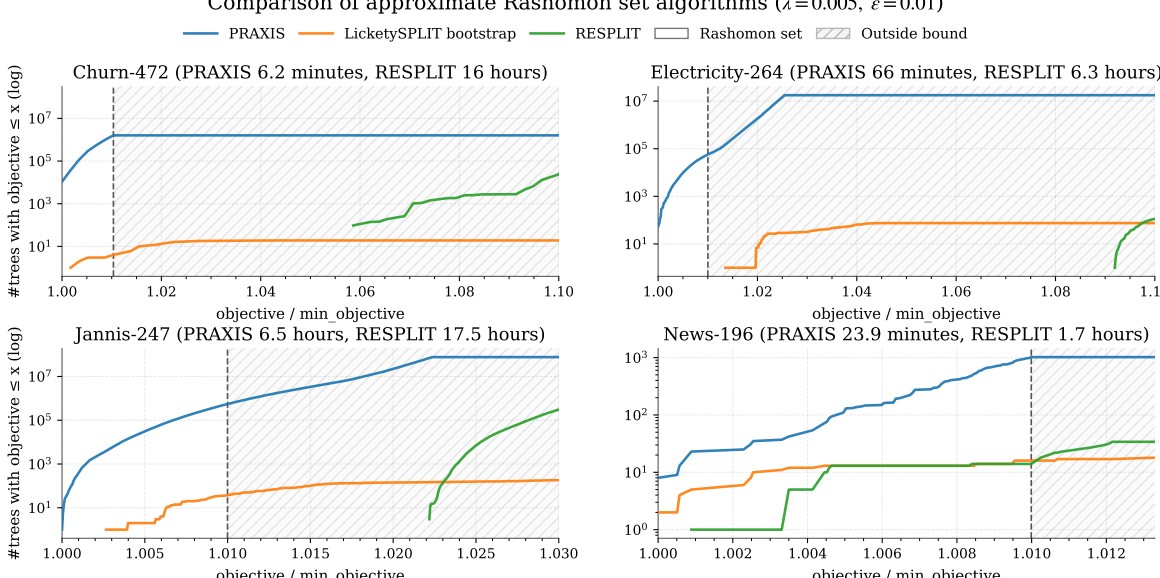

*Figure 3.* Approximation quality of non-optimal methods for the 4 datasets where optimal methods either take $> 40$ hours or $> 200$ GB of RAM. Each plot shows the number of depth 5 trees with objective value within an $x$ factor of the minimum objective found by any method. The dashed vertical line shows the Rashomon bound estimated as $(1 + \epsilon)$ times the minimum objective found by any method.

small multiplicative factor (e.g., 1.00069).

Given this, PRAXIS can be used to nearly always recover optimal decision trees and does so up to 3 orders of magnitude faster than GOSDT and STreeD (Lin et al., 2020; van der Linden et al., 2023) (see Appendix D.4). As an illustration, on News with $\lambda = 0.005$, PRAXIS computes the optimal tree in under 3 minutes; by contrast, STreeD takes more than twenty hours, and GOSDT runs out of memory.

## 5. Conclusion

We introduce PRAXIS, a fast, memory-efficient algorithm for estimating Rashomon sets of decision trees. PRAXIS efficiently prunes candidates using proxy completions as computationally light subroutines for candidate split evaluations. Leveraging these proxies and a mechanism to refine bounds during search, PRAXIS approximates the Rashomon set with near-perfect recall and delivers orders-of-magnitude speedups over existing methods. These results allow for high-quality approximations of Rashomon sets on datasets far larger than those accessible to existing methods, substantially expanding the practicality of Rashomon set computation. Future work could explore how these ideas extend to single-tree optimization and to broader classes of tree and partitioning problems in machine learning.

## Acknowledgments

We acknowledge funding from the U.S. Department of Energy under DE-SC0023194, and the National Institute On Drug Abuse of the National Institutes of Health under R01DA054994. Content does not necessarily represent official views of the United States Government nor any agency thereof. We acknowledge the support of the Natural Sciences and Engineering Research Council of Canada (NSERC). Nous remercions le Conseil de recherches en sciences naturelles et en génie du Canada (CRSNG) de son soutien. We would also like to thank Yixiao Wang and Luke Moffett for comments and suggestions on our work, as well as our reviewers.

## Impact Statement

This paper presents work on interpretable machine learning, which is essential for many ethical AI applications.

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

# Appendix Contents

# A. Theoretical Results

## A.1. Proofs of Claims

**Theorem 3.2** (PRAXIS Runtime). *Given a dataset $D$ with $n$ samples and $k$ features, depth limit $d$, leaf penalty $\gamma$, and Rashomon budget $\varepsilon_{abs}$, PRAXIS$(D, d, \gamma, \varepsilon_{abs})$ computes an estimate of the Rashomon set $R_{\varepsilon_{abs}}(D, d, \gamma)$ in time $\mathcal{O}(|R_{\varepsilon_{abs}}| \, k \, d \cdot \text{PROXYCOMPUTE}(n, k, d))$, assuming the runtime of our proxy algorithm is bounded: $\text{PROXYCOMPUTE}(n, k, d) \in \Theta(n \, g(k, d))$ for some function $g(k, d) \in \Omega(1)$ that is also nondecreasing in $d$.*

*Proof of Theorem 3.2.* We split the proof into a simpler case where iterative budget refinement does not run and then argue that whenever iterative budget refinement does run, we know a subproblem has at least one more tree than we previously expected, so the earlier bound was pessimistic and thus allows us to solve the subproblem again with a larger budget.

**Simpler Case:** First, consider the case where Algorithm 3 does not iteratively refine budgets. That is, we use Algorithm 4 in place of it. This way, we will never solve a subproblem identified by a sequence of splits more than 1 time.

---

**Algorithm 4** SINGLEPASS$(D_L, D_R, \gamma, d, \varepsilon_{\text{abs}}, P_R)$

---

**Require:** Left/right datasets $D_L, D_R$, remaining depth $d$, regularization $\gamma$, parent budget $\varepsilon_{\text{abs}}$, proxy cost $P_R$,
1: $b_L \leftarrow \varepsilon_{\text{abs}} - P_R$
2: $\mathcal{T}_L \leftarrow \text{PRAXIS}\,(D_L, d, \gamma, b_L)$
3: $m_L \leftarrow \mathcal{T}_L.min\_objective$
4: $b_R \leftarrow \varepsilon_{\text{abs}} - m_L$
5: $\mathcal{T}_R \leftarrow \text{PRAXIS}\,(D_R, d, \gamma, b_R)$
6: **return** $(\mathcal{T}_L, \mathcal{T}_R)$

---

Consider the search graph created by PRAXIS.

In this case, each node corresponds to a subproblem defined by the sequence of splits leading to it, and there is no evaluation of the same subproblem with different budgets. By construction, PRAXIS expands only those splits whose LicketySPLIT completions remain in the Rashomon set (based on the pruning procedure in Algorithm 1). Fix an AND/OR graph level $d' < d$, and let $r := d - d'$ denote the remaining depth budget at this level. Let $u_{d'}$ be the number of OR nodes (subproblems) expanded at level $d'$, and let $n_i$ be the dataset size at the $i^{th}$ such node. At each expanded node, PRAXIS evaluates all $k$ candidate splits to select the next batch of candidates. Because the proxy algorithm is linear in the local subproblem size $n_i$, the work at node $i$ is

$$\mathcal{O}\big(n_i k\big(g(k, r)\big)\big).$$

Therefore, the total cost at level $d'$ is

$$\mathcal{O}\left(\sum_{i=1}^{u_{d'}} n_i \, k\big(1 + g(k, r)\big)\right).$$

Since subproblems corresponding only to trees certified to be in the Rashomon set appear in the AND/OR graph, the combined subproblem sizes at any level $d'$ cannot exceed $n|R|$. Indeed,

$$\sum_{i=1}^{u_{d'}} n_i \leq \sum_{\text{tree } t \in R} \sum_{\text{nodes of } t \text{ at level } d'} n_{u_t} \qquad (1)$$

$$= \sum_{\text{tree } t \in R} n \qquad (2)$$

$$= n|R|. \qquad (3)$$

Substituting this bound yields that the total cost at level $d'$ is

$$\mathcal{O}\big(n|R| \, k \, \big(g(k, r)\big)\big).$$

Summing over all non-leaf levels $d' = 0, 1, \ldots, d - 1$ (equivalently, $r = 1, 2, \ldots, d$), the total runtime is

$$\mathcal{O}\left( n|R|\, k \sum_{r=1}^{d} \big(g(k, r)\big) \right).$$

If $g(k, r)$ is nondecreasing in $r$, then $\sum_{r=1}^{d} g(k, r) \leq d\, g(k, d)$, yielding the simplified bound

$$\mathcal{O}\big(n|R|\, k\, d\, \big(g(k, d)\big)\big).$$

At this point, the runtime is expressed in terms of the function $g$, which upper-bounds the proxy cost for a dataset. To express the bound directly in terms of the proxy algorithm's runtime, we require a matching lower bound.

From the assumption that

$$\text{PROXYCOMPUTE}(m, k, r) = \Theta(m\, g(k, r)),$$

we have that

$$n\, g(k, d) = \mathcal{O}(\text{PROXYCOMPUTE}(n, k, d)).$$

Substituting into the runtime bound gives:

$$\mathcal{O}(n|R|\, k\, d\, g(k, d)) = \mathcal{O}(|R|\, k\, d \cdot \text{PROXYCOMPUTE}(n, k, d)).$$

Additionally, PRAXIS may perform work at leaf nodes corresponding to $r = 0$ (equivalently, $d' = d$), where no further splits are evaluated and the algorithm considers only leaf predictions. The work performed at such a node is linear in the size of the local subproblem. Summed over all leaf nodes represented in the AND/OR graph, this contributes at most $\mathcal{O}(n|R|)$ additional work and does not affect the overall asymptotic bound.

**General Case:** Now, consider the case where Algorithm 3 does run—that is, PRAXIS is run on at least one more time on one of the child subproblems with a larger budget. If the minimum objective is still the LicketySPLIT objective, the enumeration on the left subproblem is not rerun, and we reduce to the earlier case. Thus, in order for one side to be rerun and a larger budget to have been set, it implies that there exist at least two trees for the one child (the LicketySPLIT tree and the new minimum-objective tree) that could be combined with the LicketySPLIT tree for sibling node to be within the budget for the parent subproblem.

Recall that PRAXIS has a key invariant: any solution to a subproblem not pruned is part of at least one tree in the Rashomon set (this follows from PRAXIS's construction, which only expands splits whose LicketySPLIT completions are within the budget). Thus, in our new call of PRAXIS, we know that every subproblem visited is a part of a solution with the new minimum-objective tree on the other side. And, every subproblem visited in the original PRAXIS call on one side is part of a solution with the LicketySPLIT tree on the other, so we know we're not double counting solutions.

Thus, even though we may solve a subproblem multiple times with different budgets, the number of times a subproblem (a sequence of splits) is solved is never more than the number of trees in which it appears.

This fact implies that the combined subproblem sizes at any level $d'$ cannot exceed $n|R|$, the sum of all subproblem sizes across the Rashomon-set trees found for that depth, because any distinct subproblem is solved once, and any subproblem shared by $t$ trees is solved no more than $t$ times. Thus, even in this case, the size of subproblems solved is bounded by the same quantity, and as such, the asymptotic runtime is as well.

$\square$

**Corollary 3.3** (Super-polynomial speedup over optimal dynamic programming). *Consider a dataset $D$ of size $n \times k$, and let $d$ be the depth budget. Setting $\gamma = 0$ for simplicity and assuming there is at least one tree in the Rashomon set, if*

$$\forall q, \; |R_{\varepsilon_{abs}}(D, d, \gamma)| \in o\left( \frac{\binom{k}{d}}{(kd)^q} \right),$$

*then a full dynamic programming search (i.e., Algorithm 1 from [Demirović et al., 2023](#)) for a single optima will have worst-case runtime that is a super-polynomial factor of the Rashomon set size $|R_{\varepsilon_{abs}}|$, but PRAXIS with a polynomial-time proxy algorithm satisfying Theorem [3.2](#) will not, i.e:*

$$\forall q, \frac{\text{Worst-Case Runtime(\texttt{OPT})}}{\text{Worst-Case Runtime(\texttt{PRAXIS})}} \in \omega\Big((kd)^q\Big).$$

*Proof.* Starting from the initial condition, we have:

$$\forall q, |R_{\varepsilon_{\text{abs}}}(D, d, \gamma)| \in o\big(\frac{\binom{k}{d}}{(kd)^q}\big)$$

applying the definition of little $o$, we know :

$$\forall q, \exists c, d_0, k_0, \forall k \geq k_0, d \geq d_0, |R_{\varepsilon_{\text{abs}}}(D, d, \gamma)| < c\big(\frac{\binom{k}{d}}{(kd)^q}\big)$$

$$\forall q, \exists c, d_0, k_0, \forall k \geq k_0, d \geq d_0, |R_{\varepsilon_{\text{abs}}}(D, d, \gamma)|(kd)^q < c\big(\binom{k}{d}\big)$$

$$\forall q, \exists c, d_0, k_0, \forall k \geq k_0, d \geq d_0, (kd)^q < c\big(\frac{\binom{k}{d}}{|R_{\varepsilon_{\text{abs}}}(D, d, \gamma)|}\big)$$

$$\forall q, \exists c, d_0, k_0, \forall k \geq k_0, d \geq d_0, \frac{1}{c}(kd)^q < \big(\frac{\binom{k}{d}}{|R_{\varepsilon_{\text{abs}}}(D, d, \gamma)|}\big)$$

$$\text{define } c' = \frac{1}{c}$$

$$\forall q, \exists c', d_0, k_0, \forall k \geq k_0, d \geq d_0, c'(kd)^q < \big(\frac{\binom{k}{d}}{|R_{\varepsilon_{\text{abs}}}(D, d, \gamma)|}\big)$$

$$\forall q, \exists c', d_0, k_0, \forall k \geq k_0, d \geq d_0, \big(\frac{\binom{k}{d}}{|R_{\varepsilon_{\text{abs}}}(D, d, \gamma)|}\big) > c'(kd)^q$$

divide both sides by kd times proxycompute cost:

$$\forall q, \exists c', d_0, k_0, \forall k \geq k_0, d \geq d_0, \big(\frac{\binom{k}{d}}{kd\text{PROXYCOMPUTE}(n, k, d)|R_{\varepsilon_{\text{abs}}}(D, d, \gamma)|}\big) > \frac{c'(kd)^{q-1}}{\text{PROXYCOMPUTE}(n, k, d)} \quad (4)$$

By assumption, the proxy is polynomial overall (and linear in $n$, to adhere to Theorem [3.2](#)). So we know there must be some constants $z, \tilde{c}, \tilde{d}_0, \tilde{k}_0, n_0$ for which $\text{PROXYCOMPUTE}(n, k, d) \leq \tilde{c}n(kd)^z$ for $n \geq n_0, k \geq \tilde{k}_0, d \geq \tilde{d}_0$. So, considering the constants from Equation [4](#), defining $k'_0 = \max(k_0, \tilde{k}_0), d'_0 = \max(d_0, \tilde{d}_0)$, we have

$$\forall q, \exists c', \tilde{c}, z, d'_0, k'_0, \forall k \geq k'_0, d \geq d'_0, n \geq n_0, \big(\frac{\binom{k}{d}}{kd\text{PROXYCOMPUTE}(n, k, d)|R_{\varepsilon_{\text{abs}}}(D, d, \gamma)|}\big) > \frac{c'(kd)^{q-1}}{\tilde{c}n(kd)^z}$$

$$\text{Define } q' = q - 1 - z, c'' = \frac{c'}{\tilde{c}}:$$

$$\forall q', \exists c'', d'_0, k'_0, \forall k \geq k'_0, d \geq d'_0, n \geq n_0, \frac{\binom{k}{d}}{kd\text{PROXYCOMPUTE}(n, k, d)|R_{\varepsilon_{\text{abs}}}(D, d, \gamma)|} > \frac{c''(kd)^{q'}}{n}$$

$$\forall q', \exists c'', d'_0, k'_0, \forall k \geq k'_0, d \geq d'_0, n \geq n_0, \frac{n\binom{k}{d}}{kd\text{PROXYCOMPUTE}(n, k, d)|R_{\varepsilon_{\text{abs}}}(D, d, \gamma)|} > c''(kd)^{q'}$$

Now, we lower bound the runtime of an optimal tree algorithm, and upper bound the runtime of PRAXIS. The runtime for full dynamic programming search cited in [Demirović et al. (2023)](#) is $\Theta((n + 2^d)\binom{k}{d})$(after translating that result to this paper's notation for # features and depth). So the runtime is also $\Omega(n\binom{k}{d})$, and we can write: $\exists \tilde{c}', \tilde{d}_0', \tilde{k}_0', \forall k \geq \tilde{k}_0', d \geq \tilde{d}_0', n \geq n_0'$, Worst-Case Runtime(\texttt{OPT}) $\geq \tilde{c}'n\binom{k}{d}$ So substituting $c''' = \frac{c''}{\tilde{c}'}$, $d''_0 = \max(d'_0, \tilde{d}_0')$, $k''_0 = \max(k'_0, \tilde{k}_0')$ $n''_0 = \max(n_0, n'_0)$:

$$\forall q', \exists c''', d''_0, k''_0, \forall k \geq k''_0, d \geq d''_0, n \geq n''_0, \frac{\text{Worst-Case Runtime(\texttt{OPT})}}{kd\text{PROXYCOMPUTE}(n, k, d)|R_{\varepsilon_{\text{abs}}}(D, d, \gamma)|} > c'''(kd)^{q'}$$

Now, using Theorem 3.2, we know Worst-Case Runtime(PRAXIS) $\in \mathcal{O}(|R_{\varepsilon_{\text{abs}}}|\, k\, d \cdot \text{PROXYCOMPUTE}(n, k, d))$. So,

$$\exists \tilde{c}'', \tilde{d_0}'', \tilde{k_0}'', \forall k \geq \tilde{k_0}'', d \geq \tilde{d_0}'', n \geq \tilde{n}_0, \text{Worst-Case Runtime}(\texttt{PRAXIS}) \leq \tilde{c}'' kd\text{PROXYCOMPUTE}(n, k, d)|R_{\varepsilon_{\text{abs}}}(D, d, \gamma)|.$$

So substituting $c'''' = c'''\tilde{c}''$, $d_0''' = \max(d_0'', \tilde{d_0}'')$, $k_0''' = \max(k_0'', \tilde{k_0}'')$ $n_0''' = \max(n_0'', \tilde{n}_0)$,

$$\forall q', \exists c'''', d_0''', k_0''', \forall k \geq k_0''', d \geq d_0''', n \geq n_0''', \frac{\text{Worst-Case Runtime}(\texttt{OPT})}{\text{Worst-Case Runtime}(\texttt{PRAXIS})} > c''''(kd)^{q'}$$

Meaning:

$$\forall q, \frac{\text{Worst-Case Runtime}(\texttt{OPT})}{\text{Worst-Case Runtime}(\texttt{PRAXIS})} \in \omega((kd)^q),$$

as required.

Note also the following intermediate consequence. The worst-case runtime of $\texttt{OPT}$ exceeds the Rashomon set size by more than any polynomial factor, as shown below.

$$\forall q', \exists c''', d_0'', k_0'', \forall k \geq k_0'', d \geq d_0'', n \geq n_0'', \frac{\text{Worst-Case Runtime}(\texttt{OPT})}{kd\text{PROXYCOMPUTE}(n, k, d)|R_{\varepsilon_{\text{abs}}}(D, d, \gamma)|} > c'''(kd)^{q'}$$

$$\forall q', \exists c''', d_0'', k_0'', \forall k \geq k_0'', d \geq d_0'', n \geq n_0'', \frac{\text{Worst-Case Runtime}(\texttt{OPT})}{kd\text{PROXYCOMPUTE}(n, k, d)} > c'''(kd)^{q'}|R_{\varepsilon_{\text{abs}}}(D, d, \gamma)|$$

$$\forall q', \exists c''', d_0'', k_0'', \forall k \geq k_0'', d \geq d_0'', n \geq n_0'', \text{Worst-Case Runtime}(\texttt{OPT}) > c'''(kd)^{q'}|R_{\varepsilon_{\text{abs}}}(D, d, \gamma)|$$

$$\forall q, \text{Worst-Case Runtime}(\texttt{OPT}) \in \omega((kd)^q|R_{\varepsilon_{\text{abs}}}(D, d, \gamma)|)$$

$$\Rightarrow \text{Worst-Case Runtime}(\texttt{OPT}) \in \omega(|R_{\varepsilon_{\text{abs}}}(D, d, \gamma)|Poly(k, d)) \tag{5}$$

In contrast, the worst case runtime of PRAXIS is at most a polynomial multiple of the Rashomon set size:

$$\text{Worst-Case Runtime}(\texttt{PRAXIS}) \in \mathcal{O}(|R_{\varepsilon_{\text{abs}}}(D, d, \gamma)|kd\text{PROXYCOMPUTE}(n, k, d)) \tag{6}$$

$$\Rightarrow \text{Worst-Case Runtime}(\texttt{PRAXIS}) \in \mathcal{O}(|R_{\varepsilon_{\text{abs}}}(D, d, \gamma)|nPoly(k, d)) \tag{7}$$

$\square$

**Comments on Corollary 3.3.**  For ease of discussion, we have incorporated two simplifying conditions in Corollary 3.3: (1) $\gamma = 0$, and (2) the absolute Rashomon budget $\varepsilon_{\text{abs}}$ is fixed and does not vary with depth. Both can reasonably be relaxed. $\gamma$ can be nonzero without necessarily changing the worst case behaviour of optimal trees (the applicability of bounds from $\gamma$, i.e., those in GOSDT (Lin et al., 2020), now depends on search orders and what bounds are admitted by the dataset $D$). $\varepsilon$ can be defined to scale with the depth (since the optimal objective for a fixed dataset monotonically decreases with depth budget), and/or be defined as a multiplicative factor of the proxy algorithm's call for the current $n, k, d$. Such a relaxation can be useful for defining the scaling of a Rashomon set's size with depth, as models become more accurate.

**Theorem 3.4** (PRAXIS Memory Usage). *Let $d$ be a depth budget, and $\gamma$ be a leaf penalty. Given a dataset $D$ of size $n$ with $k$ features and a proxy algorithm with $\mathcal{O}(nk)$ memory usage, a memory-efficient implementation of PRAXIS can find an estimate of the Rashomon set $R_{\varepsilon_{abs}}(D, d, \gamma)$ using $\mathcal{O}\left(nk + \sum_{t \in R_{\varepsilon_{abs}}} |t|\right)$ memory (where $|t|$ is the # of leaves in tree $t$), with the runtime complexity in Theorem 3.2.*

*Proof.* By assumption, the proxy algorithm requires only $\mathcal{O}(nk)$ memory to run. This holds for our many possible proxy algorithms. LicketySPLIT is one of them (Lemma A.11). Regardless of the proxy algorithm, it takes a single float or integer to store the output: the objective of some tree.

Now, when we run PRAXIS, at each node we visit, we just need to know which splits result in proxy objectives below the epsilon bound. So for each potential split, we need to run the proxy algorithm on the left and right subproblems (with $O(nk)$ memory total), then check if that falls within the budget $\varepsilon_{\text{abs}}$. If it does, then we will save the resulting subproblems from

this split, and later visit those nodes to continue PRAXIS. This will be an additional two nodes we need to persist in the dependency graph. However, every time this happens (increasing the total storage used by 2), the total sum of nodes in trees across the entire Rashomon set will also increase by at least 2, since at least one tree with this split (the one found by PRAXIS) will fall within the $\varepsilon_{\text{abs}}$ bound, and that tree will have at least one internal or leaf node corresponding to each of these two nodes, since it includes this split.

In order to keep the information stored in each node efficient, we use the following structure (assuming we have one global copy of the entire dataset provided as input). Consider a node to be active only if we are visiting that node currently (that is, we are mid-execution of Algorithm 1 at that node), or if we are visiting one of its child nodes. When a node is active, we persist information about the row indices corresponding to the data subset used in that node. We can still efficiently determine the row indices relevant for any child just using these row indices, the original dataset, and the binary splitting feature, so the runtime is not affected. We also maintain memory efficiency: we only have $O(d)$ nodes active at once, so the total memory required to persist these row indices is $O(nd)$; since we cannot have a depth greater than the number of binary features, that means the memory required is under $O(nk)$ and does not affect asymptotic complexity. Once we are done visiting a node, we don't need to visit it again and can safely stop persisting the memory needed for its row indices.[2]

So the total information we persist is just:

1. A single copy of the original dataset

2. the dependency graph structure, which can be constructed to include only nodes with a constant amount of storage space, where there are no more nodes in the dependency graph than there are nodes (split nodes or leaves) across the whole Rashomon set.

3. Row indices for currently active subproblems in the dependency graph.

Since the number of leaves and internal nodes is bounded by twice the number of leaves, we know object (2) is $O(\sum_{t \in R} |t|)$. We know object (1) is $O(nk)$, so combining them we have proven our claimed memory complexity. (Note that (3) is also within $O(nk)$ so does not affect the complexity).

$\square$

.

**Theorem 3.5** (Frontier cut recovery condition). *Let $t$ be a tree within the root budget $\varepsilon_{\text{abs}}$. Fix any root or internal node $n_{d'}$ of $t$ at depth $d'$, and let $n_1, n_2, \ldots, n_{d'-1}, n_{d'}$ be the nodes on the path from the root to $n_{d'}$ (excluding the root $r$). For each $n_i$ on this path, let $s_i$ denote its sibling in $t$, and let $(n_{d'})_{left}$ and $(n_{d'})_{right}$ denote the two children of $n_{d'}$. Let $d_u$ denote the remaining depth budget for internal node $u$. Suppose that*

$$\sum_{u \in \{s_1, \ldots, s_{d'}, (n_{d'})_{left}, (n_{d'})_{right}\}} \text{PROXY}\big(D_u, d_u, \gamma\big) \leq \varepsilon_{\text{abs}}.$$

*Then, PRAXIS will not prune exploration of $(n_{d'})_{left}$ and $(n_{d'})_{right}$. If this holds for all internal nodes, then $t$ will be in the set returned by PRAXIS.*

*Proof.* Fix a target tree $t$ of depth at most $d$ whose root-to-leaf paths we want to show are materialized in the AND/OR graph. For any node or tree $u$, denote the depth of that node/tree as $d_u$.

**Pruning at the Root** Let the root of $t$ split on feature $f$, yielding children $t_{\text{left}}$ and $t_{\text{right}}$. Algorithm 1 will consider feature $f$ and perform the pruning test

$$\text{PROXY}\big(D_{t_{\text{left}}}, d_{\text{root}} - 1, \gamma\big) + \text{PROXY}\big(D_{t_{\text{right}}}, d_{\text{root}} - 1, \gamma\big) \leq \varepsilon_{\text{abs}}. \tag{8}$$

If (8) holds, the split on $f$ is not pruned and the algorithm proceeds to build subtries for both children, so the prefix consisting of the root split of $t$ is materialized. (If $t$ is a single leaf, then it is handled before any split-pruning occurs.)

---

[2]This is slightly complicated by the multiple passes induced by Algorithm 3; however, it does not affect the asymptotic complexity if we reconstruct the relevant row indices again when revisiting that side for an additional pass.

Let $b_L$ and $b_R$ denote the budgets used when recursively solving the left and right child subproblems for the split on $f$. In the worst-case (no iterative refinement), these are set by subtracting the proxy estimate of the opposite side (as shown in Theorem A.3 and Corollary A.4):

$$b_L = \varepsilon_{\text{abs}} - \text{PROXY}\big(D_{t_{\text{right}}}, d_{\text{root}} - 1, \gamma\big), \qquad b_R = \varepsilon_{\text{abs}} - \text{PROXY}\big(D_{t_{\text{left}}}, d_{\text{root}} - 1, \gamma\big). \tag{9}$$

**Budget propagation down a path** Let $u$ be any internal node of $t$ at depth $d' \geq 1$ (root at depth 0). Let $n_0 = \text{root}, n_1, \ldots, n_{d'} = u$ be the nodes on the unique path from the root to $u$ in $t$. For each $i \in \{1, \ldots, d'\}$, let $s_i$ denote the sibling of $n_i$ (i.e., $s_i$ is the other child of $n_{i-1}$ not equal to $n_i$). Finally, let $u_{\text{left}}$ and $u_{\text{right}}$ denote the two children of $u$ in $t$.

In the worst case (without iterative budget refinement), we have

$$\varepsilon_u = \varepsilon_{\text{abs}} - \sum_{i=1}^{d'} \text{PROXY}(D_{s_i}, d_{s_i}, \gamma), \tag{10}$$

where $\varepsilon_u$ is the budget with which the algorithm solves the subproblem corresponding to node $u$.

**Proof of** (10) **by induction on** $d'$**.** For $d' = 1$, $u = n_1$ is a child of the root and $s_1$ is the opposite child. By (9), the budget for $u$ is exactly $\varepsilon_{\text{abs}} - \text{PROXY}(s_1)$, which matches (10).

Assume (10) holds for depth $d' - 1$, i.e.,

$$\varepsilon_{n_{d'-1}} = \varepsilon_{\text{abs}} - \sum_{i=1}^{d'-1} \text{PROXY}(D_{s_i}, d_{s_i}, \gamma).$$

At node $n_{d'-1}$, in the worst-case, we pass its chosen child $u = n_{d'}$ a budget that subtracts off the proxy algorithm on the other side (see Theorem A.3 and Corollary A.4). This gives us

$$\varepsilon_u = \varepsilon_{n_{d'-1}} - \text{PROXY}(D_{s_{d'}}, d_{s_{d'}}, \gamma)$$
$$= \left( \varepsilon_{\text{abs}} - \sum_{i=1}^{d'-1} \text{PROXY}(D_{s_i}, d_{s_i}, \gamma) \right) - \text{PROXY}(D_{s_{d'}}, d_{s_{d'}}, \gamma)$$
$$= \varepsilon_{\text{abs}} - \sum_{i=1}^{d'} \text{PROXY}(D_{s_i}, d_{s_i}, \gamma),$$

establishing (10).

**The pruning test at** $u$**.** At node $u$, the algorithm considers the split that $t$ takes at $u$ and performs the pruning check

$$\text{PROXY}\big(D_{u_{\text{left}}}, d_u - 1, \gamma\big) + \text{PROXY}\big(D_{u_{\text{right}}}, d_u - 1, \gamma\big) \leq \varepsilon_u. \tag{11}$$

Substituting (10) into (11) and rearranging yields the *frontier-cut inequality*:

$$\text{PROXY}\big(u_{\text{left}}, d_u - 1, \gamma\big) + \text{PROXY}\big(u_{\text{right}}, d_u - 1, \gamma\big) \leq \varepsilon_{\text{abs}} - \sum_{i=1}^{d'} \text{PROXY}(D_{s_i}, d_{s_i}, \gamma) \tag{12}$$

$$\iff \sum_{i=1}^{d'} \text{PROXY}(D_{s_i}, d_{s_i}, \gamma) + \text{PROXY}\big(u_{\text{left}}, d_u - 1, \gamma\big) + \text{PROXY}\big(u_{\text{right}}, d_u - 1, \gamma\big) \leq \varepsilon_{\text{abs}}. \tag{13}$$

The set

$$\{s_1, \ldots, s_{d'}, u_{\text{left}}, u_{\text{right}}\}$$

is exactly the frontier cut associated with $u$.

**Sufficiency: satisfying all frontier cuts implies $t$ is fully represented.** Assume (13) holds for every internal node $u$ of $t$ (including the root). At the root, this reduces to (8), so the root split is not pruned. Inductively, since budgets are initially allocated using proxy objectives and iterative budget refinement can only increase these budgets, the local pruning test (11) succeeds at every node $u$. Thus, every split used by $t$ is expanded, and every root-to-leaf path of $t$ is materialized in the AND/OR graph.

**On non-necessity:** The earlier analysis does not account for the fact that we may discover a tree better than the proxy certifies (in particular, Corollary A.16 guarantees recovery of trees at least as good as the pilot algorithm induced by the proxy). As a consequence, we may subtract off less than the proxy value, leading to larger children budgets and less pruning as a consequence.

Given that we instead recurse on children assuming the other side is the best objective found (which we denote $\mathrm{MinObj}$), which could be better than the proxy, then if $u$ is at depth $d'$ with path siblings $s_1, \ldots, s_{d'}$, then after refinement we have

$$\varepsilon_u \ \geq \ \varepsilon_{\mathrm{abs}} \ - \ \sum_{i=1}^{d'} \mathrm{MinObj}(s_i),$$

so a sufficient frontier-cut condition for expanding the split at $u$ is really

$$\sum_{i=1}^{d'} \mathrm{MinObj}(s_i) \ + \ \mathrm{PROXY}(u_{\mathrm{left}}) \ + \ \mathrm{PROXY}(u_{\mathrm{right}}) \ \leq \ \varepsilon_{\mathrm{abs}},$$

i.e., SOLVE_SIBLINGS refines the sibling terms from $\mathrm{PROXY}(s_i)$ to $\mathrm{MinObj}(s_i)$ while the two children at $u$ are still conservatively completed using $\mathrm{PROXY}(\cdot)$.

$\square$

**Corollary 3.6** (Slack needed for recovery under approximate proxies). *For leaf penalty $\gamma$, depth $d$ and dataset $D$, fix a tree $t_r \in \mathcal{R}_{\varepsilon_{mult}}(D, d, \gamma)$ and initialize PRAXIS with*

$$\varepsilon_{abs} \leftarrow \sigma \left(1 + \varepsilon_{mult}\right) \mathrm{PROXY}(D, d, \gamma).$$

*Define*

$$\alpha \ = \ \frac{\mathrm{PROXY}(D, d, \gamma)}{\min_{t \in \mathcal{T}_d} \mathrm{Obj}(t, D, \gamma)},$$

$$\beta \ = \ \max_{u \in \mathrm{InternalNodes}(t_r)} \left\{ \frac{\mathrm{PROXY}(D_u, d_u, \gamma)}{\min_{t \in \mathcal{T}_{d_u}} \mathrm{Obj}(t, D_u, \gamma)} \right\},$$

$$\eta_r \ = \ \frac{\mathrm{Obj}(t_r, D, \gamma)}{(1 + \varepsilon_{mult}) \min_{t \in \mathcal{T}_d} \mathrm{Obj}(t, D, \gamma)} \in \left[\frac{1}{1 + \varepsilon_{mult}}, 1\right].$$

*Then, PRAXIS will return $t_r$ with $\sigma = \max\left\{1, \frac{\beta}{\alpha} \eta_r\right\}$*

*Proof.* Fix any internal node $u$ of $t_r$ at depth $q$, and let

$$F(u) = \{s_1, \ldots, s_q, u_{\mathrm{left}}, u_{\mathrm{right}}\}$$

denote the associated frontier cut as defined in Theorem 3.5. By definition of $\beta$, for every $v \in F(u)$,

$$\frac{\mathrm{PROXY}(D_v, d_v, \gamma)}{\min_{t \in \mathcal{T}_{d_v}} \mathrm{Obj}(t, D_v, \gamma)} \leq \beta,$$

so

$$\mathrm{PROXY}(D_v, d_v, \gamma) \leq \beta \min_{t \in \mathcal{T}_{d_v}} \mathrm{Obj}(t, D_v, \gamma).$$

Summing over $v \in F(u)$ yields

$$\sum_{v \in F(u)} \text{PROXY}(D_v, d_v, \gamma) \leq \beta \sum_{v \in F(u)} \min_{t \in \mathcal{T}_{d_v}} \text{Obj}(t, D_v, \gamma). \tag{14}$$

The subtrees of $t_r$ rooted at the nodes in $F(u)$ are disjoint and collectively cover all leaves of $t_r$, so

$$\sum_{v \in F(u)} \min_{t \in \mathcal{T}_{d_v}} \text{Obj}(t, D_v, \gamma) \leq \sum_{v \in F(u)} \text{Obj}(t_r|_v, D_v, \gamma) = \text{Obj}(t_r, D, \gamma),$$

where $t_r|_v$ denotes the subtree of $t_r$ rooted at $v$. Substituting into (14) gives

$$\sum_{v \in F(u)} \text{PROXY}(D_v, d_v, \gamma) \leq \beta \, \text{Obj}(t_r, D, \gamma).$$

Using the definition of $\eta_r$,

$$\text{Obj}(t_r, D, \gamma) = \eta_r \left( (1 + \varepsilon_{\text{mult}}) \min_{t \in \mathcal{T}_d} \text{Obj}(t, D, \gamma) \right),$$

so

$$\sum_{v \in F(u)} \text{PROXY}(D_v, d_v, \gamma) \leq \beta \eta_r (1 + \varepsilon_{\text{mult}}) \min_{t \in \mathcal{T}_d} \text{Obj}(t, D, \gamma).$$

By definition of $\alpha$,

$$\min_{t \in \mathcal{T}_d} \text{Obj}(t, D, \gamma) = \frac{\text{PROXY}(D, d, \gamma)}{\alpha},$$

hence

$$\sum_{v \in F(u)} \text{PROXY}(D_v, d_v, \gamma) \leq \frac{\beta}{\alpha} \eta_r (1 + \varepsilon_{\text{mult}}) \text{PROXY}(D, d, \gamma).$$

Now multiply the standard root budget

$$\varepsilon_{\text{abs}} = (1 + \varepsilon_{\text{mult}}) \text{PROXY}(D, d, \gamma)$$

by

$$\sigma = \max \left\{ 1, \frac{\beta}{\alpha} \eta_r \right\}.$$

Then

$$\sigma \, \varepsilon_{\text{abs}} \geq \frac{\beta}{\alpha} \eta_r (1 + \varepsilon_{\text{mult}}) \text{PROXY}(D, d, \gamma) \geq \sum_{v \in F(u)} \text{PROXY}(D_v, d_v, \gamma).$$

Thus the frontier-cut inequality of Theorem 3.5 holds for $u$. Since $u$ was arbitrary, it holds for every internal node of $t_r$. Therefore, by Theorem 3.5, PRAXIS will return $t_r$. □

## A.2. Additional Theorems and Proofs

This section includes additional theoretical guarantees for PRAXIS. We show that PRAXIS can return a set of trees that are all arbitrarily better than pure greedy, prove convergence of iterative budget refinement, and provide bounds that characterize how additional slack in the root budget compensates for proxy optimality gaps. We also prove some algorithm invariants assuming a proxy algorithm is used (that is, assuming the proxy adheres to Definition 3.1), and establish key decision tree algorithms are indeed proxy algorithms.

**Simplifying Notation.** For any subproblem node $u$ with corresponding dataset $D_u$ and remaining depth budget $d_u$, define

$$\text{OPT}(u) := \min_{t \in \mathcal{T}_{d_u}} \text{Obj}(t, D_u, \gamma),$$

When $u = \text{root}$ we also write $\text{OPT}(\text{root}) = \min_{t \in \mathcal{T}_d} \text{Obj}(t, D, \gamma)$.

**Proposition A.1** (Monotonicity in the budget provided to PRAXIS)**.** *Fix a dataset $D$, depth $d$, and regularization $\gamma$. Let $0 \leq A \leq B$ be two absolute Rashomon budgets, and let $\hat{\mathcal{R}}^{\text{PRAXIS}\,(D,d,\gamma,U)}$ denote the set of trees returned by PRAXIS $(D, d, \gamma, U)$ for $U \in \{A, B\}$.*

*Then*

$$\hat{\mathcal{R}}^{\text{PRAXIS}\,(D,d,\gamma,A)} \subseteq \hat{\mathcal{R}}^{\text{PRAXIS}\,(D,d,\gamma,B)}$$

*That is, increasing the budget from $A$ to $B$ can only expand the returned AND/OR graph and set of returned trees.*

*Proof.* The proof will proceed by induction.

**Base case** ($d = 0$): At depth 0, PRAXIS considers only leaf actions. A leaf is added if and only if its objective fits within the local subproblem budget. Therefore, any leaf added under budget $A$ is also added under budget $B$, and the claim holds.

**Inductive Hypothesis:** At depth $d$, for all subproblems $D$, all non-negative leaf penalties $\gamma$, and all budgets $0 \leq A \leq B$, $\hat{\mathcal{R}}^{\text{PRAXIS}\,(D,d,\gamma,A)} \subseteq \hat{\mathcal{R}}^{\text{PRAXIS}\,(D,d,\gamma,B)}$.

**Inductive Step:** We want to show that, for all subproblems $D$, all non-negative leaf penalties $\gamma$, and all budgets $0 \leq A \leq B$, $\hat{\mathcal{R}}^{\text{PRAXIS}\,(D,d+1,\gamma,A)} \subseteq \hat{\mathcal{R}}^{\text{PRAXIS}\,(D,d+1,\gamma,B)}$. We proceed by showing each tree in $\hat{\mathcal{R}}^{\text{PRAXIS}\,(D,d+1,\gamma,A)}$ is also in $\hat{\mathcal{R}}^{\text{PRAXIS}\,(D,d+1,\gamma,B)}$. We consider different cases based on the depth of the tree in question.

**Case 1 (the tree is a leaf):** Note that trees that are just leaves at $D$ will only be in $\hat{\mathcal{R}}^{\text{PRAXIS}\,(D,d+1,\gamma,A)}$ if they are also in $\hat{\mathcal{R}}^{\text{PRAXIS}\,(D,d+1,\gamma,B)}$, for similar reasoning to the base case (leaf membership is directly determined by comparing leaf objective to the budget.)

**Case 2 (the tree is not a leaf):** All non-leaf trees are formed from calls to the SOLVE_SIBLINGS routine. In particular, for each choice of initial feature split, the returned trees come from the last call made to PRAXIS on $D_L$ and $D_R$ in SOLVE_SIBLINGS for that feature, which corresponds to the call with the most permissive budget.

To handle this case, we need a simple invariant: for each child, its final budget (the last call in SOLVE_SIBLINGS) can never be higher for initial budget $A$ than for initial budget $B$. To recover this invariant, we use the following lemma:

**Lemma A.2** (Higher initial budgets will always result in higher child budgets)**.** *Fix two subproblems $D_L, D_R$ and a depth $d \geq 0$. Fix some $\gamma > 0$. Assume that for all budgets $0 \leq A \leq B$, $\hat{\mathcal{R}}^{\text{PRAXIS}\,(D_L,d,\gamma,A)} \subseteq \hat{\mathcal{R}}^{\text{PRAXIS}\,(D_L,d,\gamma,B)}$ and $\hat{\mathcal{R}}^{\text{PRAXIS}\,(D_R,d,\gamma,A)} \subseteq \hat{\mathcal{R}}^{\text{PRAXIS}\,(D_R,d,\gamma,B)}$.*

*Then the final, largest-budget versions of $\varepsilon_L$ and $\varepsilon_R$ in SOLVE_SIBLINGS (Algorithm 3) for initial budget $B$ will be no smaller than $\varepsilon_L$ and $\varepsilon_R$ for initial budget $A$.*

*Proof.* In SOLVE_SIBLINGS, for an initial budget $U$, let $\varepsilon_L^{(i)}(U)$ and $\varepsilon_R^{(i)}(U)$ denote the left/right budgets used on the $i$-th iteration of the while-loop ($i = 0, 1, 2, \ldots$). It is now sufficient for us to show that for all $i \geq 0$ and all $0 \leq A \leq B$,

$$\varepsilon_L^{(i)}(A) \leq \varepsilon_L^{(i)}(B) \quad \text{and} \quad \varepsilon_R^{(i)}(A) \leq \varepsilon_R^{(i)}(B). \tag{15}$$

For any dataset $S$ and budget $U$, define

$$m(S, d, \gamma, U) := \min\big\{\text{obj}(t, S, \gamma) : \ t \in \hat{\mathcal{R}}^{\text{PRAXIS}\,(S,d,\gamma,U)}\big\},$$

with $m(S, d, \gamma, U) = +\infty$ if the set is empty.

By the assumption in this lemma, increasing the budget can only expand the estimated Rashomon set for depth $d$, so $m(S, d, \gamma, U)$ is monotone non-increasing in budget $U$. That is, if $A \leq B$ then

$$m(S, d, \gamma, B) \leq m(S, d, \gamma, A).$$

We prove (15) by induction on $t$.

**Base case** ($i = 0$)**.** The initialization sets

$$\varepsilon_L^{(0)}(U) = U - \text{PROXY}(D_R, d, \gamma),$$

where the proxy term is independent of $U \in \{A, B\}$. Hence $\varepsilon_L^{(0)}(A) \le \varepsilon_L^{(0)}(B)$. Next,

$$\varepsilon_R^{(0)}(U) = U - m(D_L, d, \gamma, \varepsilon_L^{(0)}(U)).$$

Since $\varepsilon_L^{(0)}(A) \le \varepsilon_L^{(0)}(B)$ and $m$ is monotone in its budget argument,

$$m\left(D_L, d, \gamma, \varepsilon_L^{(0)}(B)\right) \le m\left(D_L, d, \gamma, \varepsilon_L^{(0)}(A)\right).$$

Therefore

$$\varepsilon_R^{(0)}(A) = A - m\left(D_L, d, \gamma, \varepsilon_L^{(0)}(A)\right) \le B - m\left(D_L, d, \gamma, \varepsilon_L^{(0)}(B)\right) = \varepsilon_R^{(0)}(B).$$

**Inductive step.** Assume (15) holds at iteration $i$. The updates in SOLVE_SIBLINGS are

$$\varepsilon_R^{(i)}(U) = U - m(D_L, d, \gamma, \varepsilon_L^{(i)}(U)), \qquad \varepsilon_L^{(i+1)}(U) = U - m(D_R, d, \gamma, \varepsilon_R^{(i)}(U)).$$

Using $\varepsilon_L^{(i)}(A) \le \varepsilon_L^{(i)}(B)$ and monotonicity of $m$ yields $\varepsilon_R^{(i)}(A) \le \varepsilon_R^{(i)}(B)$. Applying monotonicity again with $\varepsilon_R^{(i)}(A) \le \varepsilon_R^{(i)}(B)$ gives $\varepsilon_L^{(i+1)}(A) \le \varepsilon_L^{(i+1)}(B)$.

Thus (15) holds for $i + 1$, completing the induction. Note that if the while loop terminates for budget $A$ but not $B$ (or vice versa), we write the update rule anyway (which does not change the return, as it is solved with the same budget as the last iteration). This needed to be handled because the smaller budget could run in fewer iterations, the same number of iterations, or more iterations than the larger budget. $\qquad \square$

Calling Lemma A.2 with $d$ (and using our inductive hypothesis to satisfy the assumption of that lemma) gives us that the final call to $D_L$ and $D_R$ in SOLVE_SIBLINGS involves a more permissive budget when the initial budget is larger. Therefore, we know that the found trees for $D_L$ and $D_R$ with the larger initial budget are supersets of those found with the smaller initial budget, due to our inductive hypothesis. That means any tree found by SOLVE_SIBLINGS with the smaller budget will also be found by SOLVE_SIBLINGS in the larger budget call, maintaining our invariant.

Since we now know all trees will only be in $\hat{\mathcal{R}}^{\text{PRAXIS }(D_S, d+1, \gamma, A)}$ if they are also in $\hat{\mathcal{R}}^{\text{PRAXIS }(D_S, d+1, \gamma, B)}$, we have shown our inductive step. Accordingly, by induction we have proven Proposition A.1. $\qquad \square$

**Theorem A.3** (PRAXIS recovers the proxy tree). *Fix any subproblem $(D, d)$ and let the proxy return an objective for a tree $f^{\text{px}} \in \mathcal{T}_d$, which we call*

$$P(D, d) := \text{PROXY}(D, d, \gamma) = \text{Obj}(f^{\text{px}}, D, \gamma).$$

*If* PRAXIS *is called with budget $\varepsilon_{abs} \ge P(D, d)$, i.e.*

$$G \leftarrow \text{PRAXIS }(D, d, \gamma, \varepsilon_{abs}),$$

*then:*

1. *the returned AND/OR graph $G$ contains (represents) the proxy tree $f^{\text{px}}$, ($G$ is nonempty);*

2. *the minimum objective stored at the root OR-node satisfies*

$$G.\text{min\_objective} \le P(D, d).$$

*In particular, the root node satisfies*

$$G.\text{min\_objective} \le P(D, d).$$

*Proof.* We argue by induction on the depth $d$.

**Base case ($d = 0$).** Then $f^{\text{px}}$ is a leaf predicting some label $b \in \{0, 1\}$, (denote this label wlog as $b'$) hence

$$P(D, 0) = C_{b'}, \tag{16}$$

one of the leaf costs computed in Algorithm 1, before any pruning. Since $\varepsilon_{\text{abs}} \geq P(D, 0)$, that leaf is added and thus

$$G.\text{min\_objective} \leq C_{b'} = P(D, 0). \tag{17}$$

**Inductive step** ($d \geq 1$). Let the proxy tree at $(D, d)$ split at the root on feature $j^\star$, inducing $(D_{f_{\text{left}}^{\text{px}}}, D_{f_{\text{right}}^{\text{px}}})$ and proxy subtrees $f_{\text{left}}^{\text{px}}, f_{\text{right}}^{\text{px}}$ of depth at most $d - 1$. Define the proxy calls made by PRAXIS at this split:

$$P_L := \text{PROXY}(D_{f_{\text{left}}^{\text{px}}}, d - 1, \gamma), \qquad P_R := \text{PROXY}(D_{f_{\text{right}}^{\text{px}}}, d - 1, \gamma). \tag{18}$$

By the refinement property (Definition 3.1) applied to the proxy tree,

$$P_L \leq \text{Obj}(f_{\text{left}}^{\text{px}}, D_{f_{\text{left}}^{\text{px}}}, \gamma), \qquad P_R \leq \text{Obj}(f_{\text{right}}^{\text{px}}, D_{f_{\text{right}}^{\text{px}}}, \gamma). \tag{19}$$

Because the objective is additive across a split,

$$P(D, d) = \text{Obj}(f^{\text{px}}, D, \gamma) \tag{20}$$
$$= \text{Obj}(f_{\text{left}}^{\text{px}}, D_{f_{\text{left}}^{\text{px}}}, \gamma) + \text{Obj}(f_{\text{right}}^{\text{px}}, D_{f_{\text{right}}^{\text{px}}}, \gamma)$$
$$\geq P_L + P_R. \tag{21}$$

Since $\varepsilon_{\text{abs}} \geq P(D, d)$, we have

$$P_L + P_R \leq \varepsilon_{\text{abs}}, \tag{22}$$

so Algorithm 1 does not prune split $j^\star$, and it calls SOLVESIBLINGS on $(D_{f_{\text{left}}^{\text{px}}}, D_{f_{\text{right}}^{\text{px}}})$.

Inside SOLVESIBLINGS, the first left budget is set to

$$\varepsilon_L = \varepsilon_{\text{abs}} - P_R. \tag{23}$$

Using $\varepsilon_{\text{abs}} \geq \text{Obj}(f_{\text{left}}^{\text{px}}, D_{f_{\text{left}}^{\text{px}}}, \gamma) + \text{Obj}(f_{\text{right}}^{\text{px}}, D_{f_{\text{right}}^{\text{px}}}, \gamma)$ and $P_R \leq \text{Obj}(f_{\text{right}}^{\text{px}}, D_{f_{\text{right}}^{\text{px}}}, \gamma)$, we obtain

$$\varepsilon_L = \varepsilon_{\text{abs}} - P_R \tag{24}$$
$$\geq \varepsilon_{\text{abs}} - \text{Obj}(f_{\text{right}}^{\text{px}}, D_{f_{\text{right}}^{\text{px}}}, \gamma)$$
$$\geq \text{Obj}(f_{\text{left}}^{\text{px}}, D_{f_{\text{left}}^{\text{px}}}, \gamma)$$
$$\geq P_L. \tag{25}$$

Therefore the recursive call

$$G_L \leftarrow \text{PRAXIS}(D_{f_{\text{left}}^{\text{px}}}, d - 1, \gamma, \varepsilon_L) \tag{26}$$

satisfies the inductive hypothesis, hence $G_L$ represents $f_{\text{left}}^{\text{px}}$ and

$$G_L.\text{min\_objective} \leq P_L \leq \text{Obj}(f_{\text{left}}^{\text{px}}, D_{f_{\text{left}}^{\text{px}}}, \gamma). \tag{27}$$

Next SOLVESIBLINGS sets the right budget to

$$\varepsilon_R = \varepsilon_{\text{abs}} - G_L.\text{min\_objective}. \tag{28}$$

Since $G_L.\text{min\_objective} \leq \text{Obj}(f_{\text{left}}^{\text{px}}, D_{f_{\text{left}}^{\text{px}}}, \gamma)$,

$$\varepsilon_R = \varepsilon_{\text{abs}} - G_L.\text{min\_objective} \tag{29}$$
$$\geq \varepsilon_{\text{abs}} - \text{Obj}(f_{\text{left}}^{\text{px}}, D_{f_{\text{left}}^{\text{px}}}, \gamma)$$
$$\geq \text{Obj}(f_{\text{right}}^{\text{px}}, D_{f_{\text{right}}^{\text{px}}}, \gamma)$$
$$\geq P_R. \tag{30}$$

Thus the call

$$G_R \leftarrow \text{PRAXIS}(D_{f_{\text{right}}^{\text{px}}}, d - 1, \gamma, \varepsilon_R) \tag{31}$$

also satisfies the inductive hypothesis, so it represents $f_R^{\mathrm{px}}$ and

$$G_R.\mathrm{min\_objective} \leq P_R. \tag{32}$$

Finally, Algorithm 1 adds this split, attaching $G_L$ and $G_R$. Therefore $G$ represents the proxy tree $f^{\mathrm{px}}$. Moreover, since $f^{\mathrm{px}}$ is a feasible tree in $G$ with objective $P(D, d)$,

$$G.\mathrm{min\_objective} \leq P(D, d). \tag{33}$$

The while-loop in SOLVESIBLINGS only increases budgets when a better (lower objective) subtree is found, so the above lower bounds on $\varepsilon_L$ and $\varepsilon_R$ are never violated by subsequent refinements (Proposition A.1 establishes this monotonicity). $\square$

We note that Theorem A.3 directly applies to PRAXIS as the budget is set as relative to the proxy algorithm at the root:

$$\varepsilon_{\mathrm{abs}} \leftarrow \left(1 + \varepsilon_{\mathrm{mult}}\right) \cdot \mathrm{PROXY}(D, d, \gamma)$$

Beyond Theorem A.3, which guarantees we find the proxy tree at the root, we also show below (Corollary A.4) that we recover the proxy tree at all explored subproblems.

**Corollary A.4** (The AND/OR subgraph for every explored subproblem contains the proxy tree). *Assume the root call to PRAXIS satisfies*

$$\varepsilon_{\mathrm{abs}} \geq P(D, d).$$

*Then every unpruned OR node in the returned AND/OR graph is explored with budget*

$$\varepsilon'_{\mathrm{abs}} \geq \mathrm{PROXY}(D', d', \gamma).$$

*Consequently, by Theorem A.3, the subgraph rooted at every OR node contains the proxy tree for that subproblem, and the minimum objective stored at that node is at most $P(D', d')$.*

*Proof.* As in Theorem A.3, we use the notation

$$P(D, d) := \mathrm{PROXY}(D, d, \gamma) = \mathrm{Obj}(f^{\mathrm{px}}, D, \gamma).$$

Let a split produce child subproblems with proxy values

$$P_L := P(D_L, d - 1), \qquad P_R := P(D_R, d - 1),$$

and suppose this split is not pruned, i.e.

$$P_L + P_R \leq \varepsilon_{\mathrm{abs}}.$$

Then the initial left budget used by SOLVESIBLINGS is

$$\varepsilon_L := \varepsilon_{\mathrm{abs}} - P_R.$$

Therefore,

$$\varepsilon_L = \varepsilon_{\mathrm{abs}} - P_R \geq P_L.$$

So the recursive call on the left subproblem is made with budget at least its proxy objective, which by Theorem A.3 is sufficient to recover the proxy tree on the left.

Now let $G_L$ denote the returned left subgraph. Since the left proxy tree is contained in $G_L$, its minimum objective satisfies

$$G_L.\mathrm{min\_objective} \leq P_L.$$

The right budget is then set to

$$\varepsilon_R := \varepsilon_{\mathrm{abs}} - G_L.\mathrm{min\_objective}.$$

Using $G_L.\text{min\_objective} \leq P_L$, we obtain

$$\varepsilon_R = \varepsilon_{\text{abs}} - G_L.\text{min\_objective} \geq \varepsilon_{\text{abs}} - P_L \geq P_R,$$

since $P_L + P_R \leq \varepsilon_{\text{abs}}$. Hence the recursive call on the right subproblem is also made with budget at least its proxy objective, so the proxy tree on the right is recovered as well.

Finally, SOLVESIBLINGS only increases budgets during subsequent refinement steps (by Lemma A.2). Thus once a child budget is at least the corresponding proxy value, this inequality is never lost (due to the monotonicity of budgets shown in Proposition A.1).

Repeated application of this logic down the AND/OR graph shows the corollary. $\qquad\square$

**Theorem A.5** (All Trees returned by PRAXIS can be Arbitrarily Better than Greedy). *Fix a depth budget $d$ and let $\varepsilon_{mult} \geq 0$ denote the multiplicative slack used in the root budget. Assume $\lambda = \gamma = 0$ (i.e., the objective is just misclassification error) and use a proxy algorithm satisfying $\forall S, \text{PROXY}(S, d, \gamma) \leq \text{Obj}(\text{LICKETYSPLIT}(S, d, \gamma))$, where LICKETYSPLIT refers to Algorithm 16. Then for every $\delta > 0$ there exists a data distribution $\mathcal{D}$ and sample size $n$ such that, with high probability over $S \sim \mathcal{D}^n$, setting $\varepsilon_{abs} = (1 + \varepsilon_{mult})\text{PROXY}(S, d, \gamma)$:*

1. *A pure greedy (information-gain) depth-$d$ tree achieves accuracy at most $\frac{1}{2} + \delta$.*

2. *Every tree $T$ returned by $PRAXIS(S, d, \gamma, \varepsilon_{abs})$ achieves accuracy at least $1 - \delta$.*

*Proof.* Fix $\delta > 0$ and define $\eta := \delta/(1 + \varepsilon_{\text{mult}})$. By Theorem A.7 of Babbar et al. (2025), for this $\eta$ and depth $d$ there exist $\mathcal{D}$ and $n$ such that, with high probability over $S \sim \mathcal{D}^n$: (i) LICKETYSPLIT returns a depth-$d$ tree with error at most $\eta$, and (ii) pure greedy achieves error at least $\frac{1}{2} - \eta$.

For (1), since $\eta \leq \delta$, we have $\text{acc}_{\text{GREEDY}} \leq \frac{1}{2} + \eta \leq \frac{1}{2} + \delta$.

For (2), let $P := \text{PROXY}(S, d, \gamma)$. By assumption,

$$P \leq \text{Obj}(\text{LICKETYSPLIT}(\mathcal{S}, d, \gamma)).$$
$$\text{applying the result discussed above, that } \text{Obj}(\text{LICKETYSPLIT}(\mathcal{S}, d, \gamma)) \leq \eta:$$
$$P \leq \eta.$$

PRAXIS is run with root budget $\varepsilon_{\text{abs}} := (1 + \varepsilon_{\text{mult}})P$, hence

$$\varepsilon_{\text{abs}} \leq (1 + \varepsilon_{\text{mult}})\eta = \delta.$$

Because every materialized tree respects the budget at the root, every tree $T$ materialized satisfies $\text{Obj}(T, S, \gamma) \leq \varepsilon_{\text{abs}} \leq \delta$. Since $\gamma = 0$, $\text{Obj}(T, S, \gamma)$ equals the misclassification error of $T$, so $\text{acc}(T) \geq 1 - \delta$. $\qquad\square$

Theorem A.5 establishes that PRAXIS can return a set of trees that are all arbitrarily better than greedy. Additionally, using any proxy algorithm that performs at least as well as greedy, the trees returned by PRAXIS will never be more than $\varepsilon$ worse than greedy, and will always include one at least as good as greedy.

**Corollary A.6** (Recovery guarantee for c-approximation proxy algorithms). *Assume PROXY is a c-approximation algorithm, i.e., for all $(D', d')$,*

$$\text{PROXY}(D', d', \gamma) \leq c \min_{t \in \mathcal{T}_{d'}} \text{Obj}(t, D', \gamma).$$

*Then, for any budget $\varepsilon_{\text{abs}}$, PRAXIS returns every tree $t_r \in \mathcal{T}_d$ such that*

$$\text{Obj}(t_r, D, \gamma) \leq \frac{\varepsilon_{\text{abs}}}{c}.$$

*Proof.* Fix any tree $t_r \in \mathcal{T}_d$ with

$$\text{Obj}(t_r, D, \gamma) \leq \frac{\varepsilon_{\text{abs}}}{c}.$$

Let $u$ be any internal node of $t_r$, let its depth be $d'$, and let

$$\{s_1, \ldots, s_{d'}, u_{\text{left}}, u_{\text{right}}\}$$

be the associated frontier cut as in Theorem 3.5. Since these subtrees partition the leaves of $t_r$, the sum of their optimal objectives is at most the objective of $t_r$:

$$\sum_{i=1}^{d'} \min_{t \in \mathcal{T}_{d_{s_i}}} \text{Obj}(t, D_{s_i}, \gamma) \;+\; \min_{t \in \mathcal{T}_{d_u - 1}} \text{Obj}(t, D_{u_{\text{left}}}, \gamma) \;+\; \min_{t \in \mathcal{T}_{d_u - 1}} \text{Obj}(t, D_{u_{\text{right}}}, \gamma) \;\leq\; \text{Obj}(t_r, D, \gamma).$$

By the $c$-approximation property of PROXY,

$$\sum_{i=1}^{d'} \text{PROXY}(D_{s_i}, d_{s_i}, \gamma) \;+\; \text{PROXY}(D_{u_{\text{left}}}, d_u - 1, \gamma) \;+\; \text{PROXY}(D_{u_{\text{right}}}, d_u - 1, \gamma) \;\leq\; c\,\text{Obj}(t_r, D, \gamma) \;\leq\; \varepsilon_{\text{abs}}.$$

Thus the frontier-cut condition of Theorem 3.5 holds at every internal node of $t_r$, so PRAXIS returns $t_r$. $\qquad\square$

**Lemma A.7** (Convergence of iterative budget refinement). *Fix a split that has been evaluated and not pruned in Algorithm 1. Let $P_L, P_R$ be the proxy objectives and $\text{OPT}_L, \text{OPT}_R$ be the optimal subtree objectives for the left and right subproblems (under the same depth constraint).*

*Then the iterative budget refinement procedure in Algorithm 3 performs at most*

$$2 \min\big(P_L - \text{OPT}_L, \; P_R - \text{OPT}_R\big) + 2$$

*recursive calls to Algorithm 1.*

*Proof.* The SOLVE_SIBLINGS procedure alternates between solving the left and right subproblems under budgets derived from the current estimate of the opposite side. Initially, the left budget is set to $\varepsilon_{\text{abs}} - P_R$. The right budget is then set to $\varepsilon_{\text{abs}} - \min_L$, where $\min_L \leq P_L$. After this point, we make at most one PRAXIS call for each improvement to these budgets.

Because objectives are integral, each successful refinement strictly decreases the current best objective on the active side by at least 1. Moreover, since the proxy objective corresponds to a realizable tree, no side can improve beyond its optimal objective:

$$\varepsilon_L \geq \varepsilon_{\text{abs}} - \text{OPT}_R, \qquad \varepsilon_R \geq \varepsilon_{\text{abs}} - \text{OPT}_L.$$

Therefore, the left side can be refined at most $P_L - \text{OPT}_L$ times (and the 1 initial call), and the right side at most $P_R - \text{OPT}_R$ times (and the 1 initial call).

To obtain the tighter bound, observe that once one side ceases to improve, at most one additional recursive call is made on the other side. Suppose the left side converges first. In this case, we incur one initial call, at most

$$2 \min\big(P_L - \text{OPT}_L, \; P_R - \text{OPT}_R\big)$$

refinement calls as the two sides alternate, and a final call to the right side with the optimal budget decremented. An analogous argument applies if the right side converges first.

Summing these contributions – the initial call, the alternating refinement calls, and the final recursive call – yields the stated bound. $\qquad\square$

**Theorem A.8** (Slack needed for perfect approximation if the proxy has additive optimality gaps). *Let $t$ be any tree. Let $D$ be a dataset, $d$ be a depth limit, and $\gamma$ be a per-leaf penalty. For each internal node $u$ of $t$ (denoted $u \in \text{Int}(t)$), let $C(u)$ denote its frontier cut. Denote nonnegative errors $\{\Delta_v\}$ such that for every subproblem $v$,*

$$\text{PROXY}(D_v, d_v, \gamma) \;\leq\; \text{OPT}(v) + \Delta_v.$$

*and define $\eta$ such that $t$ is $\eta$-suboptimal at the root:*

$$\text{Obj}(t, D, \gamma) \;\leq\; \text{OPT}(\text{root}) + \eta.$$

*If the budget is set with*

$$\varepsilon_{abs} \geq \mathrm{OPT}(\mathrm{root}) + \eta \; + \; \max_{u \in \mathrm{Int}(t)} \sum_{v \in C(u)} \Delta_v,$$

*then $t$ is fully represented in the AND/OR graph.*

*Proof.* For every internal node $u$ of $t$, the frontier cut is within budget.

$$\sum_{v \in C(u)} \mathrm{PROXY}(D_v, d_v, \gamma) \leq \sum_{v \in C(u)} \mathrm{OPT}(v) + \sum_{v \in C(u)} \Delta_v \tag{34}$$

$$\leq \sum_{v \in C(u)} \mathrm{OPT}(v) + \max_{u \in \mathrm{Int}(t)} \sum_{v \in C(u)} \Delta_v. \tag{35}$$

For the tree $t$, the frontier cut $C(u)$ partitions the remaining work below the already fixed prefix, so the sum of optimal subtree objectives below the cut is at most the objective of $t$. Hence,

$$\sum_{v \in C(u)} \mathrm{OPT}(v) \; \leq \; \mathrm{Obj}(t, D, \gamma) \; \leq \; \mathrm{OPT}(\mathrm{root}) + \eta. \tag{36}$$

Combining (35) and (36), we obtain

$$\sum_{v \in C(u)} \mathrm{PROXY}(D_v, d_v, \gamma) \leq \mathrm{OPT}(\mathrm{root}) + \eta \; + \; \max_{u \in \mathrm{Int}(t)} \sum_{v \in C(u)} \Delta_v \tag{37}$$

$$\leq \varepsilon_{\mathrm{abs}}. \tag{38}$$

$\square$

**Proposition A.9** (A greedy tree algorithm is a proxy algorithm)**.** *Let* $\mathrm{GREEDY}(D, d, \gamma)$ *be the output of the greedy algorithm in Algorithm 15 or 17 that returns the objective of a tree $t \in \mathcal{T}_d$. Define*

$$\mathrm{PROXY}_0(D, d, \gamma) \; := \; \mathrm{GREEDY}(D, d, \gamma).$$

*Then $\mathrm{PROXY}_0$ is a proxy algorithm (Definition 3.1).*

*Proof.* By recursive construction, when $\mathrm{GREEDY}$ builds $t$, the subtree $t_u$ at any node $u$ is exactly the tree returned by calling $\mathrm{GREEDY}$ on $(D_u, d_u, \gamma)$. Hence $\mathrm{PROXY}_0(D_u, d_u, \gamma) = \mathrm{Obj}(t_u, D_u, \gamma)$, which implies the stated inequality. $\square$

**Proposition A.10** (LicketySPLIT is a proxy algorithm)**.** *Let* $\mathrm{LICKETYSPLIT}(D, d, \gamma)$ *be Algorithm 16 (or Algorithm 18 with $\ell = 1$) that returns the objective of a tree $t \in \mathcal{T}_d$, and define*

$$\mathrm{PROXY}_1(D, d, \gamma) \; := \mathrm{LICKETYSPLIT}(D, d, \gamma).$$

*Then $\mathrm{PROXY}_1$ is a proxy algorithm.*

*Proof.* Algorithm 16 or 18 selects a split at the root (via greedy completions) and then *recurses by calling itself* on each child subproblem to build the left and right subtrees, finally returning the objective of that recursively built tree. Notably, these are identical recursive calls (except for updating the depth budget to pass down the constraint). Therefore, for any node $u$ in the returned tree, rerunning $\mathrm{LICKETYSPLIT}$ on $(D_u, d_u, \gamma)$ reproduces the same subtree $f_u$, so $\mathrm{PROXY}_1(D_u, d_u, \gamma) = \mathrm{Obj}(f_u, D_u, \gamma)$, implying the refinement inequality. $\square$

Likewise, this holds for our generalization of LicketySPLIT, detailed in Algorithm 18 of subsection B.5. Because the algorithm recurses with the same $\ell$ (the only added parameter), just one depth lower, the refinement inequality holds with equality for this entire family of decision tree algorithms.

**Lemma A.11** (LicketySPLIT is a memory-efficient proxy algorithm)**.** *Given a dataset of size $n$ with $k$ binary features, the memory cost of the LicketySPLIT (Babbar et al., 2025) algorithm can be limited to $\mathcal{O}(nk)$.*

*Proof.* We can show this by induction:

**Inductive hypothesis:** Let $c$ be some fixed constant. Given that LicketySPLIT called with remaining depth $d-1$ and any dataset of dimension $n_1 \times k$, with $n_1 \leq n$ takes memory no greater than $c(n_1 k + 1)$, we want to show that LicketySPLIT with depth $d$ and any dataset of dimension $n_2 \times k$, with $n_2 \leq n$, takes memory no greater than $c(n_2 k + 1)$.

Pick $c$ such that the original dataset size is $\leq cnk$ (and all subsets of size $n_i$ are similarly of size $\leq cn_i k$) and the storage required for a few constants is $\leq c$.

**Base case:** When the remaining depth budget is 0, LicketySPLIT requires no additional memory beyond its input dataset (size $\leq cn_2 k$) and a constant (size $\leq c$); all that is required is to compute the leaf objective, corresponding to the minimum of the number of positive vs negative entries in label $y$. So the memory cost is not greater than $c(nk + 1)$.

**Inductive step:** When depth is $> 0$, LicketySPLIT must call a greedy subroutine for the left and right children from every possible binary feature split. For a given potential split, the required memory is no more than $cn_2 k + c$: we need only provide the left and right training data subsets to the greedy algorithms, and store the sum of the resulting objectives. By going through one split at a time and tracking the best objective so far and the corresponding feature, LicketySPLIT can do this without persisting more than constant memory. Once the optimal split is known, LicketySPLIT then constructs the left and right subproblems corresponding to that split (still with total size matching the original dataset size, which is $\leq cn_2 k$). LicketySPLIT then must run LicketySPLIT with one fewer depth for the left and right subproblems of the selected split. Note that each of the two subproblems has a number of samples no greater than $n_2 - 1$, because the optimal split must place at least one sample in each subproblem. So, by the inductive hypothesis, each split requires $\leq c((n_2 - 1)k + 1) \leq c(n_2 k)$ memory to be solved individually. After one subproblem is solved, the memory used for it can be freed except for a single constant, and we can solve the other subproblem with $\leq c(n_2 k)$ memory. In total, we use $\leq c(n_2 k + 1)$ memory, as required. (Note that once these two LicketySPLIT approaches are ready to be called, no other information needs to be persisted in memory; LicketySPLIT will just return the sum of these two calls. So the total amount of information in memory remains bounded by $cnk + c$.)

Therefore, by induction, the total memory use for any LicketySPLIT call is bounded by $cnk + c$, and thus in $O(nk)$. $\qquad\square$

### A.3. Rashomon Set of Rule Lists

**Theorem A.12** (Modified PRAXIS returns a superset of the Rashomon set of rule lists)**.** *Let a simple rule list be defined as a tree where each split has at least one leaf child: that is, adding the constraint that for any non-leaf tree $t$, $\min(depth(t_{left}), depth(t_{right})) = 0$. Then, when the proxy algorithm in Algorithm 1 is any algorithm with performance guaranteed to at least match the objective of the majority leaf prediction, and the pruning from Algorithm 1 is adjusted to take a min instead of a sum, as in Algorithm 5, the resulting Rashomon set will include all rule lists within the depth budget.*

*Proof.* First note that a few other adjustments to the algorithm are needed, to keep its behaviour fully specified with the new pruning condition (it is now possible to explore some subproblems that do not include any trees within the budget). These changes are detailed in Algorithms 5 and 6. Intuitively, these changes mean that we have a slight adjustment to the conditions on Theorem 3.5, such that rule lists are not pruned.

Now, we must show that a valid rule list within the depth budget $d$ is never pruned. We will show this via induction.

Consider any rule list $t$ on a dataset $D$ with

$$\mathrm{Obj}(t, D, \gamma) \leq \varepsilon_{\mathrm{abs}}. \tag{39}$$

**Base Case (Leaf)**    If $t$ is a leaf with $\mathrm{Obj}(t, D, \gamma) \leq \varepsilon_{\mathrm{abs}}$, then note that adding leaves to the AND/OR graph (if they are within budget) happens before (and is unrelated to) any pruning of splits. Thus, $t$ is trivially recovered.

**Inductive Step (Non-Leaf)**    Suppose $t$ is not a leaf. Let $t$ be the root split of $t$, inducing subproblem datasets $(D_{t_{\mathrm{left}}}, D_{t_{\mathrm{right}}})$ and subtrees $(t_{\mathrm{left}}, t_{\mathrm{right}})$. Since $t$ is a rule list, at least one child is a leaf. Without loss of generality, assume $t_{\mathrm{right}}$ is a leaf and $t_{\mathrm{left}}$ is a rule list (that could be a leaf as well). Note that there is no loss of generality because the modified pruning condition ($\min(P_L, P_R)$) is symmetric. Moreover, Algorithm 6 never subtracts more than a single leaf objective from the opposite branch, regardless of which side contains the leaf.

Because $t$ is feasible,

$$\mathrm{Obj}(t, D, \gamma) = \mathrm{Obj}(t_{\mathrm{left}}, D_{t_{\mathrm{left}}}, \gamma) + \mathrm{Obj}(t_{\mathrm{right}}, D_{t_{\mathrm{right}}}, \gamma) \leq \varepsilon_{\mathrm{abs}}. \tag{40}$$

Since $t_{\mathrm{right}}$ is a leaf, its objective is at least the optimal leaf objective on $D_{t_{\mathrm{right}}}$, which (by assumption) is at least the proxy objective. Defining $P_R = \mathrm{PROXY}(D_{t_{\mathrm{right}}}, d-1, \gamma)$ and $P_L = \mathrm{PROXY}(D_{t_{\mathrm{left}}}, d-1, \gamma)$,

$$P_R \leq \mathrm{MajorityLeafObj}(D_{t_{\mathrm{right}}}) \leq \mathrm{Obj}(t_{\mathrm{right}}, D_{t_{\mathrm{right}}}, \gamma). \tag{41}$$

Combining (40) and (41) gives

$$\mathrm{Obj}(t_{\mathrm{left}}, D_{t_{\mathrm{left}}}, \gamma) \leq \varepsilon_{\mathrm{abs}} - \mathrm{Obj}(t_{\mathrm{right}}, D_{t_{\mathrm{right}}}, \gamma) \tag{42}$$

$$\leq \varepsilon_{\mathrm{abs}} - P_R. \tag{43}$$

Now, consider the new pruning condition for a split. Because $t_{\mathrm{right}}$ is a leaf, we know the following:

$$\min(P_L, P_R) \leq P_R \tag{44}$$

$$\leq \mathrm{Obj}(t_{\mathrm{right}}, D_{t_{\mathrm{right}}}, \gamma) \tag{45}$$

$$\leq \mathrm{Obj}(t, D_t, \gamma) - \gamma \tag{46}$$

$$\leq \varepsilon_{\mathrm{abs}} - \gamma. \tag{47}$$

Thus the modified pruning rule $\min(P_L, P_R) > \varepsilon_{\mathrm{abs}} - \gamma$ does not trigger, and the root split of $t$ is explored.

By (43), the recursive call on dataset $D_{t_{\mathrm{left}}}$ with budget $(\varepsilon_{\mathrm{abs}} - P_R)$ admits $t_{\mathrm{left}}$ as a feasible solution. By the inductive hypothesis, PRAXIS fully recovers $t_{\mathrm{left}}$. Since $t_{\mathrm{right}}$ is a feasible leaf, the modified sibling solver marks a valid subtree as existing and the split is retained. Hence $t$ is represented in the search graph.

$\square$

Although we design PRAXIS with the intention of approximating the full Rashomon set of sparse decision trees, we note that one could modify this proxy-approach to guarantee that the exact Rashomon set of rule lists is contained in the solution. Then, any valid cross product of these one sided decision tree branches would also be a valid solution, along with others.

In practice, the decision tree algorithms perform much better than a single leaf, so we are able to deploy more aggressive pruning to approximate the full Rashomon set faster and better than this approach (see subsection D.8).

---

**Algorithm 5** MODIFIED_PRAXIS$(D, d, \gamma, \varepsilon_{\text{abs}})$, for Theorem A.12. Changes in red

---

**Require:** Subproblem dataset $D$, remaining depth $d$, per-leaf penalty $\gamma$, budget $\varepsilon_{\text{abs}}$,

1: valid_tree_exists ← False
2: Let $G \leftarrow$ ORNODE$(\varepsilon_{\text{abs}})$ {Initialize subgraph for subtrees found within budget $\varepsilon_{\text{abs}}$ (See Appendix B.2)}
3: **for** $b \in \{0, 1\}$ **do**
4:     {For each possible leaf prediction $b$, set $C_b$ to the objective if all points were in a single leaf: $\lambda$ + misclassification error.}
5:     $C_b \leftarrow \gamma + \left|\{(x_i, y_i) \in D : y_i \neq b\}\right|$
6:     **if** $C_b \leq \varepsilon_{\text{abs}}$ **then**
7:         valid_tree_exists ← True
8:         ADDLEAF$(G, b, C_b)$ {See Appendix B.2}
9:     **end if**
10: **end for**
11: **if** $d = 0$ **or** $\varepsilon_{\text{abs}} < 2\gamma$ **then**
12:     **return** $G$ {No depth for splits or any split would exceed the budget}
13: **end if**
14: **for each** feature $j$ **do**
15:     $(D_L, D_R) \leftarrow$ PARTITION$(D, j)$
16:     **if** $D_L = \emptyset$ **or** $D_R = \emptyset$ **then**
17:         **continue** {Skip degenerate splits}
18:     **end if**
19:     $P_L \leftarrow$ PROXY$(D_L, d-1, \gamma)$ {Proxy cost on left}
20:     $P_R \leftarrow$ PROXY$(D_R, d-1, \gamma)$ {Proxy cost on right}
21:     **if** $\min(P_L, P_R) > \varepsilon_{\text{abs}} - \gamma$ **then**
22:         **continue** {Prune split if proxy completions exceed the budget}
23:     **end if**
24:     valid_subtree_exists, $G_L, G_R \leftarrow$ MODIFIED_SOLVE_SIBLINGS$(D_L, D_R, \gamma, d-1, \varepsilon_{\text{abs}}, P_L, P_R)$ {Find subgraphs for the left and right subproblems of this split; described in Algorithm 6}
25:     **if** valid_subtree_exists **then**
26:         valid_tree_exists ← True
27:         ADDSPLIT$(G, j, G_L, G_R)$ {Add $G_L, G_R$ as a split for the current subgraph $G$; see Appendix B.2}
28:     **end if**
29: **end for**
30: **return** $G$ {Rashomon graph for the subproblem}

**Usage Note:** If provided with only $\varepsilon_{\text{mult}}$ and not $\varepsilon_{\text{abs}}$, set Rashomon budget relative to proxy:

$$\varepsilon_{\text{abs}} \leftarrow \left(1 + \varepsilon_{\text{mult}}\right) \cdot \text{PROXY}(D, \lambda, d, |D|)$$

---

---

**Algorithm 6** MODIFIED_SOLVE_SIBLINGS($D_L, D_R, \gamma, d, \varepsilon_{\text{abs}}, P_L, P_R$), modified for Theorem A.12. Changes in red.

---

**Require:** Left/right datasets $D_L, D_R$, remaining depth $d$, regularization $\gamma$, parent budget $\varepsilon_{\text{abs}}$, proxy cost $P_R$

1: valid_tree_exists ← False
2: $\varepsilon_L \leftarrow -\infty$ {largest budget used for solving $D_L$ with PRAXIS ; currently we've not run on $D_L$ at all.}
3: $\varepsilon_R \leftarrow -\infty$ {largest budget used for solving $D_R$ with PRAXIS ; currently we've not run on $D_R$ at all.}
4: $\varepsilon_L^{(\text{new})} \leftarrow \varepsilon_{\text{abs}} - P_R$ {new budget to use for $D_L$ with PRAXIS }
5: $\varepsilon_R^{(\text{new})} \leftarrow \varepsilon_{\text{abs}} - P_L$ {new budget to use for $D_R$ with PRAXIS }
6: **while** $\varepsilon_L^{(\text{new})} > \varepsilon_L$ **do**
7:    $\varepsilon_L \leftarrow \varepsilon_L^{(\text{new})}$
8:    **if** $\varepsilon_L > 0$ **then**
9:       valid_subtree_exists, $G_L \leftarrow$ MODIFIED_PRAXIS($D_L, d, \gamma, \varepsilon_L$)
10:       **if** valid_subtree_exists **then**
11:          valid_tree_exists ← True
12:          $\varepsilon_R^{(\text{new})} \leftarrow \varepsilon_{\text{abs}} - G_L.min\_objective$
13:       **end if**
14:    **end if**
15:    **if** $\varepsilon_R^{(\text{new})} > \varepsilon_R$ **then**
16:       $\varepsilon_R \leftarrow \varepsilon_R^{(\text{new})}$
17:       **if** $\varepsilon_R > 0$ **then**
18:          valid_subtree_exists, $G_R \leftarrow$ MODIFIED_PRAXIS($D_L, d, \gamma, \varepsilon_R$)
19:          **if** valid_subtree_exists **then**
20:             valid_tree_exists ← True
21:             $\varepsilon_L^{(\text{new})} \leftarrow \varepsilon_{\text{abs}} - G_R.min\_objective$
22:          **end if**
23:       **end if**
24:       $G_R \leftarrow$ MODIFIED_PRAXIS($D_R, d, \gamma, \varepsilon_R, N$)
25:       $\varepsilon_L^{(\text{new})} \leftarrow \varepsilon\_\text{abs} - G_R.min\_objective$
26:    **end if**
27: **end while**
28: **return** valid_tree_exists, $G_L, G_R$

---

## A.4. Refining Proxy Guesses

Beyond being guaranteed to recover the proxy tree at each subproblem in the AND/OR graph, we are also guaranteed to find trees that would be generated from stronger decision tree algorithms. In Algorithm 7, we give a single decision tree algorithm that recursively selects splits according to proxy-completed objectives; this is precisely the pilot algorithm induced by the proxy. We note that if the proxy is a greedy tree algorithm, then this wrapper corresponds to LICKETYSPLIT (with $\ell = 1$ in our generalization). If the proxy is our default of LICKETYSPLIT($\ell = 1$), then this wrapper corresponds to LICKETYSPLIT($\ell = 2$). Having this guarantee speeds up the convergence of iterative budget refinement and improves the recovery conditions of PRAXIS .

---

**Algorithm 7** PROXYWRAPPER($D', d, \gamma$)

---

**Require:** Subproblem dataset $D' \subseteq D$, remaining depth $d$, leaf penalty $\gamma$
1: $n' \leftarrow |D'|$, $p \leftarrow$ # positives in $D'$
2: *leaf_loss* $\leftarrow \gamma + \min(p, n' - p)$
3: **if** $d = 0$ or *leaf_loss* $\leq 2\gamma$ **then**
4:     **return** *leaf_loss* {early escape: either no depth, or split can't beat leaf}
5: **end if**
6: *best_sum* $\leftarrow +\infty$, *best_feature* $\leftarrow \perp$
7: **for each** feature $j$ **do**
8:     Partition $D'$ into $(D'_L, D'_R)$ by split $x_j$; **continue** if $D'_L = \emptyset$ or $D'_R = \emptyset$
9:     $s_j \leftarrow$ PROXY($D'_L, d - 1, \gamma$) + PROXY($D'_R, d - 1, \gamma$)
10:     **if** $s_j <$ *best_sum* **then**
11:         *best_sum* $\leftarrow s_j$, *best_feature* $\leftarrow j$
12:     **end if**
13: **end for**
14: *ans* $\leftarrow$ *leaf_loss*
15: **if** *best_feature* $\neq \perp$ **then**
16:     Partition $D'$ by *best_feature* into $(D'_L, D'_R)$
17:     $L \leftarrow$ PROXYWRAPPER($D'_L, d - 1, \gamma$) {recurse only on the best split}
18:     $R \leftarrow$ PROXYWRAPPER($D'_R, d - 1, \gamma$)
19:     *ans* $\leftarrow \min(ans, L + R)$ {take better of leaf and recursive completion}
20: **end if**
21: **return** *ans*

---

We first note that Algorithm 7 is guaranteed to have an objective at least as good as the proxy algorithm that it wraps. This is formalized by the following theorem.

**Theorem A.13** (A pilot algorithm using the proxy is no worse than the proxy)**.** *Fix any subproblem* $(D', d)$ *and leaf penalty* $\gamma$. *Let* $t^{\mathrm{px}} \in \mathcal{T}_d$ *be the tree whose objective corresponds to the proxy call:*

$$\mathrm{PROXY}(D', d, \gamma) \;=\; \mathrm{Obj}(t^{\mathrm{px}}, D', \gamma).$$

*Let* PROXYWRAPPER($D', d, \gamma$) *be as in Algorithm 7.*

*Then*

$$\mathrm{PROXYWRAPPER}(D', d, \gamma) \;\leq\; \mathrm{PROXY}(D', d, \gamma)$$

*Proof.* Write

$$W(D', d) \;:=\; \mathrm{PROXYWRAPPER}(D', d, \gamma), \qquad P(D', d) \;:=\; \mathrm{PROXY}(D', d, \gamma).$$

We prove by induction on $d$ that for every dataset $D'$,

$$W(D', d) \;\leq\; P(D', d).$$

**Base case** ($d = 0$)**.** PROXYWRAPPER returns the objective of the majority leaf on $D'$. Since $P(D', 0)$ is the objective of *some* depth 0 tree (a leaf) by Definition 3.1, we have $W(D', 0) \leq P(D', 0)$.

**Inductive step.** Fix $d \geq 1$ and assume that for all datasets $A$,

$$W(A, d-1) \ \leq \ P(A, d-1).$$

Fix an arbitrary dataset $D'$, and let $t^{\mathrm{px}} \in \mathcal{T}_d$ be a tree such that

$$P(D', d) \ = \ \mathrm{Obj}(t^{\mathrm{px}}, D', \gamma).$$

If $t^{\mathrm{px}}$ is a leaf, then $P(D', d)$ equals a leaf objective on $D'$, and PROXYWRAPPER always returns at most the majority leaf objective, so $W(D', d) \leq P(D', d)$.

Otherwise, let $k$ be the root split of $t^{\mathrm{px}}$, inducing a partition $(D'_L, D'_R)$ and subtrees $t^{\mathrm{px}}_L, t^{\mathrm{px}}_R$. By additivity of the objective,

$$P(D', d) = \mathrm{Obj}(t^{\mathrm{px}}, D', \gamma) = \mathrm{Obj}(t^{\mathrm{px}}_L, D'_L, \gamma) + \mathrm{Obj}(t^{\mathrm{px}}_R, D'_R, \gamma).$$

By Definition 3.1,

$$P(D'_L, d-1) \leq \mathrm{Obj}(t^{\mathrm{px}}_L, D'_L, \gamma), \qquad P(D'_R, d-1) \leq \mathrm{Obj}(t^{\mathrm{px}}_R, D'_R, \gamma),$$

so

$$P(D'_L, d-1) + P(D'_R, d-1) \ \leq \ P(D', d).$$

Now consider the feature $j$ selected by PROXYWRAPPER at $(D', d)$ (if all splits are invalid, then PROXYWRAPPER returns the leaf and we are done as PROXY also cannot split without increasing the objective beyond a majority leaf). Let $(D_L, D_R)$ be the partition induced by $j$. By the selection rule in PROXYWRAPPER,

$$P(D_L, d-1) + P(D_R, d-1) \ \leq \ P(D'_L, d-1) + P(D'_R, d-1) \ \leq \ P(D', d).$$

Moreover, PROXYWRAPPER takes the minimum of a majority leaf and a recursive completion. Thus,

$$W(D', d) \ \leq \ W(D_L, d-1) + W(D_R, d-1).$$

Applying the induction hypothesis to $D_L$ and $D_R$ gives

$$W(D_L, d-1) + W(D_R, d-1) \ \leq \ P(D_L, d-1) + P(D_R, d-1) \ \leq \ P(D', d),$$

hence $W(D', d) \leq P(D', d)$.

$\square$

We note that wrapping a proxy algorithm in this way has additionally clean properties: the wrapped procedure itself becomes a proxy algorithm whose refinement property holds with equality.

**Proposition A.14** (A pilot algorithm using the proxy satisfies the refinement property with equality). *Let*

$$W(D, d, \gamma) := \mathrm{PROXYWRAPPER}(D, d, \gamma).$$

*Then $W$ is a proxy algorithm. Moreover, if the tree returned by PROXYWRAPPER$(D, d, \gamma)$ splits into left and right subtrees $t_L, t_R$ on child subproblems $(D_L, d-1)$ and $(D_R, d-1)$, then*

$$W(D_L, d-1, \gamma) = \mathrm{Obj}(t_L, D_L, \gamma), \qquad W(D_R, d-1, \gamma) = \mathrm{Obj}(t_R, D_R, \gamma).$$

*That is, $W$ satisfies the refinement property with equality.*

*Proof.* Consider the tree that PROXYWRAPPER implicitly enumerates. Now consider calling PROXYWRAPPER on the left and right subproblems of the root node. Note that these are exactly the recursive calls that were made during the implicit construction of the tree.

Thus, the same left and right subtrees are reproduced when PROXYWRAPPER is applied to these subproblems. In particular, the proxy values returned on the child subproblems equal the objectives of the corresponding subtrees of the original tree. Hence, the refinement property holds with equality. Notably, this conclusion does not require the original proxy algorithm to satisfy refinement with equality.

$\square$

With these properties in hand, we now show that whenever PRAXIS explores a subproblem with a budget at least the proxy value (such as in the default initialization to PRAXIS), it necessarily recovers the corresponding PROXYWRAPPER tree at the root and at every subproblem in the AND/OR graph.

**Theorem A.15** (A pilot algorithm using the proxy maps to a tree found by PRAXIS). *Fix any subproblem $(D, d)$ and leaf penalty $\gamma$.*

*Let the proxy algorithm return a tree $f^{\mathrm{px}} \in \mathcal{T}_d$ with objective*

$$P(D, d) := \mathrm{PROXY}(D, d, \gamma) = \mathrm{Obj}(f^{\mathrm{px}}, D, \gamma).$$

*Let $f^{\mathrm{pw}}(D, d)$ denote the tree implicitly enumerated by PROXYWRAPPER$(D, d, \gamma)$ (Algorithm 7)*

*Then:*

1. *the returned AND/OR graph $G$ contains the ProxyWrapper tree $f^{\mathrm{pw}}(D, d)$;*

2. *In particular,*
$$G.\mathrm{min\_objective} \leq \mathrm{Obj}(f^{\mathrm{pw}}(D, d), D, \gamma) \leq \mathrm{Obj}(f^{\mathrm{px}}, D, \gamma).$$

*Proof.* **Claim.** If PRAXIS is invoked on $(D, d)$ with $\varepsilon_{\mathrm{abs}} \geq P(D, d)$, then the returned graph contains $f^{\mathrm{pw}}(D, d)$.

**Base case:** $d = 0$.

By construction, PROXYWRAPPER returns a leaf (in fact, the majority leaf). Since $\varepsilon_{\mathrm{abs}} \geq P(D, 0)$ and PRAXIS inserts all feasible leaf actions whose objective is $\leq \varepsilon_{\mathrm{abs}}$, the leaf corresponding to $f^{\mathrm{pw}}(D, 0)$ is included (if it fits within budget). The claim holds.

**Inductive step.** Assume the claim holds for depth $d - 1$. Fix $(D, d)$.

There are two cases.

*Case 1: ProxyWrapper selects a leaf.*

Then $f^{\mathrm{pw}}(D, d)$ is a leaf with objective *leaf_loss*. Since $\varepsilon_{\mathrm{abs}} \geq P(D, d)$, PRAXIS includes that leaf option (by the same reasoning as the base case). Thus $f^{\mathrm{pw}}(D, d)$ is contained in $G$.

*Case 2: ProxyWrapper selects a split $j^{\star}$.*

Let $(D_L(j^{\star}), D_R(j^{\star}))$ be the partition of $D$ induced by $j^{\star}$.

By definition of PROXYWRAPPER,

$$j^{\star} \in \arg\min_j \left( P(D_L(j), d - 1) + P(D_R(j), d - 1) \right).$$

Let $k$ denote the root split used by the proxy tree $f^{\mathrm{px}}$. By Definition 3.1, if $f^{\mathrm{px}}$ splits on $k$ with children $t_L^{\mathrm{px}}, t_R^{\mathrm{px}}$, then

$$P(D_L(k), d - 1) \leq \mathrm{Obj}(t_L^{\mathrm{px}}, D_L(k), \gamma), \qquad P(D_R(k), d - 1) \leq \mathrm{Obj}(t_R^{\mathrm{px}}, D_R(k), \gamma).$$

Summing yields
$$P(D_L(k), d - 1) + P(D_R(k), d - 1) \leq \mathrm{Obj}(f^{\mathrm{px}}, D, \gamma) = P(D, d).$$

Since $j^{\star}$ minimizes the proxy-completed split score,

$$P(D_L(j^{\star}), d - 1) + P(D_R(j^{\star}), d - 1) \leq P(D_L(k), d - 1) + P(D_R(k), d - 1) \leq P(D, d) \leq \varepsilon_{\mathrm{abs}}.$$

Therefore PRAXIS does not prune split $j^{\star}$, because its pruning condition is

$$P(D_L, d - 1) + P(D_R, d - 1) > \varepsilon_{\mathrm{abs}}.$$

**Budget passed to children.**

PRAXIS initializes the left budget by subtracting the proxy value on the right:

$$\varepsilon_L = \varepsilon_{\text{abs}} - P(D_R, d-1).$$

Using

$$P(D_L, d-1) + P(D_R, d-1) \le \varepsilon_{\text{abs}},$$

we obtain

$$\varepsilon_L \ge P(D_L, d-1).$$

Thus, PRAXIS is invoked on $(D_L, d-1)$ with a budget at least as large as the proxy value at the subproblem. Therefore, by the inductive hypothesis, the graph returned for the left child contains $f^{\text{pw}}(D_L, d-1)$. Moreover,

$$G_L.\text{min\_objective} \le \text{Obj}(f^{\text{pw}}(D_L, d-1), D_L, \gamma) \le P(D_L, d-1).$$

The right budget is then set to

$$\varepsilon_R = \varepsilon_{\text{abs}} - G_L.\text{min\_objective} \ge \varepsilon_{\text{abs}} - P(D_L, d-1) \ge P(D_R, d-1).$$

Thus the recursive call on $(D_R, d-1)$ also receives a budget that is at least its proxy value, and by induction, the graph contains $f^{\text{pw}}(D_R, d-1)$.

Since PRAXIS attaches these child graphs under split $j$, the graph for $(D, d)$ contains $f^{\text{pw}}(D, d)$. We note that by Theorem A.13, $f^{\text{pw}}(D, d)$ will always fit within the budget, since the proxy tree itself–which is no better–always fits within the budget of an explored subproblem (Theorem A.3).

Importantly, we note that attaching these left and right PROXYWRAPPER subtrees (corresponding to the proxy rollout after selecting split $j$) yields the PROXYWRAPPER tree for $(D, d)$. This follows because PROXYWRAPPER greedily selects the best split according to the proxy algorithm and does not revisit earlier decisions (by Proposition A.14, rerunning PROXYWRAPPER on any subproblem appearing in a PROXYWRAPPER tree returns the same subtree).

**Iterative refinement.**

The sibling procedure only increases budgets supplied to recursive calls. Thus, the above argument continues to hold throughout refinement.

$\square$

Following very similar logic to Corollary A.4, we show that this fact does not just hold for the root node, but the hypothesis is true for all nodes in the AND/OR graph.

**Corollary A.16** (The tree corresponding to the pilot algorithm is recovered at every explored subproblem)**.** *Assume the root call to PRAXIS satisfies*

$$\varepsilon_{\text{abs}} \ge P(D, d).$$

*Then every OR node in the returned AND/OR graph is explored with budget*

$$\varepsilon'_{\text{abs}} \ge P(D', d').$$

*Consequently, by Theorem A.15, the subgraph rooted at every OR node contains the* PROXYWRAPPER *tree for that subproblem, and the minimum objective stored at that node is at most the objective of that* PROXYWRAPPER *tree, which is itself at most* $P(D', d')$.

*Proof.* Let a split produce child subproblems with proxy values

$$P_L := P(D_L, d-1), \qquad P_R := P(D_R, d-1),$$

and suppose this split is not pruned, i.e.

$$P_L + P_R \leq \varepsilon_{\text{abs}}.$$

Then the initial left budget used by SOLVESIBLINGS is

$$\varepsilon_L := \varepsilon_{\text{abs}} - P_R.$$

Therefore,

$$\varepsilon_L = \varepsilon_{\text{abs}} - P_R \geq P_L.$$

Thus, the recursive call on the left subproblem is made with budget at least its proxy objective, which by Theorem A.15 is sufficient to recover the PROXYWRAPPER tree on the left.

Now let $G_L$ denote the returned left subgraph. Since the left PROXYWRAPPER tree is contained in $G_L$, its minimum objective satisfies

$$G_L.\text{min\_objective} \leq P_L.$$

The right budget is then set to

$$\varepsilon_R := \varepsilon_{\text{abs}} - G_L.\text{min\_objective}.$$

Using $G_L.\text{min\_objective} \leq P_L$, we obtain

$$\varepsilon_R = \varepsilon_{\text{abs}} - G_L.\text{min\_objective} \geq \varepsilon_{\text{abs}} - P_L \geq P_R,$$

since $P_L + P_R \leq \varepsilon_{\text{abs}}$. Hence, the recursive call on the right subproblem is also made with budget at least its proxy objective, so the PROXYWRAPPER tree on the right is recovered as well.

Finally, SOLVESIBLINGS only increases budgets during subsequent refinement steps (by Lemma A.2). By Proposition A.1, no trees will ever be removed from the AND/OR graph during this budget refinement, so the hypothesis holds for the final attached subgraph. Repeated application of this argument down the AND/OR graph establishes the corollary. □

With Theorem A.13 and Theorem A.15 in place, we note how conditioning on the PROXYWRAPPER objective in addition to the PROXY objective changes earlier theoretical results. For Theorem 3.5, because we are guaranteed that the left and right AND/OR graph solutions will contain the PROXYWRAPPER tree, then when we resolve the other side in Algorithm 3, we subtract off at least PROXYWRAPPER on the other side (as opposed to PROXY). Thus, while the true recovery condition is (for each internal node $u$ of the desired tree)

$$\sum_{i=1}^{t} \text{MinObj}(s_i) \;+\; \text{PROXY}(u_{\text{left}}) \;+\; \text{PROXY}(u_{\text{right}}) \;\leq\; \varepsilon_{\text{abs}},$$

We can give a tighter sufficient condition than was given in Algorithm 3.5.

$$\sum_{i=1}^{t} \text{PROXYWRAPPER}(s_i) \;+\; \text{PROXY}(u_{\text{left}}) \;+\; \text{PROXY}(u_{\text{right}}) \;\leq\; \varepsilon_{\text{abs}},$$

Likewise, for Lemma A.7, we know that the first minimum objective query will already be PROXYWRAPPER, as opposed to simply PROXY.

## B. Implementation, Data Structures, and Caching

### B.1. Integer Objectives

Throughout this work, we minimize a regularized empirical risk objective of the form

$$\text{Obj}(f, D, \gamma) \; = \; \gamma \, |f| \; + \; \sum_{i=1}^{|D|} \mathbf{1}\{f(x_i) \neq y_i\}, \tag{48}$$

This is equivalent to minimizing the normalized objective (with $\gamma = \lambda |D|$)

$$\text{Obj}_{\text{norm}}(f, D, \lambda) \; = \; \lambda \, |f| \; + \; \frac{1}{|D|} \sum_{i=1}^{|D|} \mathbf{1}\{f(x_i) \neq y_i\},$$

For algorithmic stability and efficiency, we work exclusively with an equivalent integer-valued objective obtained by requiring $\gamma$ to be an integer (equivalently, $\lambda$ to be an integer multiple of $\frac{1}{|D|}$). As with $\lambda$, we require it to be non-negative.

This mild assumption eliminates floating-point drift, which can become problematic in extracting trees with some objective and decomposing it into an exact sum of a left and right child. Moreover, integer objectives allow additional storage in the AND/OR graph (like a histogram of objectives at each subproblem) to be stored more efficiently as there are fewer unique values, and for quicker theoretical convergence of iterative budget refinement (see Lemma A.7). Beyond these structural advantages, integer arithmetic is faster.

One advantage to defining objectives in this form is that we can decompose the objective via a simple sum.

$$\text{Obj}_{\text{int}}(f, D, \gamma) \; = \; \text{Obj}_{\text{int}}(f_L, D_L, \gamma) \; + \; \text{Obj}_{\text{int}}(f_R, D_R, \gamma). \tag{49}$$

One consequence of (49) is that the regularization parameter $\gamma$ is unchanged under recursive decomposition. As such, it therefore behaves as a global constant, and we omit it from function arguments when it is unambiguous.

If a user wishes to run our method with $\gamma$ values mapping to some $\lambda$ that is not an integer multiple of $\frac{1}{|D|}$, one can extend the algorithms from this work directly to accommodate non-integer $\gamma$. Another option is to scale up the objective, i.e.

$$\text{Obj}(f, D, \gamma) \; = \; \gamma \, |f| \; + \; C \times \sum_{i=1}^{|D|} \mathbf{1}\{f(x_i) \neq y_i\},$$

for some integer $C$, allowing a more precise match. We also observe that rounding $\lambda$ to the nearest multiple of $\frac{1}{|D|}$, for reasonably sized datasets, should be of minor consequence.

We rigorously formalize the relation between the rounded and unrounded $\gamma$ below.

**Effect of snapping $\lambda$.** $\lambda \in \mathbb{R}_{\geq 0}$. Let $\tilde{\lambda}$ be the nearest multiple of $\frac{1}{|D|}$, i.e. $\tilde{\lambda} = \frac{1}{|D|}\text{round}(\lambda|D|)$. Then

$$|\tilde{\lambda} - \lambda| \leq \frac{1}{2|D|}. \tag{50}$$

**Effect on tree objectives.** For any tree with $L$ leaves,

$$\left|\text{Obj}_{\text{norm}}(f, D, \tilde{\lambda}) - \text{Obj}_{\text{norm}}(f, D, \lambda)\right| = |\tilde{\lambda} - \lambda| \, L \; \leq \; \frac{L}{2|D|} \leq \frac{2^{d-1}}{|D|}. \tag{51}$$

**Enumerating $\lambda$ using the Rashomon Set of $\tilde{\lambda}$.** By (51), we can add a small amount of additive slack ($\frac{2^{d-1}}{|D|}$) to the Rashomon bound and recover the Rashomon set defined by $\lambda$ by solving with $\tilde{\lambda}$.

**Scaling objectives to be lossless.** Suppose $\lambda$ is specified to $p$ decimal places (commonly $p \leq 4$, as $\lambda$ is frequently chosen to be in $\{0.04, 0.02, 0.015, 0.01, 0.0075, 0.005, 0.0025, 0.001, 0\}$). Then we can write it as $\lambda = a/10^p$ for some integer $a$, so $\gamma = \lambda|D| = \frac{a}{10^p}|D| = \frac{a|D|}{10^p}$. If we scale by $\frac{10^p}{gcd(a|D|,10^p)} \leq 10^p$, the scaled objective is integer, and thus there is no loss in the integerization.

## B.2. AND/OR Graph Representation

Much like prior work (Arslan et al., 2026; Xin et al., 2022), we build out an AND/OR graph that sufficiently explores the search space to embed the Rashomon set in the graph. In this section, we provide information about this AND/OR graph, which encodes all of the information needed to materialize trees in the Rashomon set.

**Subproblems and choices.** At any point during the execution of PRAXIS, the algorithm operates on a subproblem, defined by a dataset $D$, a remaining depth $d$, and a remaining objective budget (unlike the proxy algorithm, whose subproblems do not depend on the budget). For a fixed subproblem, there are multiple possible ways to construct a tree within the budget: the algorithm may choose a leaf prediction, or it may choose to split on one of several features and recursively construct trees for the left and right children.

**OR nodes (choices at a subproblem):** We represent each subproblem by an *OR node*. Each OR node stores a collection of alternative choices, any one of which yields a valid tree for that subproblem. These choices consist of:

- **Leaf choices**, corresponding to predicting a constant label $b \in \{0, 1\}$; and

- **Split choices**, corresponding to splitting on a feature $j$, which can be used to recursively choose trees within that lowered budget for the left and right child subproblems. The AND structure arises because, once a split is chosen, both child subproblems must be recursively handled for that split.

**AND nodes (joint requirements of a split):** A split choice introduces an *AND node*: choosing a split on feature $j$ requires both a valid left subtree and a valid right subtree. Thus, a split is feasible if and only if feasible solutions exist for both children within the remaining budget.

**Graph structure:** The resulting structure is an AND/OR graph:

- OR nodes represent subproblems and store alternative choices;

- split choices introduce AND structure by linking to two child OR nodes;

- leaf choices terminate the recursion.

A decision tree corresponds to selecting exactly one outgoing choice at each OR node and, for every split choice, recursively selecting choices in both children.

**Feasibility of combinations:** For a given OR node, not every possible combination of left and right subtrees necessarily yields a feasible tree under the budget. However, our algorithms ensures a useful invariant for this representation: for every split choice stored at an OR node, *each* feasible subtree on one side of the split can be paired with *at least one* feasible subtree on the other side to form a valid tree within budget. Otherwise, that split would have been pruned by the proxy test and never added to the AND/OR graph.

**Incremental construction:** Algorithms 8 and 9 incrementally construct this AND/OR graph during the execution of PRAXIS. Unlike prior work, our AND/OR graph is significantly smaller, because the proxy algorithms have already pruned a substantial portion of the search space. A subproblem is included in our AND/OR graph if and only if it is used in some tree in the Rashomon set $R_{\varepsilon_{\text{abs}}}(\mathcal{T}, D)$. Algorithms 8 and 9 build out this AND/OR graph as we run PRAXIS. PRAXIS returns an AND/OR graph by attaching smaller AND/OR graphs using these methods. Given that PRAXIS at a subproblem is considering a split, the multipass framework in Algorithm 3 recurses on the children subproblems with a refined budget. After iterative budget refinement, we attach the AND/OR graph from the two children to the parent with Algorithm 9. Analogously, if we can fit a leaf within the budget, we add it with Algorithm 8.

---

**Algorithm 8** ADDLEAF($G, b, C_b$)

---

**Require:** ORNODE $G$, leaf prediction $b \in \{0, 1\}$, leaf objective $C_b$
1: $G.leaves \leftarrow G.leaves \cup \{(b, C_b)\}$
2: **if** $C_b < G.min\_objective$ **then**
3:     $G.min\_objective \leftarrow C_b$
4: **end if**

---

---

**Algorithm 9** ADDSPLIT($N, j, G_L, G_R$)

---

**Require:** ORNODE $G$, feature $j$, ORNODES $G_L, G_R$
1: $G.splits \leftarrow G.splits \cup \{(j, G_L, G_R)\}$
2: $m_{\text{sum}} \leftarrow G_L.min\_objective + G_R.min\_objective$
3: **if** $m_{\text{sum}} < G.min\_objective$ **then**
4:     $G.min\_objective \leftarrow m_{\text{sum}}$
5: **end if**

---

One crucial value that these methods store in each node is the $min\_objective$. This is used in Algorithm 3 to decrement budgets according to the best solution that was found.

These pieces of information are sufficient to construct the full AND/OR graph, which represents our Rashomon set approximation. However, to efficiently extract trees one at a time (without materializing the entire Rashomon set), we require additional information—specifically, a list or histogram of objectives at each subproblem. Algorithm B.3 describes this process. We do not store this information during the execution of PRAXIS, because iterative budget refinement may rebuild the AND/OR graph multiple times per subproblem. Instead, we defer this work to a single postprocessing step, as it is not needed during the algorithm's execution.

### B.3. Postprocessing Algorithms

To efficiently extract trees from the AND/OR graph, we must know how many trees on one side of a split can be paired with trees on the other side, so that we can build a one-to-one map from a tree index at the root to an actual tree in the Rashomon set. We construct this map to be monotonic, in the sense that a smaller index maps to a tree with no larger objective value (i.e., we can index into the trees in sorted order and output them one at a time). Algorithm 10 shows the histogram implementation we use: a list of (objective, count) pairs, maintained in increasing order of objective.

Algorithm 12 builds histograms of tree objectives that can be achieved within the subproblem budget. At a given subproblem (starting at the root), it recursively processes the left and right subproblems of every split node and adds any leaves to the histogram of the current subproblem. Note that we do not check whether leaves fit within the budget, as this check was already performed during the execution of PRAXIS.

Once histograms have been computed for all descendant subproblems that appear in the Rashomon set $R_{\varepsilon_{\text{abs}}}(\mathcal{T}, D)$, and leaf contributions have been added to the current histogram, the remaining step is to propagate histogram information upward through split nodes. Algorithm 11 performs a filtered Cartesian product of the histograms from the left and right children of a split node. Because objectives are additive, we sum objective values and multiply their multiplicities to obtain the total number of trees with that objective arising from the split.

The resulting (objective, count) pairs are then merged into the parent histogram. This procedure is repeated for all splits until all information has been fully propagated to the root (and then we have the distribution over objectives for the entire Rashomon set).

With histograms of objectives available at each subproblem, we can efficiently extract trees in sorted order. This procedure follows a similar indexing scheme to that of Xin et al. (2022), except that we explicitly guarantee a sorted order (Arslan et al., 2026). The high-level idea is to convert each histogram into a cumulative count of trees. Given a request for the $x$-th tree, we first consult the root histogram to determine the corresponding objective value and the index of the tree within that objective bucket. We then implement a procedure to retrieve the $y$-th tree with objective $z$.

There are many possible ways to break ties among trees with the same objective. We use a deterministic scheme without any

---

**Algorithm 10** ADDHIST($N, obj, add\_cnt$)

---

**Require:** ORNODE $N$, objective value $obj$, increment $add\_cnt \in \mathbb{N}$
1: Find the position of $obj$ in $N.hist$, keeping the list sorted by objective
2: **if** $obj$ already appears in $N.hist$ **then**
3:     Increase its count by $add\_cnt$
4: **else**
5:     Insert a new entry $(obj, add\_cnt)$ in sorted order
6: **end if**

---

**Algorithm 11** ADDSPLITANDBUILD($N, j, L, R$)

---

**Require:** ORNODE node $N$ with budget $N.budget$, feature index $j$, left child ORNODE $L$, right child ORNODE $R$
1: Initialize a new split record $s$ with $s.feature \leftarrow j$, $s.left \leftarrow L$, $s.right \leftarrow R$
2: $sum\_counts \leftarrow$ empty map from objective $\rightarrow$ count
3: **for each** $(\ell_{\mathrm{obj}}, \ell_{\mathrm{cnt}})$ in $L.hist$ **do**
4:     $rem \leftarrow N.budget - \ell_{\mathrm{obj}}$
5:     **for each** $(r_{\mathrm{obj}}, r_{\mathrm{cnt}})$ in $R.hist$ with $r_{\mathrm{obj}} \leq rem$ **do**
6:         $tot \leftarrow \ell_{\mathrm{obj}} + r_{\mathrm{obj}}$
7:         $sum\_counts[tot] \leftarrow sum\_counts[tot] + \ell_{\mathrm{cnt}} \cdot r_{\mathrm{cnt}}$
8:     **end for**
9: **end for**
10: **if** $sum\_counts$ is not empty **then**
11:     Let $tmp$ be the list of pairs $(obj, cnt)$ from $sum\_counts$, sorted by $obj$
12:     Merge the sorted lists $N.hist$ and $tmp$ into a new sorted list, summing counts for equal objectives
13:     Replace $N.hist$ by this merged list
14: **end if**
15: $s.num\_valid\_trees \leftarrow \sum_{(obj,\ cnt) \in sum\_counts} cnt$
16: Append $s$ to $N.splits$

---

meaningful interpretation, although this could be replaced with a secondary criterion.

Tie-breaking is not the focus of our work; instead, our goal is to produce trees in sorted order, allowing users to choose whether to enumerate only the top $(1 + \varepsilon)$-factor of trees or to extract the additional trees returned by PRAXIS that are within the initial budget. Either is a reasonable choice, since Table 12 shows that PRAXIS approximates the set of trees within the initial budget almost as well as the true Rashomon set.

See subsection B.4 for further details on extracting the trees in sorted order.

### B.4. Tree Extraction in sorted order

Once histograms of objective values have been computed for every subproblem via Algorithm 12, we can extract individual trees from the AND/OR graph in monotone (nondecreasing) objective order without materializing the entire Rashomon set.

Algorithm 13 is a wrapper method for tree extraction, which calls the main recursive extraction method. Given an index $i$, it scans the root histogram in increasing order of objective value, identifying the smallest objective $z$ such that the cumulative number of trees with objective at most $z$ exceeds $i$. This determines both the target objective $z$ and the within-bucket index $k$ corresponding to the $i^{\mathrm{th}}$ tree.

Algorithm 14 then recursively materializes the $k^{\mathrm{th}}$ tree with objective exactly $z$ from the AND/OR graph.

At an ORNODE $N$, the procedure first enumerates any leaf options whose loss equals $z$ (which defines a deterministic tie-breaking scheme within an objective bucket). If the desired index is not exhausted by leaf solutions, the algorithm decrements $k$ and proceeds through the split records stored at $N$. For a split $(j, L, R)$, it computes how many trees under that split achieve objective $z$ by pairing objective values from $L.hist$ and $R.hist$ whose sums equal $z$. If the desired index falls within this split, the algorithm deterministically selects the corresponding pair of left and right subtrees and recurses on the children, constructing a prediction node with feature $j$. Otherwise, $k$ is further decremented to skip all trees induced by

---

**Algorithm 12** BUILDHISTOGRAMSPOST($N$)

---

**Require:** ORNODE $N$
 1: **if** $N.hist\_built$ is **true then**
 2:     **return**
 3: **end if**
 4: **for each** split $s$ in $N.splits$ **do**
 5:     BUILDHISTOGRAMSPOST($s.left$)
 6:     BUILDHISTOGRAMSPOST($s.right$)
 7: **end for**
 8: $saved\_splits \leftarrow N.splits$
 9: $N.splits \leftarrow \emptyset$
10: $N.hist \leftarrow \emptyset$
11: **for each** leaf $(b, loss)$ in $N.leaves$ **do**
12:     ADDHIST($N, loss, 1$)
13: **end for**
14: **for each** split $s$ in $saved\_splits$ **do**
15:     ADDSPLITANDBUILD($N, s.feature, s.left, s.right$)
16: **end for**
17: $N.hist\_built \leftarrow$ **true**

---

**Algorithm 13** GETITHTREE(root, $i$)

---

**Require:** Root ORNODE root, index $i \in \{0, 1, \dots\}$
 1: BUILDHISTOGRAMSPOST(root)
 2: $cum \leftarrow 0$
 3: **for each** $(obj, ct)$ in root.$hist$ **in increasing** $obj$ **do**
 4:     **if** $i < cum + ct$ **then**
 5:         $target\_obj \leftarrow obj$
 6:         $k \leftarrow i - cum$ {index within this objective bucket}
 7:         **return** GETKTHTREEWITHOBJECTIVE(root, $target\_obj, k$)
 8:     **end if**
 9:     $cum \leftarrow cum + ct$
10: **end for**

---

that split, continuing until the target position within the bucket is reached.

In Alorithm 14, MAKELEAF($b$) denotes a leaf predicting label $b$, and MAKESPLIT($j, T_L, T_R$) denotes the decision tree whose root splits on feature $j$, with left subtree $T_L$ and right subtree $T_R$.

---

**Algorithm 14** GETKTHTREEWITHOBJECTIVE($N, z, k$)

---

**Require:** ORNODE $N$, target objective $z$, index $k$ within objective-$z$ bucket

1: BUILDHISTOGRAMSPOST($N$)
   {(1) Leaf trees at $N$ (tie-breaker: leaves are enumerated before splits).}
2: **for each** leaf $(b, \ell)$ in $N.leaves$ **do**
3:    **if** $\ell = z$ **then**
4:      **if** $k = 0$ **then**
5:        **return** MAKELEAF($b$)
6:      **else**
7:        $k \leftarrow k - 1$
8:      **end if**
9:    **end if**
10: **end for**
   {(2) Split trees at $N$.}
11: **for each** split record $s$ in $N.splits$ **do**
12:    $L \leftarrow s.left, \quad R \leftarrow s.right$
13:    BUILDHISTOGRAMSPOST($L$);  BUILDHISTOGRAMSPOST($R$)
      {Compute how many trees under this split achieve objective $z$.}
14:    $total\_here \leftarrow 0$
15:    **for each** $(\ell_{obj}, \ell_{cnt})$ in $L.hist$ **do**
16:      $r_{obj} \leftarrow z - \ell_{obj}$
17:      **if** $r_{obj}$ appears in $R.hist$ with count $r_{cnt}$ **then**
18:        $total\_here \leftarrow total\_here + \ell_{cnt} \cdot r_{cnt}$
19:      **end if**
20:    **end for**
21:    **if** $k < total\_here$ **then**
22:      {The desired tree lies under split $s$; select the $(\ell_{obj}, r_{obj})$ pair and recurse.}
23:      $running \leftarrow 0$
24:      **for each** $(\ell_{obj}, \ell_{cnt})$ in $L.hist$ **do**
25:        $r_{obj} \leftarrow z - \ell_{obj}$
26:        **if** $r_{obj}$ appears in $R.hist$ with count $r_{cnt}$ **then**
27:          $pairs \leftarrow \ell_{cnt} \cdot r_{cnt}$
28:          **if** $running + pairs > k$ **then**
29:            $rel \leftarrow k - running$
30:            $left\_idx \leftarrow \lfloor rel/r_{cnt} \rfloor$
31:            $right\_idx \leftarrow rel \bmod r_{cnt}$
32:            $T_L \leftarrow$ GETKTHTREEWITHOBJECTIVE($L, \ell_{obj}, left\_idx$)
33:            $T_R \leftarrow$ GETKTHTREEWITHOBJECTIVE($R, r_{obj}, right\_idx$)
34:            **return** MAKESPLIT($s.feature, T_L, T_R$)
35:          **end if**
36:          $running \leftarrow running + pairs$
37:        **end if**
38:      **end for**
39:    **else**
40:      $k \leftarrow k - total\_here$ {skip all objective-$z$ trees from this split}
41:    **end if**
42: **end for**

---

## B.5. LicketySPLIT modifications

We use the LicketySPLIT algorithm from Babbar et al. (2025) as our default proxy algorithm in PRAXIS. However, we provide a multitude of implementation and algorithm changes that often allow PRAXIS to run much faster than even a single call of the implementation detailed in Babbar et al. (2025).

Below are the implementations of LicketySPLIT and the greedy tree methods that it calls that are faithful to Babbar et al. (2025). Though Babbar et al. (2025)'s implementation used a modified version of the GOSDT (Lin et al., 2020) algorithm, here we represent it in a pure form. We do not present the implementation as returning an explicit tree (although Babbar et al. (2025) does so). Instead, we return only the objective value of the corresponding tree, which is sufficient for our purposes.

LicketySPLIT (Algorithm 16) introduces a one-step lookahead relative to a greedy tree algorithm. Rather than selecting a feature using information gain alone, LICKETYSPLIT evaluates each candidate split by completing both children greedily and selecting the split that minimizes the resulting summed objective. After selecting this feature, the algorithm recurses using LICKETYSPLIT itself on the children.

---

**Algorithm 15** GREEDYTREE$(D', d, \gamma)$

---

**Require:** Subproblem dataset $D'$, remaining depth $d$, per-leaf penalty $\gamma$
1: Let $n' \leftarrow |D'|$, $p \leftarrow$ number of positive labels in $D'$
2: *leaf_loss* $\leftarrow \gamma + \min(p, n' - p)$
3: **if** $d = 0$ **or** *leaf_loss* $\leq 2\gamma$ **then**
4:     **return** *leaf_loss*
5: **end if**
6: Select feature $j$ maximizing information gain on $D'$
7: Partition $D'$ into $(D'_L, D'_R)$ using feature $j$
8: **if** $D'_L = \emptyset$ **or** $D'_R = \emptyset$ **then**
9:     **return** *leaf_loss*
10: **end if**
11: $L \leftarrow$ GREEDYTREE$(D'_L, d - 1, \gamma)$
12: $R \leftarrow$ GREEDYTREE$(D'_R, d - 1, \gamma)$
13: **return** $\min($*leaf_loss*$, L + R)$

---

---

**Algorithm 16** LICKETYSPLIT$(D', d, \gamma)$ (from Babbar et al. (2025))

---

**Require:** Subproblem dataset $D'$, remaining depth $d$, per-leaf penalty $\gamma$.
 1: Let $n' \leftarrow |D'|$, $p \leftarrow$ number of positive labels in $D'$
 2: *leaf_loss* $\leftarrow \gamma + \min(p,\, n' - p)$
 3: **if** $d = 0$ **or** *leaf_loss* $\leq 2\gamma$ **then**
 4:    **return** *leaf_loss*
 5: **end if**
 6: *best_sum* $\leftarrow +\infty$, *best_feature* $\leftarrow \perp$
 7: **for each** feature $j$ **do**
 8:    Partition $D'$ into $(D'_L, D'_R)$ using feature $j$
 9:    **if** $D'_L = \emptyset$ **or** $D'_R = \emptyset$ **then**
10:       **continue**
11:    **end if**
12:    $s_j \leftarrow$ GREEDYTREE$(D'_L, d - 1, \gamma) +$ GREEDYTREE$(D'_R, d - 1, \gamma)$
13:    **if** $s_j < best\_sum$ **then**
14:       *best_sum* $\leftarrow s_j$
15:       *best_feature* $\leftarrow j$
16:    **end if**
17: **end for**
18: **if** *best_feature* $= \perp$ **then**
19:    **return** *leaf_loss*
20: **end if**
21: **if** *best_sum* $\geq$ *leaf_loss* **then**
22:    **return** *leaf_loss*
23: **end if**
24: Partition $D'$ using *best_feature* into $(D'_L, D'_R)$
25: $L \leftarrow$ LICKETYSPLIT$(D'_L, d - 1, \gamma)$
26: $R \leftarrow$ LICKETYSPLIT$(D'_R, d - 1, \gamma)$
27: **return** $L + R$

---

Now, we present changes to the LicketySPLIT and its accompanying greedy subroutine in three categories: optimal solvers of a shallow depth, caching to guarantee speed-ups, and generalizations to interpolate between greedy and optimal.

We generalize LicketySPLIT via a hierarchical rollout scheme for split selection dictated by a lookahead parameter $\ell$. Taking $\ell = 1$ in our generalization recovers LicketySPLIT, and $\ell = 0$ is taken to mean the greedy tree algorithm.

While LicketySPLIT ($\ell = 1$) evaluates candidate splits using greedy completion ($\ell = 0$), our approach generalizes split selection by defining it recursively through a hierarchy of heuristics. For a given $\ell$, candidate splits are evaluated using completions generated by a lower-tier heuristic ($\ell - 1$); for instance, the proxy algorithm at $\ell = 2$ selects splits based on LicketySPLIT completions. This construction yields a hierarchical rollout scheme in which each heuristic rolls out a cheaper heuristic, ultimately terminating at greedy induction.

We explain the modifications to LicketySPLIT, fixing $\ell = 1$ for simplicity, as it is our preferred proxy algorithm.

---

**Algorithm 17** GREEDYTREE($D', d, \gamma$) (with shared depth-1 exact solver + caching)

---

**Require:** Subproblem dataset $D' \subseteq D$, remaining depth $d$, regularization $\gamma$

1: **if** $d = 0$ **then**
2:     $ans \leftarrow$ DEPTH_D_EXACT($D', 0, \gamma$)
3:     **return** $ans$
4: **end if**

5: **if** $d = 1$ **then**
6:     **return** DEPTH_D_EXACT($D', 1, \gamma$) {last step is optimal misclassification (shared with LicketySPLIT)}
7: **end if**

8: $k \leftarrow$ KEY($D'$) {subproblem identifier (64-bit fingerprint)}
9: **if** $k \in \mathcal{C}_{\text{greedy}}[d]$ **then**
10:     **return** $\mathcal{C}_{\text{greedy}}[d][k]$
11: **end if**

12: $leaf\_loss \leftarrow$ DEPTH_D_EXACT($D', 0, \gamma$)
13: **if** $leaf\_loss \leq 2\gamma$ **then**
14:     Optionally cache: $\mathcal{C}_{\text{greedy}}[d][k] \leftarrow leaf\_loss(D')$
15:     **return** $leaf\_loss$
16: **end if**
17: Choose feature $j^\star$ maximizing information gain on $D'$
18: Split $D'$ into $(D'_L, D'_R)$ by $j^\star$; if either side is empty, return $leaf\_loss$
19: $L \leftarrow$ GREEDYTREE($D'_L, d - 1, \gamma$)
20: $R \leftarrow$ GREEDYTREE($D'_R, d - 1, \gamma$)
21: $split\_loss \leftarrow L + R$
22: $ans \leftarrow \min(leaf\_loss, split\_loss)$

23: Cache: $\mathcal{C}_{\text{greedy}}[d][k] \leftarrow ans$
24: **return** $ans$

---

---

**Algorithm 18** LICKETYSPLIT($D', d, \gamma, \ell$) (low depth exact solvers + caching + clamped lookahead + generalization)

---

**Require:** Subproblem dataset $D' \subseteq D$, remaining depth $d$, lookahead $\ell \geq 0$

1: **if** $d = 0$ **then**
2:      $ans \leftarrow$ DEPTH_D_EXACT($D', 0$)
3:      **return** $ans$
4: **end if**

5: $\ell \leftarrow \min(\ell, d - 1)$ {clamp lookahead by remaining depth to allow more caching, $\ell = d - 1$ is optimal at depth d now}

6: **if** $\ell = d - 1$ **then**
7:      **return** DEPTH_D_EXACT($D', d$) {shared with GreedyTree}
8: **end if**
9: $k \leftarrow$ KEY($D'$)
10: **if** $k \in \mathcal{C}_{\text{lickety}}[d, \ell]$ **then**
11:      **return** $\mathcal{C}_{\text{lickety}}[d, \ell][k]$
12: **end if**

13: $leaf\_loss \leftarrow$ DEPTH_D_EXACT($D', 0, \gamma$)
14: **if** $leaf\_loss \leq 2\gamma$ **then**
15:      Optionally cache: $\mathcal{C}_{\text{depth0}}[k] \leftarrow leaf\_loss(D')$
16:      **return** $leaf\_loss$
17: **end if**
18: $best\_sum \leftarrow +\infty$, $best\_feature \leftarrow \bot$
19: **for each** feature $j$ **do**
20:      Split $D'$ into ($D'_L, D'_R$) by $j$; **continue** if either side is empty
21:      **if** $\ell = 1$ **then**
22:          $s_j \leftarrow$ GREEDYTREE($D'_L, d - 1, \gamma$) + GREEDYTREE($D'_R, d - 1, \gamma$) {one-step lookahead: greedy completion}
23:      **else**
24:          $s_j \leftarrow$ LICKETYSPLIT($D'_L, d - 1, \ell - 1, \gamma$) + LICKETYSPLIT($D'_R, d - 1, \ell - 1, \gamma$) {general $\ell$: recurse with $\ell - 1$ during split evaluation}
25:      **end if**
26:      **if** $s_j < best\_sum$ **then**
27:          $best\_sum \leftarrow s_j$, $best\_feature \leftarrow j$
28:      **end if**
29: **end for**
30: $ans \leftarrow leaf\_loss$
31: **if** $best\_feature \neq \bot$ **then**
32:      Split $D'$ by $best\_feature$ into ($D'_L, D'_R$)
33:      $L \leftarrow$ LICKETYSPLIT($D'_L, d - 1, \ell, \gamma$) {recurse with constant $\ell$ in recursion}
34:      $R \leftarrow$ LICKETYSPLIT($D'_R, d - 1, \ell, \gamma$)
35:      $ans \leftarrow \min(ans, L + R)$
36: **end if**

37: Cache: $\mathcal{C}_{\text{lickety}}[d, \ell][k] \leftarrow ans$
38: **return** $ans$

---

---

**Algorithm 19** DEPTH_D_EXACT($D', d, \gamma$) (optimal depth-$d$ solver, shared for $d = 1$), (all new)

---

**Require:** Subproblem dataset $D' \subseteq D$, remaining depth $d$, regularization $\gamma$
**Ensure:** OPT($D', d$), the minimum integer objective among all trees of depth at most $d$ on $D'$
1: $k \leftarrow \text{KEY}(D')$
2: **if** $(d, k) \in \mathcal{C}_{\text{opt}}$ **then**
3:     **return** $\mathcal{C}_{\text{opt}}[d][k]$
4: **end if**
5: $n' \leftarrow |D'|$, $p \leftarrow$ number of positive labels in $D'$
6: *leaf_loss* $\leftarrow \gamma + \min(p, n' - p)$
7: **if** $d = 0$ **then**
8:     Optionally cache: $\mathcal{C}_{\text{depth0}}[k] \leftarrow$ *leaf_loss*
9:     **return** *leaf_loss*
10: **end if**
11: *best* $\leftarrow$ *leaf_loss*
12: **for each** feature $j$ **do**
13:     Split $D'$ into $(D'_L, D'_R)$ by $j$; **continue** if either side is empty
14:     $s_j \leftarrow$ DEPTH_D_EXACT($D'_L, d-1$) + DEPTH_D_EXACT($D'_R, d-1$)
15:     *best* $\leftarrow \min($*best*$, s_j)$
16: **end for**
17: Cache: $\mathcal{C}_{\text{opt}}[d][k] \leftarrow$ *best*
18: **return** *best*

---

**Optimal solvers at shallow depth.** The greedy tree solver was previously not optimal at depth 1 (the best split for information gain is not necessarily the one that minimizes misclassification error). Now, with an optimal depth 1 solver, greedy is optimal at depth 1, and LicketySPLIT is optimal at depth 2 without any additional time complexity (and in practice, computing additive objectives is cheaper than information gain).

Because we delegate solving shallow subproblems to a custom subroutine, this allows the proxy algorithm LicketySPLIT, and the subroutine of it, the greedy tree algorithm, to share caching. For instance, if they were both called at depth 1, they now use the same cache. Additionally, when the lookahead parameter $\ell \geq d - 1$, it would yield an optimal proxy algorithm even with $\ell = d - 1$, so we clamp it to share more caching. We elaborate more on caching beyond the shared optimal solvers in subsection B.6.

**Further exploration of search space:** In the original implementation of LICKETYSPLIT (see lines 21–23 of Algorithm 16; also the condition in line 1 of Algorithm 3 of (Babbar et al., 2025)), the algorithm includes an early stopping condition: if the best greedy tree completion over all candidate splits is no better than a leaf, the algorithm terminates and returns the leaf loss for the subproblem. In our implementation, we remove this early exit. While this change appears trivial, the early stopping condition is implicitly required in the implementation of Babbar et al. (2025) as a consequence of their use of GOSDT to implement the algorithm. This modification allows the algorithm to potentially find better trees (and never a worse tree as we take the minimum over it and the leaf loss) at no additional asymptotic cost.

**Caching and shared subproblem reuse within LicketySPLIT** We use a comprehensive caching strategy within LicketySPLIT: we cache every subproblem that it or its subroutine (a greedy tree algorithm) solves to report the proxy solution. This will be helpful within the larger PRAXIS algorithm, but we first observe that the LicketySPLIT algorithm benefits from caching greedy recursive subproblems as it itself recurses.

**Theorem B.1** (Caching greedy solutions at subproblems gives provable cache reuse). *Run* LICKETYSPLIT($D, d, \ell = 1$) *with remaining depth $d \geq 1$. Assume that no early stopping occurs for* LICKETYSPLIT *and its greedy subroutine (i.e. there is always a split that partitions the data and that $2\lambda <$ leaf_objective at all depths $d > 0$). Then, the total number of greedy-cache hits incurred during the execution is at least $(2^{d+1} - 4)$.*

*Proof.* Under the stated assumption that no early stopping occurs, LICKETYSPLIT($D, d, 1$) always selects a non-degenerate split at each node with positive remaining depth. Thus, the tree returned by LICKETYSPLIT is a full binary tree of depth $d$.

Equivalently, the recursion tree of the execution is a full binary tree of depth $d$. Additionally, as no early stopping occurs, the greedy tree algorithm will also return a full binary tree of depth $d$ at every subproblem with remaining depth $d$.

Now fix any non-root internal node $u$ of this recursion tree, with induced subproblem dataset $D_u$ and remaining depth $r \geq 1$ at $u$. Let $p$ be the parent of $u$. In the call at $p$, LICKETYSPLIT evaluates candidate splits by calling its greedy subroutine on the child subproblems. In particular, the candidate split at $p$ that eventually leads to $u$ necessarily triggered a greedy call on $D_u$ with remaining depth $r$. Therefore, the value for this greedy call is present in the greedy cache before the algorithm begins processing node $u$ itself. However, it isn't the greedy cache for this subproblem that matters – it is the fact that left and right greedy solutions are cached for some split at $u$ (because the greedy continues; there is no early stopping).

When LICKETYSPLIT processes node $u$, it again iterates over candidate splits and, for each split, requests two greedy completions: one for the left child subproblem and one for the right child subproblem. Consider the specific feature $j^\star$ chosen by the cached greedy run on $D_u$. By definition of the greedy recursion, that greedy call computed and cached the two recursive greedy subcalls on the children induced by $j^\star$.

When LICKETYSPLIT later evaluates feature $j^\star$ among its candidate splits at node $u$, it requests exactly those two greedy child values. Both are cache hits. Thus, every non-root internal node contributes at least two greedy-cache reuses.

It remains to count the non-root internal nodes. A full binary tree of depth $d$ has $2^d - 1$ internal nodes, including the root. Hence it has $2^d - 2$ non-root internal nodes. Since each non-root internal node contributes at least two greedy-cache hits, the total number of greedy-cache hits incurred during the execution is at least

$$2(2^d - 2) = 2^{d+1} - 4.$$

$\square$

Though we do not prove it here, we note that the proof for $\ell \geq 2$ is essentially identical. Calling LicketySPLIT($\ell = 2$) recursively calls LicketySPLIT($\ell = 1$), and $\Omega(2^d)$ of those calls will be reused in the execution of LicketySPLIT($\ell = 2$). This is in addition to the greedy reuse within each LicketySPLIT($\ell = 1$) call.

Beyond caching within the proxy algorithm (which was strictly an improvement to LicketySPLIT), this caching can also help reuse work in the recursive calls of PRAXIS as it explores subproblems within the budget $\varepsilon_{abs}$. This does not hold for every proxy algorithm, but because LicketySPLIT($\ell$) recurses with the same parameters (except decreasing the depth budget), we can reuse the work it did if PRAXIS explores its initial split. We discuss this more and quantify the benefits saved in subsection B.6.

### B.6. Proxy Algorithm Choice for Efficient Caching

In Algorithm 18, we defined a generalization of LicketySPLIT with a lookahead parameter $\ell$, with $\ell = 1$ yielding the behavior of our modified LicketySPLIT algorithm. LicketySPLIT($\ell = 1$) repeatedly selects the split whose greedy completion yields the lowest objective value. LicketySPLIT($\ell \geq 2$) repeatedly selects the split whose LicketySPLIT($\ell - 1$) completion yields the lowest objective value. Combined with $\ell = 0$ (which we define to be greedy), this allows us to interpolate between linear time proxy algorithms and optimal ones.

We choose LICKETYSPLIT for its favorable time–accuracy trade-off, further improving its runtime via caching and its accuracy via optimal solvers at shallow depths (the improvements are detailed in subsection B.5). We showed in Theorem B.1 that our modified version of LicketySPLIT allows for substantial caching within a single call of LicketySPLIT. Importantly, this proxy also exhibits a useful structure where each subtree of a LicketySPLIT tree corresponds to a LicketySPLIT call on the corresponding data subset. This property, which also holds for all values of $\ell$, allows for caching across LicketySPLIT calls in PRAXIS .

**Interaction with PRAXIS.**    Consider an execution of PRAXIS at some subproblem $(D, d)$. Suppose a split is not pruned, which occurs whenever the proxy completions for its children satisfy the budget constraint. To evaluate that split, PRAXIS invokes the proxy on both child subproblems $(D_L, d - 1)$ and $(D_R, d - 1)$.

After budget refinement, PRAXIS recurses on these child subproblems. At such a child, PRAXIS again evaluates candidate splits by invoking the proxy. One of these candidate splits corresponds exactly to the split chosen by the proxy when it was originally applied to that child subproblem. Thus, PRAXIS requests a proxy completion for a subproblem that the proxy

itself has already constructed internally. However, calling the proxy algorithm on these subproblems is not guaranteed to solve the same problem.

For example, consider SPLIT with lookahead depth 2 from Babbar et al. (2025) without any postprocessing. In this configuration, SPLIT chooses the optimal first two splits, conditioned on the tree being completed greedily after (and does not alter the greedy completion). If SPLIT is called again on a child after taking the first split of the SPLIT tree, it solves a slightly different problem. As a consequence, the SPLIT call on the two children is not fully cached (though one could imagine that many of the greedy calls would be).

To save fully on caching, we need the proxy algorithm, when implemented recursively, to recurse with the same parameters, just one depth lower (that is, it recurses with exactly the parameter set one would use to call the algorithm). This holds for LICKETYSPLIT($\ell$): although split selection depends on $\ell - 1$, recursive calls are made with $\ell$ unchanged. This behavior corresponds precisely to the equality case of the refinement property in Definition 3.1.

**Role of the refinement property.**    The refinement condition

$$
\begin{aligned}
\text{PROXY}(D_L, \gamma, d-1) &\leq \text{Obj}(f_L, \gamma, D_L) \\
\wedge \quad \text{PROXY}(D_R, \gamma, d-1) &\leq \text{Obj}(f_R, \gamma, D_R)
\end{aligned}
\tag{52}
$$

ensures that when the proxy is reapplied to a subproblem induced by its own tree, the resulting objective does not worsen. If equality holds, the subtree is preserved; if the inequality is strict, the subtree is refined to a better one. In either case, Theorem A.3 shows that all invariants of PRAXIS are maintained, including feasibility and monotonicity of minimum objectives. However, we prefer the equality case for additional caching benefits.

**Failure without refinement.**    If a proxy algorithm violates the refinement condition:

$$
\neg(\text{PROXY}(D_L, \gamma, d-1) \leq \text{Obj}(f_L, \gamma, D_L) \ \wedge \ \text{PROXY}(D_R, \gamma, d-1) \leq \text{Obj}(f_R, \gamma, D_R)) \,.
\tag{53}
$$

i.e., if it is possible that reapplying the proxy to a subtree worsens its objective, then key invariants of PRAXIS may fail. In particular, a split chosen by the proxy at a parent node may later be pruned because the sum of proxy objectives conditioned on that split exceeds the allowed budget. This breaks the invariant that the minimum objective returned by each recursive call of PRAXIS is at least as good as the proxy objective for that subproblem, as PRAXIS isn't guaranteed to recover the proxy algorithm.

**Giving other decision tree algorithms the refinement property via caching.**    Any decision tree algorithm that lacks the refinement property can be modified to satisfy it by caching all of its subtree solutions that it implicitly constructs in yielding the top-level objective. For each subproblem $(D, d)$, one would maintain a set of candidate completions produced during execution—both from direct calls to the algorithm and from subtrees constructed as part of larger trees. The proxy value is then defined as the minimum objective over this set:

$$
\text{PROXY}(D, \lambda, d) \ := \ \min\{\text{Obj}(f, D) : f \text{ constructed for } (D, d)\}.
$$

With this modification, reapplying the proxy can only preserve or improve the subtree objectives (because it always minimizes over a set that includes the old subtree), ensuring the refinement property, and restoring all invariants required by PRAXIS .

**Comparing proxy algorithms with and without these caching benefits**    In Table 3, we compare the resource usage of PRAXIS when using two different proxy algorithms with the same asymptotic time complexity, $\mathcal{O}(nk^3d^3)$.

The first proxy is our generalization of LicketySPLIT, characterized by a lookahead parameter $\ell = 2$. This proxy recursively selects the best split using LicketySPLIT completions, where each completion is itself obtained by choosing the best split according to greedy completions.

We compare this to a second decision tree algorithm, which we refer to as BlockSPLIT (we define this algorithm to make a direct comparison). BlockSPLIT is obtained by recursively applying the SPLIT procedure of Babbar et al. (2025) without

*Table 3.* Runtime and peak memory comparison between PRAXIS with $\ell=2$ and a proxy algorithm of the same time complexity without caching benefits. $\lambda = 0.01$, $\varepsilon = 0.01$, depth $= 5$. *Time (s) seconds and Peak Memory (MB) are reported for the entire script.*

| | | | PRAXIS ($\ell = 2$) | | PRAXIS with less cache friendly proxy | |
|---|---|---|---|---|---|---|
| Dataset | $n$ | $k$ | Time | Peak MB | Time | Peak MB |
| Adult | 48,842 | 209 | 2,393.00 | 5,026.95 | 4,730.00 | 21,316.95 |
| Bank | 45,211 | 217 | 962.20 | 2,061.88 | 3,130.70 | 19,391.88 |
| Bike | 17,379 | 164 | 478.80 | 2,760.59 | 1,113.30 | 14,300.59 |
| Christine | 5,418 | 231 | 9,946.00 | 139,833.82 | – | – |
| Churn | 5,000 | 472 | 13,645.90 | 76,252.34 | – | – |
| Covertype | 581,012 | 96 | 1,979.60 | 1,301.23 | 2,760.50 | 1,669.83 |
| Credit | 30,000 | 225 | 1,401.40 | 3,826.30 | 5,158.40 | 39,868.30 |
| Diabetes | 253,680 | 121 | 945.40 | 1,037.27 | 2,604.20 | 5,166.67 |
| Electricity | 38,474 | 264 | 18,210.60 | 45,175.44 | 22,793.00 | 80,906.44 |
| Helena | 65,196 | 156 | 540.50 | 1,913.96 | 1,090.60 | 7,849.96 |
| Higgs | 11,000,000 | 84 | 54,014.00 | 21,537.79 | 110,377.20 | 21,537.79 |
| Jannis | 57,580 | 247 | 43,428.00 | 150,035.83 | – | – |
| Jasmine | 2,984 | 207 | 257.00 | 4,314.29 | 1,211.70 | 30,794.29 |
| Madeline | 3,140 | 76 | 27.30 | 988.02 | 54.54 | 2,019.72 |
| Madelon | 2,000 | 186 | 929.70 | 16,384.99 | 3,809.50 | 117,292.99 |
| Magic | 19,020 | 167 | 2,349.60 | 16,237.19 | 3,032.50 | 28,859.19 |
| News | 39,644 | 196 | 7,942.00 | 20,999.32 | – | – |
| Poker | 1,025,010 | 40 | 1,129.40 | 952.10 | 1,136.10 | 952.10 |
| Shopping | 12,330 | 243 | 2,058.60 | 14,315.35 | 5,081.00 | 73,797.35 |

postprocessing, using a block size (lookahead depth) of 2. Concretely, BlockSPLIT chooses the first two splits conditioned on greedy completions, then chooses the next two splits conditioned on greedy completions, and so on.

BlockSPLIT does not satisfy the equality case of the proxy algorithm refinement property. This is because it recurses with an internal bit that tracks whether there is one remaining split to choose in the current block, or whether the algorithm is restarting a new block of two splits. Furthermore, if one refines a tree produced by BlockSPLIT, the block optimization may be offset by one level relative to how the tree was originally constructed. In this case, calling the algorithm on a node of its own tree can produce a strictly worse solution than the subtree already used.

To address this, we define a wrapper that turns BlockSPLIT into a valid proxy algorithm. Instead of directly returning the objective produced by BlockSPLIT, the wrapper checks whether an objective for the same subproblem was previously computed with the internal bit in the opposite position. Since the algorithm may encounter the same subproblem in either state – once when entered externally (with the bit indicating two more splits must be done in the block) and once internally with the bit flipped – the wrapper returns the better of the two objectives.

The results in Table 3 show that, although both proxy algorithms have the same asymptotic time complexity, using $\ell = 2$ leads to substantially better runtime and memory usage. This is because $\ell = 2$ makes much better use of caching, as motivated earlier.

### B.7. Subproblem Representation

A subproblem in a decision tree search is most naturally identified by the subset of samples it contains (together with the remaining depth). This representation is used by Xin et al. (2022), who encode subproblems as bitvectors of length $n$, but this can be prohibitively memory-intensive for large datasets.

An alternative representation, supported by Arslan et al. (2026), identifies a subproblem by the set of feature splits (and branch directions) taken to reach it. While this representation does not grow with the sample size, distinct sequences of splits can induce the same subset of samples, causing identical subproblems to be treated as different and solved repeatedly.

This implementation is commonly implemented by storing a canonical sorted list of literals. Each literal encodes a feature index $i$ and branch direction as a 16-bit integer $2i + b$, where $b \in \{0, 1\}$ indicates whether the path takes the true or false branch. Sorting enforces a canonical order, since the conjunction of split conditions is commutative and permutations of splits induce the same subproblem.

This representation is more compact than an alternative encoding using two feature-sized bitvectors – one indicating which features are split on and one encoding branch directions–because the number of literals is bounded by the depth. For a depth budget of $d = 5$, the representation stores at most five 16-bit literals, requiring 80 bits per subproblem. This is more

| Dataset | Bitvector (Exact) | | | Literal (Itemset) | | | Hash (Us) | | |
|---|---|---|---|---|---|---|---|---|---|
| | Time (s) | Memory (MB) | Cache | Time (s) | Memory (MB) | Cache | Time (s) | Memory (MB) | Cache |
| Adult-209 | 358.7 | 15,964 | 2,620,799 | 603.0 | 1,117 | 6,458,004 | **339.1** | **393** | 2,620,799 |
| Bank-97 | 13.2 | 1,759 | 300,462 | 18.2 | 248 | 523,445 | **11.8** | **203** | 300,462 |
| Bank-217 | 191.9 | 11,709 | 2,001,930 | 327.4 | 794 | 4,275,476 | **183.6** | **352** | 2,001,930 |
| Bike-43 | 1.3 | 354 | 112,903 | 1.4 | 162 | 171,325 | **1.1** | **149** | 112,903 |
| Bike-164 | 171.3 | 9,507 | 4,504,463 | 292.9 | 1,500 | 10,262,367 | **157.0** | **441** | 4,504,463 |
| Chess-50 | 1.1 | 304 | 52,036 | 3.2 | 175 | 213,184 | **0.9** | **157** | 52,036 |
| Christine-80 | 31.7 | 2,848 | 3,376,062 | 29.9 | 694 | 4,085,162 | **25.6** | **373** | 3,376,062 |
| Christine-231 | 2,580.9 | 89,977 | 108,754,093 | 2,672.2 | 20,062 | 147,725,802 | **2,298.4** | **7,924** | 108,754,093 |
| Churn-81 | 2.0 | 347 | 302,935 | 5.9 | 293 | 1,200,148 | **1.5** | **154** | 302,935 |
| Churn-472 | 657.5 | 14,979 | 19,435,936 | 4,639.5 | 30,121 | 210,901,558 | **612.1** | **1,341** | 19,435,936 |
| Covertype-96 | 1,073.2 | 75,423 | 1,217,146 | 2,477.0 | 1,364 | 3,812,675 | **1,020.8** | **1,362** | 1,217,146 |

*Table 4.* Subproblem-key comparison at depth $d = 5$, $\lambda = 0.007$, and $\varepsilon = 0.02$. Peak memory is reported in MB. Cache is the total number of cached subproblems (cache size).

| Dataset | Ratios | | | | |
|---|---|---|---|---|---|
| | Literal/BV Cache | BV/Hash MB | Lit/Hash MB | BV/Hash Time | Lit/Hash Time |
| Adult-209 | 2.464 | 40.631 | 2.842 | 1.058 | 1.778 |
| Bank-97 | 1.742 | 8.646 | 1.217 | 1.117 | 1.543 |
| Bank-217 | 2.136 | 33.233 | 2.253 | 1.045 | 1.783 |
| Bike-43 | 1.517 | 2.379 | 1.089 | 1.211 | 1.312 |
| Bike-164 | 2.278 | 21.552 | 3.401 | 1.091 | 1.865 |
| Chess-50 | 4.096 | 1.938 | 1.120 | 1.146 | 3.499 |
| Christine-80 | 1.210 | 7.642 | 1.862 | 1.238 | 1.166 |
| Christine-231 | 1.358 | 11.354 | 2.532 | 1.123 | 1.162 |
| Churn-81 | 3.962 | 2.257 | 1.909 | **1.301** | 3.810 |
| Churn-472 | **10.847** | 11.163 | **22.451** | 1.074 | **7.579** |
| Covertype-96 | 3.132 | **55.376** | 1.001 | 1.051 | 2.426 |

*Table 5.* Ratio comparison between the Literal, Bitvector (BV), and Hash subproblem representations at depth $d = 5$, $\lambda = 0.007$, and $\varepsilon = 0.02$. The maximal ratio in each column is bolded.

space-efficient than the feature-bitvector encoding whenever the number of features exceeds 40, which is the case for nearly all of our datasets.

Our approach combines the advantages of both representations. We define subproblems canonically by the subset of samples they contain, but store this information implicitly via a 64-bit fingerprint computed from the corresponding bitvector (together with the remaining depth). This compression is possible because the dominant combinatorial growth in the search space comes from the number of features and the remaining depth, rather than from the number of samples.

The only trade-off is a vanishingly small probability of hash collisions. In practice, this probability is negligible. Moreover, even in the unlikely event of a collision, the correctness of the AND/OR graph is unaffected in the sense that any tree materialized from the graph will still be valid and lie within the specified budget. A collision may cause either additional trees to be explored or some trees to be missed, but it does not invalidate the feasibility of any returned tree. This is because we do not cache subgraphs, only proxy algorithm evaluations.

For our default proxy, a modified version of LICKETYSPLIT, we maintain two separate caches: one for the greedy algorithm and one for LICKETYSPLIT. The greedy cache is substantially larger, storing up to five million entries per depth for medium to medium-challenging Rashomon set queries. With a 64-bit fingerprint, one would expect 1 in 1.5 million runs of the algorithm to have a collision (we provide a derivation at the end of the section). In contrast, collisions are virtually guaranteed with a 32-bit fingerprint at this scale. For substantially deeper queries or higher-dimensional feature spaces, the fingerprint size can be increased accordingly (noting that representations based on split sequences would also grow with either the number of features or depth of the search space, so this is not unique to us).

Table 4 and Table 5 compare the three representations on datasets whose runtimes exceed one second, using $\lambda = 0.007$, $\varepsilon = 0.02$, and depth $d = 5$. We display the number of binarized features after each dataset.

These tables make the core trade-off very clear. The bitvector dataset can be catastrophically memory-intensive for larger datasets such as Covertype. This is a difference of 75GB versus 1.3GB, and would only grow if we considered a dataset with $20\times$ more samples, such as Higgs.

The literal/itemset representation eliminates the dependence on the sample size, but Table 5 shows that this comes at the

cost of substantial cache inflation. Although each individual key is compact, distinct split sequences can correspond to the same subset of samples, requiring multiple itemset representations, whereas a single bitvector would suffice. As a result, the literal representation can actually lead to increased runtime and memory consumption.

Quantitatively, the literal representation uses more cache entries than bitvector/hash (e.g., Adult-209: $2.46\times$; Bike-164: $2.28\times$; Churn-472: $10.85\times$), reflecting that multiple distinct split sequences can map to the same subset of samples and therefore prevent reuse. This difference grows as the number of features increases.

In contrast, the hash fingerprint preserves the canonical identity while keeping keys constant in size (and thus reducing memory consumption). In these experiments, it also matches the bitvector cache sizes exactly, as we have never observed a collision that impacted the output of PRAXIS.

The core takeaway from the experiments is that the hash fingerprint is more memory efficient than both exact bitvector and itemset representations, and faster than using itemsets. In this ablation, we are also slightly faster than exact bitvectors, though this difference is not major and was not used to inform our choice of subproblem. The reason for this minor speedup is our use of interning for the ablative exact bitvector baseline. Interning is a memory-saving technique in which each distinct object is stored only once, and subsequent uses of the same object refer to that shared copy via a compact identifier. In our setting, this means storing each distinct bitvector mask exactly once and using a small integer ID in cache keys, rather than repeatedly copying the full bitvector. This approach decreases the memory consumption of the bitvector subproblem representation with a slight speed trade-off. This time-memory tradeoff via interning is insignificant compared to the larger gains we see with introducing the hash fingerprint representation.

**Derivation of hash collisions result (under uniform hashing assumptions):** When using the modified LicketySPLIT proxy algorithm, PRAXIS maintains two types of subproblem caches: one for subproblems encountered by LicketySPLIT and one for subproblems encountered by the greedy subroutine. Each cache key includes both a subproblem identifier and the remaining depth, i.e., $(\mathrm{subproblem}, \mathrm{depth})$. Equivalently, we can view this as maintaining a separate cache for each pair consisting of a cache type and a remaining depth. Since there are two cache types and a depth limit of $d$, we analyze up to $2d$ caches. Each of these caches maps a subproblem (a 64-bit fingerprint, a bitvector, or an itemset) to the corresponding objective returned by either LicketySPLIT or greedy for that subproblem and remaining depth. We do not assume that these caches are independent.

We analyze collisions under the standard idealized model in which the 64-bit hash function behaves as a uniform random mapping from distinct subproblem identifiers (bitvectors) to $\{0, \ldots, 2^{64} - 1\}$. Let $c \in \{1, \ldots, 2d\}$ index into the caches, with cache $c$ storing $n_c$ unique subproblem identifiers (bitvectors), each hashed uniformly to one of $M = 2^{64}$ different values. We consider the probability of the event $A_c$, which asks whether there exists a collision among the $n_c$ bitvectors stored in cache $c$. We are interested in bounding the probability that a collision occurs anywhere. We define the event that a collision occurs anywhere.

$$A := \bigcup_{c=1}^{2d} A_c. \tag{54}$$

By the union bound,

$$\Pr(A) = \Pr\left(\bigcup_{c=1}^{2d} A_c\right) \leq \sum_{c=1}^{2d} \Pr(A_c). \tag{55}$$

Now fix a cache $c$. Under standard hashing assumptions, the probability that no collision occurs among the $n_c$ bitvectors is

$$\Pr(A_c^c) = \prod_{i=0}^{n_c-1} \left(1 - \frac{i}{M}\right). \tag{56}$$

Consequently,

$$\Pr(A_c) = 1 - \prod_{i=0}^{n_c-1} \left(1 - \frac{i}{M}\right). \tag{57}$$

| Run | Nonzero cache sizes $(n_c)$ | Pr(collision anywhere) | Odds |
|---|---|---|---|
| Bike-164 | $\{3.68\times10^6,\ 7.20\times10^5,\ 5.29\times10^4,\ 3.74\times10^4,\ 8.81\times10^3,\ 318,\ 318,\ 1\}$ | $3.82\times10^{-7}$ | 1 in $2.6\times10^6$ |
| Chess-50 | $\{3.41\times10^4,\ 1.34\times10^4,\ 2.36\times10^3,\ 1.45\times10^3,\ 518,\ 74,\ 74,\ 1\}$ | $3.67\times10^{-11}$ | 1 in $2.7\times10^{10}$ |
| Christine-231 | $\{1.01\times10^8,\ 6.69\times10^6,\ 6.62\times10^5,\ 1.03\times10^5,\ 4.08\times10^4,\ 462,\ 462,\ 1\}$ | $2.79\times10^{-4}$ | 1 in $3.6\times10^3$ |
| Churn-472 | $\{1.49\times10^7,\ 4.24\times10^6,\ 2.24\times10^5,\ 8.83\times10^4,\ 2.43\times10^4,\ 790,\ 790,\ 1\}$ | $6.47\times10^{-6}$ | 1 in $1.5\times10^5$ |

*Table 6.* Probability of at least one 64-bit fingerprint collision across all $2d$ distinct caches

Substituting (57) into the union bound (55) yields

$$\Pr(A) \ \leq \ \sum_{c=1}^{2d} \left( 1 - \prod_{i=0}^{n_c-1} \left( 1 - \frac{i}{2^{64}} \right) \right). \tag{58}$$

As long as $n_c \ll \sqrt{M} = 2^{32}$ (which holds for all Rashomon set problems considered), the collision probability admits the standard birthday approximation:

$$1 - \prod_{i=0}^{n_c-1} \left( 1 - \frac{i}{M} \right) \approx 1 - \exp\left( -\frac{n_c(n_c-1)}{2M} \right)$$

$$\leq \frac{n_c(n_c-1)}{2M}. \tag{59}$$

The approximation follows from the classical analysis of the birthday paradox (see, e.g., Chapter 3 of Motwani & Raghavan (2013)), while the upper bound on the approximation uses the bound $(1 - e^{-x} \leq x$, which follows from $x - (1 - e^{-x})$ taking only non-negative values).

Applying (59) to (58) gives the final bound

$$\Pr(A) \ \lesssim \ \sum_{c=1}^{2d} \frac{n_c(n_c-1)}{2 \cdot 2^{64}}. \tag{60}$$

Before plugging in numbers, we note that we do not cache subproblems with remaining depth $0$ as a design choice, since these cases are fast to compute. Second, the number of cached subproblems near the root of the search is small and insignificant compared to the number of subproblems cached at later depths, so the bound is usually dominated by one term.

We evaluate this bound using the three runs with the largest cache sizes shown in Table 4, along with the smallest run. For Rashomon set tasks with 50 binary features (one of the largest feature counts considered in Arslan et al. (2026)), we expect approximately one collision in every 27.3 billion runs of PRAXIS. Scaling up to Churn with 472 binary features, the odds increase to about one in $154{,}500$. Even for the Christine dataset, which had the largest cache size, the probability of a collision is still only about one in $3{,}583$. While these odds could be further reduced by using a larger fingerprint, we find a 64-bit fingerprint to be more than sufficient for the problem sizes considered here.

### B.8. Caching Ablation

We now evaluate the importance of caching (with the 64-bit fingerprint) via an ablation. Note that the runtime result detailed in Theorem 3.2 still holds without caching, but we can provably incur practical savings within our modified LicketySPLIT proxy algorithm and within the AND/OR graph expansion.

Table 7 reports the results. Across datasets with small Rashomon sets (Diabetes, Credit, Bank), PRAXIS is able to run without any additional memory beyond the peak memory used when loading the dataset. In contrast, with aggressive caching of proxy algorithms, PRAXIS could use up to 4GB of memory. In these cases, PRAXIS typically runs an order of magnitude faster with caching. Given that 4GB of memory is not prohibitively large, we use caching in our default implementation.

For datasets with larger Rashomon sets (such as Christine), caching still leads to more memory use, but it is dwarfed by the size of the AND/OR graph needed to encode these trees. The amount of memory needed specifically for caching is

| Dataset | Trees | Cache OFF | | Cache ON | | Speedup |
|---|---|---|---|---|---|---|
| | | Time (s) | PeakMem (GB) | Time (s) | PeakMem (GB) | $\times$ |
| Bank-217 | 215,823 | 72,239 | 0.28 | 6,213 | 3.05 | 11.6 |
| Christine-231 | $9.92 \times 10^{10}$ | 4,006 | 96.58 | 855 | 97.68 | 4.7 |
| Churn-472 | $5.78 \times 10^{9}$ | 44,208 | 5.59 | 2,270 | 8.60 | 19.5 |
| Covertype-45 | $5.04 \times 10^{9}$ | 4,465 | 2.41 | 984 | 2.42 | 4.5 |
| Credit-225 | 36,140 | 62,044 | 0.23 | 5,977 | 4.24 | 10.4 |
| Diabetes-121 | 124 | 17,471 | 0.68 | 2,168 | 0.76 | 8.1 |
| Jasmine-207 | $3.11 \times 10^{9}$ | 14,378 | 10.40 | 988 | 14.45 | 14.6 |
| Wine-64 | 5,756,601 | 531 | 0.52 | 37 | 0.75 | 14.4 |

*Table 7.* PRAXIS with and without caching subproblems solved by proxy algorithms. $\lambda = 0.003, \varepsilon = 0.03, d = 5$

unlikely to determine the feasibility of the run (with the 64-bit fingerprint, it would alter feasibility with other subproblem representations such as a full bitvector).

## C. Datasets and Binarization

We provide a table of all 51 datasets and binarizations used in our experiments. For all datasets, we remove rows containing missing values and one-hot encode all categorical variables. Since all existing Rashomon set algorithms (including ours) operate on binary features, all continuous features are binarized prior to training. For most datasets, we use the threshold-guessing binarization procedure of McTavish et al. (2022), which can handle both categorical and continuous features. This procedure trains a gradient-boosted decision tree ensemble with a specified number of estimators and depth cutoffs. It then collects all of the thresholds generated, orders them by Gini variable importance, and removes the least important thresholds iteratively. To remove any dependence on the construction of the binary dataset, we use a large range of GBDT parameters: from depth 1 to depth 7, and from 15 estimators to 500 estimators. On smaller datasets such as Monk2, we took all thresholds, as it was manageable to solve using all of them. On very large datasets such as Higgs, we used quantile binarization on numerical features to reduce the computation cost of training a reference ensemble.

| Dataset | Samples | Features |
|---|---|---|
| Adult | 48842 | 14 |
| Adult | 48842 | 209 |
| Aging | 714 | 57 |
| Bank | 45211 | 97 |
| Bank | 45211 | 217 |
| Bike | 17379 | 43 |
| Bike | 17379 | 164 |
| Chess | 28056 | 50 |
| Christine | 5418 | 80 |
| Christine | 5418 | 231 |
| Churn | 5000 | 81 |
| Churn | 5000 | 472 |
| Compas | 4966 | 44 |
| Covertype | 581012 | 45 |
| Covertype | 581012 | 96 |
| Credit | 30000 | 134 |
| Credit | 30000 | 225 |
| Diabetes | 253680 | 33 |
| Diabetes | 253680 | 121 |
| Droid | 29332 | 84 |
| Electricity | 38474 | 94 |
| Electricity | 38474 | 264 |
| Heart | 297 | 42 |
| Helena | 65196 | 84 |
| Helena | 65196 | 156 |
| Heloc | 2502 | 65 |
| Higgs | 11000000 | 84 |
| IOT | 123117 | 86 |
| Jannis | 57580 | 106 |
| Jannis | 57580 | 247 |
| Jasmine | 2984 | 51 |
| Jasmine | 2984 | 207 |
| Madeline | 3140 | 76 |
| Madeline | 3140 | 451 |
| Madelon | 2000 | 73 |
| Madelon | 2000 | 186 |
| Magic | 19020 | 80 |
| Magic | 19020 | 167 |
| Monk2 | 601 | 17 |
| Mushroom | 8124 | 13 |
| News | 39644 | 196 |
| Phishing | 11055 | 44 |
| Poker | 1025010 | 40 |
| Shopping | 12330 | 112 |
| Shopping | 12330 | 243 |
| Spambase | 4601 | 24 |
| Spambase | 4601 | 67 |
| Student | 649 | 48 |
| Taxi | 1224158 | 27 |
| TicTacToe | 958 | 26 |
| Wine | 6497 | 64 |

*Table 8.* Datasets with number of features in their binarizations

**Adult** (**Becker & Kohavi, 1996**) (*48,842 samples, 14 and 209 features*)    Predict whether an individual earns more than 50,000 per year based on demographic and occupational attributes.
Two binarizations are used, produced via threshold guessing with (i) 40 depth-1 estimators and (ii) 400 depth-2 estimators.

**Aging** (**Malani et al., 2019**) (*714 samples, 57 features*)    Predict whether an individual has visited at least two doctors.
Binarized using threshold guessing with 100 estimators of depth 7.

**Bank** (**Moro et al., 2014a;b**) (*45,211 samples, 97 and 217 features*)    Predict whether a client subscribes to a term deposit following a marketing campaign.
Two binarizations are generated using (i) 100 depth-3 estimators and (ii) 400 depth-1 estimators.

**Bike** (**Fanaee-T & Gama, 2013**; **Fanaee-T, 2013**) (*17,379 samples, 43 and 164 features*)    Predict whether daily bike rental demand exceeds the median.
Two binarizations are used: (i) depth-3 trees with 250 estimators and (ii) depth-1 trees with 250 estimators.

**Chess** (**Bain & Hoff, 1994**) (*28,056 samples, 50 features*)    Determine whether White can force a win within a fixed number of plies, chosen to balance class labels.
Binarized using threshold guessing with a 100-estimator ensemble of depth 10.

**Christine** (**OpenML, 2018**; **Guyon et al., 2019**) (*5,418 samples, 80 and 231 features*)    Binary classification using the provided target column.
Two binarizations are constructed using 40 estimators with depths 2 and 3.

**Churn** (**Erickson et al., 2025**; **Marcoulides, 2005**) (*5,000 samples, 81 and 472 features*)    Predict whether a customer will churn.
Two binarizations are used, generated with 40 estimators of depth 5 and depth 3.

**COMPAS** (**Bao et al., 2021**) (*4,966 samples, 44 features*)    Predict whether a defendant will recidivate within two years.
Binarized using 400 depth-1 estimators.

**Covertype** (**Blackard, 1998**) (*581,012 samples, 45 and 96 features*)    Predict whether a forest plot belongs to cover type 2.
Two binarizations are used: 52 estimators of depth 1 and 46 estimators of depth 2.

**Credit** (**Yeh, 2009**; **Yeh & Lien, 2009**) (*30,000 samples, 134 and 225 features*)    Predict whether a client will default on their credit card payment in the following month.
Two binarizations are generated using 40 estimators with depths 4 and 3.

**Diabetes** (**Burrows et al., 2017**) (*253,680 samples, 33 and 121 features*)    Predict whether an individual is diabetic.
Continuous features are binarized using quantile thresholds, with two configurations using approximately 6 and 500 candidate quantiles per feature.

**Droid** (**Mathur et al., 2021**) (*29,332 samples, 84 features*)    Predict whether an Android application is malicious.
Binarized using threshold guessing with 500 estimators of depth 5.

**Electricity** (**OpenML, 2022a**) (*38,474 samples, 94 and 264 features*)    Predict whether electricity prices increase relative to a 24-hour moving average.
Two binarizations are used, generated with 40 estimators of depths 3 and 4.

**Heart** (**Janosi et al., 1989**; **Detrano et al., 1989**) (*297 samples, 42 features*)    Predict the presence of heart disease.
Binarized using 70 estimators of depth 2.

**Helena** (**OpenML, 2018a**; **Guyon et al., 2019**) (*65,196 samples, 84 and 156 features*)    Predict whether an instance belongs to the most frequent class.
Two binarizations are generated using (i) 35 depth-3 estimators and (ii) 40 depth-2 estimators.

**Heloc (FICO, 2018) (*2,502 samples, 65 features*)**    Predict whether an individual is high- or low-risk for a home equity line of credit.
Binarized using 200 depth-1 estimators.

**Higgs (Whiteson, 2014; Baldi et al., 2014) (*11,000,000 samples, 84 features*)**    Predict whether a particle collision corresponds to signal or background noise.
Each continuous feature is binarized at the 25th, 50th, and 75th percentiles, estimated using a 200,000-sample subsample and applied to the full dataset.

**IOT (S. & Nagapadma, 2023; Sharmila & Nagapadma, 2023) (*123,117 samples, 86 features*)**    Predict whether a network-traffic record corresponds to an attack or normal behavior.
Each numerical feature is discretized into five binary indicators using evenly spaced thresholds.

**Jannis (OpenML, 2022b; Guyon et al., 2019) (*57,580 samples, 106 and 247 features*)**    Binary classification using the provided target column.
Two binarizations are generated using 40 estimators with depths 3 and 2.

**Jasmine (OpenML, 2018b; Guyon et al., 2019) (*2,984 samples, 51 and 207 features*)**    Binary classification using the provided target column.
Two binarizations are generated using 40 estimators with depths 3 and 4.

**Madeline (OpenML, 2018c) (*3,140 samples, 76 and 451 features*)**    Binary classification using the provided target column.
Two binarizations are generated using 40 estimators with depths 2 and 4.

**Madelon (Guyon et al., 2019) (*2,000 samples, 73 and 186 features*)**    Synthetic dataset where points are labeled based on proximity to selected corners of a hypercube.
Two binarizations are generated using 40 estimators with depths 2 and 3.

**Magic (Bock, 2004) (*19,020 samples, 80 and 167 features*)**    Predict whether a Cherenkov telescope image corresponds to a gamma ray or background noise.
Two binarizations are generated using 35 estimators with depths 3 and 2.

**Monk2 (Thrun, 1991; Wnek, 1993) (*601 samples, 17 features*)**    Predict the logical rule where the label is 1 if exactly two of six attributes take value 1.
All possible thresholds are included.

**Mushroom (Audobon Society Field Guide, 1981) (*8,124 samples, 13 features*)**    Predict whether a mushroom is poisonous.
Binarized using 400 depth-1 estimators.

**News (Fernandes et al., 2015b;a) (*39,644 samples, 196 features*)**    Predict whether an online news article's number of shares exceeds the median.
Each feature is discretized into five binary indicators using quantile thresholds.

**Phishing (Mohammad & McCluskey, 2012; Mohammad et al., 2012) (*11,055 samples, 44 features*)**    Predict whether a website is phishing or legitimate.
Binarized using 200 estimators of depth 2.

**Poker (Cattral & Oppacher, 2002) (*1,025,010 samples, 40 features*)**    Predict whether a five-card hand is at least a pair.
Binarized using 40 estimators of depth 4.

**Shopping (Sakar & Kastro, 2018; Sakar et al., 2018) (*12,330 samples, 112 and 243 features*)**    Predict whether an online shopping session ends in a purchase.
Two binarizations are generated using 40 estimators with depths 3 and 4.

**Spambase (Hopkins et al., 1999) (*4,601 samples, 24 and 67 features*)**    Predict whether an email is spam.
Two binarizations are used: (i) 15 estimators of depth 4 and (ii) 40 estimators of depth 1.

**Student (Cortez, 2008; Cortez & Silva, 2008) (*649 samples, 48 features*)**    Predict whether a student passes a course
(final grade $\geq 10$).
Binarized using 500 depth-1 estimators.

**Taxi (OpenML, 2018d) (*1,224,158 samples, 27 features*)**    Predict whether a taxi ride includes a tip.
Binarized using 40 estimators of depth 2.

**Tic-Tac-Toe (Aha, 1991) (*958 samples, 26 features*)**    Predict whether a player has won given a terminal board configuration.
Binarized using 20 estimators of depth 7.

**Wine (OpenML, 2025) (*6,497 samples, 64 features*)**    Predict whether wine quality is at least 7.
Binarized using 200 depth-1 estimators.

# D. Experiments

**Experimental Setup**   All Rashomon set algorithms are run with a 90-hour time limit and 200 GB of allocated memory, unless otherwise noted. Whenever we use bootstrapping, such as in experiments evaluating the mean and standard deviation of PRAXIS's recall or in computing the Rashomon Importance Distribution (Donnelly et al., 2023), we generate 10 bootstrap resamples of the binarized dataset. For consistency with the three other Rashomon set algorithms and with PRAXIS, we allow trivial extensions of trees to be included in the Rashomon set. Trivial extensions are splits in which the leaf and its sibling make identical predictions and thus form a special case of predictive equivalence (McTavish et al., 2025). These extensions can be removed with minimal additional handling of the AND/OR graph or by post-processing the Rashomon set to discard trees that are predictively equivalent to another tree already in the set.

PRAXIS and SORTeD (Arslan et al., 2026) use a depth convention in which the root is at depth 0, whereas RESPLIT and TreeFARMS index the root at depth 1; we adjust for this difference accordingly. Additionally, for RESPLIT (Babbar et al., 2025), we set the lookahead depth parameter to exactly half of the overall tree depth. Thus, when training depth-5 and depth-7 trees (corresponding to depths 6 and 8 under RESPLIT's convention), we use lookahead depths of 3 and 4, respectively.

Additionally, because PRAXIS operates on integer-valued objectives, all reported values of $\lambda$ are snapped to the nearest multiple of $1/|D|$ for compatibility.

For all quantitative recall comparisons, we compute ground truth using PRAXIS configured with an optimal proxy and an exact subproblem representation (i.e., without 64-bit fingerprinting). This configuration is used only to define the reference Rashomon set and eliminates any theoretical risk of hash collisions or proxy-induced pruning errors. All reported experimental results for our method, including runtime, memory usage, and recall, use the modified LicketySPLIT proxy (unless stated otherwise) with a 64-bit fingerprint used to key the subproblem caches. Consequently, any observed recall differences reflect algorithmic behavior rather than numerical differences across implementations.

**Machines Used**   All experiments were performed on an institutional computing cluster. Each experiment was executed on a single compute node equipped with an AMD EPYC 9554 processor (2.75 GHz), with 64 physical cores. Resources were restricted to 1 CPU core and 200 GB of RAM.

## D.1. Rashomon Set Timing

Across all three Rashomon set configurations shown (Table 9–Table 11), PRAXIS is consistently the fastest method, often by orders of magnitude: on most datasets it completes in seconds to minutes while SORTeD and RESPLIT frequently require hours to days, and TreeFARMS does not finish the majority of tasks with a 200GB memory limit.

PRAXIS is the fastest method to complete across all datasets, binarizations, and parameter settings. In a small number of cases, however, PRAXIS exhibits higher memory usage than competing methods. When this excess exceeds a few hundred megabytes, it is always associated with particularly challenging datasets. The main example of this is Madelon, a synthetic dataset in which labels are determined by proximity to selected corners of a hypercube. Such structure does not favor heuristics, which means the initial budget is set much more loosely than optimal, leading to increased exploration. This behavior does not lead to degraded approximation quality (see subsection D.2).

This issue can be mitigated within the PRAXIS framework. As shown in subsubsection D.5.1, using a stronger proxy algorithm both reduces memory usage and shortens runtime on Madelon. Nevertheless, we retain the modified LicketySPLIT proxy as the default, since synthetic adversarial datasets such as Madelon are not typical real-world use cases. Instead, we include them to stress-test the algorithms and characterize worst-case behavior.

Finally, we note that in the few settings where RESPLIT uses less memory than PRAXIS, this comes at the cost of substantially worse approximation quality (see subsection D.11).

*Table 9.* Runtime and peak memory usage at $\lambda = 0.02$, $\varepsilon_{\text{mult}} = 0.03$, depth $= 5$. *Peak MB reports peak resident set size (RSS), including imports, dataset loading, and algorithm execution. – indicates that the method did not finish in 90 hours or with 200GB of RAM.*

| Dataset | $n$ | $k$ | PRAXIS Time | PRAXIS Peak MB | TreeFARMS Time | TreeFARMS Peak MB | SORTeD Time | SORTeD Peak MB | RESPLIT Time | RESPLIT Peak MB |
|---|---|---|---|---|---|---|---|---|---|---|
| Adult | 48842 | 14 | **0.04** | **144.27** | 0.44 | 220.77 | 2.27 | 242.91 | 4.37 | 218.53 |
| Adult | 48842 | 209 | **272.53** | **682.15** | – | – | 56440.23 | 12079.80 | 5494.42 | 5782.34 |
| Aging | 714 | 57 | **0.02** | **126.92** | 1433.52 | 9208.92 | 2.01 | 147.50 | 0.96 | 139.97 |
| Bank | 45211 | 97 | **2.43** | **195.74** | – | – | 1808.61 | 1189.15 | 164.34 | 1468.21 |
| Bank | 45211 | 217 | **19.79** | **281.88** | – | – | 39822.12 | 9455.45 | 1861.18 | 5860.19 |
| Bike | 17379 | 43 | **0.78** | **146.38** | – | – | 24.83 | 284.10 | 26.07 | 256.52 |
| Bike | 17379 | 164 | **35.40** | **359.94** | – | – | 6533.75 | 3546.60 | 1022.18 | 1302.75 |
| Chess | 28056 | 50 | **0.33** | **151.67** | – | – | 26.66 | 288.12 | 32.75 | 325.80 |
| Christine | 5418 | 80 | **31.28** | 974.84 | – | – | 252.29 | 801.67 | 373.98 | **551.52** |
| Christine | 5418 | 231 | **944.27** | 10439.32 | – | – | 38970.60 | 12709.88 | 12625.37 | **3096.83** |
| Churn | 5000 | 81 | **0.22** | **136.17** | – | – | 42.26 | 292.94 | 13.96 | 243.64 |
| Churn | 5000 | 472 | **34.84** | **279.41** | – | – | 123776.11 | 22012.99 | 2564.30 | 2792.24 |
| Compas | 4966 | 44 | **0.09** | **130.47** | 65.08 | 3742.00 | 7.23 | 163.50 | 11.90 | 159.61 |
| Covertype | 581012 | 45 | **13.11** | 579.30 | 519.70 | 27925.92 | 3123.76 | 2986.86 | 1640.53 | **2518.53** |
| Covertype | 581012 | 96 | **358.70** | **1301.23** | – | – | 64672.88 | 16107.48 | 10101.87 | 8631.58 |
| Credit | 30000 | 134 | **5.74** | **190.12** | – | – | 6194.55 | 3472.01 | 447.43 | 1728.17 |
| Credit | 30000 | 225 | **26.03** | **249.60** | – | – | 65145.23 | 14300.78 | 2316.15 | 4355.73 |
| Diabetes | 253680 | 33 | **0.79** | 291.69 | 553.31 | 28336.29 | 683.31 | 1067.69 | 48.68 | 1049.81 |
| Diabetes | 253680 | 121 | **26.59** | **677.67** | – | – | 50986.77 | 10723.03 | 2410.23 | 8037.53 |
| Droid | 29332 | 84 | **0.79** | **165.04** | – | – | 882.65 | 697.19 | 78.89 | 730.01 |
| Electricity | 38474 | 94 | **7.83** | **192.63** | – | – | 1333.20 | 1186.57 | 216.98 | 1011.17 |
| Electricity | 38474 | 264 | **306.94** | **692.10** | – | – | 114325.61 | 23777.65 | 7619.63 | 6402.99 |
| Heart | 297 | 42 | **0.05** | **130.01** | 11792.01 | 20541.30 | 3.46 | 165.31 | 4.59 | 139.74 |
| Helena | 65196 | 84 | **0.64** | 214.39 | 2.53 | 664.27 | 16.31 | 447.14 | 68.20 | 634.65 |
| Helena | 65196 | 156 | **4.74** | 290.96 | 38.32 | 1983.32 | 100.29 | 779.31 | 564.13 | 1755.30 |
| Heloc | 2502 | 65 | **0.38** | **140.59** | – | – | 52.34 | 313.95 | 8.79 | 205.18 |
| Higgs | 11000000 | 84 | **2374.91** | **21537.79** | – | – | – | – | – | – |
| IOT | 123117 | 86 | **0.63** | 302.86 | 3.73 | 936.54 | 18.29 | 688.02 | 5.33 | 805.53 |
| Jannis | 57580 | 106 | **327.54** | **1458.63** | – | – | 5277.68 | 5137.22 | 5587.19 | 2462.35 |
| Jannis | 57580 | 247 | **15352.04** | 42067.00 | – | – | 215642.66 | 57931.75 | 237926.22 | **11297.20** |
| Jasmine | 2984 | 51 | **0.35** | **142.87** | – | – | 13.72 | 211.14 | 16.78 | 193.93 |
| Jasmine | 2984 | 207 | **22.77** | **451.61** | – | – | 5211.85 | 2705.49 | 517.17 | 745.36 |
| Madeline | 3140 | 76 | **2.22** | **196.45** | – | – | 136.18 | 518.20 | 444.67 | 460.68 |
| Madeline | 3140 | 451 | **2971.65** | 29094.20 | – | – | – | – | 133189.38 | **26621.99** |
| Madelon | 2000 | 73 | **8.19** | 292.36 | – | – | 97.75 | 409.84 | 204.01 | **276.30** |
| Madelon | 2000 | 186 | **1420.33** | 29392.03 | – | – | 8760.55 | 4460.80 | 4196.68 | **1242.15** |
| Magic | 19020 | 80 | **31.05** | **446.82** | – | – | 268.66 | 850.65 | 595.96 | 527.28 |
| Magic | 19020 | 167 | **368.48** | 2481.18 | – | – | 6605.22 | 5146.42 | 8074.65 | **1771.25** |
| Monk2 | 601 | 17 | **0.00** | 125.96 | 3.81 | 414.67 | 0.09 | 133.14 | 0.14 | 130.55 |
| Mushroom | 8124 | 13 | **0.00** | 128.88 | 0.18 | 160.31 | 0.07 | 145.33 | 1.46 | 138.40 |
| News | 39644 | 196 | **596.47** | **1480.48** | – | – | 114514.40 | 30426.06 | 13544.76 | 6082.23 |
| Phishing | 11055 | 44 | **0.07** | 138.56 | 645.57 | 87245.93 | 20.98 | **191.67** | 10.20 | 195.34 |
| Poker | 1025010 | 40 | **13.71** | 952.10 | – | – | 7370.45 | 7475.91 | 657.48 | 5958.07 |
| Shopping | 12330 | 112 | **4.09** | **187.69** | – | – | 646.54 | 980.08 | 137.95 | 606.73 |
| Shopping | 12330 | 243 | **58.88** | 470.00 | – | – | 24151.41 | 7957.33 | 1760.04 | 2195.54 |
| Spambase | 4601 | 24 | **0.05** | **130.40** | 106.68 | 10617.66 | 0.98 | 155.85 | 4.46 | 148.48 |
| Spambase | 4601 | 67 | **3.67** | 232.30 | – | – | 53.30 | 351.89 | 40.33 | **223.32** |
| Student | 649 | 48 | **0.02** | **127.51** | 25801.79 | 36353.51 | 3.42 | 159.91 | 3.05 | 142.59 |
| Taxi | 1224158 | 27 | **1.18** | 894.65 | 12.50 | 2874.38 | 364.85 | 3318.65 | 32.35 | 2030.43 |
| TicTacToe | 958 | 26 | **0.17** | 145.15 | 216.40 | 9333.29 | 0.79 | 147.81 | 0.54 | **138.73** |
| Wine | 6497 | 64 | **0.24** | **135.34** | – | – | 62.98 | 307.63 | 12.54 | 255.48 |

*Table 10.* Runtime and peak memory usage at $\lambda = 0.01$, $\varepsilon_{\mathrm{mult}} = 0.02$, depth $= 5$. *Peak MB reports peak resident set size (RSS), including imports, dataset loading, and algorithm execution. – indicates that the method did not finish in 90 hours or with 200GB of RAM.*

| | | | PRAXIS | | TreeFARMS | | SORTeD | | RESPLIT | |
|---|---|---|---|---|---|---|---|---|---|---|
| Dataset | $n$ | $k$ | Time | Peak MB | Time | Peak MB | Time | Peak MB | Time | Peak MB |
| Adult | 48842 | 14 | **0.03** | **142.79** | 1.45 | 356.50 | 2.44 | 237.79 | 2.29 | 212.25 |
| Adult | 48842 | 209 | **213.43** | **329.20** | – | – | 43716.00 | 13414.49 | 5409.87 | 5889.31 |
| Aging | 714 | 57 | **0.02** | **126.82** | 3052.93 | 14589.09 | 2.12 | 153.57 | 1.24 | 140.83 |
| Bank | 45211 | 97 | **3.69** | **194.56** | – | – | 2057.39 | 1651.47 | 246.53 | 1806.91 |
| Bank | 45211 | 217 | **30.26** | **281.51** | – | – | 41199.79 | 14633.89 | 2653.47 | 7062.95 |
| Bike | 17379 | 43 | **0.87** | **145.73** | – | – | 35.19 | 304.32 | 171.08 | 378.37 |
| Bike | 17379 | 164 | **45.83** | **402.90** | – | – | 8686.16 | 4317.58 | 6840.29 | 3007.27 |
| Chess | 28056 | 50 | **0.84** | **151.67** | – | – | 31.13 | 325.01 | 255.75 | 771.57 |
| Christine | 5418 | 80 | **60.96** | 1854.50 | – | – | 318.82 | 943.47 | 176.95 | **578.80** |
| Christine | 5418 | 231 | **2688.49** | 36909.23 | – | – | 50600.24 | 15805.86 | 4764.45 | **3184.38** |
| Churn | 5000 | 81 | **1.51** | **160.04** | – | – | 65.26 | 385.35 | 55.45 | 302.81 |
| Churn | 5000 | 472 | **672.04** | 2673.30 | – | – | 207785.97 | 35154.42 | 65161.25 | 4495.50 |
| Compas | 4966 | 44 | **0.28** | **135.31** | 309.37 | 13473.63 | 7.20 | 181.21 | 12.82 | 160.25 |
| Covertype | 581012 | 45 | **16.00** | **579.65** | 3242.37 | 138008.74 | 3199.01 | 3423.12 | 1199.73 | 2586.74 |
| Covertype | 581012 | 96 | **411.17** | **1301.31** | – | – | 51888.28 | 19681.58 | 14090.82 | 9014.84 |
| Credit | 30000 | 134 | **8.49** | **193.99** | – | – | 8578.20 | 4990.32 | 623.32 | 1930.66 |
| Credit | 30000 | 225 | **37.61** | **265.08** | – | – | 78946.90 | 20408.63 | 3144.97 | 4616.44 |
| Diabetes | 253680 | 33 | **1.41** | **291.29** | – | – | 1003.21 | 1346.43 | 79.66 | 1353.52 |
| Diabetes | 253680 | 121 | **49.56** | **679.43** | – | – | 54065.58 | 17048.95 | 3854.99 | 10305.41 |
| Droid | 29332 | 84 | **1.32** | **164.86** | – | – | 905.42 | 781.80 | 99.92 | 734.73 |
| Electricity | 38474 | 94 | **167.97** | **853.92** | – | – | 1554.07 | 1714.89 | 384.58 | 1052.48 |
| Electricity | 38474 | 264 | **15226.18** | 41489.45 | – | – | 137252.76 | 31984.60 | 22809.66 | **6679.00** |
| Heart | 297 | 42 | **0.15** | **138.07** | 15562.35 | 21285.08 | 4.36 | 167.92 | 8.86 | 305.46 |
| Helena | 65196 | 84 | **4.09** | **214.65** | 3149.22 | 142928.84 | 1023.98 | 1345.18 | 140.12 | 1719.96 |
| Helena | 65196 | 156 | **53.02** | **343.79** | – | – | 19191.46 | 7918.21 | 1027.52 | 5564.16 |
| Heloc | 2502 | 65 | **1.71** | **187.75** | – | – | 63.12 | 361.53 | 17.97 | 236.79 |
| Higgs | 11000000 | 84 | **9354.19** | **21536.63** | – | – | – | – | – | – |
| IOT | 123117 | 86 | **0.69** | **305.50** | 3.73 | 939.38 | 20.49 | 688.75 | 5.34 | 809.26 |
| Jannis | 57580 | 106 | **232.05** | **1214.11** | – | – | 6107.02 | 6181.51 | 10247.63 | 2501.41 |
| Jannis | 57580 | 247 | **10085.61** | 35068.73 | – | – | 238250.82 | 71731.88 | 225254.53 | **11419.48** |
| Jasmine | 2984 | 51 | **1.76** | **184.79** | – | – | 16.98 | 224.13 | 15.75 | 188.03 |
| Jasmine | 2984 | 207 | **142.81** | 2348.67 | – | – | 6355.12 | 3245.50 | 573.81 | 745.05 |
| Madeline | 3140 | 76 | **29.26** | 1063.57 | – | – | 176.20 | **524.15** | – | – |
| Madelon | 2000 | 73 | **103.71** | 3816.83 | – | – | 116.89 | **418.05** | – | – |
| Madelon | 2000 | 186 | **4731.53** | 94685.84 | – | – | 11299.92 | **4768.32** | – | – |
| Magic | 19020 | 80 | **34.69** | **467.59** | – | – | 370.13 | 1041.29 | 370.55 | 532.23 |
| Magic | 19020 | 167 | **707.41** | 4875.07 | – | – | 9429.27 | 6714.69 | 4113.34 | **1947.57** |
| Monk2 | 601 | 17 | **0.02** | **128.17** | 3.80 | 419.89 | 0.12 | 133.92 | 0.16 | 130.15 |
| Mushroom | 8124 | 13 | **0.00** | **128.55** | 0.25 | 169.00 | 0.11 | 145.63 | 1.71 | 138.34 |
| News | 39644 | 196 | **1151.19** | **2658.81** | – | – | 136627.16 | 41543.94 | 10672.60 | 9491.36 |
| Phishing | 11055 | 44 | **0.13** | **138.57** | – | – | 22.91 | 210.77 | 14.30 | 202.65 |
| Poker | 1025010 | 40 | **770.12** | **990.07** | – | – | 10703.73 | 30587.01 | 44109.02 | 21182.09 |
| Shopping | 12330 | 112 | **5.76** | **194.20** | – | – | 1021.69 | 1438.49 | 183.90 | 723.99 |
| Shopping | 12330 | 243 | **91.16** | **575.52** | – | – | 34442.13 | 11823.67 | 2395.57 | 2560.67 |
| Spambase | 4601 | 24 | **0.03** | **128.77** | 164.15 | 14223.42 | 1.22 | 153.60 | 1.63 | 148.00 |
| Spambase | 4601 | 67 | **1.87** | **182.17** | – | – | 59.83 | 318.11 | 26.15 | 225.06 |
| Student | 649 | 48 | **0.05** | **129.01** | 26873.72 | 43952.53 | 4.71 | 168.66 | 3.23 | 143.88 |
| Taxi | 1224158 | 27 | **1.42** | **894.16** | 95.59 | 9040.93 | 1987.10 | 3496.16 | 138.84 | 2334.88 |
| TicTacToe | 958 | 26 | **0.57** | 178.92 | 243.16 | 9455.00 | 1.03 | **148.10** | 44.54 | 1135.14 |
| Wine | 6497 | 64 | **0.47** | **140.97** | – | – | 75.13 | 371.18 | 20.55 | 333.90 |

*Table 11.* Runtime and peak memory usage at $\lambda = 0.005$, $\varepsilon_{\mathrm{mult}} = 0.01$, depth = 5. *Peak MB reports peak resident set size (RSS), including imports, dataset loading, and algorithm execution. – indicates that the method did not finish in 90 hours or with 200GB of RAM.*

| Dataset | $n$ | $k$ | PRAXIS Time | PRAXIS Peak MB | TreeFARMS Time | TreeFARMS Peak MB | SORTeD Time | SORTeD Peak MB | RESPLIT Time | RESPLIT Peak MB |
|---|---|---|---|---|---|---|---|---|---|---|
| Adult | 48842 | 14 | **0.04** | **142.95** | 14.27 | 3726.86 | 3.23 | 239.91 | 2.09 | 208.89 |
| Adult | 48842 | 209 | **103.67** | **287.67** | – | – | 50692.90 | 16083.11 | 3389.43 | 5828.01 |
| Aging | 714 | 57 | **0.02** | **126.96** | 3787.89 | 15215.65 | 2.31 | 158.90 | 1.38 | 141.26 |
| Bank | 45211 | 97 | **14.84** | **195.26** | – | – | 1631.56 | 2136.69 | 604.20 | 1901.05 |
| Bank | 45211 | 217 | **258.36** | **308.94** | – | – | 50481.86 | 19501.40 | 9870.89 | 7499.14 |
| Bike | 17379 | 43 | **0.34** | **143.97** | – | – | 43.12 | 310.59 | 644.18 | 14718.25 |
| Bike | 17379 | 164 | **193.43** | **392.34** | – | – | 10165.04 | 4818.43 | – | – |
| Chess | 28056 | 50 | **0.56** | **151.82** | – | – | 42.15 | 334.95 | 311.71 | 1337.80 |
| Christine | 5418 | 80 | **9.11** | **340.74** | – | – | 366.60 | 965.29 | 59.01 | 475.03 |
| Christine | 5418 | 231 | **740.80** | 9069.65 | – | – | 58533.87 | 16757.12 | 1747.80 | **2822.89** |
| Churn | 5000 | 81 | **1.56** | **167.24** | – | – | 83.37 | 412.25 | 70.71 | 422.36 |
| Churn | 5000 | 472 | **373.36** | **668.72** | – | – | 300644.51 | 40201.41 | 57447.85 | 22535.59 |
| Compas | 4966 | 44 | **0.14** | **131.78** | 357.00 | 15292.30 | 6.95 | 172.83 | 12.91 | 165.10 |
| Covertype | 581012 | 45 | **27.02** | **577.12** | – | – | 2926.56 | 8743.22 | 800.52 | 2686.79 |
| Covertype | 581012 | 96 | **1175.58** | **1301.56** | – | – | 45014.11 | 36188.30 | 9721.09 | 9161.04 |
| Credit | 30000 | 134 | **14.08** | **187.86** | – | – | 8981.24 | 6006.17 | 788.48 | 2041.50 |
| Credit | 30000 | 225 | **65.09** | **231.32** | – | – | 96518.88 | 24726.79 | 3844.08 | 5090.96 |
| Diabetes | 253680 | 33 | **2.39** | **288.94** | – | – | 1296.70 | 1583.26 | 116.81 | 1401.60 |
| Diabetes | 253680 | 121 | **69.09** | **676.90** | – | – | 61324.21 | 24343.17 | 5082.61 | 11350.65 |
| Droid | 29332 | 84 | **1.83** | **165.44** | – | – | 864.82 | 857.08 | 111.37 | 734.25 |
| Electricity | 38474 | 94 | **33.11** | **203.75** | – | – | 1677.05 | 1896.45 | 328.27 | 1058.29 |
| Electricity | 38474 | 264 | **3936.78** | 10709.56 | – | – | 152529.97 | 35515.06 | 22561.25 | **6885.98** |
| Heart | 297 | 42 | **3.80** | 741.07 | – | – | 5.27 | **162.44** | 144.55 | 14249.28 |
| Helena | 65196 | 84 | **4.85** | **214.45** | – | – | 1582.83 | 1867.96 | 223.49 | 1864.16 |
| Helena | 65196 | 156 | **71.38** | **288.56** | – | – | 31017.18 | 11336.79 | 1639.32 | 6105.35 |
| Heloc | 2502 | 65 | **2.08** | **155.22** | – | – | 69.14 | 386.42 | 14.31 | 229.00 |
| Higgs | 11000000 | 84 | **13787.22** | **21538.12** | – | – | – | – | – | – |
| IOT | 123117 | 86 | **0.78** | **304.72** | 3.66 | 938.48 | 12.73 | 688.69 | 5.29 | 806.43 |
| Jannis | 57580 | 106 | **389.75** | **1980.20** | – | – | 5752.37 | 7342.77 | 7888.34 | 4098.71 |
| Jannis | 57580 | 247 | **23348.15** | 82284.37 | – | – | 301779.90 | 86424.74 | 63050.14 | **22226.05** |
| Jasmine | 2984 | 51 | **1.28** | **148.84** | – | – | 18.84 | 244.91 | 8.07 | 191.91 |
| Jasmine | 2984 | 207 | **248.60** | 1136.76 | – | – | 6700.77 | 3744.66 | 354.05 | **751.01** |
| Madeline | 3140 | 76 | **4.29** | **242.38** | – | – | 190.28 | 510.69 | – | – |
| Madelon | 2000 | 73 | **11.09** | 1751.20 | – | – | 130.92 | **436.17** | – | – |
| Magic | 19020 | 80 | **24.47** | **378.42** | – | – | 313.07 | 941.38 | 216.06 | 534.37 |
| Magic | 19020 | 167 | **453.49** | **682.59** | – | – | 9165.80 | 6907.26 | 2277.17 | 2181.43 |
| Monk2 | 601 | 17 | **0.16** | 207.18 | – | – | 0.14 | **133.68** | 0.16 | 129.73 |
| Mushroom | 8124 | 13 | **0.00** | **127.17** | 0.36 | 172.73 | 0.19 | 144.43 | 1.47 | 136.66 |
| News | 39644 | 196 | **1432.08** | 3948.98 | – | – | 162757.59 | 47046.81 | 5987.42 | 11158.40 |
| Phishing | 11055 | 44 | **0.15** | **138.13** | – | – | 18.30 | 216.72 | 13.78 | 201.65 |
| Poker | 1025010 | 40 | **450.94** | **989.50** | – | – | 8472.44 | 14107.55 | 61478.45 | 112157.77 |
| Shopping | 12330 | 112 | **9.37** | **161.82** | – | – | 1137.70 | 1835.37 | 226.42 | 769.48 |
| Shopping | 12330 | 243 | **121.89** | **303.47** | – | – | 43659.86 | 15684.98 | 2416.49 | 2734.83 |
| Spambase | 4601 | 24 | **0.03** | **128.27** | 179.34 | 15334.18 | 1.69 | 155.17 | 4338.20 | 92571.09 |
| Spambase | 4601 | 67 | **1.00** | **136.66** | – | – | 70.93 | 334.90 | 26.00 | 223.70 |
| Student | 649 | 48 | **0.12** | **127.36** | 24661.38 | 44943.45 | 5.54 | 174.04 | 8.23 | 144.50 |
| Taxi | 1224158 | 27 | **1.65** | 893.99 | – | – | 1672.81 | 3575.89 | 146.76 | 2709.54 |
| TicTacToe | 958 | 26 | **0.08** | **133.65** | 208.98 | 9563.60 | 1.27 | 144.92 | 214.33 | 8655.65 |
| Wine | 6497 | 64 | **0.61** | **142.35** | – | – | 78.07 | 428.43 | 22.27 | 348.78 |

### D.2. Rashomon Set Recall

We now report approximation quality over a broader range of parameter settings than in Table 2. In particular, we vary the sparsity parameter from $\lambda = 0.0025$ to $\lambda = 0.02$ and evaluate recall under two regimes: Multiplicative, which considers trees within a $1 + \varepsilon_{\text{mult}}$ factor of the optimal objective, and Absolute, which considers trees within $1 + \varepsilon_{\text{mult}}$ factor of the proxy algorithm. Because trees are enumerated in sorted order, the multiplicative set can always be recovered and is a subset (or equal to) the full set returned by PRAXIS. Regardless, if recall remains high on the larger absolute set, these additional trees may be useful for downstream applications.

Table 12 reports Rashomon set recall (mean $\pm$ standard deviation over bootstraps). Across most datasets and $\lambda$ values, PRAXIS achieves essentially perfect recall (often $1.000 \pm 0$) under both variants, indicating close agreement with ground-truth enumeration. In cases where multiplicative recall falls below 1, absolute recall differs only slightly and may be marginally higher or lower; these differences are not substantial. This suggests that trees farther from optimal are not necessarily harder to recover, even though their higher objectives leave less slack before a completion exceeds the pruning budget. Additionally, the datasets with a recall below 1 tend to be smaller datasets, which can be solved exactly with little computational cost.

In subsubsection D.5.3, we report recall for using a greedy proxy algorithm, which has reasonable, but inferior, recall compared to using modified LicketySPLIT as the proxy algorithm.

In the main body of the paper, we calculate recall relative to two algorithms: PRAXIS with an optimal proxy and an exact subproblem representation, which corresponds to full recall of the Rashomon set, and SORTeD (Xin et al., 2022). For each method, we compute the histogram of tree objectives within the Rashomon set. Then, for each objective value, we take the maximum count reported by either method. We use these maximum counts to estimate the ground-truth number of optimal trees. This approach is more conservative, yielding lower recall for our method, than using either PRAXIS with an optimal proxy alone or SORTeD alone as the ground truth. We use this procedure because, in some cases, the two existing exact solvers (TreeFARMS (Xin et al., 2022) and SORTeD (Arslan et al., 2026)) disagree even when run on identical datasets with identical parameter settings. Additionally, TreeFARMS does not always finish within our 90-hour time limit and 200GB memory limit.

We believe the missed trees in SORTeD are caused solely by column deduplication in SORTeD. For example, if

$$\texttt{years\_of\_education} \geq 16 \quad \text{and} \quad \texttt{income} \geq 70000$$

induce identical bitvectors (this can happen with quantile binarization or Threshold Guessing (McTavish et al., 2022)), SORTeD arbitrarily chooses one binary column to keep, even though the two splits represent different features and should both be included in the Rashomon set (their inclusion can affect downstream analyses such as variable-importance conclusions). We have not observed any cases where PRAXIS with optimal parameters finds fewer trees than SORTeD.

Note that the occurrence of this kind of collision is not indicative of a flaw in binarization when we are working with Rashomon sets rather than trying to find a single optimum. Consider, for example, the exhaustive binarization needed to evaluate the full Rashomon set for continuous features: each individual feature becomes, in the worst case, $n$ binary features. If any two features are sufficiently correlated, it is quite likely that there exist thresholds for these two features with the same support. Even for fully decorrelated features, each continuous feature will have a split with support size one (actually two disjoint ones: where all but one sample is below a threshold, or all but one is above), and now we have the birthday paradox with $n$ possibilities and $k$ assignments.

Using the same birthday paradox approximation as in Section B.7, we have a probability of $\frac{k(k-1)}{2n}$ of a collision; for 20 continuous features and 5000 samples, that gives a 7.6% chance of at least one collision among just this subset of features with support 1. When the collision occurs, every instance of the preserved split can be replaced with the removed split while maintaining Rashomon membership, leading to a substantial number of missed trees.

*Table 12.* Recall comparison between the Absolute-$\varepsilon$ and Multiplicative-$\varepsilon$ (Mult) Rashomon sets across $\lambda \in \{0.02, 0.01, 0.005, 0.0025\}$ (depth $= 5$, $\varepsilon = 0.03$). Values are mean $\pm$ std over 10 bootstraps (rounded to 3 decimals for means and 2 decimals for std). If some but not all bootstraps run out of memory, the mean and std reported are for those runs that finished. If a dataset has more than one binarization, the smaller one is displayed in this table.

| Dataset | $\lambda = 0.02$ | | $\lambda = 0.01$ | | $\lambda = 0.005$ | | $\lambda = 0.0025$ | |
|---|---|---|---|---|---|---|---|---|
| | Abs | Mult | Abs | Mult | Abs | Mult | Abs | Mult |
| Adult-14 | 1.000±0.00 | 1.000±0.00 | 1.000±0.00 | 1.000±0.00 | 1.000±0.00 | 1.000±0.00 | 1.000±0.00 | 1.000±0.00 |
| Aging-57 | 1.000±0.00 | 1.000±0.00 | 1.000±0.00 | 1.000±0.00 | 1.000±0.00 | 1.000±0.00 | 0.998±0.00 | 0.999±0.00 |
| Bank-97 | 1.000±0.00 | 1.000±0.00 | 1.000±0.00 | 1.000±0.00 | 1.000±0.00 | 1.000±0.00 | 1.000±0.00 | 1.000±0.00 |
| Bike-43 | 1.000±0.00 | 1.000±0.00 | 1.000±0.00 | 1.000±0.00 | 0.979±0.03 | 0.978±0.03 | 0.994±0.01 | 0.993±0.01 |
| Christine-80 | 0.993±0.01 | 0.993±0.01 | 0.992±0.01 | 0.994±0.01 | 0.997±0.00 | 0.998±0.01 | 1.000±0.00 | 1.000±0.00 |
| Churn-81 | 1.000±0.00 | 1.000±0.00 | 1.000±0.00 | 1.000±0.00 | 0.977±0.03 | 0.979±0.03 | 0.991±0.02 | 0.989±0.03 |
| Compas-44 | 1.000±0.00 | 1.000±0.00 | 0.999±0.00 | 0.999±0.00 | 1.000±0.00 | 1.000±0.00 | 1.000±0.00 | 1.000±0.00 |
| Covertype-45 | 1.000±0.00 | 1.000±0.00 | 1.000±0.00 | 1.000±0.00 | 1.000±0.00 | 1.000±0.00 | 1.000±0.00 | 1.000±0.00 |
| Diabetes-33 | 1.000±0.00 | 1.000±0.00 | 1.000±0.00 | 1.000±0.00 | 1.000±0.00 | 1.000±0.00 | 1.000±0.00 | 1.000±0.00 |
| Droid-84 | 1.000±0.00 | 1.000±0.00 | 1.000±0.00 | 1.000±0.00 | 0.975±0.03 | 1.000±0.00 | 0.987±0.01 | 0.993±0.01 |
| Electricity-94 | 1.000±0.00 | 1.000±0.00 | 0.921±0.04 | 0.963±0.06 | 0.982±0.03 | 0.980±0.03 | 0.950±0.00 | 0.987±0.02 |
| Heart-42 | 0.995±0.02 | 1.000±0.00 | 0.946±0.13 | 0.917±0.22 | 0.948±0.14 | 0.952±0.13 | 0.948±0.14 | 0.952±0.13 |
| Helena-84 | 1.000±0.00 | 1.000±0.00 | 1.000±0.00 | 1.000±0.00 | 1.000±0.00 | 1.000±0.00 | 1.000±0.00 | 1.000±0.00 |
| Heloc-65 | 1.000±0.00 | 1.000±0.00 | 1.000±0.00 | 1.000±0.00 | 1.000±0.00 | 1.000±0.00 | 1.000±0.00 | 1.000±0.00 |
| IOT-86 | 1.000±0.00 | 1.000±0.00 | 1.000±0.00 | 1.000±0.00 | 1.000±0.00 | 1.000±0.00 | 1.000±0.00 | 1.000±0.00 |
| Jasmine-51 | 1.000±0.00 | 1.000±0.00 | 0.985±0.02 | 0.997±0.01 | 0.994±0.01 | 0.995±0.01 | 0.992±0.02 | 0.992±0.02 |
| Madeline-76 | 0.991±0.01 | 1.000±0.00 | 0.986±0.04 | 0.997±0.01 | 0.999±0.00 | 1.000±0.00 | 0.866±0.26 | 0.905±0.23 |
| Madelon-73 | 0.975±0.01 | 0.997±0.01 | 0.996±0.01 | 0.999±0.00 | 1.000±0.00 | 1.000±0.00 | 0.995±0.01 | 0.999±0.00 |
| Magic-80 | 0.983±0.03 | 0.991±0.02 | 0.985±0.01 | 0.991±0.01 | 0.990±0.01 | 0.992±0.01 | 0.999±0.00 | 0.999±0.00 |
| Monk2-17 | 1.000±0.00 | 1.000±0.00 | 0.901±0.17 | 0.943±0.15 | 0.874±0.26 | 0.914±0.20 | 0.888±0.13 | 0.887±0.15 |
| Mushroom-13 | 1.000±0.00 | 1.000±0.00 | 1.000±0.00 | 1.000±0.00 | 1.000±0.00 | 1.000±0.00 | 1.000±0.00 | 1.000±0.00 |
| Phishing-44 | 1.000±0.00 | 1.000±0.00 | 1.000±0.00 | 1.000±0.00 | 1.000±0.00 | 1.000±0.00 | 0.987±0.04 | 0.987±0.04 |
| Shopping-112 | 1.000±0.00 | 1.000±0.00 | 1.000±0.00 | 1.000±0.00 | 1.000±0.00 | 1.000±0.00 | 0.996±0.01 | 0.996±0.01 |
| Spambase-24 | 0.992±0.02 | 0.992±0.02 | 0.993±0.02 | 0.998±0.01 | 0.967±0.08 | 0.963±0.09 | 0.986±0.03 | 0.992±0.02 |
| Student-48 | 1.000±0.00 | 1.000±0.00 | 0.985±0.04 | 0.987±0.04 | 0.991±0.01 | 0.992±0.01 | 0.986±0.04 | 0.987±0.04 |
| Taxi-27 | 1.000±0.00 | 1.000±0.00 | 1.000±0.00 | 1.000±0.00 | 1.000±0.00 | 1.000±0.00 | 0.990±0.01 | 0.991±0.01 |
| TicTacToe-26 | 1.000±0.00 | 1.000±0.00 | 0.967±0.06 | 0.988±0.03 | 0.967±0.08 | 0.965±0.09 | 0.982±0.06 | 0.995±0.02 |
| Wine-64 | 1.000±0.00 | 1.000±0.00 | 1.000±0.00 | 1.000±0.00 | 1.000±0.00 | 1.000±0.00 | 0.998±0.01 | 0.998±0.00 |

## D.3. What Trees Are Missing?

We next examine the datasets from Table 12 for which PRAXIS does not achieve perfect recall on every bootstrap. Our goal is to understand which Rashomon-set trees are missed.

Fix a $(\lambda, d, \varepsilon_{\text{mult}})$ configuration. For each bootstrap, consider a tree that belongs to the true Rashomon set (within a $1 + \varepsilon_{\text{mult}}$ factor of the minimum objective) but is not returned by our approximation. We normalize its objective value by dividing by the optimal objective on that bootstrap, mapping it into the interval $[1, 1 + \varepsilon_{\text{mult}}]$. We then aggregate these normalized objectives across all bootstraps and analyze the distribution of missingness on the interval $[1, 1 + \varepsilon_{\text{mult}}]$.

We summarize the distribution of missing objectives using three statistics. The *top third* reports the proportion of missing trees whose normalized objective lies in $[1, 1 + \varepsilon_{\text{mult}}/3)$, while the *bottom third* reports the proportion in $[1 + 2\varepsilon_{\text{mult}}/3, 1 + \varepsilon_{\text{mult}}]$. In addition, we report the proportion of missing trees that occur at the *very end*. Unlike the quantile-based summaries, this corresponds to a single objective value: the largest objective observed in the full Rashomon set on that bootstrap. This statistic captures whether the trees we miss lie (essentially) right at the Rashomon boundary, where there is no slack to absorb proxy error.

Table 13 aggregates these statistics across all datasets and values of $\lambda$ (with depth fixed to $d = 5$ and $\varepsilon_{\text{mult}} = 0.03$). Across all configurations, PRAXIS rarely misses trees in the best third of the objective interval, consistent with the fact that it can be used as an optimal tree finder (see subsection D.4). Instead, the missed trees are overwhelmingly concentrated near the Rashomon boundary, and in many cases a substantial fraction occur exactly at the worst possible objective. This suggests that the observed recall losses are driven primarily by trees whose objectives are very close to the Rashomon threshold, motivating the use of slack in PRAXIS's budget to improve recall for a slightly smaller budget (see subsection D.6). However, this is typically unnecessary, as PRAXIS frequently recovers all or nearly all of the Rashomon set, yielding a strong approximation anyway.

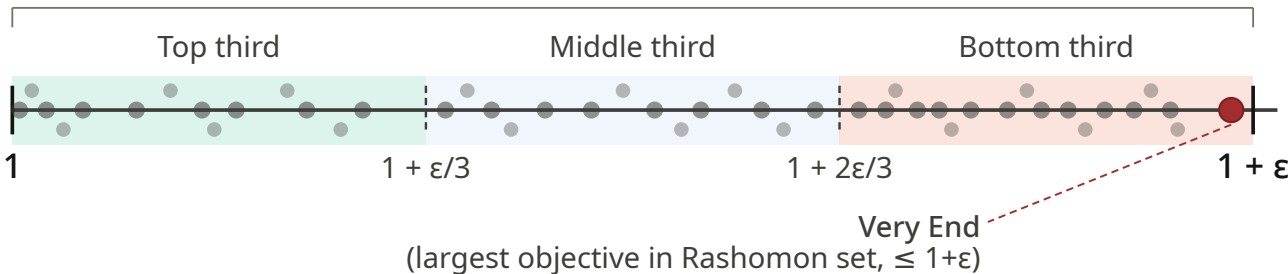

Table 13. Distribution of the *missed* trees for various $\lambda$ configurations ($d = 5$, $\varepsilon_{\text{mult}} = 0.03$). The number line illustrates the regions used in the table. For each missed tree, we normalize its objective by the optimal objective, so values lie in $[1, 1 + \varepsilon_{\text{mult}}]$. The entries report *where the missed trees occur*, not the miss rate within each region: for example, 0.90 in Bottom Third means that 90% of the missed trees lie in the bottom third, not that 90% of bottom-third trees are missed. Top Third denotes $[1, 1 + \varepsilon_{\text{mult}}/3)$, Bottom Third denotes $[1 + 2\varepsilon_{\text{mult}}/3, 1 + \varepsilon_{\text{mult}}]$, and Very End denotes missed trees at the largest objective in the Rashomon set. See Appendix D.2 for overall approximation quality; very few trees are missed.

| $\lambda$ | Dataset | Top Third | Bottom Third | Very End |
|---|---|---|---|---|
| 0.0025 | Aging-57 | 0.0000 | 1.0000 | 0.9780 |
| | Bank-97 | 0.0000 | 1.0000 | 0.1316 |
| | Bike-43 | 0.0000 | 0.9882 | 0.1188 |
| | Christine-80 | 0.0000 | 0.9997 | 0.3870 |
| | Churn-81 | 0.0000 | 0.9514 | 0.3918 |
| | Compas-44 | 0.0000 | 0.9992 | 0.2661 |
| | Covertype-45 | 0.0000 | 0.9996 | 0.0050 |
| | Droid-84 | 0.0000 | 0.9333 | 0.0000 |
| | Electricity-94 | 0.0000 | 0.9949 | 0.0453 |
| | Heart-42 | 0.0000 | 1.0000 | 1.0000 |
| | Heloc-65 | 0.0000 | 0.9997 | 0.6642 |
| | Jasmine-51 | 0.0000 | 0.9937 | 0.5966 |
| | Madeline-76 | 0.0000 | 0.9944 | 0.5591 |
| | Madelon-73 | 0.0000 | 0.9940 | 0.6998 |
| | Magic-80 | 0.0000 | 0.9951 | 0.1204 |
| | Monk2-17 | 0.0004 | 0.9843 | 0.8440 |
| | Phishing-44 | 0.0000 | 0.8452 | 0.0476 |
| | Shopping-112 | 0.0000 | 0.9399 | 0.1796 |
| | Spambase-24 | 0.0000 | 1.0000 | 0.4434 |
| | Student-48 | 0.0000 | 0.9993 | 0.9713 |
| | Taxi-27 | 0.0000 | 1.0000 | 0.0625 |
| | Tictactoe-26 | 0.0000 | 0.9923 | 0.8454 |
| | Wine-64 | 0.0000 | 0.9929 | 0.3229 |
| 0.005 | Bike-43 | 0.0000 | 0.9581 | 0.1321 |
| | Christine-80 | 0.0000 | 0.9980 | 0.2632 |
| | Churn-81 | 0.0048 | 0.9286 | 0.3360 |
| | Compas-44 | 0.0000 | 1.0000 | 0.1123 |
| | Covertype-45 | 0.0000 | 1.0000 | 0.0000 |
| | Electricity-94 | 0.0023 | 0.9698 | 0.0490 |
| | Heart-42 | 0.0000 | 1.0000 | 1.0000 |
| | Heloc-65 | 0.0000 | 0.9903 | 0.3829 |
| | Jasmine-51 | 0.0000 | 0.9767 | 0.3878 |
| | Madeline-76 | 0.0000 | 1.0000 | 0.4695 |
| | Madelon-73 | 0.0000 | 0.9899 | 0.5539 |
| | Magic-80 | 0.0003 | 0.9773 | 0.0949 |
| | Monk2-17 | 0.0000 | 0.9959 | 0.7115 |
| | Spambase-24 | 0.0000 | 0.9091 | 0.5152 |
| | Student-48 | 0.0000 | 0.9881 | 0.9486 |
| | Tictactoe-26 | 0.0313 | 0.8504 | 0.6104 |
| | Wine-64 | 0.0000 | 1.0000 | 0.6364 |
| 0.01 | Christine-80 | 0.0000 | 0.9846 | 0.1919 |
| | Compas-44 | 0.0000 | 1.0000 | 0.3636 |
| | Electricity-94 | 0.0113 | 0.9016 | 0.0160 |
| | Heart-42 | 0.0000 | 0.9925 | 0.9925 |
| | Heloc-65 | 0.0000 | 1.0000 | 1.0000 |
| | Jasmine-51 | 0.0000 | 1.0000 | 0.0000 |
| | Madeline-76 | 0.0000 | 0.9952 | 0.3692 |
| | Madelon-73 | 0.0000 | 0.9558 | 0.4575 |
| | Magic-80 | 0.0000 | 0.9880 | 0.0901 |
| | Monk2-17 | 0.0000 | 0.9981 | 0.5102 |
| | Spambase-24 | 0.0000 | 0.0000 | 0.0000 |
| | Student-48 | 0.0000 | 1.0000 | 1.0000 |
| | Tictactoe-26 | 0.0000 | 1.0000 | 0.7812 |
| 0.02 | Christine-80 | 0.0000 | 0.8000 | 0.2000 |
| | Madelon-73 | 0.0000 | 1.0000 | 0.1000 |
| | Magic-80 | 0.0000 | 0.9712 | 0.0481 |
| | Spambase-24 | 0.0000 | 0.6000 | 0.0000 |

## D.4. Optimal Tree Finder

Across all 3 sparsity settings ($\lambda \in \{0.02, 0.01, 0.005\}$), PRAXIS almost always achieves the same minimum objective as STreeD (and GOSDT when available) while being typically 2-3 orders of magnitude faster. In the rare failure cases, the returned objective exceeds the optimal value by a small multiplicative factor and by an even smaller additive amount.

*Table 14.* Minimum objective values at $\lambda = 0.005$, depth $= 5$ and $\varepsilon_{\text{mult}} \in \{0.0, 0.01, 0.03\}$, with runtimes in seconds

| Dataset | STreeD | PRAXIS ($\varepsilon_{\text{mult}}$=0.0) | PRAXIS ($\varepsilon_{\text{mult}}$=0.01) | PRAXIS ($\varepsilon_{\text{mult}}$=0.03) | Time (GOSDT) | Time (STreeD) | Time (PRAXIS, $\varepsilon_{\text{mult}}$=0.0) |
|---|---|---|---|---|---|---|---|
| Adult-14 | **8677** | **8677** | **8677** | **8677** | 1.16 | 2.17 | **0.03** |
| Adult-209 | **8677** | **8677** | **8677** | **8677** | – | 49434.59 | **43.85** |
| Aging-57 | **135** | **135** | **135** | **135** | – | 1.88 | **0.02** |
| Bank-97 | **5273** | **5273** | **5273** | **5273** | – | 3142.87 | **6.35** |
| Bank-217 | **5273** | **5273** | **5273** | **5273** | – | 48896.44 | **83.77** |
| Bike-43 | **2977** | **2977** | **2977** | **2977** | 430.16 | 51.90 | **0.32** |
| Bike-164 | **2889** | **2889** | **2889** | **2889** | – | 5333.04 | **68.07** |
| Chess-50 | **6321** | **6321** | **6321** | **6321** | 445.12 | 20.32 | **0.59** |
| Christine-80 | **1706** | **1706** | **1706** | **1706** | – | 346.66 | **2.05** |
| Christine-231 | **1709** | **1709** | **1709** | – | – | 16139.66 | **63.66** |
| Churn-81 | **580** | **580** | **580** | **580** | 1648.47 | 81.57 | **0.55** |
| Churn-472 | **580** | **580** | **580** | **580** | – | 61895.81 | **101.92** |
| Compas-44 | **1724** | **1724** | **1724** | **1724** | 41.20 | 1.34 | **0.09** |
| Covertype-45 | **164668** | **164668** | **164668** | **164668** | 1423.34 | 849.66 | **4.17** |
| Covertype-96 | **164668** | **164668** | **164668** | **164668** | – | 19457.18 | **74.15** |
| Credit-134 | **5712** | **5712** | **5712** | **5712** | – | 3665.61 | **10.54** |
| Credit-225 | **5712** | **5712** | **5712** | **5712** | – | 28223.63 | **68.63** |
| Diabetes-33 | **36614** | **36614** | **36614** | **36614** | 1565.15 | 608.10 | **2.00** |
| Diabetes-121 | **36614** | **36614** | **36614** | **36614** | – | 23273.75 | **102.40** |
| Droid-84 | **2715** | **2715** | **2715** | **2715** | – | 305.52 | **1.59** |
| Electricity-94 | **9406** | 9519 | **9406** | **9406** | – | 536.49 | **6.35** |
| Electricity-264 | **9376** | **9376** | **9376** | **9376** | – | 44936.91 | **672.79** |
| Heart-42 | **39** | **39** | **39** | **39** | 301.39 | **1.20** | 1.32 |
| Helena-84 | **3246** | **3246** | **3246** | **3246** | – | 404.41 | **4.67** |
| Helena-156 | **3246** | **3246** | **3246** | **3246** | – | 16671.62 | **30.47** |
| Heloc-65 | **770** | **770** | **770** | **770** | 6502.01 | 19.70 | **0.61** |
| Higgs-84 | – | **4159657** | **4159657** | **4159657** | – | – | **4466.14** |
| IOT-86 | **1356** | **1356** | **1356** | **1356** | **0.27** | 1.06 | 0.74 |
| Jannis-106 | **17339** | 17351 | **17339** | **17339** | – | 3787.99 | **46.88** |
| Jannis-247 | **17311** | **17311** | **17311** | – | – | 120205.06 | **14468.23** |
| Jasmine-51 | **659** | **659** | **659** | **659** | 352.79 | 17.23 | **0.73** |
| Jasmine-207 | **659** | **659** | **659** | **659** | – | 1620.51 | **97.47** |
| Madeline-76 | **679** | **679** | **679** | **679** | 5488.74 | 49.20 | **1.17** |
| Madeline-451 | **652** | – | – | – | – | 183383.52 | – |
| Madelon-73 | **456** | **456** | **456** | **456** | 5573.99 | 30.73 | **25.52** |
| Madelon-186 | **435** | **435** | – | – | – | **2374.22** | 6330.09 |
| Magic-80 | **3903** | **3903** | **3903** | **3903** | – | 183.60 | **9.82** |
| Magic-167 | **3903** | 3915 | 3905 | **3903** | – | 4171.23 | **39.28** |
| Monk2-17 | **162** | **162** | **162** | **162** | 1.02 | **0.06** | 0.18 |
| Mushroom-13 | **253** | **253** | **253** | **253** | 0.18 | 0.06 | **0.00** |
| News-196 | **15716** | **15716** | **15716** | – | – | 72221.13 | **177.13** |
| Phishing-44 | **1136** | **1136** | **1136** | **1136** | 199.00 | 7.93 | **0.16** |
| Poker-40 | **473585** | **473585** | **473585** | **473585** | – | 5270.71 | **61.15** |
| Shopping-112 | **1456** | **1456** | **1456** | **1456** | – | 545.95 | **4.29** |
| Shopping-243 | **1456** | **1456** | **1456** | **1456** | – | 12060.33 | **40.47** |
| Spambase-24 | **564** | **564** | **564** | **564** | 9.39 | 0.57 | **0.02** |
| Spambase-67 | **570** | **570** | **570** | **570** | 1038.04 | 18.88 | **0.39** |
| Student-48 | **115** | **115** | **115** | **115** | 255.05 | 1.78 | **0.07** |
| Taxi-27 | **100623** | **100623** | **100623** | **100623** | 268.61 | 427.25 | **1.69** |
| TicTacToe-26 | **168** | **168** | **168** | **168** | 18.45 | 0.44 | **0.03** |
| Wine-64 | **1260** | **1260** | **1260** | **1260** | 1492.83 | 29.50 | **0.52** |

*Table 15.* Minimum objective values at $\lambda = 0.01$, depth $= 5$ and $\varepsilon_{\text{mult}} \in \{0.0, 0.01, 0.03\}$, with runtimes in seconds

| Dataset | STreeD | PRAXIS ($\varepsilon_{\text{mult}}$=0.0) | PRAXIS ($\varepsilon_{\text{mult}}$=0.01) | PRAXIS ($\varepsilon_{\text{mult}}$=0.03) | Time (GOSDT) | Time (STreeD) | Time (PRAXIS , $\varepsilon_{\text{mult}}$=0.0) |
|---|---|---|---|---|---|---|---|
| Adult-14 | **9653** | **9653** | **9653** | **9653** | 0.31 | 1.06 | **0.03** |
| Adult-209 | **9653** | **9653** | **9653** | **9653** | – | 9611.06 | **37.43** |
| Aging-57 | **138** | **138** | **138** | **138** | 56.40 | 2.24 | **0.02** |
| Bank-97 | **5741** | **5741** | **5741** | **5741** | – | 2265.54 | **2.80** |
| Bank-217 | **5741** | **5741** | **5741** | **5741** | – | 33338.40 | **28.43** |
| Bike-43 | **3454** | **3454** | **3454** | **3454** | 304.76 | 43.90 | **0.36** |
| Bike-164 | **3454** | **3454** | **3454** | **3454** | – | 5459.65 | **12.53** |
| Chess-50 | **7133** | **7133** | **7133** | **7133** | 356.90 | 20.73 | **0.46** |
| Christine-80 | **1851** | **1851** | **1851** | **1851** | – | 305.20 | **4.06** |
| Christine-231 | **1860** | **1860** | **1860** | **1860** | – | 13646.96 | **80.84** |
| Churn-81 | **736** | **736** | **736** | **736** | 1005.89 | 77.91 | **0.69** |
| Churn-472 | **736** | **736** | **736** | **736** | – | 48780.84 | **149.35** |
| Compas-44 | **1845** | **1845** | **1845** | **1845** | 29.27 | 1.24 | **0.05** |
| Covertype-45 | **173383** | **173383** | **173383** | **173383** | 923.09 | 535.43 | **3.33** |
| Covertype-96 | **173383** | **173383** | **173383** | **173383** | – | 16413.62 | **60.55** |
| Credit-134 | **6012** | **6012** | **6012** | **6012** | – | 3483.04 | **8.42** |
| Credit-225 | **6012** | **6012** | **6012** | **6012** | – | 24738.49 | **37.78** |
| Diabetes-33 | **37883** | **37883** | **37883** | **37883** | 618.10 | 426.37 | **1.40** |
| Diabetes-121 | **37883** | **37883** | **37883** | **37883** | – | 20655.07 | **49.53** |
| Droid-84 | **3285** | **3285** | **3285** | **3285** | – | 177.58 | **1.15** |
| Electricity-94 | **10526** | **10526** | **10526** | **10526** | – | 524.95 | **32.23** |
| Electricity-264 | **10526** | **10526** | **10526** | **10526** | – | 38287.22 | **2732.79** |
| Heart-42 | **62** | **62** | **62** | **62** | 211.61 | 1.14 | **0.09** |
| Helena-84 | **4224** | **4224** | **4224** | **4224** | 153.80 | 18.99 | **1.84** |
| Helena-156 | **4224** | **4224** | **4224** | **4224** | – | 1634.69 | **14.12** |
| Heloc-65 | **818** | **818** | **818** | **818** | 5534.76 | 17.18 | **0.61** |
| Higgs-84 | – | **4449359** | **4449359** | **4449359** | – | – | **2285.68** |
| IOT-86 | **2586** | **2586** | **2586** | **2586** | 0.26 | 2.23 | **0.67** |
| Jannis-106 | **18789** | **18789** | **18789** | **18789** | – | 3182.88 | **26.58** |
| Jannis-247 | **18771** | **18771** | **18771** | **18771** | – | 97224.37 | **321.67** |
| Jasmine-51 | **725** | **725** | **725** | **725** | 257.59 | 6.07 | **0.45** |
| Jasmine-207 | **725** | **725** | **725** | **725** | – | 1289.89 | **30.33** |
| Madeline-76 | **889** | **889** | **889** | **889** | 4159.87 | 49.33 | **4.51** |
| Madeline-451 | **858** | **858** | **858** | **858** | – | 174318.25 | **19950.96** |
| Madelon-73 | **577** | **577** | **577** | **577** | 4212.66 | 30.28 | 40.14 |
| Madelon-186 | **560** | **560** | **560** | **560** | – | 2235.10 | **1697.94** |
| Magic-80 | **4498** | 4500 | **4498** | **4498** | – | 155.68 | **3.21** |
| Magic-167 | **4490** | **4490** | **4490** | **4490** | – | 3412.59 | **28.78** |
| Monk2-17 | **208** | 212 | 209 | **208** | 0.81 | 0.06 | **0.00** |
| Mushroom-13 | **444** | **444** | **444** | **444** | 0.16 | 0.03 | **0.00** |
| News-196 | **16496** | **16496** | **16496** | **16496** | – | 65250.51 | **131.71** |
| Phishing-44 | **1360** | **1360** | **1360** | **1360** | 143.64 | 7.70 | **0.12** |
| Poker-40 | **509567** | **509567** | **509567** | **509567** | – | 5076.40 | **49.22** |
| Shopping-112 | **1578** | **1578** | **1578** | **1578** | – | 349.46 | **2.85** |
| Shopping-243 | **1578** | **1578** | **1578** | **1578** | – | 8196.00 | **27.35** |
| Spambase-24 | **702** | **702** | **702** | **702** | 5.18 | 0.44 | **0.02** |
| Spambase-67 | **708** | **708** | **708** | **708** | 762.07 | 17.37 | **0.30** |
| Student-48 | **125** | **125** | **125** | **125** | 163.02 | 1.45 | **0.04** |
| Taxi-27 | **112865** | **112865** | **112865** | **112865** | 5.38 | 140.52 | **1.41** |
| TicTacToe-26 | **244** | **244** | **244** | **244** | 12.80 | 0.45 | **0.36** |
| Wine-64 | **1326** | **1326** | **1326** | **1326** | 1008.05 | 25.99 | **0.36** |

*Table 16.* Minimum objective values at $\lambda = 0.02$, depth = 5 and $\varepsilon_{\text{mult}} \in \{0.0, 0.01, 0.03\}$, with runtimes in seconds

| Dataset | STreeD | PRAXIS ($\varepsilon_{\text{mult}}$=0.0) | PRAXIS ($\varepsilon_{\text{mult}}$=0.01) | PRAXIS ($\varepsilon_{\text{mult}}$=0.03) | Time (GOSDT) | Time (STreeD) | Time (PRAXIS , $\varepsilon_{\text{mult}}$=0.0) |
|---|---|---|---|---|---|---|---|
| Adult-14 | **11557** | **11557** | **11557** | **11557** | 0.08 | 0.26 | **0.02** |
| Adult-209 | **11557** | **11557** | **11557** | **11557** | – | 32301.16 | **28.89** |
| Aging-57 | **145** | **145** | **145** | **145** | 24.88 | 1.64 | **0.02** |
| Bank-97 | **6193** | **6193** | **6193** | **6193** | 204.54 | 1324.78 | **1.92** |
| Bank-217 | **6193** | **6193** | **6193** | **6193** | – | 14266.24 | **19.94** |
| Bike-43 | **4138** | **4138** | **4138** | **4138** | 140.38 | 28.09 | **0.20** |
| Bike-164 | **4138** | **4138** | **4138** | **4138** | – | 2071.78 | **6.63** |
| Chess-50 | **8253** | **8253** | **8253** | **8253** | 251.54 | 18.39 | **0.31** |
| Christine-80 | **2053** | **2053** | **2053** | **2053** | 3960.06 | 237.72 | **4.10** |
| Christine-231 | **2054** | **2054** | **2054** | **2054** | – | 27891.23 | **63.42** |
| Churn-81 | **807** | **807** | **807** | **807** | 63.67 | 44.96 | **0.22** |
| Churn-472 | **807** | **807** | **807** | **807** | – | 30923.96 | **34.95** |
| Compas-44 | **1992** | **1992** | **1992** | **1992** | 3.85 | 0.69 | **0.04** |
| Covertype-45 | **190813** | **190813** | **190813** | **190813** | 216.55 | 6.80 | **2.83** |
| Covertype-96 | **190813** | **190813** | **190813** | **190813** | – | 10531.68 | **50.26** |
| Credit-134 | **6612** | **6612** | **6612** | **6612** | – | 5998.63 | **5.78** |
| Credit-225 | **6612** | **6612** | **6612** | **6612** | – | 17247.67 | **26.36** |
| Diabetes-33 | **40420** | **40420** | **40420** | **40420** | 3.54 | 29.18 | **0.79** |
| Diabetes-121 | **40420** | **40420** | **40420** | **40420** | – | 40707.22 | **26.85** |
| Droid-84 | **4167** | **4167** | **4167** | **4167** | – | 245.28 | **0.80** |
| Electricity-94 | **11769** | **11769** | **11769** | **11769** | – | 430.66 | **2.62** |
| Electricity-264 | **11769** | **11769** | **11769** | **11769** | – | 29572.30 | **61.22** |
| Heart-42 | **77** | **77** | **77** | **77** | 147.28 | 0.88 | **0.03** |
| Helena-84 | **5309** | **5309** | **5309** | **5309** | **0.29** | 0.45 | 0.64 |
| Helena-156 | **5309** | **5309** | **5309** | **5309** | **0.82** | 1.00 | 4.78 |
| Heloc-65 | **875** | **875** | **875** | **875** | 3731.96 | 14.37 | **0.25** |
| Higgs-84 | – | **4763119** | **4763119** | **4763119** | – | – | **1073.52** |
| IOT-86 | **5048** | **5048** | **5048** | **5048** | **0.27** | 0.97 | 0.63 |
| Jannis-106 | **20963** | **20963** | **20963** | **20963** | – | 2502.86 | **19.77** |
| Jannis-247 | **20986** | **20986** | **20986** | **20986** | – | 71091.52 | **275.90** |
| Jasmine-51 | **828** | **828** | **828** | **828** | 151.62 | 5.07 | **0.09** |
| Jasmine-207 | **828** | **828** | **828** | **828** | – | 1192.10 | **3.43** |
| Madeline-76 | **1110** | **1110** | **1110** | **1110** | 2783.98 | 110.12 | **0.65** |
| Madeline-451 | **1104** | **1104** | **1104** | **1104** | – | 159124.31 | **405.31** |
| Madelon-73 | **734** | **734** | **734** | **734** | 2402.67 | 28.34 | **7.01** |
| Madelon-186 | **730** | **730** | **730** | **730** | – | 2258.98 | **214.28** |
| Magic-80 | **5226** | **5226** | **5226** | **5226** | 2206.13 | 123.31 | **4.69** |
| Magic-167 | **5225** | **5225** | **5225** | **5225** | – | 2341.04 | **25.46** |
| Monk2-17 | **218** | **218** | **218** | **218** | 0.55 | 0.05 | **0.00** |
| Mushroom-13 | **768** | **768** | **768** | **768** | 0.12 | 0.02 | **0.00** |
| News-196 | **17675** | **17675** | **17675** | **17675** | – | 47814.68 | **75.18** |
| Phishing-44 | **1670** | **1670** | **1670** | **1670** | 1.71 | 5.76 | **0.06** |
| Poker-40 | **531808** | **531808** | **531808** | **531808** | – | 4081.42 | **13.91** |
| Shopping-112 | **1826** | **1826** | **1826** | **1826** | 152.18 | 123.01 | **1.38** |
| Shopping-243 | **1826** | **1826** | **1826** | **1826** | – | 3321.37 | **13.68** |
| Spambase-24 | **950** | **950** | **950** | **950** | 1.31 | 0.25 | **0.01** |
| Spambase-67 | **945** | **945** | **945** | **945** | 501.69 | 42.46 | **0.37** |
| Student-48 | **143** | **143** | **143** | **143** | 84.21 | 1.08 | **0.02** |
| Taxi-27 | **137347** | **137347** | **137347** | **137347** | 1.82 | 4.85 | **1.17** |
| TicTacToe-26 | **304** | **304** | **304** | **304** | 6.57 | 0.35 | **0.07** |
| Wine-64 | **1407** | **1407** | **1407** | **1407** | 532.82 | 19.85 | **0.25** |

## D.5. Other Proxy Algorithms

### D.5.1. OTHER PROXY ALGORITHM TIMINGS

In subsection B.6, we demonstrated how less cache-friendly proxies can hurt runtime and memory usage. Here, we evaluate three proxy regimes for PRAXIS : (i) a greedy tree builder ($\ell=0$), (ii) our default modified LicketySPLIT algorithm ($\ell=1$), and (iii) our LicketySPLIT generalization ($\ell=2$, detailed in subsection B.5), all of which have cache-friendly properties. Each increase in $\ell$ adds a factor of $\mathcal{O}(kd)$ to the runtime of each proxy call. In practice, this means that each increase in $\ell$ makes PRAXIS an order of magnitude slower, though this is not a universal rule. Table 17–Table 19 show these observations. For instance, on the Christine, Electricity, Jannis, and Madelon datasets, using a greedy algorithm as the proxy algorithm can lead PRAXIS to run out of memory, as the initial bound was set too loose (see Table 19). These cases are a motivating reason for why the modified LicketySPLIT algorithm is our default proxy algorithm: only on Madelon is it ever slower than using the $\ell = 2$ generalization, which is a very challenging dataset that pushes beyond typical real-world uses. Additionally, it is generally expected that increasing $\ell$ will correspond to better recall for the set of trees within $\varepsilon_{\text{mult}}$ of optimal. While this does not always have to hold due to the budget initialization in PRAXIS, we find that it holds practically (see subsubsection D.5.2). In particular, we find that there are some cases where using a greedy proxy can lead to significantly worse recall than using modified LicketySPLIT (which leads to perfect recall the majority of the time). One explanation for this is that choosing the optimal split based on more information is more robust, so the proxy objective at the root better reflects the performance of the proxy objective at other subproblems, so Corollary 3.6 and other theoretical results are more likely to hold.

*Table 17.* Runtime and peak memory usage at $\lambda = 0.02$, $\varepsilon_{\mathrm{mult}} = 0.03$, depth $= 5$. *Peak MB reports peak resident set size (RSS), including imports, dataset loading, and algorithm execution. – indicates that the method did not finish.*

| Dataset | $n$ | $k$ | PRAXIS ($\ell{=}0$) | | PRAXIS ($\ell{=}1$) | | PRAXIS ($\ell{=}2$) | |
|---|---|---|---|---|---|---|---|---|
| | | | Time | Peak MB | Time | Peak MB | Time | Peak MB |
| Adult | 48842 | 14 | **0.01** | **144.27** | 0.04 | **144.27** | 0.11 | **144.27** |
| Adult | 48842 | 209 | **7.84** | **286.95** | 272.53 | 682.15 | 4709.93 | 14651.75 |
| Aging | 714 | 57 | **0.00** | **126.92** | 0.02 | **126.92** | 0.17 | 134.71 |
| Bank | 45211 | 97 | **0.05** | **195.74** | 2.43 | **195.74** | 30.17 | 298.87 |
| Bank | 45211 | 217 | **0.19** | **281.88** | 19.79 | **281.88** | 627.68 | 1476.60 |
| Bike | 17379 | 43 | 1.17 | 149.96 | **0.78** | 146.38 | 5.02 | 206.43 |
| Bike | 17379 | 164 | 42.20 | **332.17** | 35.40 | 359.94 | 605.43 | 4125.13 |
| Chess | 28056 | 50 | **0.03** | **151.67** | 0.33 | **151.67** | 2.64 | 159.85 |
| Christine | 5418 | 80 | 66.84 | 1417.77 | **31.28** | **974.84** | 199.96 | 5516.96 |
| Christine | 5418 | 231 | 2285.95 | 21360.87 | **944.27** | **10439.32** | – | – |
| Churn | 5000 | 81 | **0.01** | **136.17** | 0.22 | **136.17** | 2.92 | 199.82 |
| Churn | 5000 | 472 | **0.11** | **162.34** | 34.84 | 279.41 | 2139.13 | 14412.20 |
| Compas | 4966 | 44 | **0.01** | **130.47** | 0.09 | **130.47** | 0.42 | 137.01 |
| Covertype | 581012 | 45 | **2.51** | **579.30** | 13.11 | **579.30** | 41.02 | **579.30** |
| Covertype | 581012 | 96 | **21.69** | **1301.23** | 358.70 | **1301.23** | 2760.71 | 1350.85 |
| Credit | 30000 | 134 | **0.10** | **189.42** | 5.74 | 190.12 | 142.21 | 764.59 |
| Credit | 30000 | 225 | **0.25** | **230.30** | 26.03 | 249.60 | 985.63 | 3665.34 |
| Diabetes | 253680 | 33 | **0.09** | **291.69** | 0.79 | **291.69** | 5.53 | **291.69** |
| Diabetes | 253680 | 121 | **0.54** | **677.67** | 26.59 | **677.67** | 545.65 | 848.44 |
| Droid | 29332 | 84 | **0.03** | **165.04** | 0.79 | **165.04** | 18.15 | 254.45 |
| Electricity | 38474 | 94 | **0.28** | **183.05** | 7.83 | 192.63 | 72.14 | 590.03 |
| Electricity | 38474 | 264 | **4.22** | **290.44** | 306.94 | 692.10 | 7059.97 | 16147.89 |
| Heart | 297 | 42 | **0.01** | **126.41** | 0.05 | 130.01 | 0.44 | 158.48 |
| Helena | 65196 | 84 | **0.04** | **214.39** | 0.64 | **214.39** | 8.91 | 218.62 |
| Helena | 65196 | 156 | **0.11** | **290.96** | 4.74 | **290.96** | 126.88 | 644.31 |
| Heloc | 2502 | 65 | **0.23** | **133.98** | 0.38 | 140.59 | 5.08 | 335.07 |
| Higgs | 11000000 | 84 | **71.33** | **21537.79** | 2374.91 | **21537.79** | 37609.94 | **21537.79** |
| IOT | 123117 | 86 | **0.06** | **302.86** | 0.63 | **302.86** | 9.43 | **302.86** |
| Jannis | 57580 | 106 | 14.44 | **221.44** | 327.54 | 1458.63 | 2283.14 | 14792.75 |
| Jannis | 57580 | 247 | 485.68 | **757.78** | 15352.04 | 42067.00 | – | – |
| Jasmine | 2984 | 51 | **0.02** | **129.66** | 0.35 | 142.87 | 3.00 | 254.38 |
| Jasmine | 2984 | 207 | **0.37** | **141.29** | 22.77 | 451.61 | 460.97 | 8297.02 |
| Madeline | 3140 | 76 | 115.00 | 9451.59 | **2.22** | **196.45** | 19.29 | 710.83 |
| Madeline | 3140 | 451 | – | – | **2971.65** | **29094.20** | – | – |
| Madelon | 2000 | 73 | 40.80 | 1640.36 | 8.19 | 175.66 | **8.15** | **166.89** |
| Madelon | 2000 | 186 | 2448.54 | 53989.45 | **1420.33** | **29392.03** | 1967.57 | 35821.65 |
| Magic | 19020 | 80 | 21.23 | **299.13** | 31.05 | 446.82 | 115.34 | 1355.47 |
| Magic | 19020 | 167 | 404.96 | **2118.31** | **368.48** | 2481.18 | 3190.99 | 29738.90 |
| Monk2 | 601 | 17 | **0.00** | **125.96** | 0.00 | **125.96** | 0.01 | 126.65 |
| Mushroom | 8124 | 13 | 0.00 | **128.88** | **0.00** | **128.88** | 0.00 | **128.88** |
| News | 39644 | 196 | **6.49** | **249.32** | 596.47 | 1480.48 | 16288.08 | 61919.68 |
| Phishing | 11055 | 44 | **0.01** | **138.56** | 0.07 | **138.56** | 0.79 | 146.90 |
| Poker | 1025010 | 40 | **0.62** | **952.10** | 13.71 | **952.10** | 150.82 | **952.10** |
| Shopping | 12330 | 112 | **2.51** | **162.80** | 4.09 | 187.69 | 57.09 | 1001.80 |
| Shopping | 12330 | 243 | **36.44** | **308.27** | 58.88 | 470.00 | 1609.78 | 14590.64 |
| Spambase | 4601 | 24 | **0.01** | **128.88** | 0.05 | 130.40 | 0.19 | 137.54 |
| Spambase | 4601 | 67 | **0.36** | **136.48** | 3.67 | 232.30 | 22.89 | 754.88 |
| Student | 649 | 48 | **0.00** | **125.95** | 0.02 | 127.51 | 0.23 | 141.20 |
| Taxi | 1224158 | 27 | **0.29** | **894.65** | 1.18 | **894.65** | 4.72 | **894.65** |
| TicTacToe | 958 | 26 | **0.02** | **128.40** | 0.17 | 145.15 | 0.16 | 142.17 |
| Wine | 6497 | 64 | **0.01** | **135.34** | 0.24 | **135.34** | 3.18 | 206.77 |

*Table 18.* Runtime and peak memory usage at $\lambda = 0.01$, $\varepsilon_{\text{mult}} = 0.02$, depth $= 5$. *Peak MB reports peak resident set size (RSS), including imports, dataset loading, and algorithm execution. – indicates that the method did not finish.*

| Dataset | $n$ | $k$ | PRAXIS ($\ell$=0) | | PRAXIS ($\ell$=1) | | PRAXIS ($\ell$=2) | |
|---|---|---|---|---|---|---|---|---|
| | | | Time | Peak MB | Time | Peak MB | Time | Peak MB |
| Adult | 48842 | 14 | 0.04 | **142.79** | **0.03** | **142.79** | 0.11 | **142.79** |
| Adult | 48842 | 209 | 68.23 | **287.65** | 213.43 | 329.20 | 4202.96 | 9429.84 |
| Aging | 714 | 57 | **0.00** | 126.82 | 0.02 | **126.82** | 0.22 | 138.23 |
| Bank | 45211 | 97 | **0.10** | **194.56** | 3.69 | **194.56** | 46.95 | 374.95 |
| Bank | 45211 | 217 | **0.37** | **281.51** | 30.26 | **281.51** | 969.46 | 2103.91 |
| Bike | 17379 | 43 | **0.41** | **144.00** | 0.87 | 145.73 | 5.68 | 207.70 |
| Bike | 17379 | 164 | **13.67** | **225.51** | 45.83 | 402.91 | 707.02 | 4586.06 |
| Chess | 28056 | 50 | **0.10** | **151.67** | 0.84 | **151.67** | 4.95 | 175.17 |
| Christine | 5418 | 80 | 240.18 | 5638.40 | **60.96** | **1854.50** | 218.22 | 5511.67 |
| Christine | 5418 | 231 | 8532.86 | 95097.52 | **2688.49** | **36909.23** | – | – |
| Churn | 5000 | 81 | **0.43** | **137.02** | 1.51 | 160.04 | 14.76 | 413.08 |
| Churn | 5000 | 472 | **79.49** | **399.20** | 672.04 | 2673.30 | 22093.47 | 127553.72 |
| Compas | 4966 | 44 | **0.17** | **133.18** | 0.28 | 135.31 | 0.91 | 145.99 |
| Covertype | 581012 | 45 | **6.12** | **579.65** | 16.00 | **579.65** | 49.36 | **579.65** |
| Covertype | 581012 | 96 | **59.59** | **1301.31** | 411.17 | **1301.31** | 2953.70 | 1556.26 |
| Credit | 30000 | 134 | **0.14** | **189.78** | 8.49 | 194.00 | 203.41 | 1014.44 |
| Credit | 30000 | 225 | **0.35** | **231.85** | 37.61 | 265.08 | 1407.14 | 3913.87 |
| Diabetes | 253680 | 33 | **0.11** | **291.29** | 1.41 | **291.29** | 9.97 | **291.29** |
| Diabetes | 253680 | 121 | **1.50** | **679.43** | 49.56 | **679.43** | 960.79 | 1039.69 |
| Droid | 29332 | 84 | **0.05** | **164.86** | 1.32 | **164.86** | 23.14 | 285.92 |
| Electricity | 38474 | 94 | 64.11 | 635.72 | 167.97 | 853.92 | 206.93 | 1074.71 |
| Electricity | 38474 | 264 | **5691.44** | **27635.33** | 15226.18 | 41489.45 | 24916.21 | 69089.77 |
| Heart | 297 | 42 | **0.02** | **127.28** | 0.15 | 138.06 | 0.97 | 182.99 |
| Helena | 65196 | 84 | **1.17** | **214.65** | 4.09 | **214.65** | 46.84 | 392.11 |
| Helena | 65196 | 156 | **12.65** | **290.91** | 53.02 | 343.79 | 986.02 | 3744.70 |
| Heloc | 2502 | 65 | **0.05** | **129.84** | 1.71 | 187.75 | 18.10 | 666.60 |
| Higgs | 11000000 | 84 | 393.56 | **21536.63** | 9354.19 | **21536.63** | 73165.26 | **21536.63** |
| IOT | 123117 | 86 | **0.06** | **305.50** | 0.69 | **305.50** | 10.90 | **305.50** |
| Jannis | 57580 | 106 | 310.42 | 1347.28 | **232.05** | **1214.11** | 1690.06 | 8275.83 |
| Jannis | 57580 | 247 | 11780.88 | 38579.31 | **10085.61** | **35068.73** | – | – |
| Jasmine | 2984 | 51 | **0.24** | **134.40** | 1.76 | 184.79 | 4.23 | 253.73 |
| Jasmine | 2984 | 207 | **11.70** | **268.94** | 142.81 | 2348.67 | 584.55 | 8660.03 |
| Madeline | 3140 | 76 | – | – | **29.26** | **1063.57** | 35.10 | 1115.99 |
| Madelon | 2000 | 73 | – | – | 103.71 | 3816.83 | **34.16** | **1026.64** |
| Madelon | 2000 | 186 | – | – | 4731.53 | 94685.84 | **1150.60** | **21008.90** |
| Magic | 19020 | 80 | **1.69** | **155.94** | 34.69 | 467.59 | 157.95 | 1492.90 |
| Magic | 19020 | 167 | **167.85** | **1377.71** | 707.41 | 4875.07 | 4224.45 | 32126.71 |
| Monk2 | 601 | 17 | **0.00** | **125.15** | 0.02 | 128.17 | 0.02 | 127.72 |
| Mushroom | 8124 | 13 | 0.00 | **128.55** | **0.00** | **128.55** | 0.01 | **128.55** |
| News | 39644 | 196 | 52.38 | 302.44 | 1151.19 | 2658.81 | 25423.37 | 120160.33 |
| Phishing | 11055 | 44 | **0.01** | **138.57** | 0.13 | **138.57** | 1.48 | 159.73 |
| Poker | 1025010 | 40 | **457.95** | **990.07** | 770.12 | **990.07** | 1509.56 | **990.07** |
| Shopping | 12330 | 112 | 5.18 | **175.07** | 5.76 | 194.20 | 87.15 | 1188.32 |
| Shopping | 12330 | 243 | **76.70** | **415.36** | 91.16 | 575.52 | 2516.95 | 16797.06 |
| Spambase | 4601 | 24 | 0.05 | 130.68 | **0.03** | **128.77** | 0.13 | 133.84 |
| Spambase | 4601 | 67 | **1.49** | **174.85** | 1.87 | 182.17 | 13.96 | 556.00 |
| Student | 649 | 48 | **0.00** | **126.16** | 0.05 | 129.01 | 0.41 | 151.35 |
| Taxi | 1224158 | 27 | **0.31** | **894.16** | 1.42 | **894.16** | 5.63 | **894.16** |
| TicTacToe | 958 | 26 | **0.01** | **127.37** | 0.57 | 178.93 | 0.20 | 144.43 |
| Wine | 6497 | 64 | **0.01** | **135.97** | 0.47 | 140.97 | 5.71 | 258.30 |

*Table 19.* Runtime and peak memory usage at $\lambda = 0.005$, $\varepsilon_{\text{mult}} = 0.01$, depth $= 5$ across 3 proxy options. *Peak MB reports peak resident set size (RSS), including imports, dataset loading, and algorithm execution. – indicates that the method did not finish.*

| Dataset | $n$ | $k$ | PRAXIS ($\ell{=}0$) Time | Peak MB | PRAXIS ($\ell{=}1$) Time | Peak MB | PRAXIS ($\ell{=}2$) Time | Peak MB |
|---|---|---|---|---|---|---|---|---|
| Adult | 48842 | 14 | **0.04** | **142.95** | **0.04** | **142.95** | 0.13 | **142.95** |
| Adult | 48842 | 209 | **32.36** | **287.67** | 103.67 | **287.67** | 2428.60 | 4861.11 |
| Aging | 714 | 57 | **0.00** | **126.96** | 0.02 | **126.96** | 0.25 | 138.39 |
| Bank | 45211 | 97 | **12.10** | **195.26** | 14.84 | **195.26** | 144.98 | 671.35 |
| Bank | 45211 | 217 | **224.29** | **299.06** | 258.36 | 308.94 | 4200.01 | 7761.75 |
| Bike | 17379 | 43 | **0.02** | **143.97** | 0.34 | **143.97** | 3.22 | 184.31 |
| Bike | 17379 | 164 | **0.22** | **170.06** | 193.43 | 392.35 | 386.16 | 2333.43 |
| Chess | 28056 | 50 | 1.24 | **151.82** | **0.56** | **151.82** | 4.57 | 171.81 |
| Christine | 5418 | 80 | 2823.78 | 93080.18 | **9.11** | **340.74** | 148.76 | 3683.37 |
| Christine | 5418 | 231 | – | – | **740.80** | **9069.65** | – | – |
| Churn | 5000 | 81 | **1.15** | **160.91** | 1.56 | 167.24 | 14.48 | 405.19 |
| Churn | 5000 | 472 | **313.91** | 1614.43 | 373.36 | **668.72** | 16226.72 | 120081.82 |
| Compas | 4966 | 44 | 0.83 | 171.84 | **0.14** | **131.78** | 0.56 | 141.74 |
| Covertype | 581012 | 45 | 203.48 | **577.12** | **27.02** | **577.12** | 73.76 | **577.12** |
| Covertype | 581012 | 96 | 1983.47 | 1394.22 | **1175.58** | **1301.56** | 5246.79 | 2086.85 |
| Credit | 30000 | 134 | **0.21** | **187.86** | 14.08 | **187.86** | 254.82 | 1120.93 |
| Credit | 30000 | 225 | **0.44** | **231.32** | 65.09 | **231.32** | 1715.57 | 4348.59 |
| Diabetes | 253680 | 33 | **0.15** | **288.94** | 2.39 | **288.94** | 14.38 | **288.94** |
| Diabetes | 253680 | 121 | **1.35** | **676.90** | 69.09 | **676.90** | 1370.85 | 1290.02 |
| Droid | 29332 | 84 | **0.22** | **165.44** | 1.83 | **165.44** | 29.04 | 310.22 |
| Electricity | 38474 | 94 | – | – | **33.11** | **203.75** | 363.33 | 1528.72 |
| Electricity | 38474 | 264 | – | – | **3936.78** | **10709.56** | 17162.87 | 39449.11 |
| Heart | 297 | 42 | 41.15 | 17781.65 | 3.80 | 741.07 | **0.65** | **164.29** |
| Helena | 65196 | 84 | **1.46** | **214.45** | 4.85 | **214.45** | 63.93 | 427.16 |
| Helena | 65196 | 156 | **24.05** | **288.56** | 71.38 | **288.56** | 1135.65 | 3719.30 |
| Heloc | 2502 | 65 | **0.13** | **129.70** | 2.08 | 155.22 | 24.39 | 786.97 |
| Higgs | 11000000 | 84 | **9941.68** | **21538.12** | 13787.22 | **21538.12** | 65312.94 | **21538.12** |
| IOT | 123117 | 86 | **0.06** | **304.72** | 0.78 | **304.72** | 12.29 | **304.72** |
| Jannis | 57580 | 106 | – | – | **389.75** | **1980.20** | 1174.44 | 4765.17 |
| Jannis | 57580 | 247 | – | – | **23348.15** | **82284.37** | – | – |
| Jasmine | 2984 | 51 | 4.15 | 383.18 | **1.28** | **148.84** | 5.36 | 269.31 |
| Jasmine | 2984 | 207 | 916.47 | 13211.78 | **248.60** | **1136.76** | 710.45 | 10160.48 |
| Madeline | 3140 | 76 | – | – | **4.29** | **242.38** | 43.02 | 1052.55 |
| Madelon | 2000 | 73 | 452.75 | 77155.76 | **11.09** | 1751.20 | 22.42 | **773.55** |
| Magic | 19020 | 80 | **1.18** | **150.76** | 24.47 | 378.41 | 138.15 | 1336.18 |
| Magic | 19020 | 167 | **19.67** | **192.93** | 453.49 | 682.60 | 3298.74 | 22249.79 |
| Monk2 | 601 | 17 | 0.31 | 289.94 | 0.16 | 207.18 | **0.03** | **128.70** |
| Mushroom | 8124 | 13 | **0.00** | **127.17** | **0.00** | **127.17** | 0.01 | **127.17** |
| News | 39644 | 196 | **770.88** | **520.43** | 1432.08 | 3948.98 | 26120.80 | 120072.61 |
| Phishing | 11055 | 44 | **0.12** | **138.13** | 0.15 | **138.13** | 1.64 | 160.26 |
| Poker | 1025010 | 40 | 4874.99 | 2703.79 | **450.94** | **989.50** | 1251.19 | **989.50** |
| Shopping | 12330 | 112 | **1.17** | **152.06** | 9.37 | 161.83 | 124.63 | 1402.99 |
| Shopping | 12330 | 243 | **13.05** | **171.57** | 121.89 | 303.47 | 2774.69 | 17096.23 |
| Spambase | 4601 | 24 | **0.07** | 134.62 | **0.03** | **128.27** | 0.16 | **134.46** |
| Spambase | 4601 | 67 | 3.93 | 310.89 | **1.00** | **136.66** | 14.37 | 554.75 |
| Student | 649 | 48 | **0.01** | **124.62** | 0.12 | 127.36 | 0.88 | 175.46 |
| Taxi | 1224158 | 27 | **0.32** | **893.99** | 1.65 | **893.99** | 6.75 | **893.99** |
| TicTacToe | 958 | 26 | **0.03** | 134.17 | 0.08 | **133.65** | 0.27 | 149.93 |
| Wine | 6497 | 64 | 1.61 | **141.50** | **0.61** | 142.35 | 7.67 | 285.83 |

D.5.2. TREES FOUND USING VARIOUS PROXY ALGORITHMS

We display approximation results comparing the same three proxy algorithms for $\lambda = 0.005$, $\varepsilon_{\text{mult}} = 0.03$, and depth $d = 5$. We exclude cases where the Rashomon set size is believed to be small ($< 10$ trees).

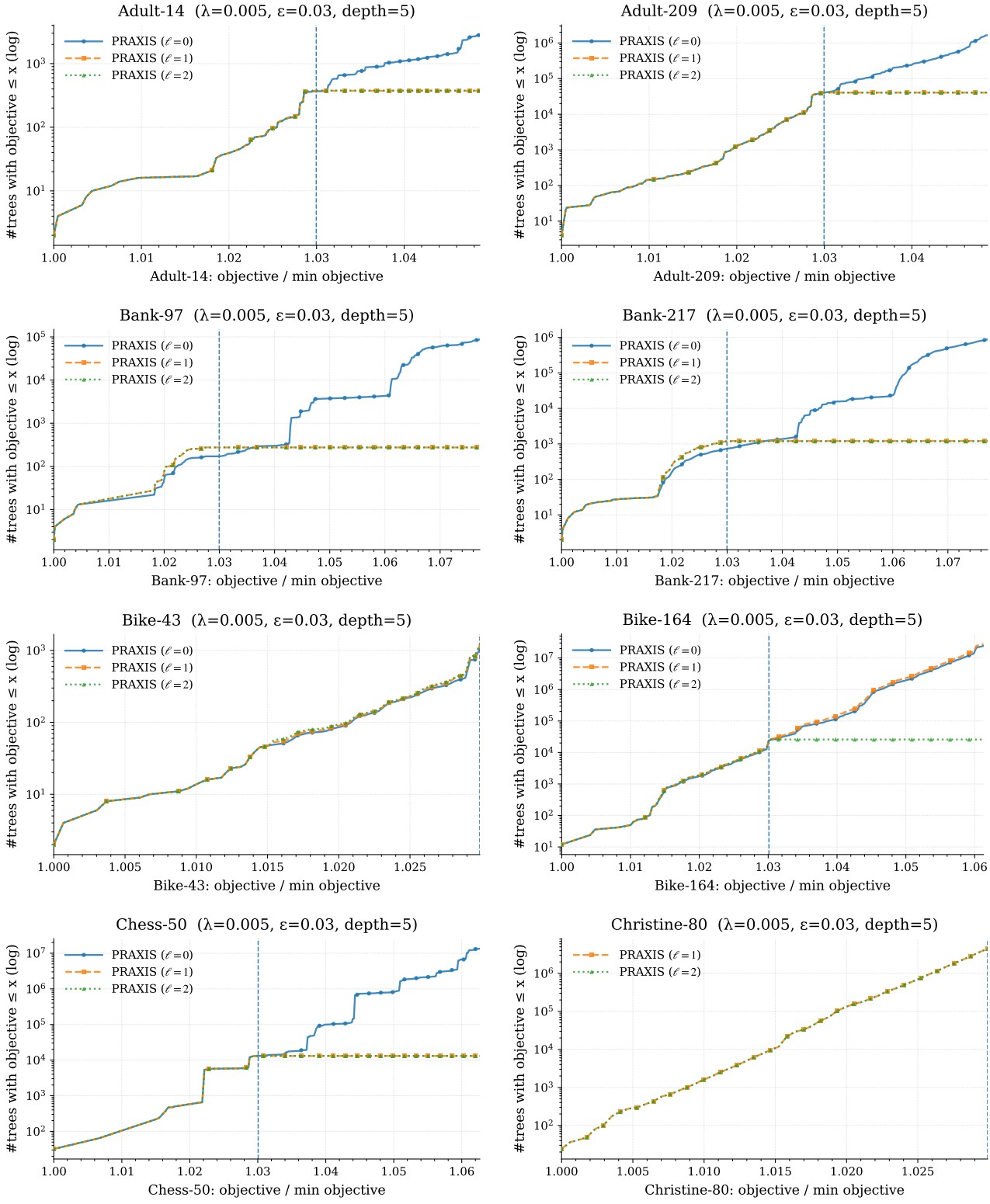

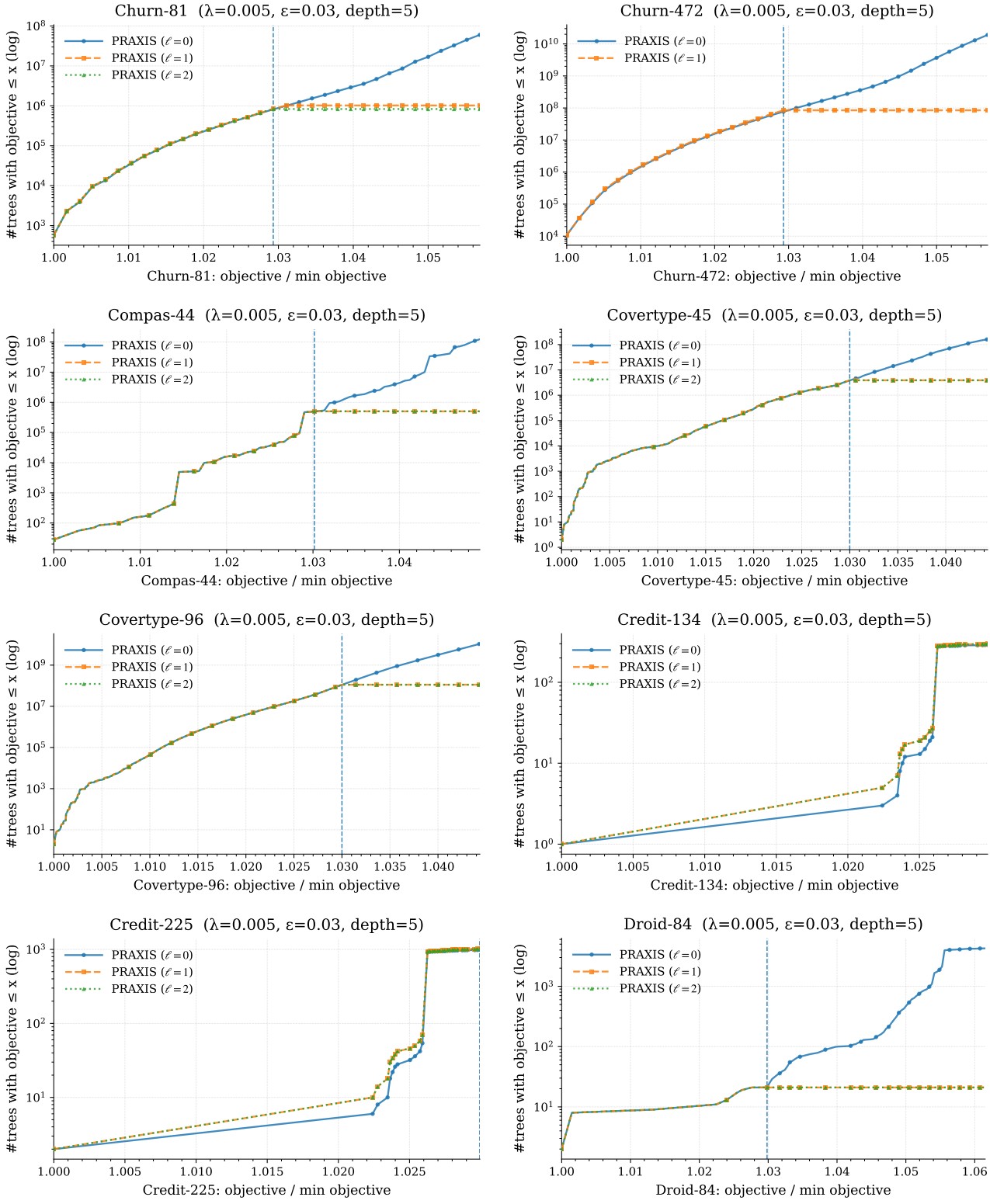

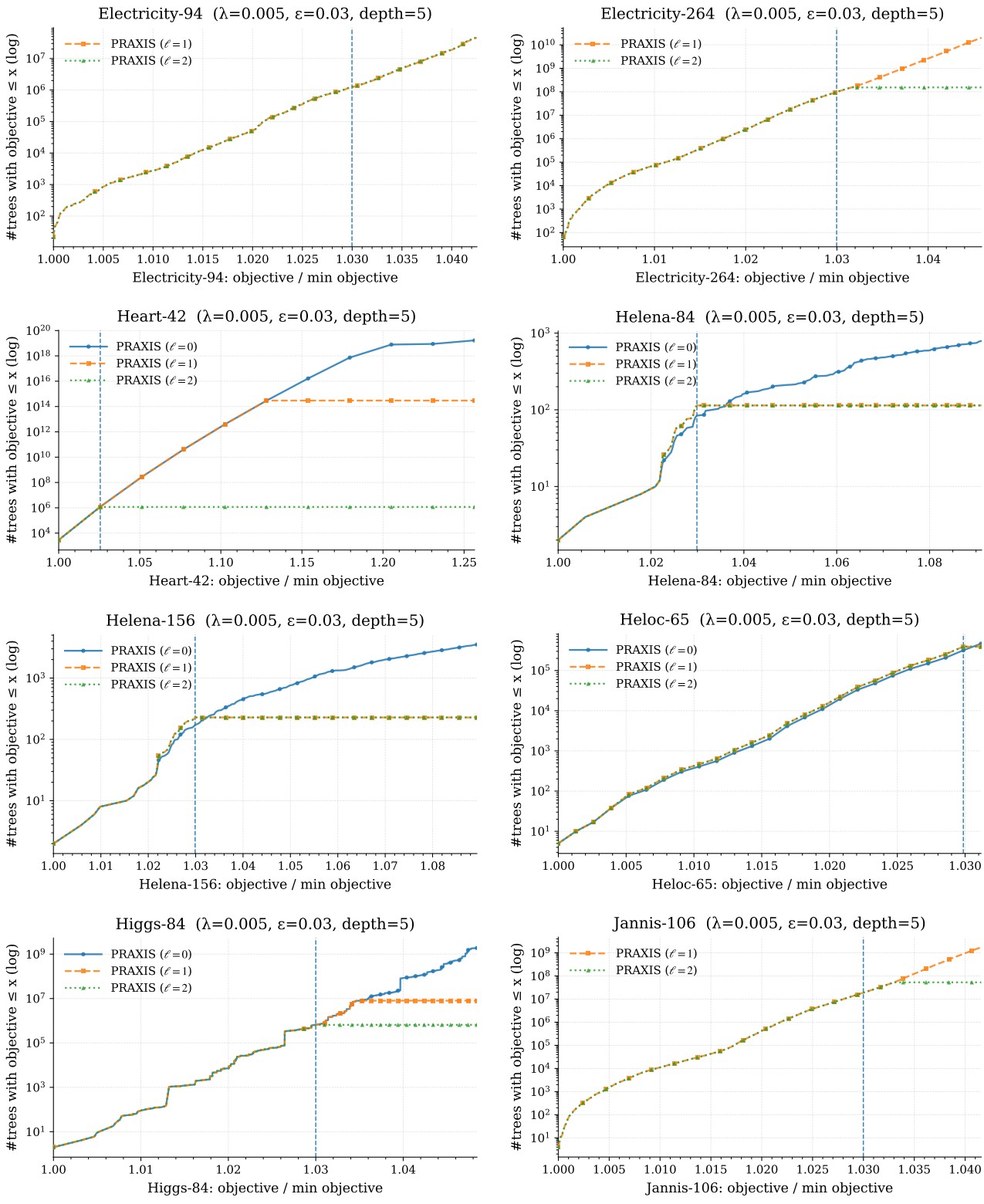

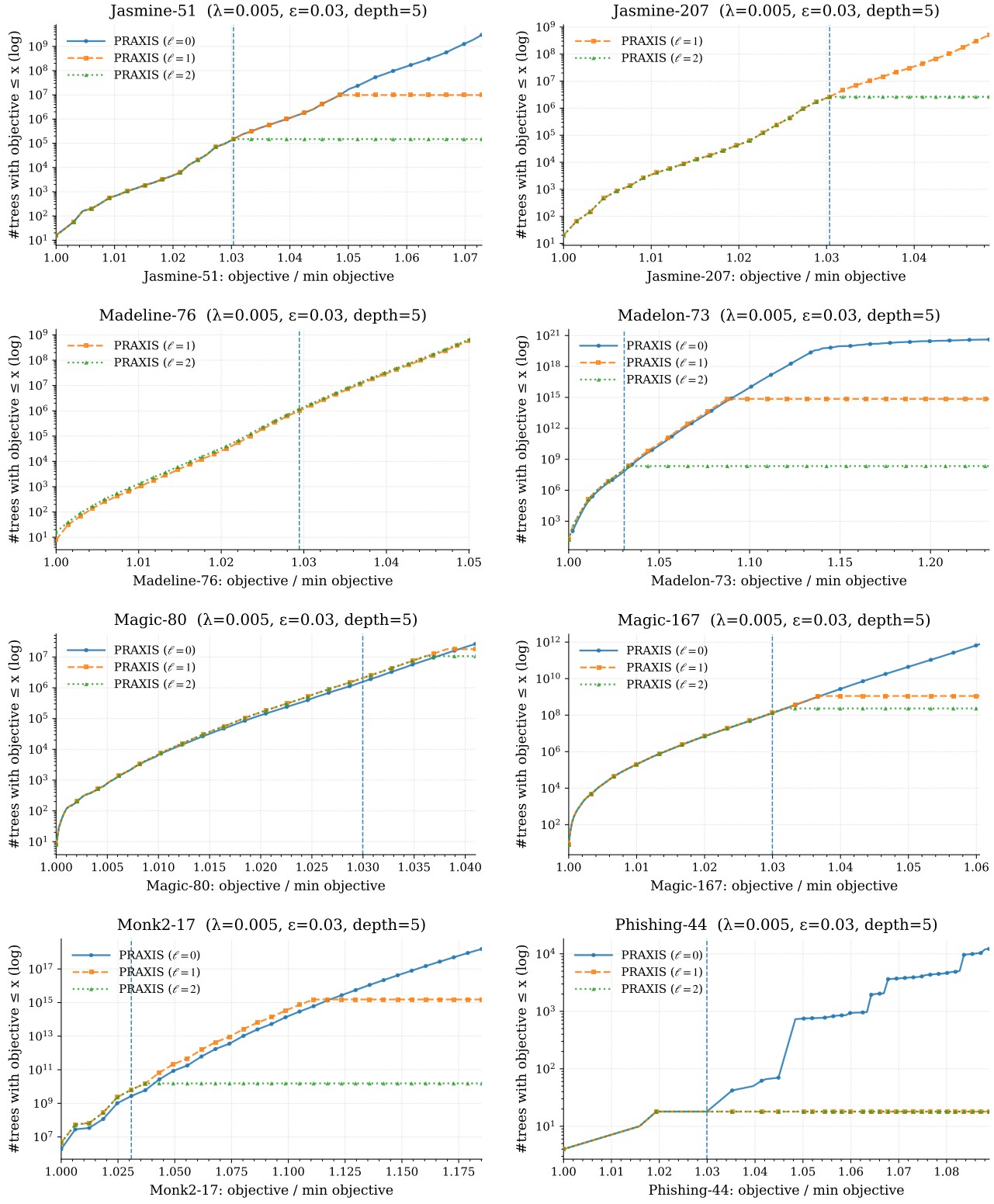

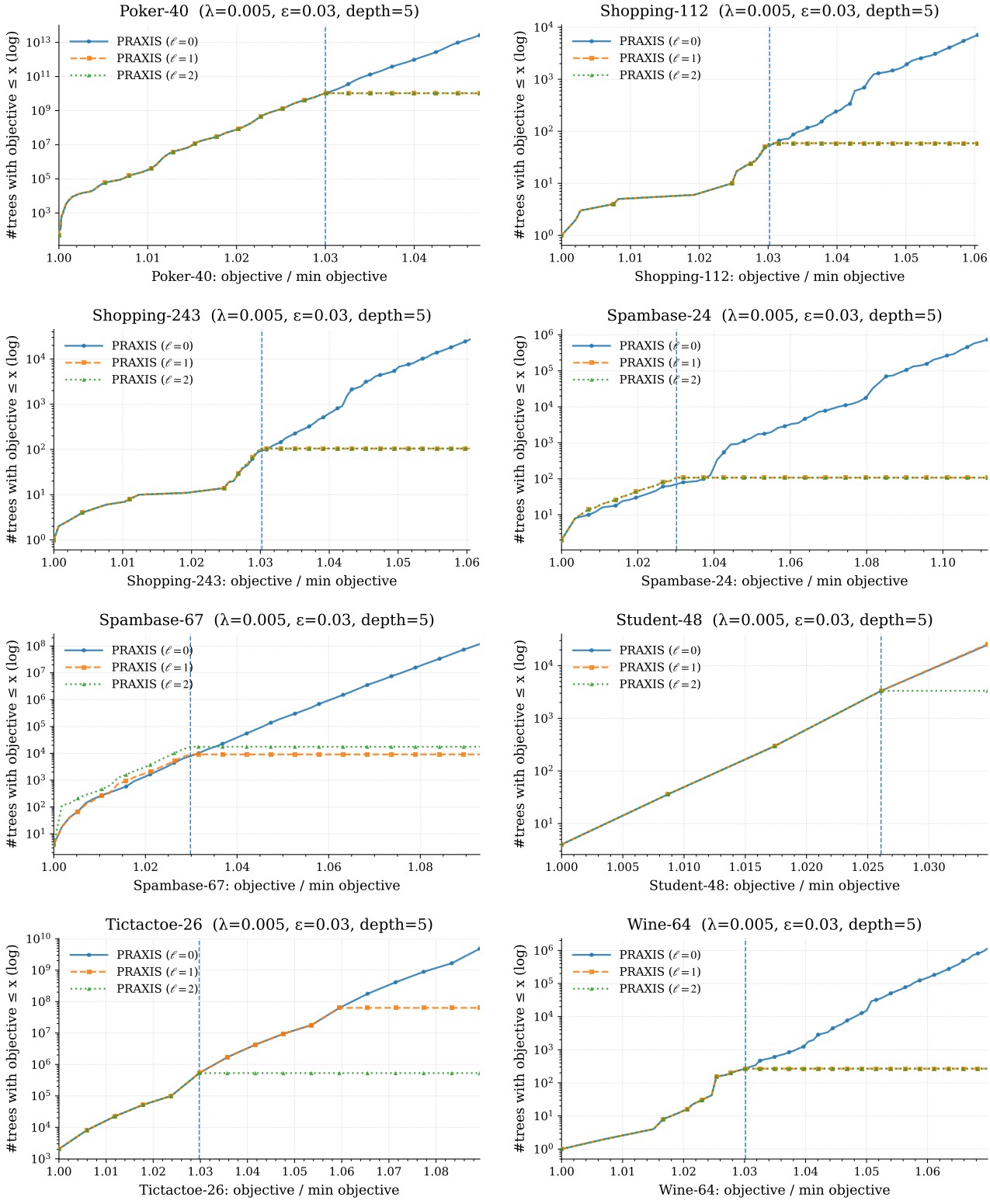

D.5.3. RECALL WITH GREEDY PROXY ALGORITHM

Table 20 shows that using the greedy proxy algorithm in PRAXIS can perform significantly worse than using modified LicketyRESPLIT, which was shown in Table 12. While the majority of datasets had recall above $0.85$, there existed datasets (Madelon and TicTacToe) where PRAXIS recovered a small fraction or none of the true Rashomon set. Additionally, some datasets (particularly at lower $\lambda$ values) were unable to complete using the greedy proxy algorithm due to the suboptimality of the objectives it returned on them.

*Table 20.* Multiplicative-$\varepsilon_{\text{mult}}$ (Mult) Rashomon recall across $\lambda \in \{0.02, 0.01, 0.005, 0.0025\}$ (depth $= 5$, $\varepsilon_{\text{mult}} = 0.03$) using the greedy proxy algorithm. Values are mean $\pm$ std over 10 bootstraps. Missing (–) entries indicate no completed runs. For the Bank dataset at $\lambda = 0.0025$, no standard deviation is reported because only a single bootstrap run completed within the 200 GB memory limit.

| Dataset | $\lambda = 0.02$ | $\lambda = 0.01$ | $\lambda = 0.005$ | $\lambda = 0.0025$ |
|---|---|---|---|---|
| Adult-14 | 0.987±0.021 | 1.000±0.000 | 0.979±0.011 | 0.996±0.003 |
| Aging-57 | 1.000±0.000 | 1.000±0.000 | 0.912±0.158 | 0.967±0.057 |
| Bank-97 | 1.000±0.000 | 1.000±0.000 | 0.742±0.164 | 0.997±– |
| Bike-43 | 0.955±0.142 | 0.972±0.044 | 0.847±0.127 | 0.902±0.122 |
| Christine-80 | 0.694±0.180 | 0.933±0.099 | 0.981±0.015 | |
| Churn-81 | 1.000±0.000 | 0.946±0.083 | 0.979±0.048 | 0.982±0.051 |
| Compas-44 | 0.958±0.059 | 0.944±0.085 | 0.990±0.009 | 0.995±0.012 |
| Covertype-45 | 1.000±0.000 | 0.990±0.014 | 0.998±0.001 | 1.000±0.000 |
| Diabetes-33 | 1.000±0.000 | 1.000±0.000 | 1.000±0.000 | 0.995±0.016 |
| Droid-84 | 1.000±0.000 | 1.000±0.000 | 1.000±0.000 | 1.000±0.000 |
| Electricity-94 | 1.000±0.000 | 0.867±0.097 | – | – |
| Heart-42 | 0.874±0.214 | 0.871±0.295 | 0.863±0.304 | 0.863±0.304 |
| Helena-84 | 1.000±0.000 | 0.787±0.120 | 0.669±0.241 | 0.562±0.267 |
| Heloc-65 | 0.983±0.036 | 0.942±0.066 | 0.959±0.061 | – |
| IOT-86 | 1.000±0.000 | 1.000±0.000 | 1.000±0.000 | 1.000±0.000 |
| Jasmine-51 | 0.861±0.110 | 0.911±0.132 | 0.912±0.121 | 0.939±0.042 |
| Madeline-76 | 0.861±0.321 | – | – | – |
| Madelon-73 | 0.867±0.238 | 0.014±0.020 | 0.000±– | 0.000±0.000 |
| Magic-80 | 0.865±0.157 | 0.971±0.042 | 0.948±0.043 | – |
| Monk2-17 | 1.000±0.000 | 0.612±0.443 | 0.813±0.277 | 0.783±0.359 |
| Mushroom-13 | 1.000±0.000 | 1.000±0.000 | 1.000±0.000 | 0.943±0.074 |
| Phishing-44 | 1.000±0.000 | 1.000±0.000 | 1.000±0.000 | 1.000±0.000 |
| Shopping-112 | 1.000±0.000 | 1.000±0.000 | 0.913±0.100 | 0.953±0.045 |
| Spambase-24 | 0.955±0.078 | 1.000±0.000 | 1.000±0.000 | 0.996±0.005 |
| Student-48 | 1.000±0.000 | 0.900±0.162 | 0.908±0.180 | 0.990±0.031 |
| Taxi-27 | 1.000±0.000 | 1.000±0.000 | 1.000±0.000 | 0.975±0.050 |
| TicTacToe-26 | 0.240±0.382 | 0.324±0.393 | 0.692±0.438 | 0.730±0.434 |
| Wine-64 | 1.000±0.000 | 0.945±0.127 | 0.985±0.017 | 0.984±0.007 |

### D.5.4. PROXIES WITH FEATURE SELECTION

An immediate implication of Theorem 3.2 is that PRAXIS scales linearly in the number of binary features outside of the work performed by the proxy algorithm. While an ideal Rashomon set approximation would retain a large feature representation to fully capture the Rashomon effect, any single sparse decision tree would use a small fraction of those features. Given this, it may not be necessary for the proxy algorithm itself to operate over the entire feature set to obtain accurate subproblem estimates.

We therefore study a hybrid regime in which the AND/OR graph is constructed using a large binarization (on the order of hundreds of thresholds), while the proxy algorithm is restricted to a much smaller subset of features. In our experiments, we obtain both the small and large binarizations using Threshold Guessing (McTavish et al., 2022), which prunes binary threshold features to the minimal subset sufficient to match the predictions of a strong reference ensemble (a gradient boosted decision tree). To be precise, for the datasets where we generated two binarizations (with two different parameter sets for GBDTs), we pass the dataset with more columns into PRAXIS, but we modify our proxy algorithm (modified LicketySPLIT) to use only those features in the intersection of the two binarizations. By limiting the proxy to this reduced feature set while preserving the full binarization for search, our goal is to approximate the Rashomon set equally well but with even less of a computational burden.

In Table 21 and Table 22, we see that using feature selection in the proxy algorithm is frequently an order of magnitude faster than using all the features in the larger binarization, while rarely sacrificing in the approximation of the Rashomon set. This fact does have some nuance, however. For datasets such as Christine (with $d = 7$), the proxy solution at the root was significantly worse when using feature selection, resulting in a much longer runtime and higher memory consumption. Interestingly, on Christine with $d = 5$, the feature selection version actually yielded more trees within the bound (a consequence of the initial budget having more slack).

However, there are some risks to this approach. If the number of features in the intersection is too small (e.g., Spambase), PRAXIS with a restricted proxy will run out of memory ($d = 7$, due to a very poor budget initialization) or take longer than the full variant ($d = 5$).

We include this experiment to illustrate that PRAXIS can accommodate extremely large feature spaces (giving much better handling of continuous features, something that is not possible in any other Rashomon set work), provided the proxy is constrained in a principled way.

We additionally compute the Rashomon Importance Distribution (RID; Donnelly et al., 2023) using PRAXIS to assess the effect of binarization size. Table 23 reports, for each dataset feature, whether it receives zero or nonzero importance under the RID when using a larger versus smaller binarization in the proxy algorithm. We group all thresholds corresponding to the same continuous feature into a single importance score (through the RID binning map), while treating each one-hot encoded categorical variable as a distinct feature.

Across both datasets, the larger binarization identifies a substantial number of features that receive nonzero importance in the Rashomon set but are absent under the smaller binarization. At the same time, it also reveals that some features highlighted by the smaller binarization are in fact irrelevant to all near-optimal models. As a result, disagreements in variable importance between the RID computed on the smaller and larger binarizations comprise a substantial fraction of the features in the union of the two binarizations (22.3% for Christine and 29.9% for Jannis).

*Table 21.* Effect of feature selection on runtime, memory, and Rashomon set size. Memory is reported as peak RSS usage (Peak MB). $\lambda = 0.01, \varepsilon_{\mathrm{mult}} = 0.03, d = 5$. Trees are counted only if they are within a $1 + \varepsilon_{\mathrm{mult}}$ factor of the best tree either method found.

| | Features | | Feature Selection | | | Full Features | | |
|---|---|---|---|---|---|---|---|---|
| Dataset | FS | Full | Time (s) | Peak MB | Trees | Time (s) | Peak MB | Trees |
| Adult | 14 | 209 | 4.26 | 144.27 | 128 | 290.24 | 675.64 | 128 |
| Bank | 78 | 217 | 16.27 | 235.65 | 6 | 56.24 | 323.10 | 6 |
| Bike | 42 | 164 | 11.16 | 209.24 | 594 | 67.22 | 571.40 | 594 |
| Covertype | 45 | 96 | 233.35 | 579.30 | 1,592 | 748.10 | 1,301.23 | 1,592 |
| Christine | 61 | 231 | 1,291.73 | 11,056.51 | 110,338 | 8,670.74 | 137,812.63 | 108,301 |
| Churn | 48 | 472 | 29.99 | 505.58 | 18,756 | 1,294.29 | 5,493.56 | 19,461 |
| Credit | 86 | 225 | 11.92 | 189.42 | 2 | 37.13 | 263.52 | 2 |
| Electricity | 76 | 264 | 4,721.27 | 17,737.12 | 240,818 | 28,839.68 | 77,920.45 | 240,818 |
| Helena | 32 | 156 | 5.38 | 249.57 | 74 | 60.02 | 361.47 | 74 |
| Jannis | 45 | 247 | 5,548.03 | 8,489.09 | 502,754 | 29,538.39 | 120,010.49 | 503,636 |
| Jasmine | 40 | 207 | 16.02 | 449.68 | 310 | 253.28 | 4,734.81 | 310 |
| Madeline | 41 | 451 | – | – | – | – | – | – |
| Madelon | 43 | 186 | – | – | – | 8,075.54 | 183,972.04 | 423,264 |
| Magic | 54 | 167 | 264.80 | 1,287.62 | 630,949 | 1,554.83 | 10,724.28 | 755,584 |
| Shopping | 74 | 243 | 20.74 | 259.70 | 15 | 116.08 | 675.98 | 15 |
| Spambase | 10 | 67 | 13.33 | 2,374.39 | 3,911 | 2.93 | 209.73 | 4,901 |

*Table 22.* Effect of feature selection on runtime, memory, and Rashomon set size. Memory is reported as peak RSS usage (Peak MB). $\lambda = 0.0075, \varepsilon_{\mathrm{mult}} = 0.01, d = 7$. Trees are counted only if they are within a $1 + \varepsilon_{\mathrm{mult}}$ factor of the best tree either method found.

| | Features | | Feature Selection | | | Full Features | | |
|---|---|---|---|---|---|---|---|---|
| Dataset | FS | Full | Time (s) | Peak MB | Trees | Time (s) | Peak MB | Trees |
| Adult | 14 | 209 | 3.64 | 144.27 | 22 | 438.45 | 491.71 | 22 |
| Bank | 78 | 217 | 19.75 | 246.61 | 1 | 110.42 | 302.35 | 1 |
| Bike | 42 | 164 | 80.66 | 649.51 | 264 | 671.85 | 1,542.75 | 264 |
| Covertype | 45 | 96 | 171.48 | 579.30 | 114 | 1,083.31 | 1,301.23 | 114 |
| Christine | 61 | 231 | 6,570.90 | 57,737.20 | 893 | 7,088.70 | 21,987.82 | 893 |
| Churn | 48 | 472 | 23.54 | 432.37 | 4,800 | 919.79 | 1,810.98 | 4,800 |
| Credit | 86 | 225 | 35.90 | 238.98 | 2 | 170.45 | 325.19 | 2 |
| Electricity | 76 | 264 | 2,996.05 | 5,528.33 | 140,517 | 22,176.67 | 12,875.72 | 140,517 |
| Helena | 32 | 156 | 3.07 | 242.51 | 8 | 64.56 | 309.66 | 8 |
| Jannis | 45 | 247 | 84,690.03 | 133,152.96 | 3,553,370 | – | – | – |
| Jasmine | 40 | 207 | 109.08 | 2,259.58 | 76 | 1,400.28 | 7,145.45 | 76 |
| Madeline | 41 | 451 | – | – | – | – | – | – |
| Madelon | 43 | 186 | – | – | – | – | – | – |
| Magic | 54 | 167 | 227.91 | 1,432.90 | 11,547 | 1,578.62 | 3,224.09 | 11,547 |
| Shopping | 74 | 243 | 28.07 | 337.92 | 7 | 227.76 | 762.61 | 7 |
| Spambase | 10 | 67 | – | – | – | 2.72 | 163.99 | 216 |

*Table 23.* Feature-level agreement between binarization thresholds

**Christine** (157 features (union of both binarization))

| | **Smaller Binarization** | |
|---|---|---|
| **Larger Binarization** | Not Important | Yes Important |
| Not Important | 86 | 24 |
| Yes Important | 11 | 36 |
| *Agreement metrics* | | |
| Disagreement | 35 (22.3%) | |
| Both nonzero | 36 features | |
| Jaccard similarity | 0.507 | |

**Jannis** (127 features (union of both binarization))

| | **Smaller Binarization** | |
|---|---|---|
| **Larger Binarization** | Not Important | Yes Important |
| Not Important | 69 | 34 |
| Yes Important | 4 | 20 |
| *Agreement metrics* | | |
| Disagreement | 38 (29.9%) | |
| Both nonzero | 20 features | |
| Jaccard similarity | 0.345 | |

## D.6. Extracting smaller Rashomon sets

In subsection D.4, we showed that PRAXIS can recover exactly optimal trees when the budget initialization is set with $\varepsilon_{\text{mult}} = 0$ nearly all of the time, or, in every case tested, when one sets $\varepsilon_{\text{mult}} = 0.03$. Here, we show that a larger $\varepsilon_{\text{mult}}$ will contain a great approximation of the Rashomon set corresponding to a smaller $\varepsilon_{\text{mult}}$. In particular, we run PRAXIS with $\varepsilon_{\text{mult}} = 0.03$ and evaluate the recall of the Rashomon set characterized by $\varepsilon_{\text{mult}} = 0.01$ across a range of $\lambda$ values.

*Table 24.* Running PRAXIS with $\varepsilon_{\text{mult}} = 0.03$ but evaluating recall for the Rashomon set characterized by $\varepsilon_{\text{mult}} = 0.01$.

| | $\lambda = 0.02$ | $\lambda = 0.01$ | $\lambda = 0.005$ | $\lambda = 0.0025$ |
|---|---|---|---|---|
| Dataset | Subset Recall | Subset Recall | Subset Recall | Subset Recall |
| Adult-14 | 1.000±0.000 | 1.000±0.000 | 1.000±0.000 | 1.000±0.000 |
| Aging-57 | 1.000±0.000 | 1.000±0.000 | 1.000±0.000 | 1.000±0.000 |
| Bank-97 | 1.000±0.000 | 1.000±0.000 | 1.000±0.000 | 1.000±0.000 |
| Bike-43 | 1.000±0.000 | 1.000±0.000 | 1.000±0.000 | 1.000±0.000 |
| Christine-80 | 1.000±0.000 | 1.000±0.000 | 1.000±0.000 | 1.000±0.000 |
| Churn-81 | 1.000±0.000 | 1.000±0.000 | 0.998±0.008 | 0.998±0.007 |
| Compas-44 | 1.000±0.000 | 1.000±0.000 | 1.000±0.000 | 1.000±0.000 |
| Covertype-45 | 1.000±0.000 | 1.000±0.000 | 1.000±0.000 | 1.000±0.000 |
| Diabetes-33 | 1.000±0.000 | 1.000±0.000 | 1.000±0.000 | 1.000±0.000 |
| Droid-84 | 1.000±0.000 | 1.000±0.000 | 1.000±0.000 | 1.000±0.000 |
| Electricity-94 | 1.000±0.000 | 0.993±0.015 | 0.981±0.045 | 0.990±0.018 |
| Heart-42 | 1.000±0.000 | 0.999±0.004 | 0.900±0.316 | 0.900±0.316 |
| Helena-84 | 1.000±0.000 | 1.000±0.000 | 1.000±0.000 | 1.000±0.000 |
| Heloc-65 | 1.000±0.000 | 1.000±0.000 | 1.000±0.000 | 1.000±0.000 |
| IOT-86 | 1.000±0.000 | 1.000±0.000 | 1.000±0.000 | 1.000±0.000 |
| Jasmine-51 | 1.000±0.000 | 1.000±0.000 | 1.000±0.000 | 0.996±0.013 |
| Madeline-76 | 1.000±0.000 | 1.000±0.000 | 1.000±0.000 | 0.890±0.272 |
| Madelon-73 | 1.000±0.000 | 1.000±0.000 | 1.000±0.000 | 0.998±0.005 |
| Magic-80 | 1.000±0.000 | 1.000±0.000 | 0.999±0.002 | 1.000±0.000 |
| Monk2-17 | 1.000±0.000 | 0.967±0.104 | 0.997±0.010 | 0.871±0.203 |
| Mushroom-13 | 1.000±0.000 | 1.000±0.000 | 1.000±0.000 | 1.000±0.000 |
| Phishing-44 | 1.000±0.000 | 1.000±0.000 | 1.000±0.000 | 0.950±0.158 |
| Shopping-112 | 1.000±0.000 | 1.000±0.000 | 1.000±0.000 | 1.000±0.000 |
| Spambase-24 | 1.000±0.000 | 1.000±0.000 | 1.000±0.000 | 1.000±0.000 |
| Student-48 | 1.000±0.000 | 1.000±0.000 | 0.992±0.027 | 0.991±0.028 |
| Taxi-27 | 1.000±0.000 | 1.000±0.000 | 1.000±0.000 | 1.000±0.000 |
| TicTacToe-26 | 1.000±0.000 | 1.000±0.000 | 0.986±0.043 | 1.000±0.000 |
| Wine-64 | 1.000±0.000 | 1.000±0.000 | 1.000±0.000 | 1.000±0.000 |

## D.7. Allowing Non-Majority Leaf Predictions

All existing Rashomon set algorithms enumerate decision trees under the restriction that each leaf predicts the majority class of the samples it contains. While this convention simplifies enumeration, it excludes trees that are still within the Rashomon bound.

PRAXIS removes this restriction by constructing a more flexible AND/OR graph representation that supports enumeration and extraction of trees with any feasible leaf predictions (for a finite number of classes), provided that the resulting tree remains within the specified objective budget. This additional flexibility is important for downstream analyses that depend on the full diversity of feasible models, such as the Rashomon Importance Distribution (Donnelly et al., 2023).

Table 25 compares the size of the Rashomon set returned by PRAXIS when restricting leaves to majority-class predictions versus allowing any leaf prediction that satisfies the budget. Across datasets, permitting non-majority leaf predictions expands the Rashomon set by up to 40%.

Importantly, enabling non-majority leaf predictions leaves runtime essentially unchanged, because no additional subproblems are explored, since alternative leaf predictions are evaluated only at already-discovered subproblems. Moreover, any non-majority leaf prediction is guaranteed to have an objective no better than the majority-class prediction at that node, so allowing these predictions cannot introduce new splits or alter the structure of the AND/OR search graph.

We note that all comparisons with existing methods are conducted using the majority-leaf-only setting to ensure fairness, though PRAXIS will allow non-majority leaves by default.

| Dataset | Trees (Majority Leaves) | Trees (Any Leaves) | Ratio |
|---|---|---|---|
| Adult-14 | 374 | 388 | 1.037 |
| Adult-209 | 40,445 | 46,707 | 1.155 |
| Aging-57 | 58 | 58 | 1.000 |
| Bank-97 | 275 | 275 | 1.000 |
| Bank-217 | 1,209 | 1,209 | 1.000 |
| Bike-43 | 1,185 | 1,216 | 1.026 |
| Bike-164 | 28,051,413 | 33,375,316 | 1.190 |
| Chess-50 | 13,060 | 13,844 | 1.060 |
| Christine-80 | 4,455,530 | 5,023,805 | 1.128 |
| Churn-81 | 1,022,472 | 1,022,472 | 1.000 |
| Churn-472 | 84,800,256 | 84,800,256 | 1.000 |
| Compas-44 | 502,296 | 631,411 | 1.257 |
| Covertype-45 | 3,887,020 | 4,277,705 | 1.101 |
| Covertype-96 | 111,933,980 | 118,746,673 | 1.061 |
| Credit-134 | 299 | 337 | 1.127 |
| Credit-225 | 1,018 | 1,166 | 1.145 |
| Diabetes-33 | 1 | 1 | 1.000 |
| Diabetes-121 | 1 | 1 | 1.000 |
| Droid-84 | 21 | 21 | 1.000 |
| Electricity-94 | 45,425,460 | 50,308,970 | 1.108 |
| Electricity-264 | 20,085,242,492 | 22,739,283,641 | 1.132 |
| Heart-42 | 295,467,637,808,347 | 325,114,015,524,404 | 1.100 |
| Helena-84 | 114 | 114 | 1.000 |
| Helena-156 | 228 | 228 | 1.000 |
| Heloc-65 | 392,314 | 459,669 | 1.172 |
| Higgs-84 | 8,021,852 | 8,607,868 | 1.073 |
| IOT-86 | 1 | 1 | 1.000 |
| Jannis-106 | 1,749,341,033 | 2,088,723,863 | 1.194 |
| Jasmine-51 | 9,813,787 | 12,282,077 | 1.252 |
| Jasmine-207 | 513,101,630 | 624,323,888 | 1.217 |
| Madeline-76 | 596,704,313 | 627,765,128 | 1.052 |
| Madelon-73 | 715,653,012,747,000 | 1,012,247,433,898,600 | **1.414** |
| Magic-80 | 17,957,431 | 19,585,383 | 1.091 |
| Magic-167 | 1,120,242,116 | 1,198,581,794 | 1.070 |
| Monk2-17 | 1,531,085,513,715,984 | 1,933,476,627,234,832 | 1.263 |
| Mushroom-13 | 18 | 18 | 1.000 |
| Phishing-44 | 18 | 18 | 1.000 |
| Poker-40 | 10,470,399,019 | 14,767,465,903 | **1.410** |
| Shopping-112 | 59 | 59 | 1.000 |
| Shopping-243 | 105 | 105 | 1.000 |
| Spambase-24 | 108 | 108 | 1.000 |
| Spambase-67 | 9,048 | 9,048 | 1.000 |
| Student-48 | 25,784 | 26,996 | 1.047 |
| Taxi-27 | 3 | 3 | 1.000 |
| TicTacToe-26 | 63,198,720 | 64,643,200 | 1.023 |
| Wine-64 | 268 | 300 | 1.119 |

*Table 25.* Effect of majority-leaf-only restriction on Rashomon set size.

## D.8. Enumerating the full set of Rule Lists

We now compare PRAXIS with the rule-list variant from Theorem A.12, which guarantees that the full Rashomon set of rule lists is contained as a subset of the returned decision trees when the proxy is at least as good as a majority leaf prediction.

We run PRAXIS with a greedy proxy using

$$\lambda = 0.01, \quad \varepsilon_{\text{mult}} = 0.1, \quad d = 5,$$

and record the resulting budget. We then rerun the algorithm in rule-list mode. There are 4 different rule list variants we consider deploying: for one, we could either use a majority leaf proxy algorithm or a greedy tree algorithm. Additionally, the result of Theorem A.12 does not require iterative budget refinement, so we also consider just subtracting the leaf objective for the other side when we recurse.

Table 26 reports results on eight representative datasets without iterative budget refinement. Using a greedy proxy instead of a majority-leaf predictor recovers substantially more non–rule-list decision trees. Moreover, for several datasets (Bike, Churn, and Covertype), no rule lists exist within 10% of the greedy tree objective.

Overall, the rule-list variant is not competitive with the default PRAXIS configuration. The default configuration is often up to two orders of magnitude faster, although in some datasets the difference narrows to only a few-fold. We also observe no advantage to using the greedy rule-list variant (without iterative budget refinement) over directly running PRAXIS. The

*Table 26.* Comparison of rule list enumeration under a fixed budget with $\lambda = 0.01$, $\varepsilon_{\mathrm{mult}} = 0.1$, and depth $d = 5$. Budgets are defined relative to the greedy tree objective. The final two columns report the ratio of the number of trees enumerated by each rule list variant relative to PRAXIS .

| Dataset | Greedy Rule List | | Leaf Rule List | | PRAXIS (Greedy) | | Tree Ratio vs. PRAXIS | |
|---|---|---|---|---|---|---|---|---|
| | Trees | Time (s) | Trees | Time (s) | Trees | Time (s) | Greedy RL | Leaf RL |
| Adult-14 | 103,481 | 1.606 | 498 | 0.914 | 108,844 | 0.321 | 0.951 | 0.005 |
| Aging-57 | 5,768 | 72.892 | 5,768 | 67.056 | 5,768 | 0.095 | 1.000 | 1.000 |
| Bike-43 | 21,493,743 | 138.512 | 0 | 32.365 | 23,485,289 | 13.986 | 0.915 | 0.000 |
| Chess-50 | 6,867,464 | 322.358 | 0 | 89.991 | 6,883,172 | 7.528 | 0.998 | 0.000 |
| Churn-81 | 25,248,640 | 995.635 | 2,691,075 | 766.393 | 25,917,923 | 18.556 | 0.974 | 0.104 |
| Compas-44 | 11,456,202,024 | 115.098 | 1,066,678,125 | 50.406 | 11,926,699,093 | 62.703 | 0.961 | 0.089 |
| Covertype-45 | 271,299,290 | 5,025.532 | 0 | 3,393.202 | 279,368,958 | 2,686.531 | 0.971 | 0.000 |
| Helena-84 | 26,968 | 824.717 | 1,898 | 638.739 | 30,079 | 16.277 | 0.897 | 0.063 |

*Table 27.* Rule list enumeration with iterative budget refinement (RLIBR) under a fixed budget with $\lambda = 0.01$, $\varepsilon_{\mathrm{mult}} = 0.1$, and depth $d = 5$. Budgets are defined relative to the greedy tree objective. The final two columns report the ratio of the number of trees enumerated by RLIBR relative to PRAXIS (Greedy).

| Dataset | RLIBR Greedy | | RLIBR Leaf | | PRAXIS (Greedy) | | Tree Ratio vs. PRAXIS | |
|---|---|---|---|---|---|---|---|---|
| | Trees | Time (s) | Trees | Time (s) | Trees | Time (s) | Greedy | Leaf |
| Adult-14 | 109,147 | 1.772 | 109,079 | 1.751 | 108,844 | 0.321 | 1.003 | 1.002 |
| Aging-57 | 5,768 | 76.568 | 5,768 | 68.720 | 5,768 | 0.095 | 1.000 | 1.000 |
| Bike-43 | 23,644,937 | 188.044 | 23,321,274 | 79.218 | 23,485,289 | 13.986 | 1.007 | 0.993 |
| Chess-50 | 6,883,708 | 327.658 | 6,847,876 | 143.788 | 6,883,172 | 7.528 | 1.000 | 0.995 |
| Churn-81 | 26,094,667 | 1,122.606 | 26,062,803 | 1,161.384 | 25,917,923 | 18.556 | 1.007 | 1.006 |
| Compas-44 | 11,926,719,436 | 133.500 | 11,866,219,859 | 133.909 | 11,926,699,093 | 62.703 | 1.000 | 0.996 |
| Covertype-45 | 279,650,349 | 5,971.026 | 271,114,007 | 4,607.987 | 279,368,958 | 2,686.531 | 1.001 | 0.970 |
| Helena-84 | 31,622 | 863.677 | 31,622 | 656.378 | 30,079 | 16.277 | 1.051 | 1.051 |

larger number of trees returned by PRAXIS arises from its iterative budget refinement. Performance improves with iterative budget refinement (see Table 27), which increases the number of trees returned, but at the cost of additional runtime.

### D.9. Depth 7 Rashomon Sets

Even for approximate algorithms, Rashomon set computation becomes increasingly challenging as depth grows, since the search space expands exponentially. Despite this, PRAXIS remains practical at depths where the approximation offered by RESPLIT struggles. For example, on Magic with 167 binary features, PRAXIS completes in under 7 hours, whereas RESPLIT was projected to require over 150 days before timing out.

More broadly, PRAXIS enables depth-7 Rashomon sets to be approximated efficiently across a wide range of datasets. For instance, Bank with 97 binary features completes in just over 2 minutes (compared to nearly 70 hours for RESPLIT), while Churn with 472 binary features finishes in under 2 hours, where RESPLIT fails to complete at all. Similar behavior is observed across many other datasets.

Beyond being up to four orders of magnitude more efficient than RESPLIT in both runtime and memory, PRAXIS also yields substantially better Rashomon set approximations. Table 28, on 100% of datasets with a sufficient number of features (and where both methods finish), PRAXIS returned more trees within the estimated Rashomon bound. RESPLIT frequently returns zero trees within the bounds, even when hundreds of thousands of feasible trees exist and were found by PRAXIS .

We additionally ran SORTeD on each of the datasets shown in the tables for 148 hours. Many datasets ran out of time; the full list is shown in Table 30. This list includes Rashomon sets that PRAXIS approximated in 156 seconds (Bike), 386 seconds (Covertype), 55 seconds (Credit), 11 seconds (Droid) and 54 seconds (Helena).

*Table 28.* Number of trees within the shared Rashomon bound ($\lambda = 0.003$, $\varepsilon_{\text{mult}} = 0.01$, depth $= 7$). The bound is computed as $(1 + \varepsilon_{\text{mult}}) \cdot$ min objective across PRAXIS and RESPLIT for each dataset. We display only the datasets where both methods finished, and for datasets with at least 40 binary features (to allow the algorithm to build out a reasonably full tree, as depth 7 trees have up to 127 splits).

| Dataset | PRAXIS Count | RESPLIT Count | RESPLIT / PRAXIS |
|---|---|---|---|
| Aging-57 | **1** | **1** | 1.00 |
| Bank-97 | **237** | **237** | 1.00 |
| Christine-80 | **97,383** | 0 | 0.00 |
| Churn-81 | **444,984** | 0 | 0.00 |
| Compas-44 | **11,777** | 1,058 | 0.09 |
| Covertype-45 | **651,749** | 82,042 | 0.13 |
| Droid-84 | **20** | 4 | 0.20 |
| Electricity-94 | **338,384** | 0 | 0.00 |
| Helena-84 | **90** | **90** | 1.00 |
| Heloc-65 | **8,955** | 58 | 0.01 |
| IOT-86 | **1** | **1** | 1.00 |
| Jasmine-51 | **6,306** | 352 | 0.06 |
| Jasmine-207 | **251,240** | 8,328 | 0.03 |
| Magic-80 | **332,550** | 0 | 0.00 |
| Phishing-44 | **44** | 32 | 0.73 |
| Shopping-112 | **118** | 46 | 0.39 |
| Spambase-67 | **383** | 0 | 0.00 |
| Student-48 | **56,920,256** | 0 | 0.00 |
| Wine-64 | **255** | 47 | 0.18 |

*Table 29.* Runtime and peak memory at $\lambda = 0.003$, $\varepsilon_{\text{mult}} = 0.01$, depth $= 7$.

| Dataset | $n$ | $k$ | PRAXIS Time | PRAXIS Peak MB | RESPLIT Time | RESPLIT Peak MB |
|---|---|---|---|---|---|---|
| Adult | 48842 | 14 | **0.19** | **145.69** | 18.11 | 415.54 |
| Adult | 48842 | 209 | **7461.51** | **2774.41** | – | – |
| Aging | 714 | 57 | **0.07** | **130.70** | 80.95 | 342.25 |
| Bank | 45211 | 97 | **141.28** | **290.55** | 247881.42 | 91926.96 |
| Bank | 45211 | 217 | **3535.03** | **1661.61** | – | – |
| Bike | 17379 | 43 | **2.55** | **146.74** | – | – |
| Bike | 17379 | 164 | **156.06** | **376.99** | – | – |
| Chess | 28056 | 50 | **6.12** | **159.23** | – | – |
| Christine | 5418 | 80 | **3460.69** | **19184.12** | 35727.41 | 31740.47 |
| Churn | 5000 | 81 | **7.09** | **204.37** | 20299.75 | 113299.24 |
| Churn | 5000 | 472 | **6761.06** | **15688.87** | – | – |
| Compas | 4966 | 44 | **1.15** | **147.32** | 1859.00 | 459.10 |
| Covertype | 581012 | 45 | **385.79** | **591.83** | 47552.42 | 22275.11 |
| Covertype | 581012 | 96 | **107591.75** | **24887.65** | – | – |
| Credit | 30000 | 134 | **54.93** | **218.25** | – | – |
| Credit | 30000 | 225 | **241.45** | **323.15** | – | – |
| Diabetes | 253680 | 33 | **7.87** | **288.39** | 6797.93 | 24142.36 |
| Diabetes | 253680 | 121 | **373.83** | **678.28** | – | – |
| Droid | 29332 | 84 | **11.08** | **166.18** | 17210.01 | 32359.53 |
| Electricity | 38474 | 94 | **1212.86** | **2107.44** | 26917.56 | 36508.97 |
| Helena | 65196 | 84 | **53.73** | **221.91** | 27537.87 | 103029.52 |
| Helena | 65196 | 156 | **945.48** | **585.24** | – | – |
| Heloc | 2502 | 65 | **92.73** | **1291.25** | 4953.27 | 5481.90 |
| IOT | 123117 | 86 | **0.75** | **306.32** | 5.35 | 808.59 |
| Jasmine | 2984 | 51 | **273.50** | 9100.83 | 767.51 | **2055.16** |
| Jasmine | 2984 | 207 | **1651.24** | **6359.39** | 249459.32 | 78873.87 |
| Madeline | 3140 | 76 | **20.29** | **329.56** | – | – |
| Magic | 19020 | 80 | **2607.54** | **5061.03** | 40294.21 | 17221.75 |
| Magic | 19020 | 167 | **24503.25** | **14662.51** | – | – |
| Monk2 | 601 | 17 | **0.10** | **137.16** | – | – |
| Mushroom | 8124 | 13 | **0.01** | **130.81** | 1.79 | 145.43 |
| News | 39644 | 196 | **61652.99** | **24006.25** | – | – |
| Phishing | 11055 | 44 | **0.91** | **139.52** | 968.78 | 2222.81 |
| Poker | 1025010 | 40 | **9883.75** | **990.20** | – | – |
| Shopping | 12330 | 112 | **393.77** | **662.75** | 69335.79 | 43121.48 |
| Shopping | 12330 | 243 | **11203.95** | **6963.14** | – | – |
| Spambase | 4601 | 24 | **0.17** | **130.68** | 39.61 | 377.48 |
| Spambase | 4601 | 67 | **53.20** | **433.59** | 2271.98 | 3803.36 |
| Student | 649 | 48 | **2.43** | **197.89** | 362.59 | 1410.65 |
| Taxi | 1224158 | 27 | **18.39** | **846.56** | 13034.83 | 14111.06 |
| TicTacToe | 958 | 26 | **17.45** | **4087.46** | – | – |
| Wine | 6497 | 64 | **41.15** | **400.14** | 5735.99 | 11738.09 |

*Table 30.* PRAXIS and SORTeD Runtime at $\lambda = 0.003$, $\varepsilon_{\mathrm{mult}} = 0.01$, depth $= 7$.

| | | | Runtime | |
|---|---|---|---|---|
| Dataset | $n$ | $k$ | PRAXIS | SORTeD |
| Adult | 48842 | 14 | **0.19** | 39.53 |
| Adult | 48842 | 209 | **7461.51** | $> 532800$ |
| Aging | 714 | 57 | **0.07** | 414.39 |
| Bank | 45211 | 97 | **141.28** | $> 200$ GB |
| Bank | 45211 | 217 | **3535.03** | $> 200$ GB |
| Bike | 17379 | 43 | **2.55** | 4103.58 |
| Bike | 17379 | 164 | **156.06** | $> 532800$ |
| Chess | 28056 | 50 | **6.12** | 5841.20 |
| Christine | 5418 | 80 | **3460.69** | $> 200$ GB |
| Churn | 5000 | 81 | **7.09** | 41797.70 |
| Churn | 5000 | 472 | **6761.06** | $> 532800$ |
| Compas | 4966 | 44 | **1.15** | 1352.45 |
| Covertype | 581012 | 45 | **385.79** | $> 532800$ |
| Covertype | 581012 | 96 | **107591.75** | $> 532800$ |
| Credit | 30000 | 134 | **54.93** | $> 532800$ |
| Credit | 30000 | 225 | **241.45** | $> 532800$ |
| Diabetes | 253680 | 33 | **7.87** | 104729.81 |
| Diabetes | 253680 | 121 | **373.83** | $> 532800$ |
| Droid | 29332 | 84 | **11.08** | $> 532800$ |
| Electricity | 38474 | 94 | **1212.86** | $> 532800$ |
| Helena | 65196 | 84 | **53.73** | $> 532800$ |
| Helena | 65196 | 156 | **945.48** | $> 532800$ |
| Heloc | 2502 | 65 | **92.73** | 54344.65 |
| IOT | 123117 | 86 | **0.75** | 24.84 |
| Jasmine | 2984 | 51 | **273.50** | 7167.92 |
| Jasmine | 2984 | 207 | **1651.24** | $> 532800$ |
| Madeline | 3140 | 76 | **20.29** | 150090.54 |
| Magic | 19020 | 80 | **2607.54** | 152162.28 |
| Magic | 19020 | 167 | **24503.25** | $> 532800$ |
| Monk2 | 601 | 17 | **0.10** | 2.68 |
| Mushroom | 8124 | 13 | **0.01** | 0.96 |
| News | 39644 | 196 | **61652.99** | $> 532800$ |
| Phishing | 11055 | 44 | **0.91** | 6195.96 |
| Poker | 1025010 | 40 | **9883.75** | $> 200$ GB |
| Shopping | 12330 | 112 | **393.77** | $> 200$ GB |
| Shopping | 12330 | 243 | **11203.95** | $> 200$ GB |
| Spambase | 4601 | 24 | **0.17** | 79.16 |
| Spambase | 4601 | 67 | **53.20** | 61253.28 |
| Student | 649 | 48 | **2.43** | 1545.37 |
| Taxi | 1224158 | 27 | **18.39** | 138645.19 |
| TicTacToe | 958 | 26 | **17.45** | 137.47 |
| Wine | 6497 | 64 | **41.15** | 43200.34 |

## D.10. Results on Fully Binarized Datasets

In Table 31, we also evaluate Rashomon set computation on fully binarized versions of several datasets. For each continuous feature, we generate binary threshold features using the midpoints between consecutive unique values. This produces a fully binarized feature space. As in our earlier experiments, PRAXIS achieves near-perfect approximation quality, while requiring substantially less time than prior methods. We additionally ran TreeFARMS on these datasets, but it did not complete on any instance within the prescribed time and memory limits.

| Dataset | PRAXIS | | RESPLIT | | SORTeD |
|---|---|---|---|---|---|
| | Time | Recall | Time | Recall | Time |
| | | | | $\lambda = 0.02,\ \varepsilon = 0.03$ | |
| Bike-279 | 187.3 | 1 | 4434.1 | 0.2857 | 67188.5 |
| Compas-108 | 1.44 | 1 | 75.76 | 1 | 277.9 |
| Diabetes-185 | 106.8 | — | 7622.2 | — | — |
| Droid-85 | 0.89 | 1 | 75.5 | 1 | 933.7 |
| Heloc-1496 | 3996 | — | 98764.5 | — | — |
| Student-101 | 0.27 | 1 | 18.32 | 1 | 163.8 |
| | | | | $\lambda = 0.005,\ \varepsilon = 0.01$ | |
| Bike-279 | 938.2 | 1 | — | — | 93207.2 |
| Compas-108 | 2.24 | 1 | 79.4 | 0.0254 | 239.5 |
| Diabetes-185 | 273.4 | — | 15121.1 | — | — |
| Droid-85 | 2.36 | 1 | 136.9 | 0.125 | 737.5 |
| Heloc-1496 | 34842.1 | — | 94000.7 | — | — |
| Student-101 | 2.08 | 0.9817 | 46.86 | 0.2073 | 321.8 |
| | | | | $\lambda = 0.005,\ \varepsilon = 0.03$ | |
| Bike-279 | 2809.4 | 1 | — | — | 104518.2 |
| Compas-108 | 34.3 | 0.9999 | 830.2 | 0.0023 | 308.1 |
| Diabetes-185 | 273.6 | — | 15138.9 | — | — |
| Droid-85 | 2.64 | 1 | 139.6 | 0.0476 | 739.6 |
| Student-101 | 13.51 | 0.9992 | 786.8 | 0.1134 | 340.7 |

*Table 31.* Timing and approximation results on fully binarized continuous datasets at depth 5. Time is reported in seconds. Results are shown only for runs that completed within 144,000 seconds and 200GB RAM. TreeFARMS did not complete on any of these instances within the same resource budget.

## D.11. Approximation Quality Comparisons

| Dataset | PRAXIS Recall | | RESPLIT Recall | |
|---|---|---|---|---|
| | $\lambda{=}0.005$ | $\lambda{=}0.02$ | $\lambda{=}0.005$ | $\lambda{=}0.02$ |
| Adult-209 | 1.000±0.000 | 1.000±0.000 | 0.001±0.001 | 0.265±0.039 |
| Bank-97 | 1.000±0.000 | 1.000±0.000 | 0.511±0.202 | 1.000±0.000 |
| Bike-164 | 0.986±0.010 | 1.000±0.000 | 0.617±0.000 | 0.299±0.075 |
| Christine-231 | 1.000±0.000 | 0.994±0.011 | 0.617±0.000 | 0.435±0.091 |
| Churn-472 | 0.997±0.005 | 1.000±0.000 | 0.000±0.000 | 0.955±0.071 |
| Compas-44 | 1.000±0.000 | 1.000±0.000 | 0.007±0.007 | 0.916±0.177 |
| Covertype-96 | 1.000±0.000 | 1.000±0.000 | 0.003±0.001 | 1.000±0.000 |
| Credit-225 | 1.000±0.000 | 1.000±0.000 | 1.000±0.000 | 1.000±0.000 |
| Diabetes-33 | 1.000±0.000 | 1.000±0.000 | 1.000±0.000 | 1.000±0.000 |
| Droid-84 | 1.000±0.000 | 1.000±0.000 | 0.065±0.027 | 1.000±0.000 |
| Electricity-264 | 0.994±0.003 | 1.000±0.000 | 0.067±0.000 | 1.000±0.000 |
| Helena-156 | 1.000±0.000 | 1.000±0.000 | 0.202±0.122 | 1.000±0.000 |
| Heloc-65 | 0.999±0.000 | 1.000±0.000 | 0.001±0.002 | 0.905±0.211 |
| Jannis-106 | 0.995±0.008 | 1.000±0.000 | 0.001±0.000 | 0.232±0.021 |
| Jasmine-207 | 0.993±0.009 | 1.000±0.000 | 0.001±0.001 | 0.489±0.469 |
| Madelon-73 | 1.000±0.000 | 1.000±0.000 | 0.000±0.000 | 0.009±0.016 |
| Magic-80 | 0.997±0.004 | 0.998±0.004 | 0.001±0.001 | 0.301±0.022 |
| News-196 | 1.000±0.000 | 1.000±0.000 | 0.003±0.002 | 0.938±0.043 |
| Poker-40 | 1.000±0.000 | 1.000±0.000 | 0.111±0.000 | 1.000±0.000 |
| Shopping-243 | 0.984±0.034 | 1.000±0.000 | 0.091±0.080 | 0.987±0.030 |
| Taxi-27 | 1.000±0.000 | 1.000±0.000 | 1.000±0.000 | 1.000±0.000 |
| Wine-64 | 1.000±0.000 | 1.000±0.000 | 0.313±0.278 | 1.000±0.000 |

*Table 32.* Rashomon set recall (mean ± standard deviation over 5 bootstraps) comparing PRAXIS (PRAXIS) and RESPLIT across datasets for $\varepsilon_{\mathrm{mult}} = 0.03$ and depth $d = 5$. Format: Dataset-NumBinaryFeatures.

**Recall compared to RESPLIT.** In Table 2, we show recall results for PRAXIS across two sparsity values: $\lambda = 0.02$ and $\lambda = 0.005$. In Table 32, we additionally provide the recall results for RESPLIT. We note that RESPLIT substantially degrades in performance for smaller values of $\lambda$; this is in stark contrast to PRAXIS. We also show approximation results

for PRAXIS under an even smaller value of $\lambda$: 0.0025 in Table 33.

| Dataset | Recall |
|---|---|
| Adult-209 | 1.000±0.000 |
| Bank-97 | 1.000±0.001 |
| Bike-164 | 1.000±0.001 |
| Churn-472 | 0.995±0.009 |
| Compas-44 | 1.000±0.000 |
| Credit-225 | 1.000±0.000 |
| Diabetes-33 | 1.000±0.000 |
| Droid-84 | 0.994±0.010 |
| Helena-156 | 1.000±0.000 |
| Heloc-65 | 1.000±0.000 |
| Madelon-73 | 0.997±0.006 |
| Magic-80 | 0.999±0.001 |
| Poker-40 | 1.000±0.000 |
| Shopping-243 | 0.997±0.004 |
| Taxi-27 | 0.989±0.007 |
| Wine-64 | 0.997±0.005 |

*Table 33.* PRAXIS recall (mean $\pm$ standard deviation over 5 bootstraps) for $\lambda = 0.0025$, $\varepsilon = 0.003$. We exclude rows where the ground truth could not be enumerated.

**Qualitative Comparison of Rashomon Set Approximations.** We display results for Rashomon set approximations for $\lambda = 0.005$, $\varepsilon_{\text{mult}} = 0.01$ and $d = 5$. We show a subset of the 51 datasets and binarizations here because we exclude cases where the Rashomon set is believed to be small (on the order of $10^1$ trees) or where neither PRAXIS or RESPLIT ran. Across all figures, blue denotes PRAXIS using the modified LicketySPLIT proxy, green denotes RESPLIT, and orange denotes bootstrapped LicketySPLIT for as long as PRAXIS ran. As in Figure 3, the dashed vertical line marks the estimated Rashomon bound, defined as $(1 + \varepsilon)$ times the minimum objective found by any method. Trees returned beyond this threshold should not be counted as improving coverage of the target Rashomon set, since they have objectives outside the requested quality range. If such lower-quality trees are of interest, this should instead be reflected by requesting a larger Rashomon set, i.e., by increasing $\varepsilon$.

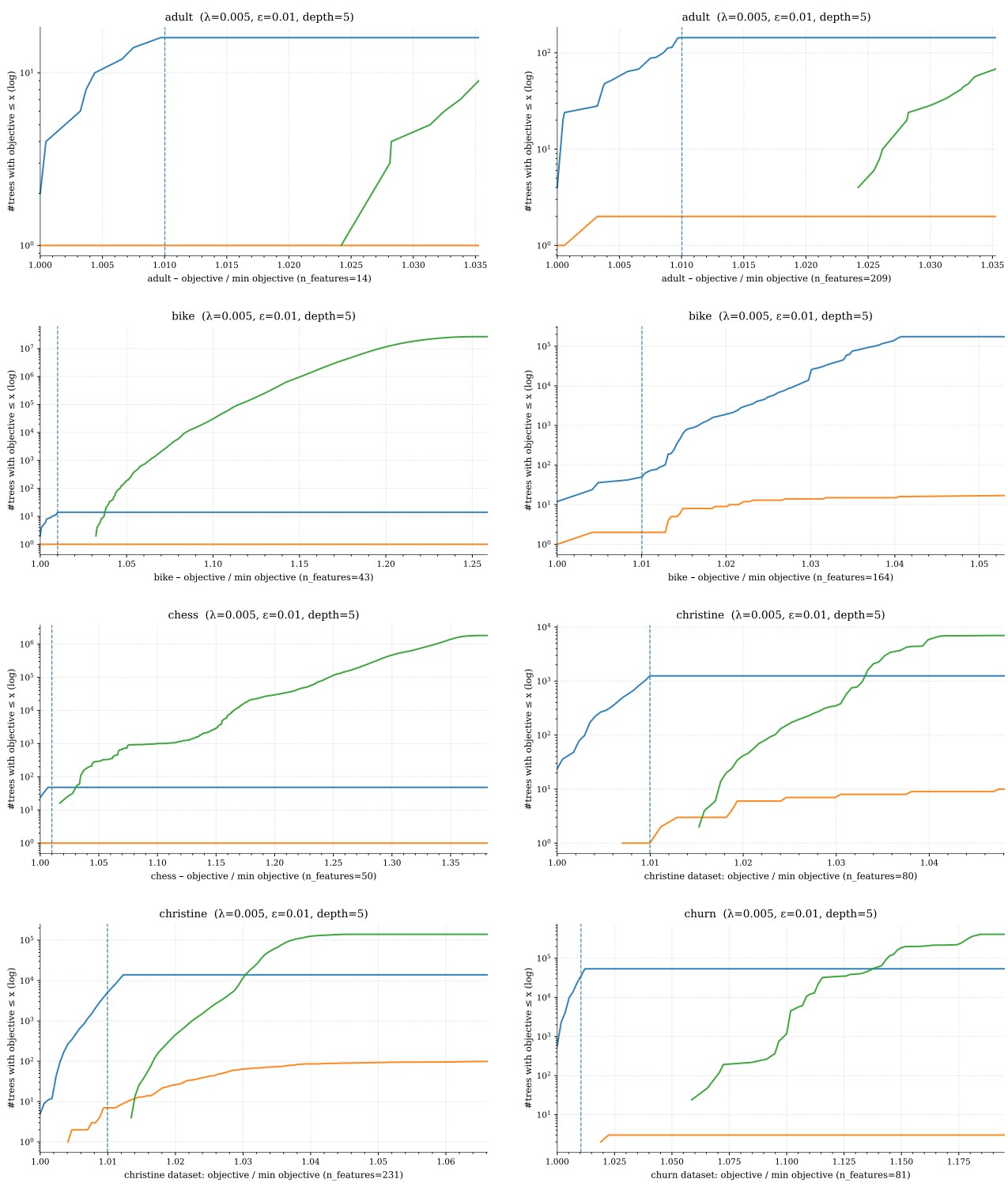

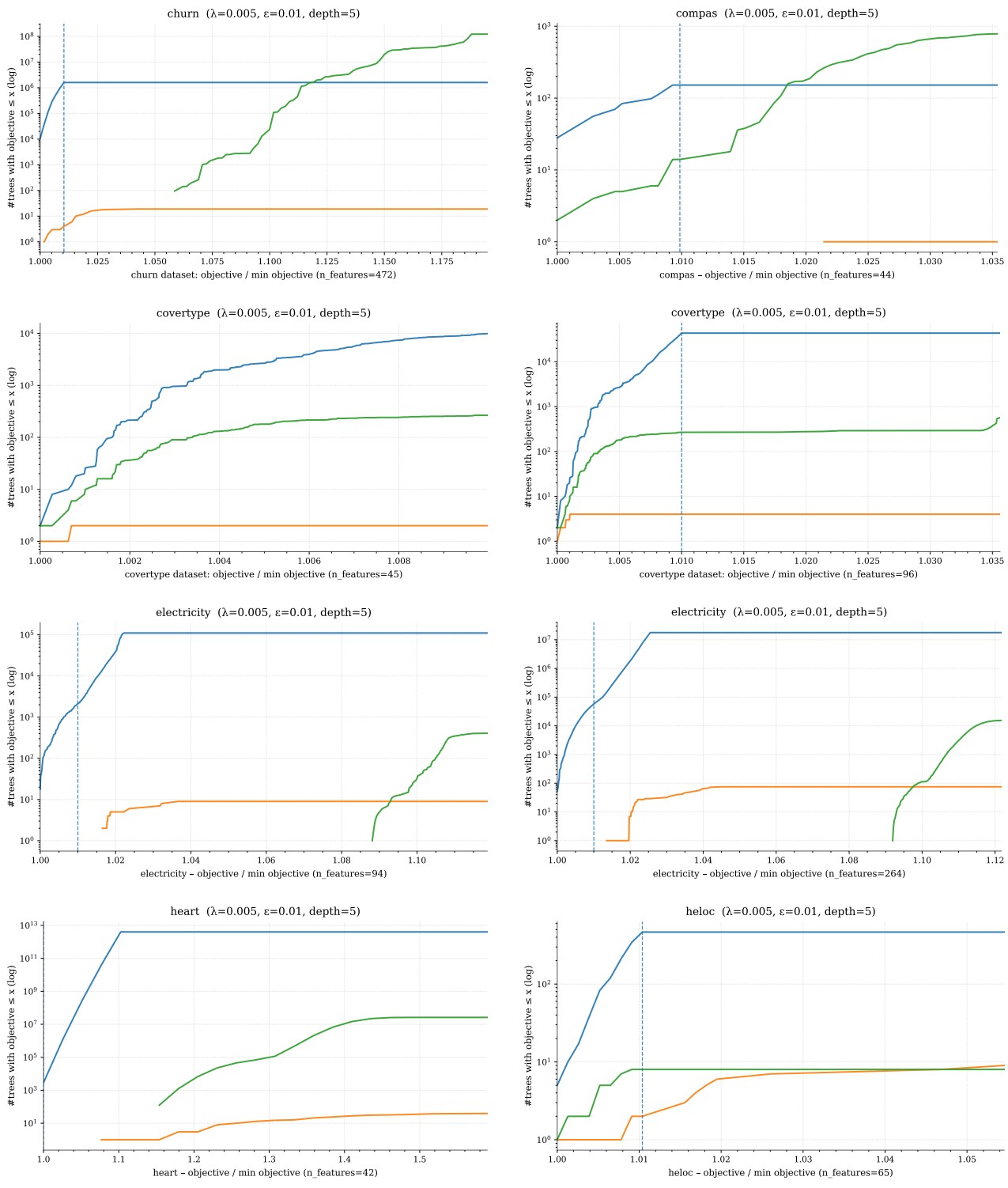

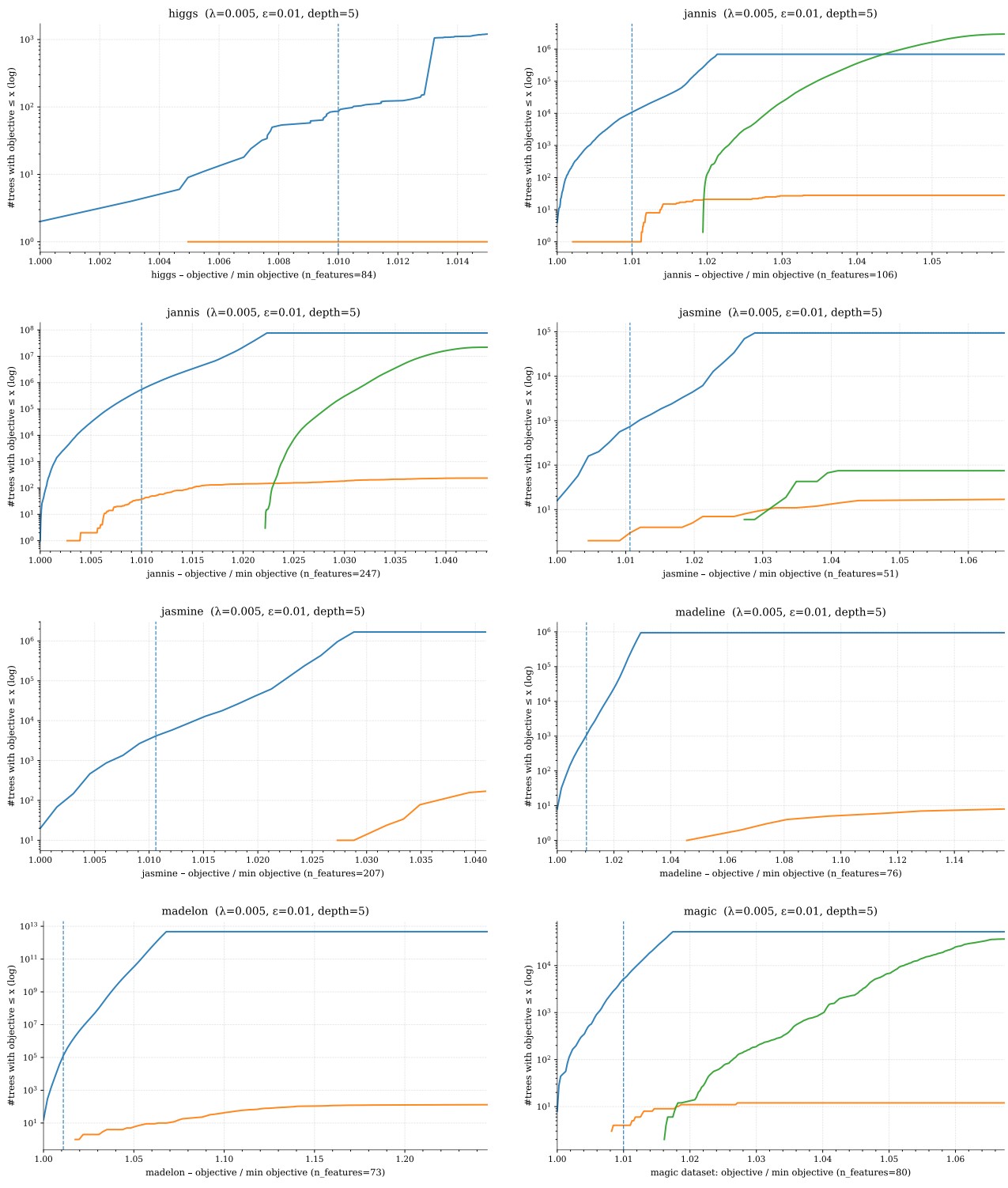

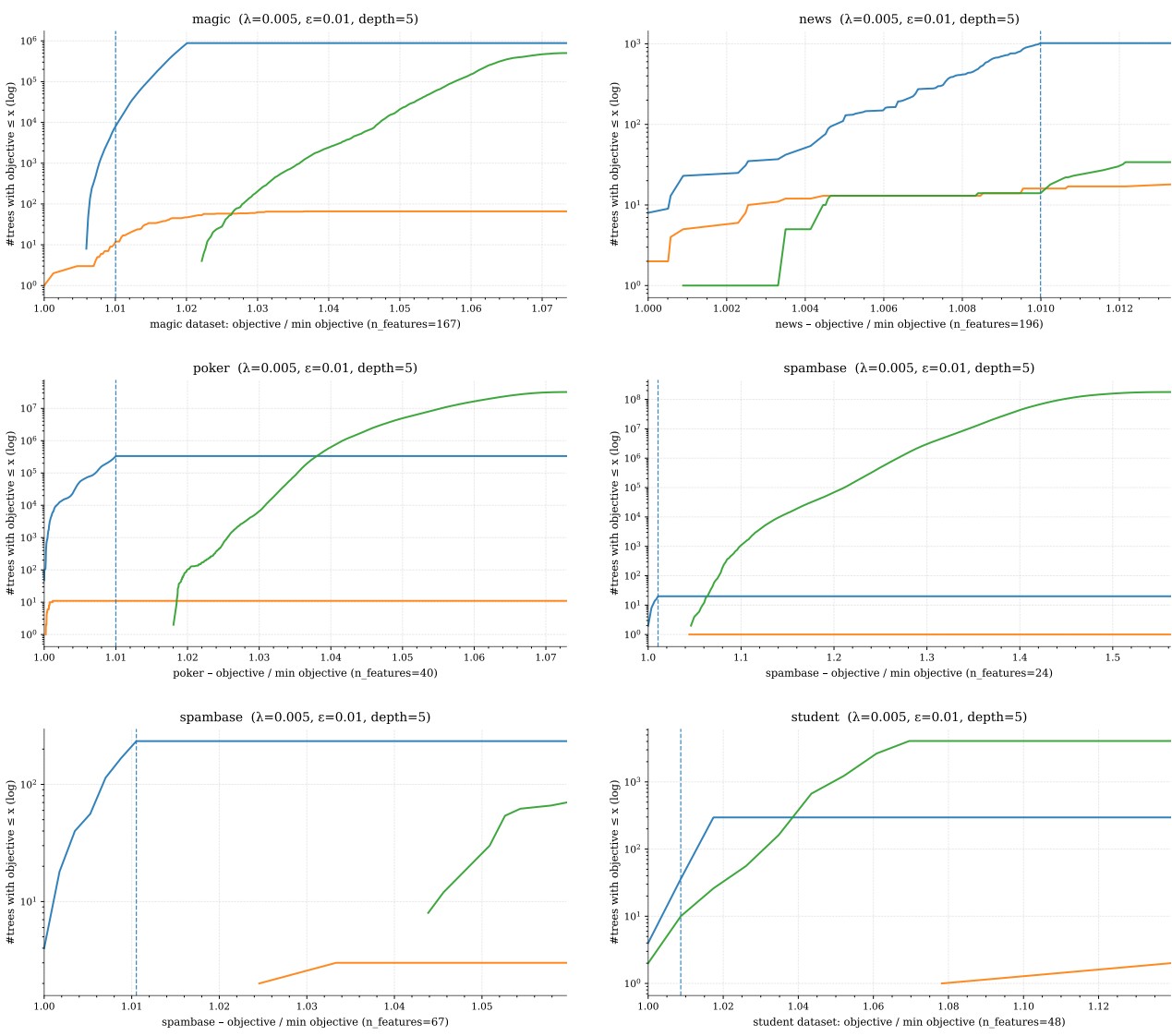

## D.12. Stopping Algorithms Early

One algorithm we compare against, SORTeD (Arslan et al., 2026), supports saving intermediate solutions. Because it enumerates the Rashomon set in sorted order of objective value, truncating this process is equivalent to computing the Rashomon set for a smaller $\varepsilon_{\text{mult}}$. However, before any solutions can be returned, SORTeD must first identify the optimal tree, which constitutes a major computational bottleneck. As shown in Figure 4, computing the optimal tree dominates the runtime; consequently, early termination still incurs most of the computational cost while recovering only a small fraction of the Rashomon set.

SORTeD Intermediate Results ($\lambda = 0.005$, $\varepsilon = 0.03$)

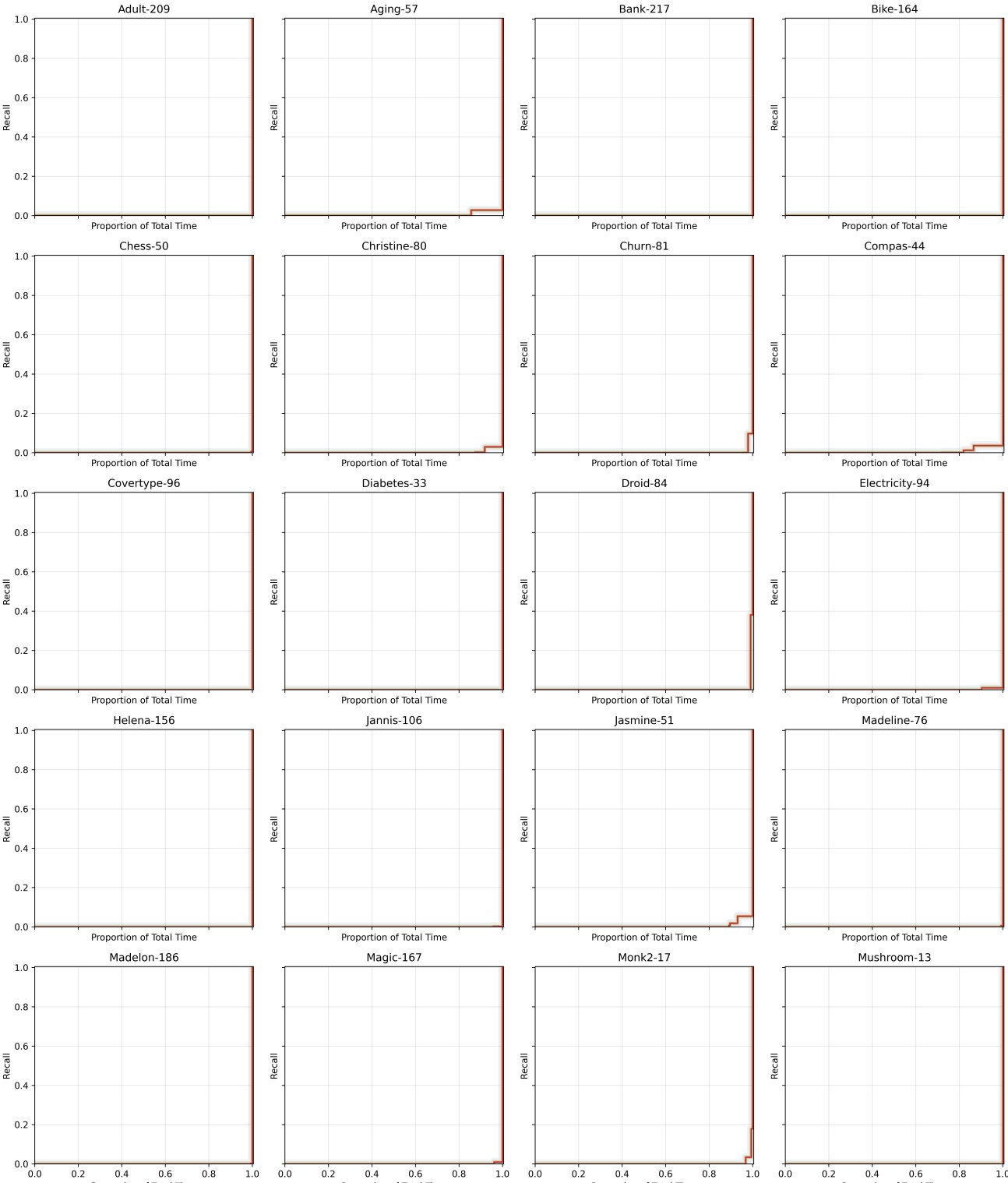

*Figure 4.* Runtime vs. fraction of the Rashomon set ($\lambda = 0.005$, $\varepsilon_{\mathrm{mult}} = 0.03$) recovered when stopping SORTeD early. The majority of time is spent computing the optimal tree, while early stopping yields only a small fraction of the set.

Similarly, PRAXIS could be run with a smaller $\varepsilon_{\mathrm{mult}}$ and then reuse all of the caches from the proxy algorithm to approximate a Rashomon set for a larger $\varepsilon_{\mathrm{mult}}$.

### D.13. Example Decision Trees

We present example decision trees discovered by PRAXIS for several datasets used in the paper. Note that the $\gamma$ (or equivalently, $\lambda$) penalty on the number of leaves encourages sparse trees rather than fully grown trees. We set $\varepsilon_{\mathrm{mult}} = 0.03$ and $\lambda = 0.005$ for these examples. The examples highlight the variety of rules and tree sizes that can arise while achieving similar objective values, illustrating the Rashomon effect in practice.

At each internal node, samples are routed to the left child when the binary feature evaluates to `True` and to the right child otherwise.

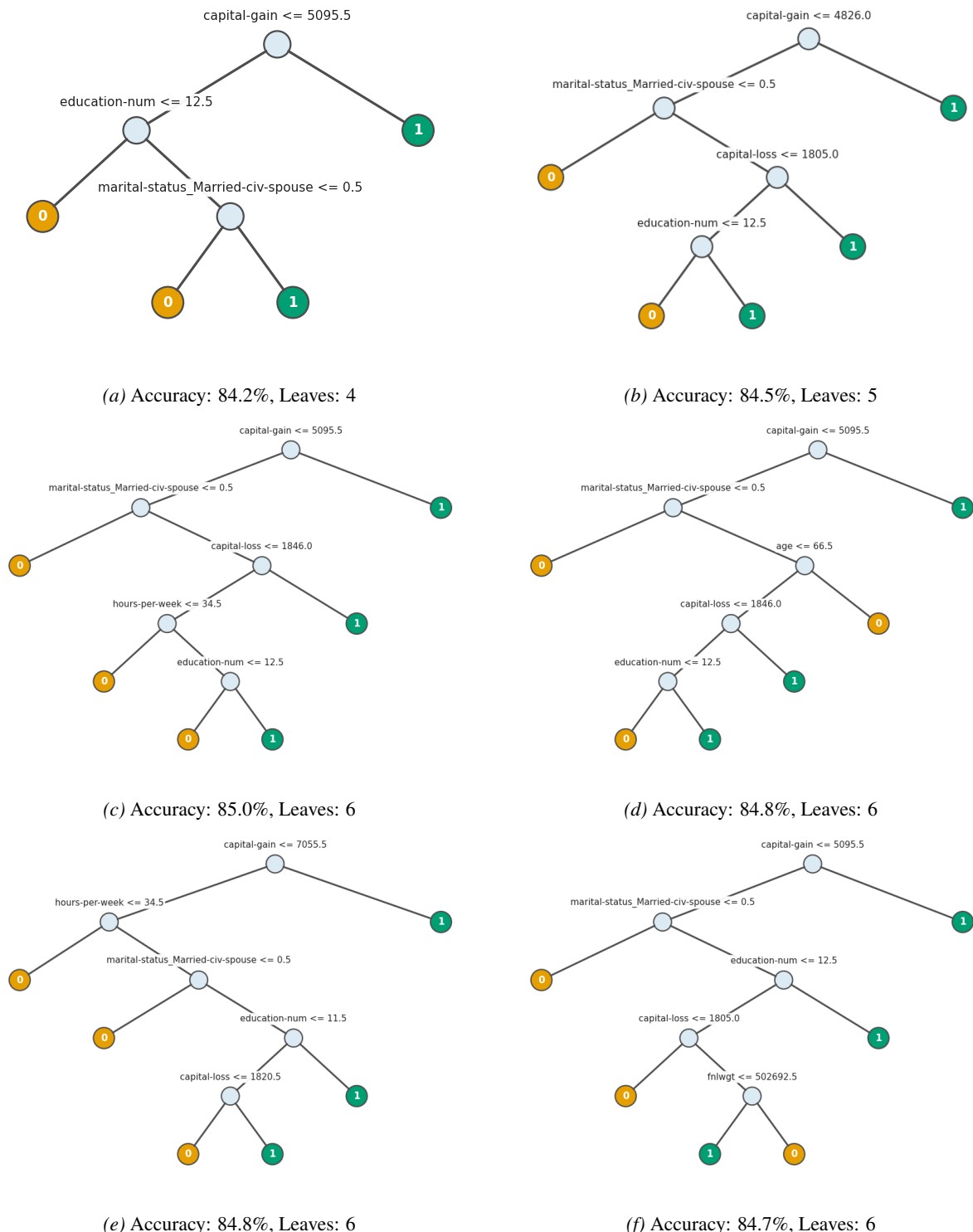

*(a)* Accuracy: 84.2%, Leaves: 4

*(b)* Accuracy: 84.5%, Leaves: 5

*(c)* Accuracy: 85.0%, Leaves: 6

*(d)* Accuracy: 84.8%, Leaves: 6

*(e)* Accuracy: 84.8%, Leaves: 6

*(f)* Accuracy: 84.7%, Leaves: 6

*Figure 5.* **Adult**. Example near-optimal trees with accuracy and number of leaves shown below each tree.

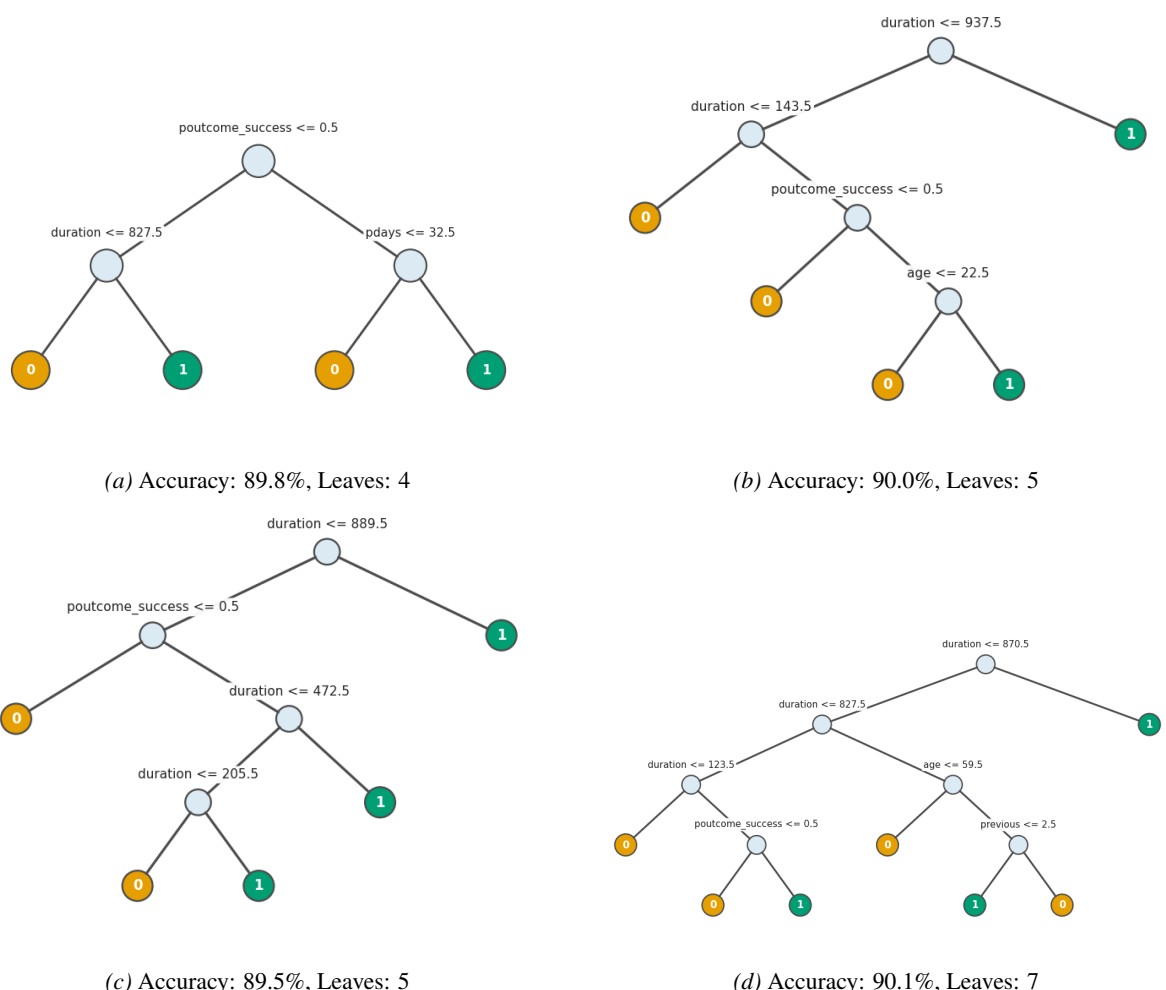

*(a)* Accuracy: 89.8%, Leaves: 4

*(b)* Accuracy: 90.0%, Leaves: 5

*(c)* Accuracy: 89.5%, Leaves: 5

*(d)* Accuracy: 90.1%, Leaves: 7

*Figure 6.* **Bank**. Example near-optimal trees with accuracy and number of leaves shown below each tree.

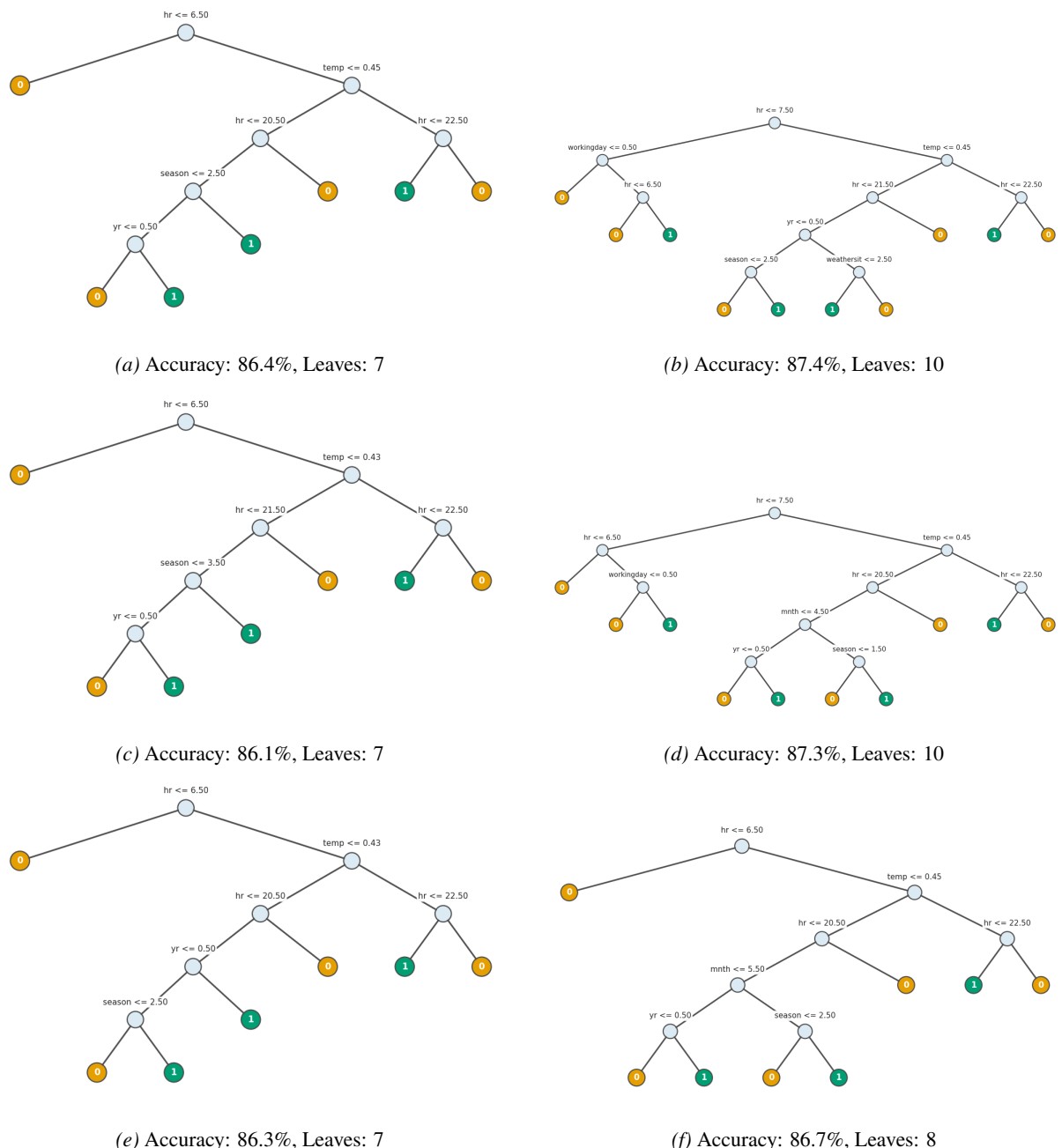

*(a)* Accuracy: 86.4%, Leaves: 7

*(b)* Accuracy: 87.4%, Leaves: 10

*(c)* Accuracy: 86.1%, Leaves: 7

*(d)* Accuracy: 87.3%, Leaves: 10

*(e)* Accuracy: 86.3%, Leaves: 7

*(f)* Accuracy: 86.7%, Leaves: 8

*Figure 7.* **Bike**. Example near-optimal trees with accuracy and number of leaves shown below each tree.

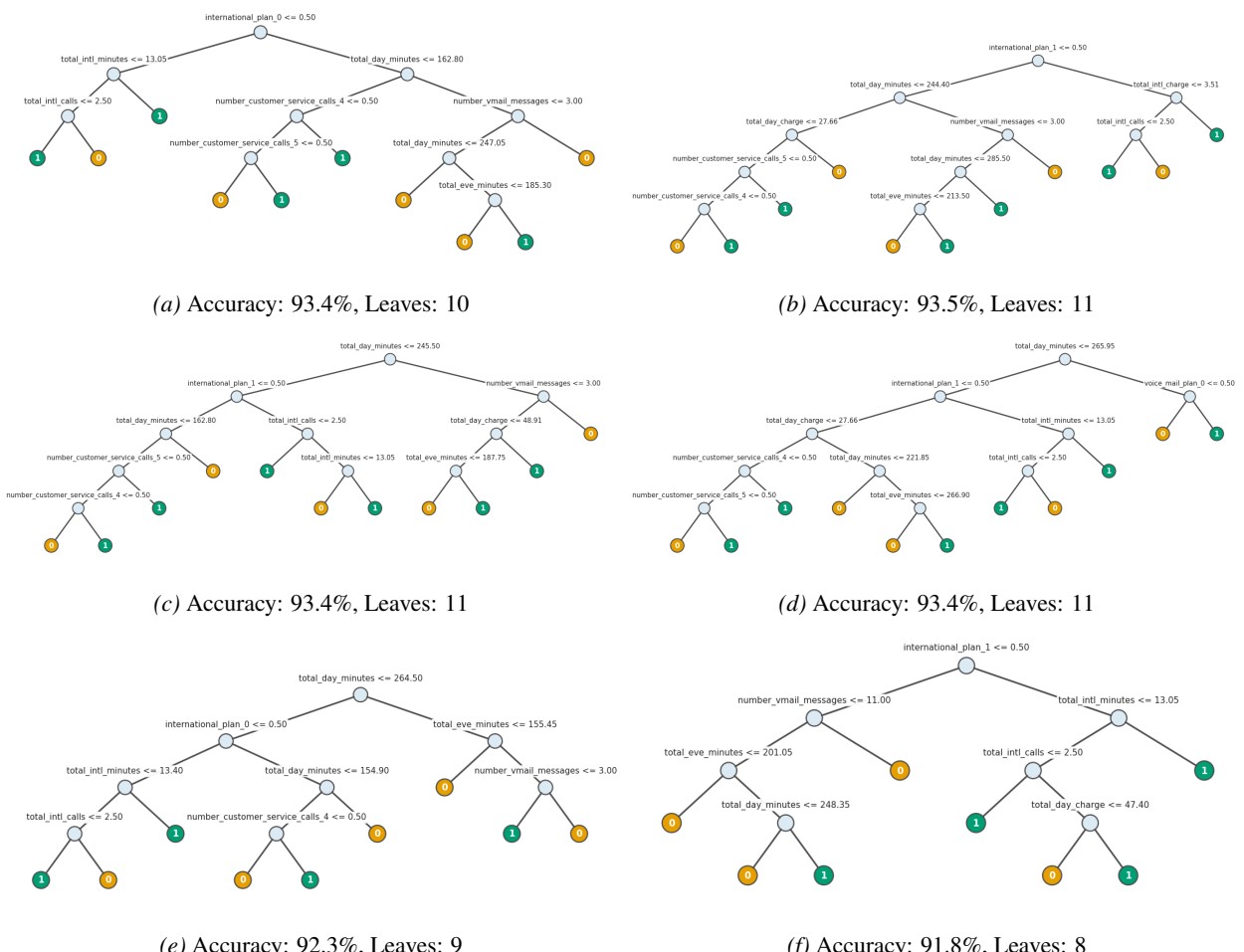

*(a)* Accuracy: 93.4%, Leaves: 10

*(b)* Accuracy: 93.5%, Leaves: 11

*(c)* Accuracy: 93.4%, Leaves: 11

*(d)* Accuracy: 93.4%, Leaves: 11

*(e)* Accuracy: 92.3%, Leaves: 9

*(f)* Accuracy: 91.8%, Leaves: 8

*Figure 8.* **Churn**. Example near-optimal trees with accuracy and number of leaves shown below each tree.

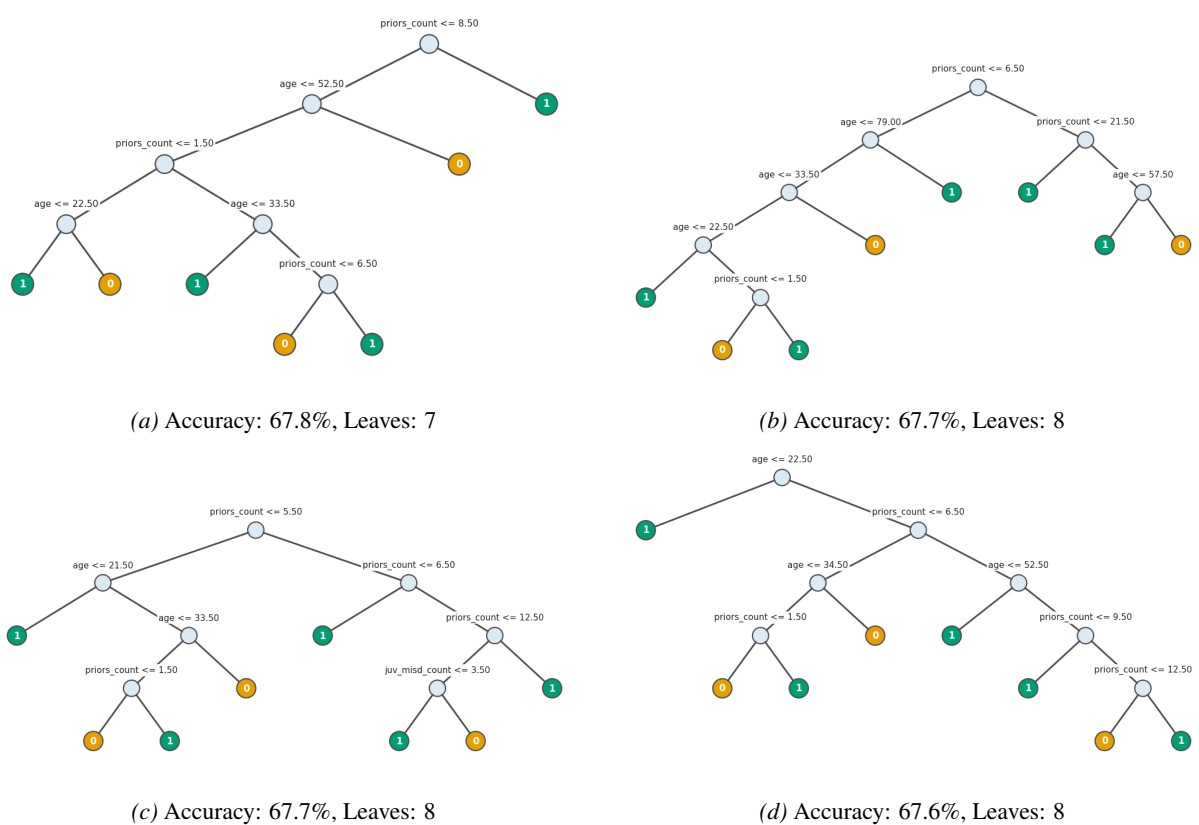

*(a)* Accuracy: 67.8%, Leaves: 7

*(b)* Accuracy: 67.7%, Leaves: 8

*(c)* Accuracy: 67.7%, Leaves: 8

*(d)* Accuracy: 67.6%, Leaves: 8

*Figure 9.* **Compas**. Example near-optimal trees with accuracy and number of leaves shown below each tree.

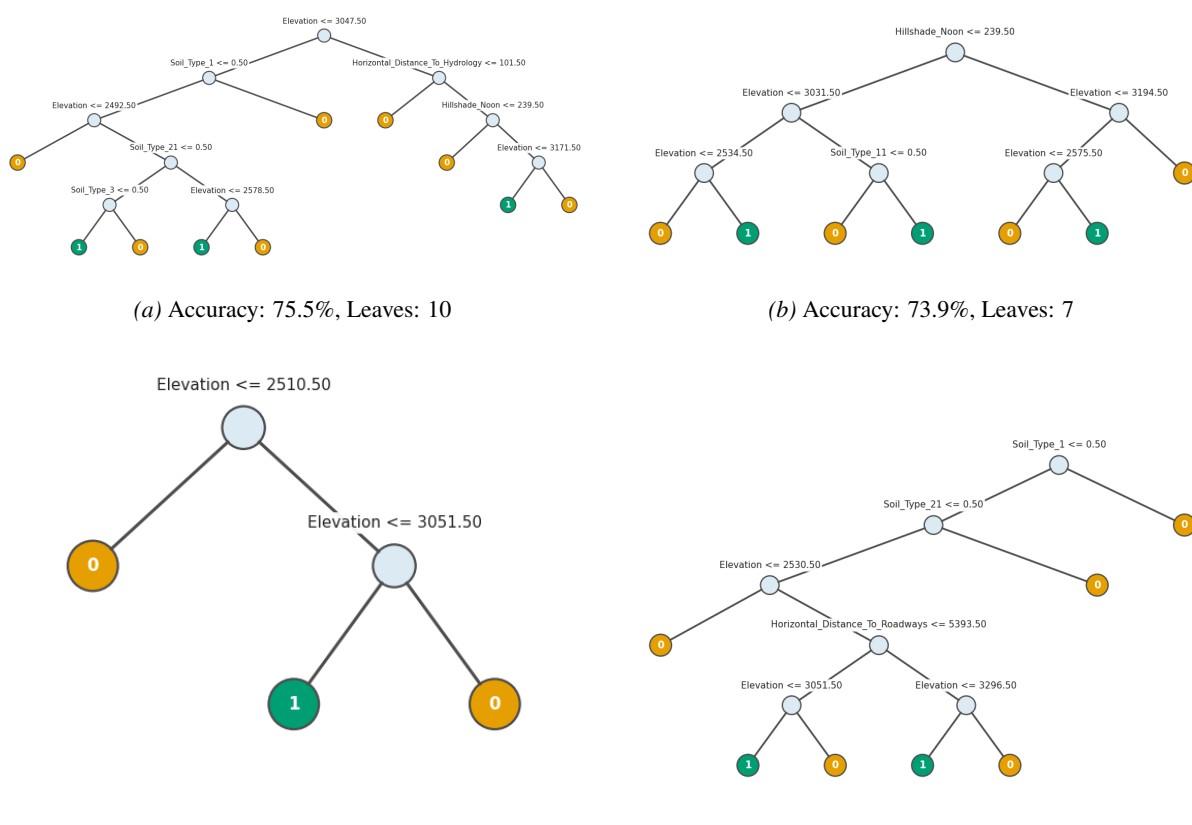

*(a)* Accuracy: 75.5%, Leaves: 10

*(b)* Accuracy: 73.9%, Leaves: 7

*(c)* Accuracy: 73.2%, Leaves: 3

*(d)* Accuracy: 74.5%, Leaves: 7

*Figure 10.* **Covertype**. Example near-optimal trees with accuracy and number of leaves shown below each tree.

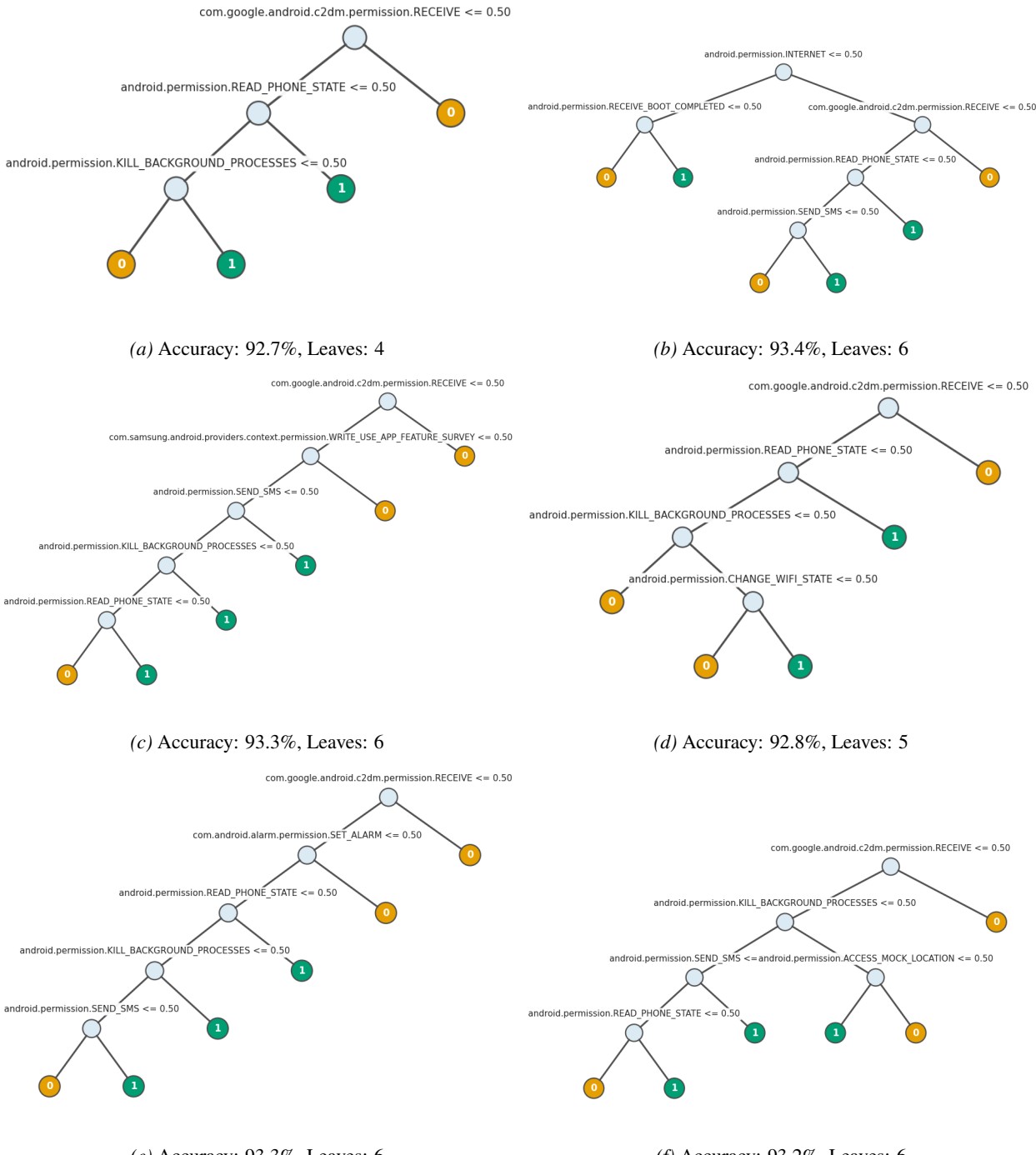

*(a)* Accuracy: 92.7%, Leaves: 4

*(b)* Accuracy: 93.4%, Leaves: 6

*(c)* Accuracy: 93.3%, Leaves: 6

*(d)* Accuracy: 92.8%, Leaves: 5

*(e)* Accuracy: 93.3%, Leaves: 6

*(f)* Accuracy: 93.2%, Leaves: 6

*Figure 11.* **Droid**. Example near-optimal trees with accuracy and number of leaves shown below each tree.

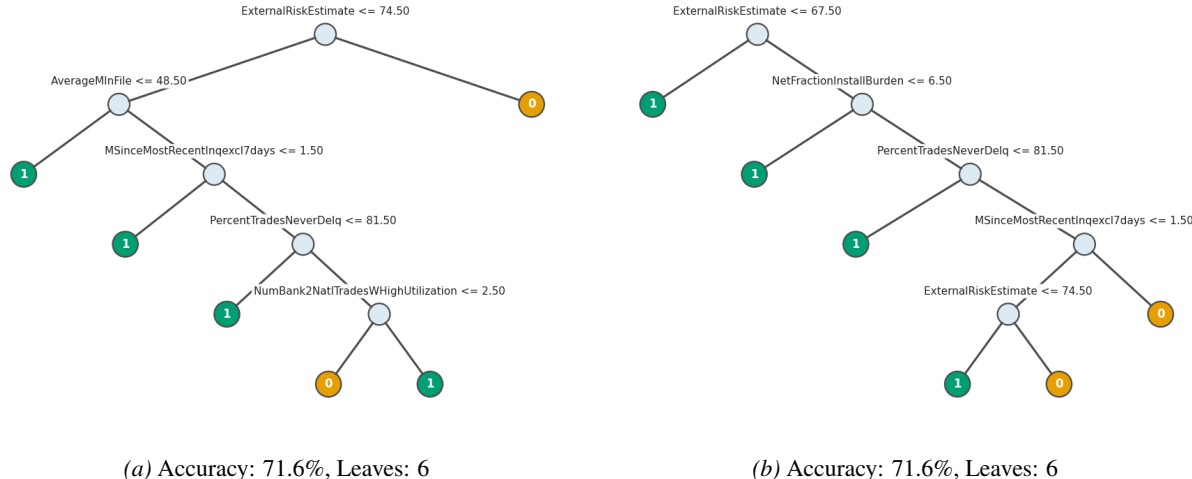

*(a)* Accuracy: 71.6%, Leaves: 6          *(b)* Accuracy: 71.6%, Leaves: 6

*Figure 12.* **Heloc**. Example near-optimal trees with accuracy and number of leaves shown below each tree.

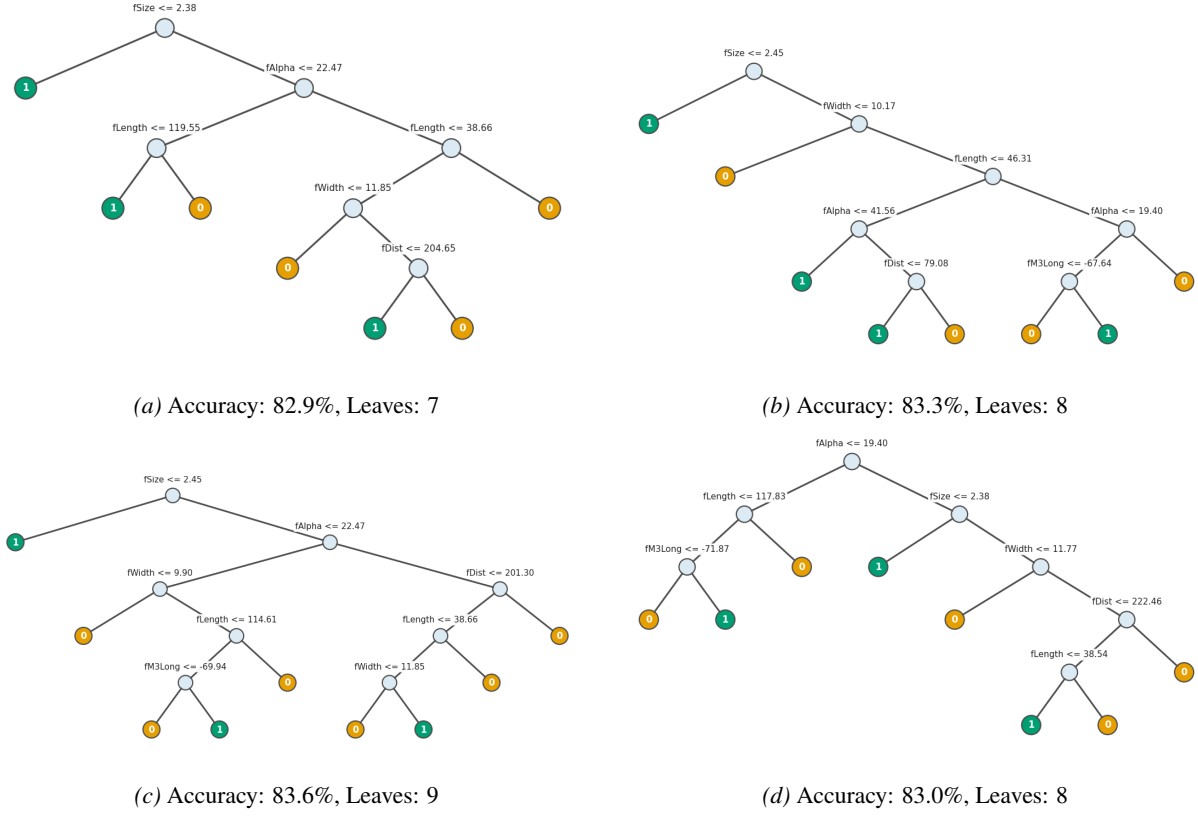

*(a)* Accuracy: 82.9%, Leaves: 7          *(b)* Accuracy: 83.3%, Leaves: 8

*(c)* Accuracy: 83.6%, Leaves: 9          *(d)* Accuracy: 83.0%, Leaves: 8

*Figure 13.* **Magic**. Example near-optimal trees with accuracy and number of leaves shown below each tree.

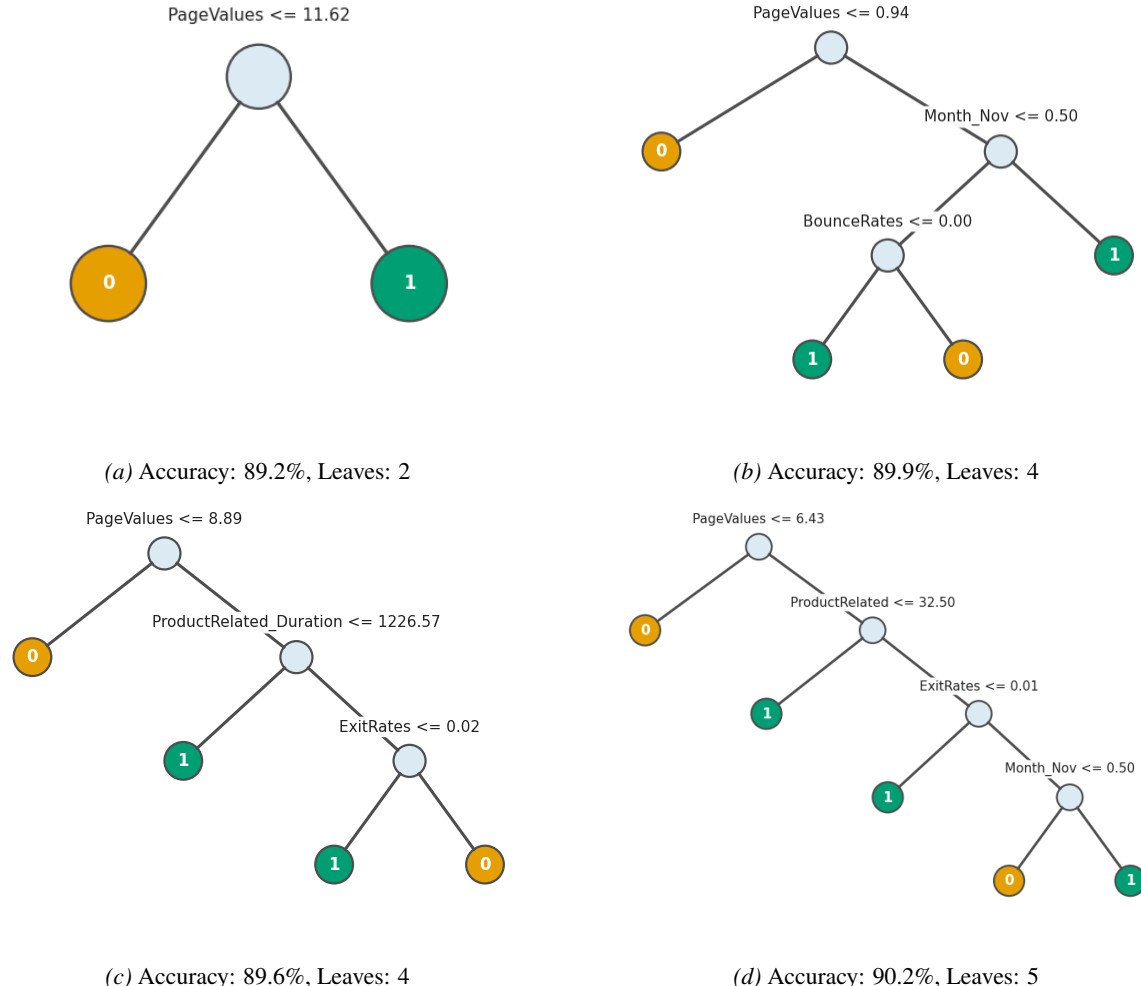

*(a)* Accuracy: 89.2%, Leaves: 2

*(b)* Accuracy: 89.9%, Leaves: 4

*(c)* Accuracy: 89.6%, Leaves: 4

*(d)* Accuracy: 90.2%, Leaves: 5

*Figure 14.* **Shopping**. Example near-optimal trees with accuracy and number of leaves shown below each tree.

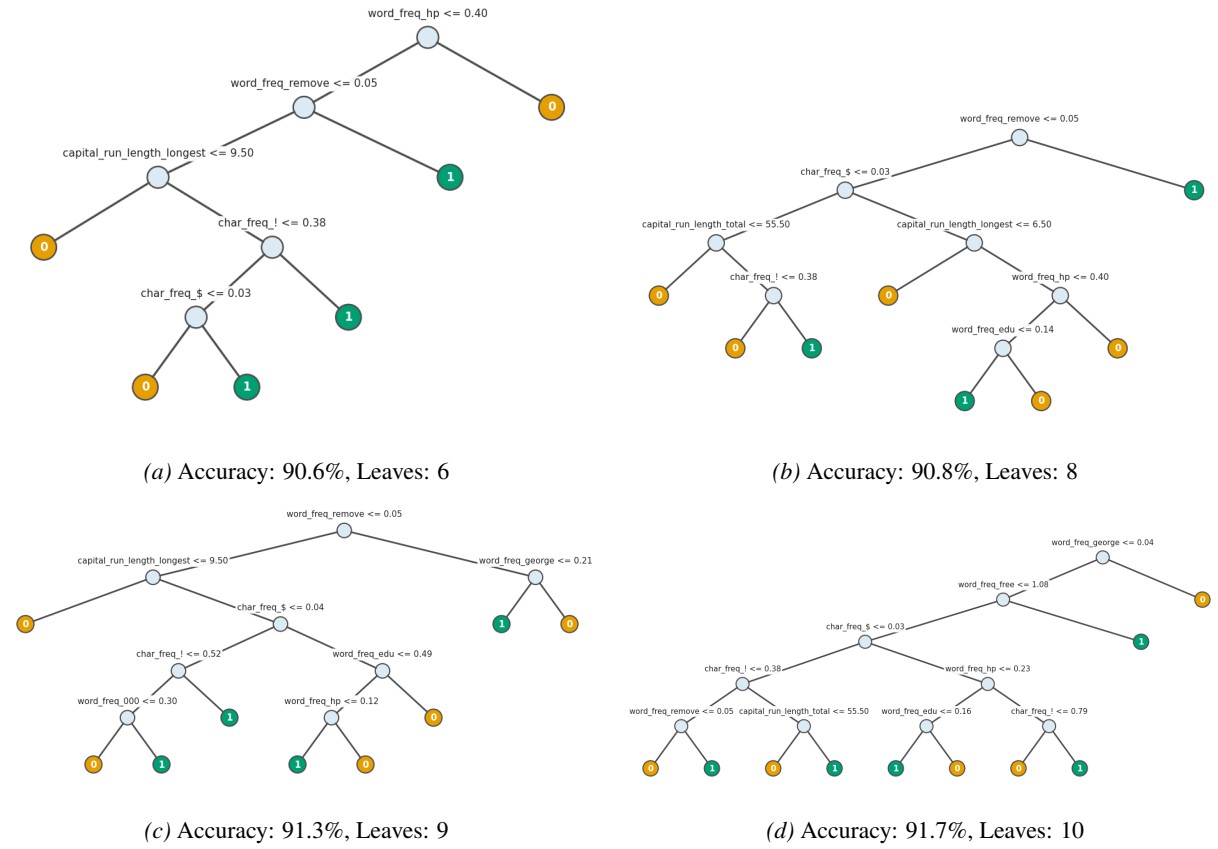

*(a)* Accuracy: 90.6%, Leaves: 6

*(b)* Accuracy: 90.8%, Leaves: 8

*(c)* Accuracy: 91.3%, Leaves: 9

*(d)* Accuracy: 91.7%, Leaves: 10

*Figure 15.* **Spambase**. Example near-optimal trees with accuracy and number of leaves shown below each tree.

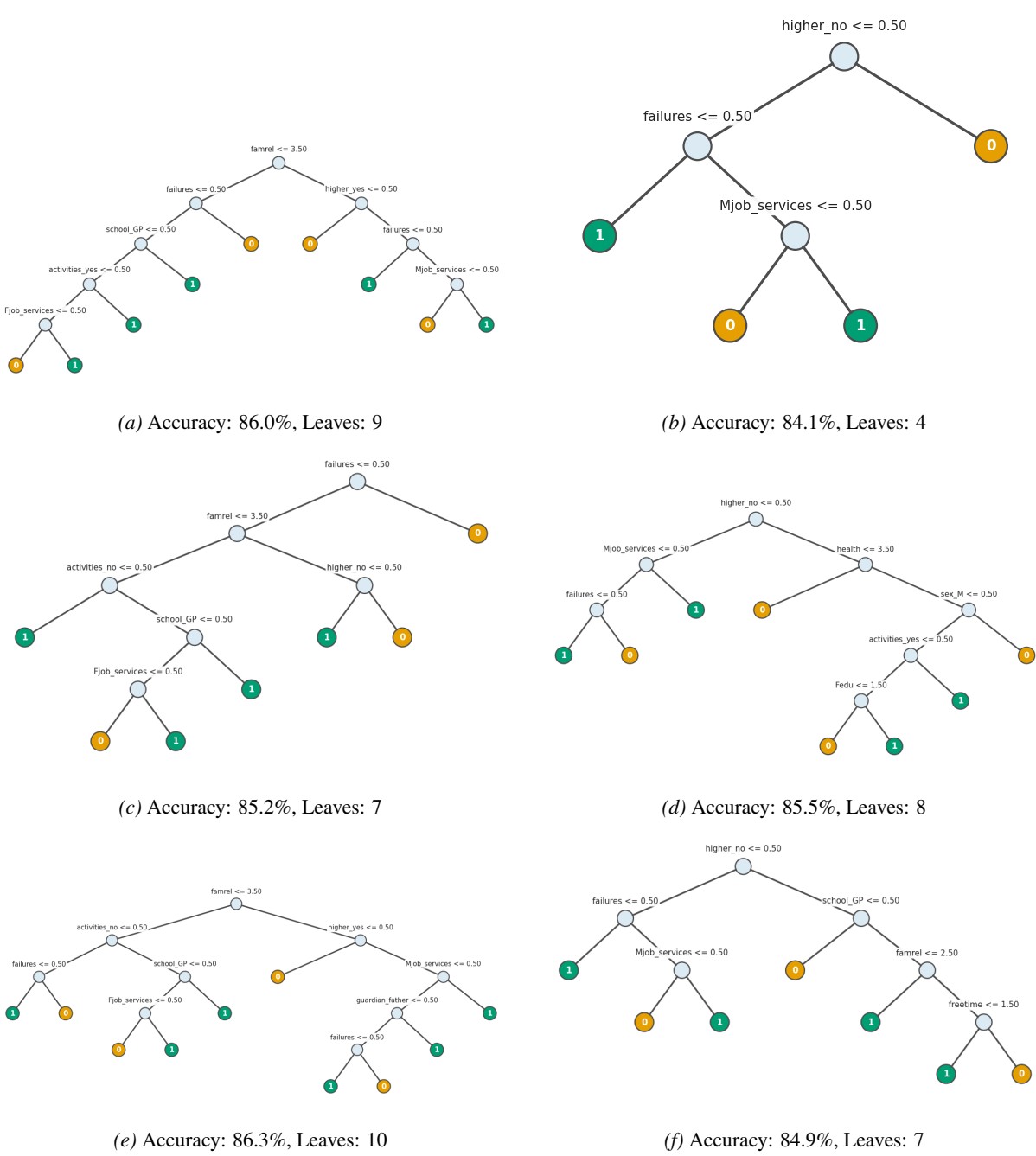

*(a)* Accuracy: 86.0%, Leaves: 9

*(b)* Accuracy: 84.1%, Leaves: 4

*(c)* Accuracy: 85.2%, Leaves: 7

*(d)* Accuracy: 85.5%, Leaves: 8

*(e)* Accuracy: 86.3%, Leaves: 10

*(f)* Accuracy: 84.9%, Leaves: 7

*Figure 16.* **Student**. Example near-optimal trees with accuracy and number of leaves shown below each tree.

