# OpenReview forum: "From Rashomon Theory to PRAXIS: Efficient Decision Tree Rashomon Sets"
_ICML.cc/2026/Conference — ICML 2026 regular_

### Official Review · Reviewer_KYCy · 2026-03-08

**Soundness:** 3
**Presentation:** 2
**Significance:** 3
**Originality:** 2
**Overall Recommendation:** 5
**Confidence:** 3

**Summary:**

The paper presents a novel approach to approximating the Rashomon set of decision trees learned on binary data.  The idea underlying it is to use an efficient method to approximate the quality of a split that enables (substantial) pruning of solutions that violate the given budget. This approximation gives wins in terms of both run time and memory usage.  The authors evaluate the approach on 50 datasets involving varying numbers of examples and features.

**Compliance With Llm Reviewing Policy:**

Affirmed.

**Final Justification:**

The paper is clearly above the bar. The paper mixes theoretical and empirical analysis. The results are strong and interesting. It is well written and accessible. The authors answered all questions in the rebuttal.

**Key Questions For Authors:**

How much of the memory savings vs. e.g, RESPLIT comes from the hash based representation of the caching?

Can you briefly summarize on an intuitively level the key changes made to Licketysplit to incorporate it into praxis?

Do you have a sense about how much of the memory win comes from the hash based caching vs. previous schemes?

Where’s the overhead in memory usage in the dataload coming from?

**Limitations:**

Yes.

**Strengths And Weaknesses:**

The paper contains a nice mix of theoretical and empirical analysis.

Section 2 gives some good intuitions of the algorithm. The algorithm itself is densely explained and notationally heavy, which makes it more difficult / time consuming to get through. E.g., I’ve seen simpler (and shorter) definitions of a binary decision tree.
It seems that the key wins on memory come from a combination of the pruning that the proxy routine enables and the hash-based caching scheme.

Some of the material pushed to the appendix is crucial for understanding the paper whereas the paper has some components that are not essential (i.e., belong in the appendix) or take up a lot of space without adding much (imo).
*The three paragraphs on Finding optimal trees could go to the appendix; this isn’t related to any of the central claims of the paper and isn’t mentioned in the intro or conclusions. It could be a sentence or two with a pointer to the appendix
*Fig 1 also does not add much imo, i.e. it did not help me understand the claim or further convince me (the text + tables were sufficient to see this point).

In contrast, the following points are essential:
*Describing the modifications to LicketySplit. This seems to be a key point of the algorithm but it is only in the appendix
*Aspects of Algorithm 1 are deferred to the appendix

A more minor point in this regard would be that summarizing the and/or graph representation would make the paper a bit more self contained.

Given that both RESPLIT and PRAXIS make use of licketsplit, saying a bit more than one sentence about their differences would be helpful.

Technically, you would need some extra memory to run the algorithm (it can indeed to be negligible) so it would seem that 0.00 should be < 0.01. Unless the Load MB has some of the data structures for the algorithms baked in. The load MB seems higher than would be necessary for the datasets.  E.g., ~50k examples  and 209 features for adult should be more like 50 MB if you use full 4 byte ints to represent each feature whereas table 1 indicates ~270 MB.

The paper could also acknowledge that RESPLIT does use less memory than PRAXIS on 10 of the datasets.

Minor points:
both exact decision tree Rashomon set works -> sentence missing an and?

Timing and Memory. Table ?? -> missing table ref

---

> ### Author Rebuttal · Authors · 2026-03-31
>
> Thank you for your thoughtful review.
>
>
> > Can you briefly summarize on an intuitive level the key changes made to Licketysplit to incorporate it into praxis?
>
> We implemented LicketySPLIT in C++ (removing dependencies on GOSDT/SPLIT) and improved it in 3 key ways. None of these are necessary for LicketySPLIT to be used as a proxy algorithm, but they are natural modifications to improve its efficiency and/or accuracy.
> - We implement subproblem caching in LicketySPLIT and its greedy subroutine for efficiency; these persist across the execution of PRAXIS, not just the LicketySPLIT call.
> - The greedy subroutine takes the objective minimizing split when the remaining depth is 1, where such behaviour is optimal (but it still uses entropy minimization for other depths).
> - We removed an early stopping condition in LicketySPLIT to improve the proxy quality (if the greedy subroutine made no splits on a subproblem, then the original LicketySPLIT would recurse no further; we now take the better of the recursive solution and a leaf).
>
> > How much of the memory savings vs. e.g, RESPLIT comes from the hash based representation of the caching?
> >
> > Do you have a sense about how much of the memory win comes from the hash based caching vs. previous schemes?
>
> We use roughly 2-3x more memory when we disable hash-based caching and use the next-best alternative. Tables 4 and 5 in the appendix have more details on the ablation and a breakdown of these results per dataset.
>
> > Where’s the overhead in memory usage in the dataload coming from?
> >
> > The load MB seems higher than would be necessary for the datasets. E.g., ~50k examples and 209 features for adult…
>
> Thank you for bringing this up. We explain below how we made these measures, what overheads were potential confounds, and how we plan to clarify this.
>
> Load MB in table 1 was measured as peak resident set size just after importing the requisite libraries and loading the dataset with pandas. This means that load_MB also captures (a) overhead from importing packages and (b) inefficiencies in the pandas read_csv() mechanism (we believe pandas is reading the csv temporarily, and then converting the data types out of place, thereby reserving twice as much memory as needed).
>
> Below, we show the peak and current resident set sizes at three stages: (a) after imports and just before loading the data (peak/curr_mb_lib), (b) just after loading the data and before running PRAXIS (peak/curr_mb_load), and (c) just after running PRAXIS (peak/curr_mb_end), for adult as well as one of our small and one of our large datasets. The dataframe size itself is similar to what you had anticipated for this dataset (with some additional pandas representation overhead). The actual memory being used just after loading the data is roughly the data for imports + twice the memory for the dataset.
>
>
> | Dataset   | curr_mb_lib | curr_mb_load | curr_mb_end | df_size_mb | peak_mb_lib | peak_mb_load | peak_mb_end |
> |-----------|------------:|-------------:|------------:|-----------:|------------:|-------------:|------------:|
> | Adult     | 124.883     | 282.617      | 682.824     | 78.253     | 124.883     | 287.309      | 682.824     |
> | Compas    | 124.492     | 127.176      | 130.582     | 1.705      | 124.492     | 131.051      | 131.051     |
> | Covertype | 125.773     | 987.773      | 987.773     | 429.979    | 125.773     | 1302.488     | 1302.488    |
>
> We will update the paper to make the method of measuring memory clearer, and better separate out dataset size from other components of overhead - thank you for flagging this as important to clarify. Load_MB + $\Delta MB$ from Table 1 does remain a valid upper bound for each method’s peak memory usage (and gives the minimum amount of memory needed to run the entire script with pandas).
>
> > Given that both RESPLIT and PRAXIS make use of licketsplit, saying a bit more than one sentence about their differences would be helpful.
>
> We are happy to provide additional characterization of this difference in the paper. Note that unlike PRAXIS, RESPLIT does not make use of LicketySPLIT; it replaces the leaf base case of TreeFARMS with a greedy solver to find the Rashomon set conditioned on greedy completions, then postprocesses those once to find the shallow depth Rashomon set for each prefix leaf. Importantly, the number of shallow depth Rashomon sets one must find to replace each prefix leaf can be potentially exponentially large.
>
> > Some of the material pushed to the appendix is crucial for understanding the paper…
>
> Thank you for the comments here. We are happy to use some of the extra page in our camera ready version for this if accepted.

---

> > ### Author Rebuttal · Reviewer_KYCy · 2026-04-01
> >
> > Thanks for the thoughtful response, as stated I think the paper should be accepted.
> >
> > Thanks for digging into the memory issues, I figured there was something with libraries or preallocations. It would be good to clarify this for a final version (even though I agree it does not affect your argumentation).

---

### Official Review · Reviewer_a3pW · 2026-03-12

**Soundness:** 4
**Presentation:** 3
**Significance:** 4
**Originality:** 3
**Overall Recommendation:** 5
**Confidence:** 2

**Summary:**

The paper "Efficient Rashomon Set Approximation for Decision Trees" presents a novel approach to approximate the Rashomon Set of Decision Trees. To do so, the authors propose to use the fairly general concept of Proxy Algorithms that is then distilled into a practical algorithm (called PRAXIS) for DT building under budget constraints (depth, error budget, and some penalty). There are two key components: first, the PROXY submethod offers reasonable candidates in a reasonable runtime (required by a subsequent theorem), and second, the error budget is carefully controlled during the runtime of the algorithm. The authors continue to show that with their algorithm, they can approximate the Rashomon set in a "reasonable" runtime (polynomial per tree times the size of the Rashomon Set) and with a reasonable approximation guarantee.

**Compliance With Llm Reviewing Policy:**

Affirmed.

**Final Justification:**

I am in favor of accepting the paper. I was in favor before the rebuttal and I am in favor after it. It is good stuff.

**Key Questions For Authors:**

1) Do you have a formal guarantee for your approximation quality? If so, what is it?
2) The cool part of a Rashomon Set is that we guarantee to find the $1-\varepsilon$ set of trees. In practice, since the set might be large, we will not look at all trees. Hence, it is natural to approximate it and present a user, e.g. "only" 1000 trees instead of 100 000 trees (which they will not review anyway). Since we are now losing any formal guarantee anyway , why not just compute 100 trees (i.e. via some randomization) that are in the $1-\varepsilon$ set and we are done? What is the main purpose of approximating the Rashomon Set?

**Limitations:**

yes

**Strengths And Weaknesses:**

Strength:
- Extensive paper with impressive overall contribution
- Interesting and valuable topic

Weaknesses:
- Approximation ratio is not clearly summarized, and it is debatable if approximating the Rashomon Set has an inherent value or not.


Detailed Review:

I had some trouble reviewing this paper. Not because of its topic or because of its writing-style, but because of its amount of content: Lets be honest, I am an overworked reviewer that needs to review 6 papers in 6 weeks, with on _average_ 34 pages. With 79 pages this paper is overly long and not really suited for the conference format. I understand that we as a field have now converged to this publishing style, and ICML allows an unlimited appendix, but I do not have the capacity to review such a long paper in a few weeks. Hence, my job as reviewer changes from making sure a paper is correct to making sure a paper is believable. As mentioned, there are many papers in my batch that are overly long, and I try not to view this as a negative for this paper, but my review is simply not as in-depth as one might have hoped. In particular, I have no idea if any of the math is correct, and I trust the authors that they did a good job here. Now that being said: Writing flow is good, and methodology seems sound, although the it would be extremely helpful to know the exact PROXY algorithm that is used in PRAXIS. There seems to be some graph algorithm going on, but I dont know. Evaluation is also impressive, and most of the results speak for themselves. The only real negative point I found with this paper is the fact, that there does not seem to be a guarantee of how good the approximation is in terms of quality, although I might have missed that.

---

> ### Author Rebuttal · Authors · 2026-03-31
>
> Thank you for your thoughtful review, especially for taking the time to review despite your heavy reviewer workload with so many particularly long papers.
>
> > it would be extremely helpful to know the exact PROXY algorithm that is used in PRAXIS
>
> We use a modified version of the LicketySPLIT [5] algorithm. We summarize the changes made to it in our reply to Reviewer KYCy (full details are in Algorithm 16 in the appendix). If the work is accepted, we would be happy to use the extra page in our camera ready version to discuss our default proxy algorithm in more detail.
>
> > Do you have a formal guarantee for your approximation quality? If so, what is it?
>
> We have several guarantees for recovery (and thereby quality of the Rashomon set approximation), based on aspects of the proxy algorithm. For instance, if the proxy algorithm’s optimality gap is largest at the root, we are guaranteed to recover the entire Rashomon set. Please see our response to UJ52 for some additional discussion. As you’ve highlighted, we are most excited about the strong empirical results for recovery, rather than theoretical ones: for many real-world datasets, we recover the entire Rashomon set for a perfect approximation.
>
> > The cool part of a Rashomon Set is that we guarantee to find the  set of trees. In practice, since the set might be large, we will not look at all trees. Hence, it is natural to approximate it and present a user, e.g. "only" 1000 trees instead of 100 000 trees (which they will not review anyway). Since we are now losing any formal guarantee anyway , why not just compute 100 trees (i.e. via some randomization) that are in the  set and we are done? What is the main purpose of approximating the Rashomon Set?
>
> Approximating the whole Rashomon set can be useful, for example, when we want to use Rashomon-based variable importance [1, 2]. In such applications, the user does not need to inspect each tree individually, so there is a benefit to covering the Rashomon set more broadly. Also, to the best of our knowledge, there is currently no known method for uniformly randomly sampling from decision tree Rashomon sets without first computing the whole set. Thus, understanding the distribution of the Rashomon set (or having a reasonable probability of recovering specific members satisfying secondary optimality criteria) requires approximating the whole set.
>
> We explore finding Rashomon members via randomization in Figure 3. We find that, for the same computational budget in which PRAXIS can recover the full approximate set, the randomized baseline recovers few or no Rashomon members (~10 members for three of four datasets; 0 members for the fourth). Our randomization method for this experiment runs a proxy algorithm on bootstraps of the data; other randomization methods (i.e.,  randomized CART, random forest weak learners) have been explored in prior work [3, 4], with similarly negative results for the randomization method.
>
>
>
> [1] Donnelly et al. "The Rashomon importance distribution: Getting rid of unstable, single model-based variable importance." Neurips, 2023.
>
> [2] Laberge et al. "Partial order in chaos: consensus on feature attributions in the Rashomon set."  JMLR, 2023.
>
> [3] Xin et al. "Exploring the whole Rashomon set of sparse decision trees." Neurips, 2022.
>
> [4] Arslan et al. "SORTeD Rashomon Sets of Sparse Decision Trees: Anytime Enumeration." Neurips 2025.
>
> [5] Babbar et al. "Near optimal decision trees in a SPLIT second." ICML 2025.

---

> > ### Author Rebuttal · Reviewer_a3pW · 2026-04-01
> >
> > All my questions have been answered.

---

### Official Review · Reviewer_nZN7 · 2026-03-12

**Soundness:** 4
**Presentation:** 3
**Significance:** 4
**Originality:** 3
**Overall Recommendation:** 5
**Confidence:** 4

**Summary:**

This paper introduces PROXIS, an algorithm for building Rashomon Sets of sparse decision trees (sparsity being expressed as a regularization over the number of leaves) in an approximate manner. The core idea of the approach is to use proxy completions to efficiently prune subproblems in the search tree. While the algorithm is relatively simple (iteratively exploring each possible candidate split and verifying the resulting left and right subproblems with iterative refinement of their budget), the proposed bounds and caching strategies are sound, and theoretical results are provided regarding the time and memory complexity of the algorithm and the conditions under which a given tree is guaranteed to be part of the Rashomon Set approximation.

The experiments demonstrate that the proposed approach can approximate Rashomon Sets for large datasets (up to millions of examples with hundreds of binary features) using orders of magnitude less memory and time than existing exact methods (SORTeD and TreeFARMS). At the same time, it achieves substantially higher approximation quality than existing approximate approaches (RESPLIT), practically recovering all or most (above 98%) trees in the Rashomon Set.

**Compliance With Llm Reviewing Policy:**

Affirmed.

**Key Questions For Authors:**

-	Appendix D.2 mentions cases where PRAXIS (or RESPLIT) finds trees that do not appear in the Rashomon Set returned by SORTeD or TreeFARMS, despite satisfying the prescribed tolerance. Does this suggest potential implementation issues in these libraries, or is there another explanation? (e.g., pruning strategies based on symmetries or functional equivalence)

-	How could the proposed algorithm be extended to handle continuous features (e.g., by adding a nested loop between lines 12 and 13 of Algorithm 1 on the possible split levels of each feature) ?

-	Can the optimality gap induced by attribute binarization be bounded theoretically, or at least be estimated empirically ?

-	Did you observe consistent trends regarding the effect of the leaf regularization parameter on the recall of PRAXIS? While recall is reported for different parameter values, the results in the main paper do not seem to reveal a clear trend.

**Limitations:**

The main limitation is the reliance on binary (or binarized) features, but it is also an assumption made by all existing methods for building Rashomon Sets of decision trees to the best of my knowledge.

**Strengths And Weaknesses:**

This is a good paper on a timely topic. Rashomon Sets are increasingly studied as an important tool for trustworthy ML, in particular for interpretable, tree-based models. This paper makes a significant contribution to this field by scaling up their computation by orders of magnitude, offering a real-world tool to tackle this problem.
Strengths :

- The paper is well structured and written

- The proposed approach is relatively simple but technically sound, and a thorough theoretical characterization is also provided

- In real-world applications where Rashomon Sets are used to derive more robust statistics (such as feature importance), the proposed approximation is likely to yield the same conclusions as using the exact Rashomon Set, but at a fraction of the computational cost, making it a highly promising tool for practical use

- The method’s ability to (i) provably generate the Rashomon Set of rule lists and (ii) generate optimal or near-optimal decision trees also highlights its flexibility and demonstrates its broader potential

- Several interesting complementary analyses are provided in the Appendix, for example, on how the choice of proxy algorithm affects the performance of PRAXIS and on which trees are missed by the PRAXIS approximation. I would recommend pointing more explicitly to these analyses from the main body of the paper.

Weaknesses :

- The approach is limited to binary features. To the best of my knowledge, it is also the case for the other existing approaches building Rashomon Sets of decision trees (TreeFARMS, SORTeD and RESPLIT), but it could still be interesting to quickly discuss how the proposed algorithm could be extended (or not) to handle continuous features (e.g., by adding a nested loop between lines 12 and 13 of Algorithm 1 ?), and how much this limitation may affect optimality (with respect to the true, non-binarized dataset)

Minor :

- References Hu et al. (2019a) and Hu et al. (2019b) are duplicates

- Algorithm 2: typo on line 11 ($\epsilon_{abs}$)

- Section 4, page 6: “Table ??” (should probably be “Table 1”)

---

> ### Author Rebuttal · Authors · 2026-03-31
>
> Thank you for your thoughtful review.
>
> > Appendix D.2 mentions cases where PRAXIS (or RESPLIT) finds trees that do not appear in the Rashomon Set returned by SORTeD or TreeFARMS, despite satisfying the prescribed tolerance. Does this suggest potential implementation issues in these libraries, or is there another explanation?
>
> Since submitting this manuscript, we have discovered that SORTeD’s errors are fully resolved by accounting for one preprocessing choice: when two binarized features are identical with respect to the training set, the method will keep only one of the binarized features (even if they are splits on different continuous features), and will find the Rashomon set without consideration of the other feature. This preprocessing is perfectly reasonable for finding a single optimal tree (so presumably this is an artifact of the StreeD library upon which SORTeD is built), but the choice leads to a number of missed trees in the Rashomon set. If we constrain all our datasets not to contain binarized features that are identical with respect to the training set, SORTeD exactly matches PRAXIS with an optimal proxy. We have unfortunately not yet isolated the cause for TreeFARMS, and suspect it is an issue with that package’s implementation; for this reason we did not use TreeFARMS for recall computations.
>
> Since the comparisons reported in the main paper were already computed conservatively (we used our own implementation with an optimal proxy to catch any potential missing Rashomon members), the bug we have found in SORTeD does not affect our reported recall results.
> > How could the proposed algorithm be extended to handle continuous features (e.g., by adding a nested loop between lines 12 and 13 of Algorithm 1 on the possible split levels of each feature)?
>
> We view this as a particularly exciting direction for future work.
>
> We should first note that the current version of PRAXIS can recover an exact solution for continuous features with optimality-preserving binarization (that is, taking every possible cutpoint which separates training samples in the data); we present results for ($\lambda, \varepsilon$) showing that this is substantially more feasible for PRAXIS than for TreeFARMS or SORTED in tables 1-3 at this link: https://docs.google.com/document/d/e/2PACX-1vTyS7j8ACEiAqmyHiNAxNGNVyGjcMxkTD-w2Fha0RyePYNmCFKmzReXwAsOAjkDjMrrCR_67q-bSV6g/pub .
>
> We can handle continuous features better with the nested-for loop you mentioned, and the following:
>
> - Proxies that are designed to work efficiently with continuous features, such as CART (or the greedy component of a LicketySPLIT algorithm) can be designed to leverage those benefits and work directly on the continuous features.
> - As we explore in Appendix D.5.4, it is possible to use different levels of binarization for the proxy and for the main loop in PRAXIS; this gives the ability to optimize tradeoffs between the search space explored (which could be equal or close to an exhaustive binarization) and the computational complexity of the proxies (which could be minimized by taking fewer thresholds).
>
> > Can the optimality gap induced by attribute binarization be bounded theoretically, or at least be estimated empirically?
>
> For binning into m quantiles, one can show (with a union bound) that the optimality gap is below the number of leaves divided by 2m (for misclassification scores in [0,1]); we can provide a proof if this is of interest.
>
> There are a few empirical estimates of the impact of threshold guessing in existing work [1, 2, 3]. Tables 4 and 5 in the appendix of [1] show that usually there is minimal drop in train and test accuracy for optimal trees (and sometimes a benefit to test accuracy, when tuning the threshold guessing hyperparameters). Table 6 in the appendix of [3] shows that threshold guessing frequently yields comparable or better test objectives for binary classification problems. Figure 13 in the appendix of [2] shows little to no gap when running SPLIT and LicketySPLIT with versus without this binarization; meaning our default proxy approach is likely to be minimally affected.
>
> > Did you observe consistent trends regarding the effect of the leaf regularization parameter on the recall of PRAXIS?
>
> We ran some additional small regularization experiments in Table 4 at the above link. We do not observe consistent trends regarding the effect of leaf regularization on recall for PRAXIS. We also provide recall results for the baseline RESPLIT in our response to reviewer UJ52; these may be of interest to you because we see that, in contrast to PRAXIS, the baseline RESPLIT’s recall does degrade as regularization decreases.
>
> [1] McTavish et al. "Fast sparse decision tree optimization via reference ensembles." AAAI, 2022.
>
> [2] Babbar et al. "Near optimal decision trees in a SPLIT second." ICML 2025.
>
> [3] Brița et al. "Optimal classification trees for continuous feature data using dynamic programming with branch-and-bound." AAAI 2025.

---

> > ### Author Rebuttal · Reviewer_nZN7 · 2026-04-03
> >
> > Thank you for your detailed answer (in particular, for the additional experiments with continuous features with optimality-preserving binarization). I confirm that it appropriately addresses all my questions. Since my original evaluation of the paper was already “5 – Accept”, I am maintaining my rating.
> >
> > Yet, I find it a bit weird that two binarized features are identical across the entire training set. I won’t argue for this, as it does not affect the paper’s findings, but it is a strong signal that binarization might have been performed a bit aggressively.
> >
> > On the positive side, I would also like to point out that I found the discussion with Reviewer UJ52 regarding how the proxy algorithm's approximation guarantees translate to approximation guarantees (in terms of recall) for the Rashomon Set very interesting (and worth including in the final paper). Stated as is, it gives a more intuitive interpretation of Theorem 3.5's implications, and I think this comment is shared across reviewers, as Reviewer a3pW also noted that the guarantees stated in the original version of the paper regarding approximation quality were a bit unclear (“that there does not seem to be a guarantee of how good the approximation is in terms of quality”).

---

### Official Review · Reviewer_UJ52 · 2026-03-12

**Soundness:** 2
**Presentation:** 3
**Significance:** 3
**Originality:** 3
**Overall Recommendation:** 5
**Confidence:** 4

**Summary:**

The paper proposes PRAXIS, a novel approximation algorithm for computing the Rashomon Set of decision trees by utilizing a search graph representation of the Rashomon Set and a PROXY algorithm that can compute decision trees. The paper proposes various theoretical guarantees on the behaviour of PRAXIS, including w.r.t the runtime. Experiments validate that the proposed approach shows orders of magnitude lower runtime and memory compared to optimal approaches (SORTeD, TreeFARMS), and also outperforms a previous approximation algorithm (RESPLIT).

**Compliance With Llm Reviewing Policy:**

Affirmed.

**Final Justification:**

My concerns have been addressed. I Increased my evaluation.

**Key Questions For Authors:**

See "weaknesses" above.

**Limitations:**

While there is no discussion of limitations, I did not identify anything concrete that needs to be addressed.

**Strengths And Weaknesses:**

## Strengths:
- Novel approximation algorithm for computing the Rashomon Set for decision trees.
- The paper provides various theoretical guarantees for the algorithm, including w.r.t the run time (however, it seems, not w.r.t to performance, i.e., recall - see under weaknesses).
- Experiments show PRAXIS outperforms the baselines in both time and memory efficiency, often by order of magnitude, compared to optimal approaches as well as existing approximation (RESPLIT). Further, it is shown that PRAXIS obtains a high (near perfect) recall.
  * Importantly PRAXIS is also shown to scale better with depth compared to RESPLIT.
- Paper is mostly clear and organized well.


## Weaknesses:

Concerns regarding to the proposed methodology:
  - The proposed approximation does not seem to provide theoretical guarantees w.r.t the quality of the generated Rashomon Set, i.e., the recall. Specifically, a non-optimal PROXY can cause incorrect pruning. It is not clear whether assuming a PROXY algorithm that satisfies a given approximation bound will enable to provide guarantees w.r.t PRAXIS’s recall. Perhaps it will at least allow principled computation of the “slack” proposed at the end of Section 3.3.
  - Theorem 3.5: it is not clear what is $T$ (this notation does not seem to be defined earlier).
  - It is not clear whether this approach is limited to classification or can easily be extended to regression trees?
  - The framework requires binarization of features which likely reduce the size of the Rashomon Set (compared to the unbinarized Rashomon Set).

Concerns regarding experimental analysis:
  - The comparison in table 1 assumes baselines run until completion, but it is not clear if that is needed. For example, if they generate intermediate solutions and can be stopped at each point and outputs the existing set, then a comparison of performance over time would be more appropriate (an algorithm should not be penalized if it can, but not must, run for longer and produce potentially better results).
  - It is not clear why approximation quality (Table 2) is not reported for RESPLIT.

Writing/Minor:
  - The paper does not actually motivates the computation of Rashomon Set (e.g., by demonstrating the benefits of computing the full Rashomon set). While it does rely on existing literature to establish the significance of the problem, it could be better highlighted in the paper (e.g., via examples, references for actual use cases, etc.)
- Table reference missing line 318 under Timing and Memory: “Table ??”

---

> ### Author Rebuttal · Authors · 2026-03-31
>
> Thank you for your thoughtful review.
>
> >It is not clear whether assuming a PROXY algorithm that satisfies a given approximation bound will enable to provide guarantees w.r.t PRAXIS’s recall. Perhaps it will at least allow principled computation of the “slack” proposed at the end of Section 3.3.
>
> We would be happy to add the below claim to the appendix:
>
> Let $c$ be the guaranteed approximation ratio of some proxy algorithm, and let $\\varepsilon_\textrm{abs}^\textrm{true}$ be the target Rashomon set bound. Then, PRAXIS with this proxy will recover this Rashomon set (that is, achieve recall of 1) if provided an absolute bound of $\\varepsilon_\textrm{abs} = c \\varepsilon_\textrm{abs}^\textrm{true}$.
>
>
> The proof is omitted here due to space constraints. We are happy to provide it upon request during the author-reviewer discussion period; it follows from applying Theorem 3.5. Note that this is a conservative claim; when an algorithm is a c-approximation, we can still provably recover the entire Rashomon set if the proxy is a no-worse approximation for shallower subproblems (via Corollary 3.6).
>
> > Theorem 3.5: it is not clear what is $T$ (this notation does not seem to be defined earlier).
>
> $T$ in theorem 3.5 should be lower case t, the tree under consideration. Apologies for the typo.
>
> > It is not clear whether this approach is limited to classification or can easily be extended to regression trees?
>
> This approach can trivially be extended to piecewise regression (by changing the PRAXIS base case to predict the mean, and by using a proxy that works for regression).
>
> > The framework requires binarization of features which likely reduce the size of the Rashomon Set (compared to the unbinarized Rashomon Set).
>
> Binarization does not necessarily lose information -  it is possible to find the unbinarized Rashomon set by binarizing conservatively - that is, considering thresholds between all realized feature values in the training set.  Please see our response to reviewer nZN7 for experimental results and a brief discussion of the benefits from PRAXIS relative to baselines in this setting. A common limitation of all current approaches for Rashomon sets is their scaling with the number of binarized features, so in practice (as in this paper), we tend to binarize less conservatively to run faster. Finding ways to more efficiently handle all split points in continuous features is an excellent direction for future work (we also discuss some directions here in our response to reviewer nZN7).
>
>
>
> > The comparison in table 1 assumes baselines run until completion, but it is not clear if that is needed. For example, if they generate intermediate solutions and can be stopped at each point and outputs the existing set, then a comparison of performance over time would be more appropriate (an algorithm should not be penalized if it can, but not must, run for longer and produce potentially better results).
>
> Among our baselines, the only support for intermediate results we’re aware of is for SORTeD, which can give the top k trees rather than returning the whole set. However, to get even the first tree, SORTeD needs to first run an optimal tree algorithm to completion, which dominates the runtime relative to extracting trees. In Figure 1 of the linked results (https://docs.google.com/document/d/e/2PACX-1vQ0sHCX6zXq5S_inawBoNwsIldQikeAGwWh0ChmYnljQqG_BmyVNoztWSvl1Avcp18CMw7RbAUtURKn/pub ), we show a plot of runtime-recall tradeoffs for SORTeD when we ask for a range of trees (from just one to the full Rashomon set). Note that often the runtime is so similar for different numbers of requested trees, relative to the full runtime, that the plots show up as a single indicator at recall = 1. In other words, getting intermediate solutions comes at almost no runtime benefit but substantially hurts recall.
>
> For completeness, we note that SORTeD also has a parameter to exit early if it exceeds a time limit; however, the fit() call consistently freezes when using this parameter if we attempt to set a timeout below the time needed to recover all trees equal in objective to the optimum.
>
> > It is not clear why approximation quality (Table 2) is not reported for RESPLIT.
>
> There is a qualitative analysis of approximation quality in Appendix D.10. For completeness, we provide the RESPLIT results for Table 2 from the main paper in the linked results (https://docs.google.com/document/d/e/2PACX-1vQ0sHCX6zXq5S_inawBoNwsIldQikeAGwWh0ChmYnljQqG_BmyVNoztWSvl1Avcp18CMw7RbAUtURKn/pub).
>
> > Writing/Minor
>
> We appreciate these corrections/suggestions, and are happy to incorporate them!

---

> > ### Author Rebuttal · Reviewer_UJ52 · 2026-04-02
> >
> > My concerns have been addressed.

---

### Decision · Program_Chairs · 2026-04-30

**Decision:**

Accept (regular)

**Comment:**

A very nice paper, for which all reviewers unanimously agreed on the quality and importance of the results. I, for myself, appreciated the extensive experiments. Congratulations to the authors for their work.

In the final version, I would recommend that the authors acknowledge the work done by reviewers. See for example nZN7 as an illustration of the reviewers diligence into not just getting their opinion into shape, but also using others' to elevate their own.

From a technical standpoint, it is also important to include in the final version some feedback: memory considerations (KYCy) and try to develop on the argument raised by nZN7 on binarization (I agree with the reviewer's remark).